# HOPFIELD NETWORKS IS ALL YOU NEED

**Hubert Ramsauer**[*]   **Bernhard Schäfl**[*]   **Johannes Lehner**[*]   **Philipp Seidl**[*]
**Michael Widrich**[*]   **Thomas Adler**[*]   **Lukas Gruber**[*]   **Markus Holzleitner**[*]
**David Kreil**[†]   **Michael Kopp**[†]   **Günter Klambauer**[*]   **Johannes Brandstetter**[*]
**Sepp Hochreiter**[*,†]

[*]ELLIS Unit Linz, LIT AI Lab, Institute for Machine Learning,
 Johannes Kepler University Linz, Austria
[†]Institute of Advanced Research in Artificial Intelligence (IARAI)
Email: {ramsauer,schaefl,brandstetter,hochreit}@ml.jku.at

## ABSTRACT

We introduce a modern Hopfield network with continuous states and a corresponding update rule. The new Hopfield network can store exponentially (with the dimension of the associative space) many patterns, retrieves the pattern with one update, and has exponentially small retrieval errors. It has three types of energy minima (fixed points of the update): (1) global fixed point averaging over all patterns, (2) metastable states averaging over a subset of patterns, and (3) fixed points which store a single pattern. The new update rule is equivalent to the attention mechanism used in transformers. This equivalence enables a characterization of the heads of transformer models. These heads perform in the first layers preferably global averaging and in higher layers partial averaging via metastable states. The new modern Hopfield network can be integrated into deep learning architectures as layers to allow the storage of and access to raw input data, intermediate results, or learned prototypes. These Hopfield layers enable new ways of deep learning, beyond fully-connected, convolutional, or recurrent networks, and provide pooling, memory, association, and attention mechanisms. We demonstrate the broad applicability of the Hopfield layers across various domains. Hopfield layers improved state-of-the-art on three out of four considered multiple instance learning problems as well as on immune repertoire classification with several hundreds of thousands of instances. On the UCI benchmark collections of small classification tasks, where deep learning methods typically struggle, Hopfield layers yielded a new state-of-the-art when compared to different machine learning methods. Finally, Hopfield layers achieved state-of-the-art on two drug design datasets. The implementation is available at: https://github.com/ml-jku/hopfield-layers

## 1  INTRODUCTION

The deep learning community has been looking for alternatives to recurrent neural networks (RNNs) for storing information. For example, linear memory networks use a linear autoencoder for sequences as a memory (Carta et al., 2020). Additional memories for RNNs like holographic reduced representations (Danihelka et al., 2016), tensor product representations (Schlag & Schmidhuber, 2018; Schlag et al., 2019) and classical associative memories (extended to fast weight approaches) (Schmidhuber, 1992; Ba et al., 2016a;b; Zhang & Zhou, 2017; Schlag et al., 2021) have been suggested. Most approaches to new memories are based on attention. The neural Turing machine (NTM) is equipped with an external memory and an attention process (Graves et al., 2014). Memory networks (Weston et al., 2014) use an arg max attention by first mapping a query and patterns into a space and then retrieving the pattern with the largest dot product. End to end memory networks (EMN) make this attention scheme differentiable by replacing arg max through a softmax (Sukhbaatar et al., 2015a;b). EMN with dot products became very popular and implement a key-value attention (Daniluk et al., 2017) for self-attention. An enhancement of EMN is the transformer (Vaswani et al., 2017a;b) and its

extensions (Dehghani et al., 2018). The transformer has had a great impact on the natural language processing (NLP) community, in particular via the BERT models (Devlin et al., 2018; 2019).

**Contribution of this work:** (i) introducing novel deep learning layers that are equipped with a memory via modern Hopfield networks, (ii) introducing a novel energy function and a novel update rule for continuous modern Hopfield networks that are differentiable and typically retrieve patterns after one update. Differentiability is required for gradient descent parameter updates and retrieval with one update is compatible with activating the layers of deep networks.

We suggest using modern Hopfield networks to store information or learned prototypes in different layers of neural networks. Binary Hopfield networks were introduced as associative memories that can store and retrieve patterns (Hopfield, 1982). A query pattern can retrieve the pattern to which it is most similar or an average over similar patterns. Hopfield networks seem to be an ancient technique, however, new energy functions improved their properties. The stability of spurious states or metastable states was sensibly reduced (Barra et al., 2018). The largest and most impactful successes are reported on increasing the storage capacity of Hopfield networks. In a $d$-dimensional space, the standard Hopfield model can store $d$ uncorrelated patterns without errors but only $Cd/\log(d)$ random patterns with $C < 1/2$ for a fixed stable pattern or $C < 1/4$ if all patterns are stable (McEliece et al., 1987). The same bound holds for nonlinear learning rules (Mazza, 1997). Using tricks-of-trade and allowing small retrieval errors, the storage capacity is about $0.138d$ (Crisanti et al., 1986; Hertz et al., 1991; Torres et al., 2002). If the learning rule is not related to the Hebb rule, then up to $d$ patterns can be stored (Abu-Mostafa & StJacques, 1985). For Hopfield networks with non-zero diagonal matrices, the storage can be increased to $Cd\log(d)$ (Folli et al., 2017). In contrast to the storage capacity, the number of energy minima (spurious states, stable states) of Hopfield networks is exponential in $d$ (Tanaka & Edwards, 1980; Bruck & Roychowdhury, 1990; Wainrib & Touboul, 2013).

The standard binary Hopfield network has an energy function that can be expressed as the sum of interaction functions $F$ with $F(x) = x^2$. Modern Hopfield networks, also called "dense associative memory" (DAM) models, use an energy function with interaction functions of the form $F(x) = x^n$ and, thereby, achieve a storage capacity proportional to $d^{n-1}$ (Krotov & Hopfield, 2016; 2018). The energy function of modern Hopfield networks makes them robust against adversarial attacks (Krotov & Hopfield, 2018). Modern binary Hopfield networks with energy functions based on interaction functions of the form $F(x) = \exp(x)$ even lead to storage capacity of $2^{d/2}$, where all stored binary patterns are fixed points but the radius of attraction vanishes (Demircigil et al., 2017). However, in order to integrate Hopfield networks into deep learning architectures, it is necessary to make them differentiable, that is, we require continuous Hopfield networks (Hopfield, 1984; Koiran, 1994).

Therefore, we generalize the energy function of Demircigil et al. (2017) that builds on exponential interaction functions to continuous patterns and states and obtain a new modern Hopfield network. We also propose a new update rule which ensures global convergence to stationary points of the energy (local minima or saddle points). We prove that our new modern Hopfield network typically retrieves patterns in one update step ($\epsilon$-close to the fixed point) with an exponentially low error and has a storage capacity proportional to $c^{\frac{d-1}{4}}$ (reasonable settings for $c = 1.37$ and $c = 3.15$ are given in Theorem 3). The retrieval of patterns with one update is important to integrate Hopfield networks in deep learning architectures, where layers are activated only once. Surprisingly, our new update rule is also the key-value attention as used in transformer and BERT models (see Fig. 1). Our modern Hopfield networks can be integrated as a new layer in deep learning architectures for pooling, memory, prototype learning, and attention. We test these new layers on different benchmark datasets and tasks like immune repertoire classification.

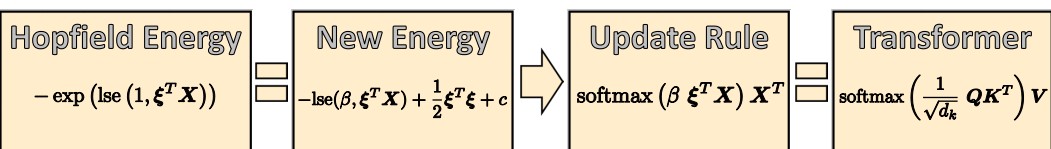

Figure 1: We generalize the energy of binary modern Hopfield networks to continuous states while keeping fast convergence and storage capacity properties. We also propose a new update rule that minimizes the energy. The new update rule is the attention mechanism of the transformer. Formulae are modified to express $\mathrm{softmax}$ as row vector. "="-sign means "keeps the properties".

## 2 MODERN HOPFIELD NETS WITH CONTINUOUS STATES

**New energy function for continuous state Hopfield networks.** In order to integrate modern Hopfield networks into deep learning architectures, we have to make them continuous. To allow for continuous states, we propose a new energy function that is a modification of the energy of modern Hopfield networks (Demircigil et al., 2017). We also propose a new update rule which can be proven to converge to stationary points of the energy (local minima or saddle points).

We have $N$ stored (key) patterns $\boldsymbol{x}_i \in \mathbb{R}^d$ represented by the matrix $\boldsymbol{X} = (\boldsymbol{x}_1, \ldots, \boldsymbol{x}_N)$ with the largest pattern $M = \max_i \|\boldsymbol{x}_i\|$. The state (query) pattern is $\boldsymbol{\xi} \in \mathbb{R}^d$. For exponential interaction functions, we need the *log-sum-exp function* (lse) for $0 < \beta$

$$\text{lse}(\beta, \boldsymbol{x}) \; = \; \beta^{-1} \log \left( \sum_{i=1}^{N} \exp(\beta x_i) \right) \, , \tag{1}$$

which is convex (see appendix Eq. (461), and Lemma A22). The energy function E of the modern Hopfield networks for binary patterns $\boldsymbol{x}_i$ and a binary state pattern $\boldsymbol{\xi}$ is $\text{E} = -\sum_{i=1}^{N} F\left(\boldsymbol{\xi}^T \boldsymbol{x}_i\right)$ (Krotov & Hopfield, 2016). Here, $F(x) = x^n$ is the interaction function, where $n = 2$ gives the classical Hopfield network. The storage capacity is proportional to $d^{n-1}$ (Krotov & Hopfield, 2016). This model was generalized by Demircigil et al. (2017) to exponential interaction functions $F(x) = \exp(x)$ which gives the energy $\text{E} = -\exp(\text{lse}(1, \boldsymbol{X}^T\boldsymbol{\xi}))$. This energy leads to an exponential storage capacity of $N = 2^{d/2}$ for binary patterns. Furthermore, with a single update, the fixed point is recovered with high probability for random patterns. However, still this modern Hopfield network has binary states.

We generalize this energy function to continuous-valued patterns while keeping the properties of the modern Hopfield networks like the exponential storage capacity and the extremely fast convergence (see Fig. 1). For the new energy we take the logarithm of the negative energy of modern Hopfield networks and add a quadratic term of the current state. The quadratic term ensures that the norm of the state vector $\boldsymbol{\xi}$ remains finite and the energy is bounded. Classical Hopfield networks do not require to bound the norm of their state vector, since it is binary and has fixed length. We define the novel energy function E as

$$\text{E} \; = \; -\text{lse}(\beta, \boldsymbol{X}^T\boldsymbol{\xi}) \; + \; \frac{1}{2}\boldsymbol{\xi}^T\boldsymbol{\xi} \; + \; \beta^{-1}\log N \; + \; \frac{1}{2}M^2 \, . \tag{2}$$

We have $0 \leqslant \text{E} \leqslant 2M^2$ (see appendix Lemma A1). Using $\boldsymbol{p} = \text{softmax}(\beta\boldsymbol{X}^T\boldsymbol{\xi})$, we define a novel update rule (see Fig. 1):

$$\boldsymbol{\xi}^{\text{new}} \; = \; f(\boldsymbol{\xi}) \; = \; \boldsymbol{X}\boldsymbol{p} \; = \; \boldsymbol{X}\text{softmax}(\beta\boldsymbol{X}^T\boldsymbol{\xi}) \, . \tag{3}$$

The next theorem states that the update rule Eq. (3) converges globally. The proof uses the Concave-Convex Procedure (CCCP) (Yuille & Rangarajan, 2002; 2003), which is equivalent to Legendre minimization (Rangarajan et al., 1996; 1999) algorithms (Yuille & Rangarajan, 2003).

**Theorem 1.** *The update rule Eq. (3) converges globally: For $\boldsymbol{\xi}^{t+1} = f(\boldsymbol{\xi}^t)$, the energy $\text{E}(\boldsymbol{\xi}^t) \to \text{E}(\boldsymbol{\xi}^*)$ for $t \to \infty$ and a fixed point $\boldsymbol{\xi}^*$.*

*Proof.* The update rule in Eq. (3) is the CCCP for minimizing the energy E, which is the sum of the convex $1/2\boldsymbol{\xi}^T\boldsymbol{\xi}$ and concave $-\text{lse}$ (see details in appendix Theorem 1). Theorem 2 in Yuille & Rangarajan (2002) yields the global convergence property. Also, in Theorem 2 in Sriperumbudur & Lanckriet (2009) the global convergence of CCCP is proven via a rigorous analysis using Zangwill's global convergence theory of iterative algorithms. $\square$

The global convergence theorem only assures that for the energy $\text{E}(\boldsymbol{\xi}^t) \to \text{E}(\boldsymbol{\xi}^*)$ for $t \to \infty$ but not $\boldsymbol{\xi}^t \to \boldsymbol{\xi}^*$. The next theorem strengthens Zangwill's global convergence theorem (Meyer, 1976) and gives convergence results similar to those known for expectation maximization (Wu, 1983).

**Theorem 2.** *For the iteration Eq. (3) we have $\text{E}\left(\boldsymbol{\xi}^t\right) \to \text{E}\left(\boldsymbol{\xi}^*\right) = \text{E}^*$ as $t \to \infty$, for some stationary point $\boldsymbol{\xi}^*$. Furthermore, $\left\|\boldsymbol{\xi}^{t+1} - \boldsymbol{\xi}^t\right\| \to 0$ and either $\{\boldsymbol{\xi}^t\}_{t=0}^{\infty}$ converges or, in the other case, the set of limit points of $\{\boldsymbol{\xi}^t\}_{t=0}^{\infty}$ is a connected and compact subset of $\mathcal{L}\left(\text{E}^*\right)$, where $\mathcal{L}\left(a\right) = \{\boldsymbol{\xi} \in \mathcal{L} \mid \text{E}\left(\boldsymbol{\xi}\right) = a\}$ and $\mathcal{L}$ is the set of stationary points of the iteration Eq. (3). If $\mathcal{L}\left(\text{E}^*\right)$ is finite, then any sequence $\{\boldsymbol{\xi}^t\}_{t=0}^{\infty}$ generated by the iteration Eq. (3) converges to some $\boldsymbol{\xi}^* \in \mathcal{L}\left(\text{E}^*\right)$.*

For a proof, see appendix Theorem 2. Therefore, all the limit points of any sequence generated by the iteration Eq. (3) are stationary points (local minima or saddle points) of the energy function E. Either the iteration converges or, otherwise, the set of limit points is a connected and compact set.

The next theorem gives the results on the storage capacity of our new continuous state modern Hopfield network. We first define what we mean by storing and retrieving patterns using a modern Hopfield network with continuous states.

**Definition 1** (Pattern Stored and Retrieved). *We assume that around every pattern $\boldsymbol{x}_i$ a sphere $\mathrm{S}_i$ is given. We say $\boldsymbol{x}_i$ is stored if there is a single fixed point $\boldsymbol{x}_i^* \in \mathrm{S}_i$ to which all points $\boldsymbol{\xi} \in \mathrm{S}_i$ converge, and $\mathrm{S}_i \cap \mathrm{S}_j = \emptyset$ for $i \neq j$. We say $\boldsymbol{x}_i$ is retrieved for a given $\epsilon$ if iteration (update rule) Eq. (3) gives a point $\tilde{\boldsymbol{x}}_i$ that is at least $\epsilon$-close to the single fixed point $\boldsymbol{x}_i^* \in \mathrm{S}_i$. The retrieval error is $\|\tilde{\boldsymbol{x}}_i - \boldsymbol{x}_i\|$.*

As with classical Hopfield networks, we consider patterns on the sphere, i.e. patterns with a fixed norm. For randomly chosen patterns, the number of patterns that can be stored is exponential in the dimension $d$ of the space of the patterns ($\boldsymbol{x}_i \in \mathbb{R}^d$).

**Theorem 3.** *We assume a failure probability $0 < p \leqslant 1$ and randomly chosen patterns on the sphere with radius $M := K\sqrt{d-1}$. We define $a := \frac{2}{d-1}(1 + \ln(2\beta K^2 p(d-1)))$, $b := \frac{2K^2\beta}{5}$, and $c := \frac{b}{W_0(\exp(a + \ln(b)))}$, where $W_0$ is the upper branch of the Lambert W function (Olver et al., 2010, (4.13)), and ensure $c \geq \left(\frac{2}{\sqrt{p}}\right)^{\frac{4}{d-1}}$. Then with probability $1 - p$, the number of random patterns that can be stored is*

$$N \;\geqslant\; \sqrt{p}\, c^{\frac{d-1}{4}} \;. \tag{4}$$

*Therefore it is proven for $c \geq 3.1546$ with $\beta = 1$, $K = 3$, $d = 20$ and $p = 0.001$ ($a + \ln(b) > 1.27$) and proven for $c \geq 1.3718$ with $\beta = 1$, $K = 1$, $d = 75$, and $p = 0.001$ ($a + \ln(b) < -0.94$).*

For a proof, see appendix Theorem A5.

The next theorem states that the update rule typically retrieves patterns after one update. Retrieval of a pattern $\boldsymbol{x}_i$ for fixed point $\boldsymbol{x}_i^*$ and query $\boldsymbol{\xi}$ is defined via an $\epsilon$ by $\|f(\boldsymbol{\xi}) - \boldsymbol{x}_i^*\| < \epsilon$, that is, the update is $\epsilon$-close to the fixed point. Retrieval with one update is crucial to integrate modern Hopfield networks into deep learning architectures, where layers are activated only once. First we need the concept of separation of a pattern. For pattern $\boldsymbol{x}_i$ we define its separation $\Delta_i$ to other patterns by:

$$\Delta_i \;:=\; \min_{j, j \neq i} \left(\boldsymbol{x}_i^T \boldsymbol{x}_i - \boldsymbol{x}_i^T \boldsymbol{x}_j\right) \;=\; \boldsymbol{x}_i^T \boldsymbol{x}_i - \max_{j, j \neq i} \boldsymbol{x}_i^T \boldsymbol{x}_j \;. \tag{5}$$

The update rule retrieves patterns with one update for well separated patterns, that is, patterns with large $\Delta_i$.

**Theorem 4.** *With query $\boldsymbol{\xi}$, after one update the distance of the new point $f(\boldsymbol{\xi})$ to the fixed point $\boldsymbol{x}_i^*$ is exponentially small in the separation $\Delta_i$. The precise bounds using the Jacobian $\mathrm{J} = \frac{\partial f(\boldsymbol{\xi})}{\partial \boldsymbol{\xi}}$ and its value $\mathrm{J}^m$ in the mean value theorem are:*

$$\|f(\boldsymbol{\xi}) - \boldsymbol{x}_i^*\| \;\leqslant\; \|\mathrm{J}^m\|_2 \, \|\boldsymbol{\xi} - \boldsymbol{x}_i^*\| \;, \tag{6}$$

$$\|\mathrm{J}^m\|_2 \;\leqslant\; 2\,\beta\,N\,M^2\,(N-1)\exp(-\,\beta\,(\Delta_i \,-\, 2\,\max\{\|\boldsymbol{\xi} - \boldsymbol{x}_i\|, \|\boldsymbol{x}_i^* - \boldsymbol{x}_i\|\}\,M)) \;. \tag{7}$$

*For given $\epsilon$ and sufficient large $\Delta_i$, we have $\|f(\boldsymbol{\xi}) - \boldsymbol{x}_i^*\| < \epsilon$, that is, retrieval with one update.*

See proof in appendix Theorem A8.

At the same time, the retrieval error decreases exponentially with the separation $\Delta_i$.

**Theorem 5** (Exponentially Small Retrieval Error). *The retrieval error $\|f(\boldsymbol{\xi}) - \boldsymbol{x}_i\|$ of pattern $\boldsymbol{x}_i$ is bounded by*

$$\|f(\boldsymbol{\xi}) - \boldsymbol{x}_i\| \;\leqslant\; 2\,(N-1)\,\exp(-\,\beta\,(\Delta_i \,-\, 2\,\max\{\|\boldsymbol{\xi} - \boldsymbol{x}_i\|, \|\boldsymbol{x}_i^* - \boldsymbol{x}_i\|\}\,M))\,M \tag{8}$$

*and for $\|\boldsymbol{x}_i - \boldsymbol{x}_i^*\| \leqslant \frac{1}{2\,\beta\,M}$ together with $\|\boldsymbol{x}_i - \boldsymbol{\xi}\| \leqslant \frac{1}{2\,\beta\,M}$ by*

$$\|\boldsymbol{x}_i - \boldsymbol{x}_i^*\| \;\leqslant\; 2\,e\,(N-1)\,M\,\exp(-\,\beta\,\Delta_i) \;. \tag{9}$$

See proof in appendix Theorem A9.

**Metastable states and one global fixed point.** So far, we considered patterns $\boldsymbol{x}_i$ that are well separated and the iteration converges to a fixed point which is near a pattern $\boldsymbol{x}_i$. If no pattern $\boldsymbol{x}_i$ is well separated from the others, then the iteration converges to a global fixed point close to the arithmetic mean of the vectors. In this case the softmax vector $\boldsymbol{p}$ is close to uniform, that is, $p_i = 1/N$. If some vectors are similar to each other and well separated from all other vectors, then a metastable state near the similar vectors exists. Iterations that start near the metastable state converge to this metastable state, also if initialized by one of the similar patterns. For convergence proofs to one global fixed point and to metastable states see appendix Lemma A7 and Lemma A12, respectively.

**Hopfield update rule is attention of the transformer.** The Hopfield network update rule is the attention mechanism used in transformer and BERT models (see Fig. 1). To see this, we assume $N$ stored (key) patterns $\boldsymbol{y}_i$ and $S$ state (query) patterns $\boldsymbol{r}_i$ that are mapped to the Hopfield space of dimension $d_k$. We set $\boldsymbol{x}_i = \boldsymbol{W}_K^T \boldsymbol{y}_i, \boldsymbol{\xi}_i = \boldsymbol{W}_Q^T \boldsymbol{r}_i$, and multiply the result of our update rule with $\boldsymbol{W}_V$. The matrices $\boldsymbol{Y} = (\boldsymbol{y}_1, \ldots, \boldsymbol{y}_N)^T$ and $\boldsymbol{R} = (\boldsymbol{r}_1, \ldots, \boldsymbol{r}_S)^T$ combine the $\boldsymbol{y}_i$ and $\boldsymbol{r}_i$ as row vectors. We define the matrices $\boldsymbol{X}^T = \boldsymbol{K} = \boldsymbol{Y} \boldsymbol{W}_K, \boldsymbol{\Xi}^T = \boldsymbol{Q} = \boldsymbol{R} \boldsymbol{W}_Q$, and $\boldsymbol{V} = \boldsymbol{Y} \boldsymbol{W}_K \boldsymbol{W}_V = \boldsymbol{X}^T \boldsymbol{W}_V$, where $\boldsymbol{W}_K \in \mathbb{R}^{d_y \times d_k}, \boldsymbol{W}_Q \in \mathbb{R}^{d_r \times d_k}, \boldsymbol{W}_V \in \mathbb{R}^{d_k \times d_v}$. If $\beta = 1/\sqrt{d_k}$ and softmax $\in \mathbb{R}^N$ is changed to a row vector, we obtain for the update rule Eq. (3) multiplied by $\boldsymbol{W}_V$:

$$\boldsymbol{Z} = \text{softmax}\left(1/\sqrt{d_k}\, \boldsymbol{Q}\, \boldsymbol{K}^T\right)\, \boldsymbol{V} = \text{softmax}\left(\beta\, \boldsymbol{R}\, \boldsymbol{W}_Q\, \boldsymbol{W}_K^T \boldsymbol{Y}^T\right)\, \boldsymbol{Y}\, \boldsymbol{W}_K \boldsymbol{W}_V . \tag{10}$$

The left part of Eq. (10) is the transformer attention. In the transformer self-attention $\boldsymbol{R} = \boldsymbol{Y}$, and $\boldsymbol{W}_K \boldsymbol{W}_V$ replaced by just $\boldsymbol{W}_V$. Besides the attention mechanism, Hopfield networks allow for other functionalities in deep network architectures, which we introduce via specific layers in the next section. The right part of Eq. (10) serves to explain these specific layers.

## 3   NEW HOPFIELD LAYERS FOR DEEP LEARNING

Modern Hopfield networks with continuous states can be integrated into deep learning architectures, because they are continuous and differentiable with respect to their parameters. Furthermore, they typically retrieve patterns with one update, which is conform to deep learning layers that are activated only once. For these two reasons, modern Hopfield networks can serve as specialized layers in deep networks to equip them with memories. Below, we introduce three types of Hopfield layers: `Hopfield`, `HopfieldPooling`, and `HopfieldLayer`. Possible applications of Hopfield layers in deep network architectures comprise:

- multiple instance learning (MIL) (Dietterich et al., 1997),

- processing of and learning with point sets (Qi et al., 2017a;b; Xu et al., 2018),

- set-based and permutation invariant learning (Guttenberg et al., 2016; Ravanbakhsh et al., 2016; Zaheer et al., 2017; Korshunova et al., 2018; Ilse et al., 2018; Zhai et al., 2020),

- attention-based learning (Vaswani et al., 2017a),

- deep learning with associative memories (Graves et al., 2014; Weston et al., 2014; Ba et al., 2016a;b; Schlag & Schmidhuber, 2018; Schlag et al., 2019),

- natural language processing (Devlin et al., 2018; 2019),

- sequence analysis and time series prediction (Hochreiter, 1991; Hochreiter & Schmidhuber, 1997; Cho et al., 2014), and

- storing and retrieving reference data, e.g. the training data, outliers, high error data points, prototypes or cluster centers, support vectors & border cases.

Hopfield network layers can substitute existing layers like pooling layers, permutation equivariant layers (Guttenberg et al., 2016; Ravanbakhsh et al., 2016), GRU (Cho et al., 2014) & LSTM (Hochreiter, 1991; Hochreiter & Schmidhuber, 1997) layers, and attention layers (Vaswani et al., 2017a;b; Bahdanau et al., 2014).

**Types of neural networks.** We consider two types of feed-forward neural networks: (I) Neural networks that propagate an activation vector from the input layer to the output layer. Examples are fully-connected or convolutional neural networks. (II) Neural networks that propagate a set of vectors from the input layer to the output layer, where each layer applies the same operation to each element of the set and the output layer may summarize the set via a vector. An example is the transformer. Recurrent neural networks are networks of type (I), which are iteratively applied to a set or a sequence, where intermediate results are stored in a memory and can be reused. Modern Hopfield networks can be integrated into both types of neural network architectures and enable to equip each of their layers with associative memories. See Fig. 2.



Figure 2: Left: A standard deep network with layers (■) propagates either a vector or a set of vectors from the input to the output. Right: A deep network, where layers (■) are equipped with associative memories via Hopfield layers (■).

**Types of new Hopfield layers.** We introduce three types of Hopfield layers: `Hopfield`, `HopfieldPooling`, and `HopfieldLayer`. The continuous modern Hopfield network results in a plethora of new deep learning architectures, since we can (a) propagate sets or single vectors, (b) propagate queries, stored patterns, or both, (c) learn static queries or stored patterns, (d) fill the memory by training sets, prototypes, or external data. Next, we provide three useful types of Hopfield layers. The implementation is available at: https://github.com/ml-jku/hopfield-layers

**(1)** Layer `Hopfield` for networks that **propagate sets of vectors via state (query) patterns $R$ and stored (key) patterns $Y$**. The layer `Hopfield` is the realization of formula (10). The memory of the `Hopfield` layer can be *filled with sets from the input or previous layers*, see Fig. 3. The memory may be filled with a reference set, which is covered by providing the reference set as additional input. Thus, the layer `Hopfield` allows the association of two sets. A prominent example of a layer that performs such association is the transformer attention mechanism, which associates keys and queries, e.g. two point sets that have to be compared. This layer allows for different kinds of sequence-to-sequence learning, point set operations, and retrieval-based methods. The layer `Hopfield` with skip connections in a ResNet architecture is identical to the popular transformer and BERT models. In the experiments, we analyzed these Hopfield layers in transformer architectures. In our experiments in which we compare machine learning methods on small datasets of the UCI benchmark collection the layer `Hopfield` is also used.

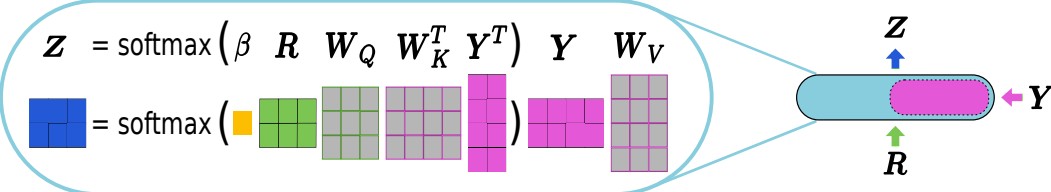

Figure 3: The layer `Hopfield` allows the association of two sets $R$ (■) and $Y$ (■). It can be integrated into deep networks that propagate sets of vectors. The Hopfield memory is filled with a set from either the input or previous layers. The output is a set of vectors $Z$ (■).

**(2)** Layer `HopfieldPooling` for networks that **propagate patterns via the stored (key) patterns $Y$**. This layer performs a pooling or summarization of sets $Y$ obtained from queries in previous layers or the input. The memory of the `HopfieldPooling` layer is *filled with sets from the input or previous layers*. The `HopfieldPooling` layer uses the queries to search for patterns in the memory, the stored set. If more patterns are similar to a particular search pattern (query), then the result is an average over these patterns. The state (query) patterns of each layer are static and can be learned. Multiple queries supply a set to the next layer, where each query corresponds to one element of the set. Thus, the layer `HopfieldPooling` enables fixed pattern search, pooling operations, and memories like LSTMs or GRUs. The static pattern functionality is typically needed if particular patterns must be identified in the data.

A single `HopfieldPooling` layer allows for multiple instance learning. Static state (query)

patterns together with position encoding in the keys allows for performing pooling operations. The position encoding can be two-dimensional, where standard convolutional filters can be constructed as in convolutional neural networks (CNNs). The `HopfieldPooling` layer can substitute pooling, averaging, LSTM, and permutation equivariant layers. See Fig. 4. The layer `HopfieldPooling` is used for experiments with multiple instance learning tasks, e.g. for immune repertoire classification in the experiments.

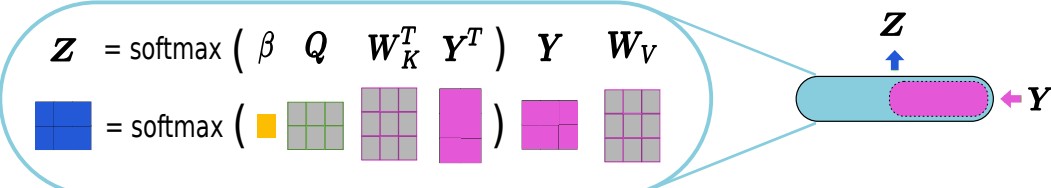

Figure 4: The layer `HopfieldPooling` enables pooling or summarization of sets, which are obtained from the input or from previous layers. The input $\boldsymbol{Y}$ (■) can be either a set or a sequence. The query patterns of each layer are static and can be learned. The output is a set of vectors $\boldsymbol{Z}$ (■), where the number of vectors equals the number of query patterns. The layer `HopfieldPooling` can realize multiple instance learning.

**(3)** Layer `HopfieldLayer` for networks that **propagate a vector or a set of vectors via state (query) patterns** $\boldsymbol{R}$. The queries $\boldsymbol{R}$ can be input vectors or queries that are computed from the output of previous layers. The memory of the `HopfieldLayer` layer is *filled with a fixed set*, which can be the training set, a reference set, prototype set, or a learned set (a learned matrix). The stored (key) patterns are static and can be learned. If the training set is stored in the memory, then each layer constructs a new set of queries based on the query results of previous layers. The stored patterns can be initialized by the training set or a reference set and then learned, in which case they deviate from the training set. The stored patterns can be interpreted as weights from the state (query) to hidden neurons that have a softmax activation function (Krotov & Hopfield, 2020). The layer `HopfieldLayer` can substitute a fully connected layer, see Fig. 5. A single `HopfieldLayer` layer also allows for approaches similar to support vector machines (SVMs), approaches similar to $k$-nearest neighbor, approaches similar to learning vector quantization, and pattern search. For classification, the raw data $\boldsymbol{y}_i = (\boldsymbol{z}_i, \boldsymbol{t}_i)$ can be the concatenation of input $\boldsymbol{z}_i$ and target $\boldsymbol{t}_i$. In this case, the matrices $\boldsymbol{W}_K$ and $\boldsymbol{W}_V$ can be designed such that inside the softmax the input $\boldsymbol{z}_i$ is used and outside the softmax the target $\boldsymbol{t}_i$. Thus, the softmax provides a weighted average of the target vectors based on the similarity between the query and the inputs. Also SVM models, $k$-nearest neighbor, and learning vector quantization can be considered as weighted averages of the targets. The encoder-decoder attention layer of the transformers are a `HopfieldLayer` layer, where the memory is filled with the encoder output set. In our experiments with the drug design benchmark datasets, the layer `HopfieldLayer` has been applied and compared to other machine learning methods.

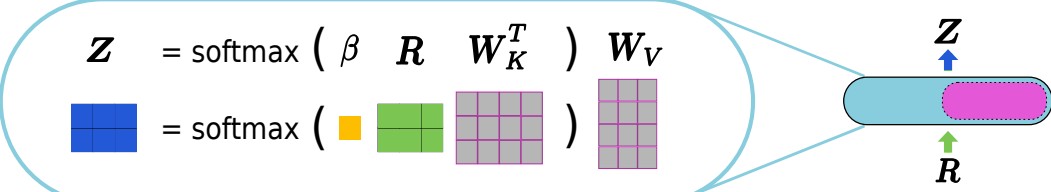

Figure 5: The layer `HopfieldLayer` enables multiple queries of the training set, a reference set, prototype set, or a learned set (a learned matrix). The queries for each layer are computed from the results of previous layers. The input is a set of vectors $\boldsymbol{R}$ (■). The output is also a set of vectors $\boldsymbol{Z}$ (■), where the number of output vectors equals the number of input vectors. The layer `HopfieldLayer` can realize SVM models, $k$-nearest neighbor, and LVQ.

**Additional functionality of new Hopfield layers.** The insights about energy, convergence, and storage properties provide all new Hopfield layers with additional functionalities: i) *multiple updates*

to control how precise fixed points are found without additional parameters needed. ii) *variable $\beta$* to determine the kind of fixed points such as the size of metastable states. The variable $\beta$ controls over how many patterns is averaged. As observed in the experiments, the variable is relevant in combination with the learning rate to steer the learning dynamics. The parameter $\beta$ governs the fixed point dynamics and can be learned, too. iii) *controlling the storage capacity* via the dimension of the associative space. The storage capacity can be relevant for tasks with a huge number of instances as in the immune repertoire classification experiment. iv) *pattern normalization* controls, like the layernorm, the fixed point dynamics by the norm and shift of the patterns. For more details see appendix, Section A.6.

## 4 EXPERIMENTS

We show that our proposed Hopfield layers can be applied successfully to a wide range of tasks. The tasks are from natural language processing, contain multiple instance learning problems, a collection of small classification tasks, and drug design problems.

**Analysis of transformer and BERT models.** Transformer and BERT models can be implemented by the layer `Hopfield`. The kind of fixed point of the Hopfield net is determined by how the pattern $\boldsymbol{x}_i$ is separated from others patterns. (a) *a global fixed point*: no separation of a pattern from the others, (b) *a fixed point close to a single pattern*: pattern is separated from other patterns, (c) *metastable state*: some patterns are similar to each other and well separated from all other vectors. We observed that the attention heads of transformer and BERT models are predominantly in metastable states, which are categorized into four classes: (I) averaging over a very large number of patterns (very large metastable state or fixed point (a)), (II) averaging over a large number of patterns (large metastable state), (III) averaging over a medium number of patterns (medium metastable state), (IV) averaging over a small number of patterns (small metastable state or fixed point (c)). For analyzing the metastable states, we calculated the minimal number $k$ of $\mathrm{softmax}$ values required to sum up to $0.90$. Hence, $k$ indicates the size of a metastable state. To determine in which of the four classes a head is mainly operating, we computed the distribution of $k$ across sequences. Concretely, for $N$ tokens and for $\bar{k}$ as the median of the distribution, a head is classified as operating in class (I) if $1/2N < \bar{k}$, as operating in class (II) if $1/8N < \bar{k} \leqslant 1/2N$, as operating in class (III) if $1/32N < \bar{k} \leqslant 1/8N$, and as operating in class (IV) if $\bar{k} \leqslant 1/32N$. We analyzed pre-trained BERT models from Hugging Face Inc. (Wolf et al., 2019) according to these operating classes. In Fig. A.3 in the appendix the distribution of the pre-trained bert-base-cased model is depicted (for other models see appendix Section A.5.1.4). Operating classes (II) (large metastable states) and (IV) (small metastable states) are often observed in the middle layers. Operating class (I) (averaging over a very large number of patterns) is abundant in lower layers. Similar observations have been reported in other studies (Toneva & Wehbe, 2019a;b; Tay et al., 2020). Operating class (III) (medium metastable states) is predominant in the last layers.

**Multiple Instance Learning Datasets.** For multiple instance learning (MIL) (Dietterich et al., 1997), we integrate our new Hopfield network via the layer `HopfieldPooling` into deep learning architectures. Recently, deep learning methods have been applied to MIL problems (Ilse et al., 2018), but still the performance on many datasets lacks improvement. Thus, MIL datasets still pose an interesting challenge, in which Hopfield layers equipped with memory are a promising approach.

•*Immune Repertoire Classification.* The first MIL task is immune repertoire classification, where a deep learning architecture with `HopfieldPooling` (DeepRC) was used (Widrich et al., 2020a;b). Immune repertoire classification (Emerson et al., 2017) typically requires to extract few patterns from a large set of sequences, the repertoire, that are indicative for the respective immune status. The datasets contain $\approx 300{,}000$ instances per immune repertoire, which represents one of the largest multiple instance learning experiments ever conducted (Carbonneau et al., 2018). Most MIL methods fail due the large number of instances. This experiment comprises real-world and simulated datasets. Simulated datasets are generated by implanting sequence motifs (Akbar et al., 2019; Weber et al., 2020) with low frequency into simulated or experimentally-observed immune receptor sequences. The performance of DeepRC was compared with other machine learning methods: (i) known motif, (ii) SVM using $k$-mers and MinMax or Jaccard kernel, (iii) $K$-Nearest Neighbor (KNN) with $k$-mers, (iv) logistic regression with $k$-mers, (v) burden test with $k$-mers, and (vi) logistic multiple

| Method | tiger | fox | elephant | UCSB |
|---|---|---|---|---|
| Hopfield (ours) | $\mathbf{91.3 \pm 0.5}$ | $64.05 \pm 0.4$ | $\mathbf{94.9 \pm 0.3}$ | $\mathbf{89.5 \pm 0.8}$ |
| Path encoding (Küçükaşcı & Baydoğan, 2018) | $91.0 \pm 1.0^a$ | $71.2 \pm 1.4^a$ | $94.4 \pm 0.7^a$ | $88.0 \pm 2.2^a$ |
| MInD (Cheplygina et al., 2016) | $85.3 \pm 1.1^a$ | $70.4 \pm 1.6^a$ | $93.6 \pm 0.9^a$ | $83.1 \pm 2.7^a$ |
| MILES (Chen et al., 2006) | $87.2 \pm 1.7^b$ | $\mathbf{73.8 \pm 1.6}^a$ | $92.7 \pm 0.7^a$ | $83.3 \pm 2.6^a$ |
| APR (Dietterich et al., 1997) | $77.8 \pm 0.7^b$ | $54.1 \pm 0.9^b$ | $55.0 \pm 1.0^b$ | — |
| Citation-kNN (Wang, 2000) | $85.5 \pm 0.9^b$ | $63.5 \pm 1.5^b$ | $89.6 \pm 0.9^b$ | $70.6 \pm 3.2^a$ |
| DD (Maron & Lozano-Pérez, 1998) | $84.1^b$ | $63.1^b$ | $90.7^b$ | — |

Table 1: Results for MIL datasets Tiger, Fox, Elephant, and UCSB Breast Cancer in terms of AUC. Results for all methods except the first are taken from either [a](Küçükaşcı & Baydoğan, 2018) or [b](Carbonneau et al., 2016), depending on which reports the higher AUC.

instance learning (lMIL). On the real-world dataset DeepRC achieved an AUC of $0.832 \pm 0.022$, followed by the SVM with MinMax kernel (AUC $0.825 \pm 0.022$) and the burden test with an AUC of $0.699 \pm 0.041$. Across datasets, DeepRC outperformed all competing methods with respect to average AUC (Widrich et al., 2020a;b).

•*MIL benchmark datasets.* We apply Hopfield layers to further MIL datasets (Ilse et al., 2018; Küçükaşcı & Baydoğan, 2018; Cheplygina et al., 2016): Elephant, Fox and Tiger for image annotation (Andrews et al., 2003). These datasets consist of color images from the Corel dataset that have been preprocessed and segmented. An image consists of a set of segments (or blobs), each characterized by color, texture and shape descriptors. The datasets have 100 positive and 100 negative example images. The latter have been randomly drawn from a pool of photos of other animals. Elephant comprises 1,391 instances and 230 features, Fox 1,320 instances and 230 features, and Tiger has 1,220 instances and 230 features. Furthermore, we use the UCSB breast cancer classification (Kandemir et al., 2014) dataset, which consists of 2,002 instances across 58 input objects. An instance represents a patch of a histopathological image of cancerous or normal tissue. The layer `HopfieldPooling` is used, which allows for computing a per-input-object representation by extracting an average of instances that are indicative for one of the two classes. The input to the layer `HopfieldPooling` is a set of embedded instances $\boldsymbol{Y}$. A trainable but fixed state (query) pattern $\boldsymbol{Q}$ is used for averaging over class-indicative instances. This averaging enables a compression of variable-sized bags to a fixed-sized representation to discriminate the bags. More details in appendix Sec. A.5.2. Our approach has set a new state-of-the-art and has outperformed other methods (Küçükaşcı & Baydoğan, 2018; Carbonneau et al., 2016) on the datasets Tiger, Elephant and UCSB Breast Cancer (see Table 1).

**UCI Benchmark Collection.** So far deep learning struggled with small datasets. However, Hopfield networks are promising for handling small datasets, since they can store the training data points or their representations to perform similarity-based, nearest neighbor, or learning vector quantization methods. Therefore, we test the Hopfield layer `Hopfield` on the small datasets of the UC Irvine (UCI) Machine Learning Repository that have been used to benchmark supervised learning methods (Fernández-Delgado et al., 2014; Wainberg et al., 2016; Khan et al., 2018) and also feed-forward neural networks (Klambauer et al., 2017a; Wu et al., 2018), where our Hopfield networks could exploit their memory. The whole 121 datasets in the collection vary strongly with respect to their size, number of features, and difficulties (Fernández-Delgado et al., 2014), such that they have been divided into 75 "small datasets" with less than 1,000 samples and 45 "large datasets" with more than or equal to 1,000 samples in Klambauer et al. (2017a).

On the 75 small datasets, Random Forests (RFs) and Support Vector Machines (SVM) are highly accurate, whereas on the large datasets, deep learning methods and neural networks are in the lead (Klambauer et al., 2017a;b; Wu et al., 2018). We applied a modern Hopfield network via the layer `HopfieldLayer`, where a self-normalizing net (SNN) maps the input vector to $\boldsymbol{Y}$ and $\boldsymbol{R}$. The output $\boldsymbol{Z}$ of `HopfieldLayer` enters a softmax output. We compared our modern Hopfield networks against deep learning

| Method | avg. rank diff. | $p$-value |
|---|---|---|
| Hopfield (ours) | $\mathbf{-3.92}$ | — |
| SVM | $-3.23$ | $0.15$ |
| SNN | $-2.85$ | $0.10$ |
| RandomForest | $-2.79$ | $0.05$ |
| . . . | . . . | . . . |
| Stacking | $8.73$ | $1.2e{-}11$ |

Table 2: Results on 75 small datasets of the UCI benchmarks given as difference to average rank.

methods (e.g. SNNs, resnet), RFs, SVMs, boosting, bagging, and many other machine learning methods of Fernández-Delgado et al. (2014). Since for each method, multiple variants and implementations had been included, we used method groups and representatives as defined by Klambauer et al. (2017a). For each dataset, a ranking of the methods was calculated which is presented in Table 2. We found that Hopfield networks outperform all other methods on the small datasets, setting a new state-of-the-art for 10 datasets. The difference is significant except for the first three runner-up methods (Wilcoxon signed rank test). See appendix Section A.5.3 for details.

**Drug Design Benchmark Datasets.** We test the Hopfield layer `HopfieldLayer`, on four drug design datasets. These datasets represent four main areas of modeling tasks in drug design, concretely to develop accurate models for predicting a) new anti-virals (HIV) by the Drug Therapeutics Program (DTP) AIDS Antiviral Screen, b) new protein inhibitors, concretely human $\beta$-secretase (BACE) inhibitors by Subramanian et al. (2016), c) metabolic effects as blood-brain barrier permeability (BBBP) (Martins et al., 2012) and d) side effects of a chemical compound from the Side Effect Resource (SIDER) Kuhn et al. (2016). We applied the Hopfield layer `HopfieldLayer`, where the training data is used as stored patterns $Y$, the input vector as state pattern $R$, and the corresponding training label to project the output of the Hopfield layer $YW_V$. Our architecture with `HopfieldLayer` has reached state-of-the-art for predicting side effects on SIDER $0.672 \pm 0.019$ as well as for predicting $\beta$-secretase BACE $0.902 \pm 0.023$. For details, see Table A.5 in the appendix.

**Conclusion.** We have introduced a modern Hopfield network with continuous states and the corresponding new update rule. This network can store exponentially many patterns, retrieves patterns with one update, and has exponentially small retrieval errors. We analyzed the attention heads of BERT models. The new modern Hopfield networks have been integrated into deep learning architectures as layers to allow the storage of and access to raw input data, intermediate results, or learned prototypes. These Hopfield layers enable new ways of deep learning, beyond fully-connected, convolutional, or recurrent networks, and provide pooling, memory, association, and attention mechanisms. Hopfield layers that equip neural network layers with memories improved state-of-the-art in three out of four considered multiple instance learning problems and on immune repertoire classification, and on two drug design dataset. They yielded the best results among different machine learning methods on the UCI benchmark collections of small classification tasks.

## ACKNOWLEDGMENTS

The ELLIS Unit Linz, the LIT AI Lab and the Institute for Machine Learning are supported by the Land Oberösterreich, LIT grants DeepToxGen (LIT-2017-3-YOU-003), and AI-SNN (LIT-2018-6-YOU-214), the Medical Cognitive Computing Center (MC3), Janssen Pharmaceutica, UCB Biopharma, Merck Group, Audi.JKU Deep Learning Center, Audi Electronic Venture GmbH, TGW, Primal, S3AI (FFG-872172), Silicon Austria Labs (SAL), Anyline, FILL, EnliteAI, Google Brain, ZF Friedrichshafen AG, Robert Bosch GmbH, TÜV Austria, DCS, and the NVIDIA Corporation. IARAI is supported by Here Technologies.

## A  APPENDIX

This appendix consists of six sections (A.1–A.6). Section A.1 introduces the new modern Hopfield network with continuous states and its update rule. Furthermore, Section A.1 provides a thorough and profound theoretical analysis of this new Hopfield network. Section A.2 provides the mathematical background for Section A.1. Section A.3 reviews *binary* Modern Hopfield Networks of Krotov & Hopfield. Section A.4 shows that the Hopfield update rule is the attention mechanism of the transformer. Section A.5 gives details on the experiments. Section A.6 describes the PyTorch implementation of layers based on the new Hopfield networks and how to use them.

### CONTENTS OF THE APPENDIX

### LIST OF THEOREMS

## LIST OF DEFINITIONS

## LIST OF FIGURES

## LIST OF TABLES

### A.1   CONTINUOUS STATE MODERN HOPFIELD NETWORKS (A NEW CONCEPT)

#### A.1.1   INTRODUCTION

In Section A.1 our new modern Hopfield network is introduced. In Subsection A.1.2 we present the new energy function. Then in Subsection A.1.3, our new update rule is introduced. In Subsection A.1.4, we show that this update rule ensures global convergence. We show that all the limit points of any sequence generated by the update rule are the stationary points (local minima or saddle points) of the energy function. In Section A.1.5, we consider the local convergence of the update rule and see that patterns are retrieved with one update. In Subsection A.1.6, we consider the properties of the fixed points that are associated with the stored patterns. In Subsection A.1.6.1, we show that exponentially many patterns can be stored. The main result is given in Theorem A5: For random

patterns on a sphere we can store and retrieve exponentially (in the dimension of the Hopfield space) many patterns. Subsection A.1.6.2 reports that patterns are typically retrieved with one update step and that the retrieval error is exponentially small.

In Subsection A.1.7, we consider how associations for the new Hopfield networks can be learned. In Subsection A.1.7.2, we analyze if the association is learned directly by a bilinear form. In Subsection A.1.7.3, we analyze if stored patterns and query patterns are mapped to the space of the Hopfield network. Therefore, we treat the architecture of the transformer and BERT. In Subsection A.1.8, we introduce a temporal component into the new Hopfield network that leads to a forgetting behavior. The forgetting allows us to treat infinite memory capacity in Subsection A.1.8.1. In Subsection A.1.8.2, we consider the controlled forgetting behavior.

In Section A.2, we provide the mathematical background that is needed for our proofs. In particular we give lemmas on properties of the softmax, the log-sum-exponential, the Legendre transform, and the Lambert $W$ function.

In Section A.3, we review the new Hopfield network as introduced by Krotov and Hopfield in 2016. However in contrast to our new Hopfield network, the Hopfield network of Krotov and Hopfield is binary, that is, a network with binary states. In Subsection A.3.1, we give an introduction to neural networks equipped with associative memories and new Hopfield networks. In Subsection A.3.1.1, we discuss neural networks that are enhanced by an additional external memory and by attention mechanisms. In Subsection A.3.1.2, we give an overview over the modern Hopfield networks. Finally, in Subsection A.3.2, we present the energy function and the update rule for the modern, binary Hopfield networks.

### A.1.2 NEW ENERGY FUNCTION

We have patterns $\boldsymbol{x}_1, \ldots, \boldsymbol{x}_N$ that are represented by the matrix

$$\boldsymbol{X} = (\boldsymbol{x}_1, \ldots, \boldsymbol{x}_N) . \tag{11}$$

The largest norm of a pattern is

$$M = \max_i \|\boldsymbol{x}_i\| . \tag{12}$$

The query or state of the Hopfield network is $\boldsymbol{\xi}$.

The energy function E in the new type of Hopfield models of Krotov and Hopfield is $\mathrm{E} = -\sum_{i=1}^N F\left(\boldsymbol{\xi}^T \boldsymbol{x}_i\right)$ for binary patterns $\boldsymbol{x}_i$ and binary state $\boldsymbol{\xi}$ with interaction function $F(x) = x^n$, where $n = 2$ gives classical Hopfield model (Krotov & Hopfield, 2016). The storage capacity is proportional to $d^{n-1}$ (Krotov & Hopfield, 2016). This model was generalized by Demircigil et al. (Demircigil et al., 2017) to exponential interaction functions $F(x) = \exp(x)$, which gives the energy $\mathrm{E} = -\exp(\mathrm{lse}(1, \boldsymbol{X}^T \boldsymbol{\xi}))$. This energy leads to an exponential storage capacity of $N = 2^{d/2}$ for binary patterns. Furthermore, with a single update the fixed point is recovered with high probability. See more details in Section A.3.

In contrast to the these binary modern Hopfield networks, we focus on modern Hopfield networks with *continuous states* that can store *continuous patterns*. We generalize the energy of Demircigil et al. (Demircigil et al., 2017) to continuous states while keeping the lse properties which ensure high storage capacity and fast convergence. Our new energy E for a continuous query or state $\boldsymbol{\xi}$ is defined

as

$$\mathrm{E} \;=\; -\operatorname{lse}(\beta, \boldsymbol{X}^T \boldsymbol{\xi}) \;+\; \frac{1}{2}\boldsymbol{\xi}^T \boldsymbol{\xi} \;+\; \beta^{-1} \ln N \;+\; \frac{1}{2}M^2 \tag{13}$$

$$=\; -\beta^{-1} \ln \left( \sum_{i=1}^{N} \exp(\beta \boldsymbol{x}_i^T \boldsymbol{\xi}) \right) \;+\; \beta^{-1} \ln N \;+\; \frac{1}{2}\boldsymbol{\xi}^T \boldsymbol{\xi} \;+\; \frac{1}{2}M^2 \tag{14}$$

$$=\; -\beta^{-1} \ln \left( \frac{1}{N} \sum_{i=1}^{N} \exp\left( -\frac{1}{2}\,\beta\,\left( M^2 \;-\; \|\boldsymbol{x}_i\|^2 \right) \right) \exp\left( -\frac{1}{2}\,\beta\,\|\boldsymbol{x}_i \;-\; \boldsymbol{\xi}\|^2 \right) \right) . \tag{15}$$

First let us collect and prove some properties of E. The next lemma gives bounds on the energy E.

**Lemma A1.** *The energy* E *is larger than zero:*

$$0 \;\leqslant\; \mathrm{E}\,. \tag{16}$$

*For $\boldsymbol{\xi}$ in the simplex defined by the patterns, the energy* E *is upper bounded by:*

$$\mathrm{E} \;\leqslant\; \beta^{-1} \ln N \;+\; \frac{1}{2}\,M^2\,, \tag{17}$$

$$\mathrm{E} \;\leqslant\; 2\,M^2\,. \tag{18}$$

*Proof.* We start by deriving the lower bound of zero. The pattern most similar to query or state $\boldsymbol{\xi}$ is $\boldsymbol{x}_{\boldsymbol{\xi}}$:

$$\boldsymbol{x}_{\boldsymbol{\xi}} \;=\; \boldsymbol{x}_k\,, \quad k \;=\; \arg\max_i \boldsymbol{\xi}^T \boldsymbol{x}_i\,. \tag{19}$$

We obtain

$$\mathrm{E} \;=\; -\beta^{-1} \ln \left( \sum_{i=1}^{N} \exp(\beta \boldsymbol{x}_i^T \boldsymbol{\xi}) \right) \;+\; \beta^{-1} \ln N \;+\; \frac{1}{2}\boldsymbol{\xi}^T \boldsymbol{\xi} \;+\; \frac{1}{2}\,M^2 \tag{20}$$

$$=\; -\beta^{-1} \ln \left( \frac{1}{N} \sum_{i=1}^{N} \exp(\beta \boldsymbol{x}_i^T \boldsymbol{\xi}) \right) \;+\; \frac{1}{2}\boldsymbol{\xi}^T \boldsymbol{\xi} \;+\; \frac{1}{2}\,M^2$$

$$\geq\; -\beta^{-1} \ln \left( \frac{1}{N} \sum_{i=1}^{N} \exp(\beta \boldsymbol{x}_i^T \boldsymbol{\xi}) \right) \;+\; \frac{1}{2}\,\boldsymbol{\xi}^T \boldsymbol{\xi} \;+\; \frac{1}{2}\,\boldsymbol{x}_{\boldsymbol{\xi}}^T \boldsymbol{x}_{\boldsymbol{\xi}}$$

$$\geq\; -\beta^{-1} \ln \left( \exp(\beta \boldsymbol{x}_{\boldsymbol{\xi}}^T \boldsymbol{\xi}) \right) \;+\; \frac{1}{2}\boldsymbol{\xi}^T \boldsymbol{\xi} \;+\; \frac{1}{2}\,\boldsymbol{x}_{\boldsymbol{\xi}}^T \boldsymbol{x}_{\boldsymbol{\xi}}$$

$$=\; -\boldsymbol{x}_{\boldsymbol{\xi}}^T \boldsymbol{\xi} \;+\; \frac{1}{2}\,\boldsymbol{\xi}^T \boldsymbol{\xi} \;+\; \frac{1}{2}\,\boldsymbol{x}_{\boldsymbol{\xi}}^T \boldsymbol{x}_{\boldsymbol{\xi}}$$

$$=\; \frac{1}{2}\,(\boldsymbol{\xi} \;-\; \boldsymbol{x}_{\boldsymbol{\xi}})^T \,(\boldsymbol{\xi} \;-\; \boldsymbol{x}_{\boldsymbol{\xi}}) \;=\; \frac{1}{2}\,\|\boldsymbol{\xi} \;-\; \boldsymbol{x}_{\boldsymbol{\xi}}\|^2 \;\geq\; 0\,.$$

The energy is zero and, therefore, the bound attained, if all $\boldsymbol{x}_i$ are equal, that is, $\boldsymbol{x}_i = \boldsymbol{x}$ for all $i$ and $\boldsymbol{\xi} = \boldsymbol{x}$.

For deriving upper bounds on the energy E, we require the the query $\boldsymbol{\xi}$ to be in the simplex defined by the patterns, that is,

$$\boldsymbol{\xi} \;=\; \sum_{i=1}^{N} p_i\,\boldsymbol{x}_i\,, \quad \sum_{i=1}^{N} p_i \;=\; 1\,, \quad \forall_i : 0 \;\leqslant\; p_i\,. \tag{21}$$

The first upper bound is.

$$\mathrm{E} \;=\; -\beta^{-1} \ln \left( \sum_{i=1}^{N} \exp(\beta \boldsymbol{x}_i^T \boldsymbol{\xi}) \right) \;+\; \frac{1}{2}\,\boldsymbol{\xi}^T \boldsymbol{\xi} \;+\; \beta^{-1} \ln N \;+\; \frac{1}{2}\,M^2 \tag{22}$$

$$\leqslant\; -\sum_{i=1}^{N} p_i\,(\boldsymbol{x}_i^T \boldsymbol{\xi}) \;+\; \frac{1}{2}\,\boldsymbol{\xi}^T \boldsymbol{\xi} \;+\; \beta^{-1} \ln N \;+\; \frac{1}{2}\,M^2$$

$$=\; -\frac{1}{2}\,\boldsymbol{\xi}^T \boldsymbol{\xi} \;+\; \beta^{-1} \ln N \;+\; \frac{1}{2}\,M^2 \;\leqslant\; \beta^{-1} \ln N \;+\; \frac{1}{2}\,M^2\,.$$

For the first inequality we applied Lemma A19 to $-\mathrm{lse}(\beta, \boldsymbol{X}^T\boldsymbol{\xi})$ with $\boldsymbol{z} = \boldsymbol{p}$ giving

$$- \mathrm{lse}(\beta, \boldsymbol{X}^T\boldsymbol{\xi}) \;\leqslant\; - \sum_{i=1}^{N} p_i \, (\boldsymbol{x}_i^T\boldsymbol{\xi}) \;+\; \beta^{-1} \sum_{i=1}^{N} p_i \ln p_i \;\leqslant\; - \sum_{i=1}^{N} p_i \, (\boldsymbol{x}_i^T\boldsymbol{\xi}) \,, \qquad (23)$$

as the term involving the logarithm is non-positive.

Next we derive the second upper bound, for which we need the mean $\boldsymbol{m_x}$ of the patterns

$$\boldsymbol{m_x} \;=\; \frac{1}{N} \sum_{i=1}^{N} \boldsymbol{x}_i \,. \qquad (24)$$

We obtain

$$\mathrm{E} \;=\; - \beta^{-1} \ln \left( \sum_{i=1}^{N} \exp(\beta \boldsymbol{x}_i^T\boldsymbol{\xi}) \right) \;+\; \frac{1}{2}\,\boldsymbol{\xi}^T\boldsymbol{\xi} \;+\; \beta^{-1} \ln N \;+\; \frac{1}{2}\,M^2 \qquad (25)$$

$$\leqslant\; - \sum_{i=1}^{N} \frac{1}{N}\,\boldsymbol{x}_i^T\boldsymbol{\xi} \;+\; \frac{1}{2}\,\boldsymbol{\xi}^T\boldsymbol{\xi} \;+\; \frac{1}{2}\,M^2$$

$$=\; - \boldsymbol{m_x}^T\boldsymbol{\xi} \;+\; \frac{1}{2}\,\boldsymbol{\xi}^T\boldsymbol{\xi} \;+\; \frac{1}{2}\,M^2$$

$$\leqslant\; \|\boldsymbol{m_x}\| \, \|\boldsymbol{\xi}\| \;+\; \frac{1}{2}\,\|\boldsymbol{\xi}\|^2 \;+\; \frac{1}{2}\,M^2$$

$$\leqslant\; 2\,M^2 \,,$$

where for the first inequality we again applied Lemma A19 with $\boldsymbol{z} = (1/N, \ldots, 1/N)$ and $\beta^{-1} \sum_i 1/N \ln(1/N) = -\beta^{-1} \ln(N)$. This inequality also follows from Jensen's inequality. The second inequality uses the Cauchy-Schwarz inequality. The last inequality uses

$$\|\boldsymbol{\xi}\| \;=\; \left\| \sum_i p_i\,\boldsymbol{x}_i \right\| \;\leqslant\; \sum_i p_i\,\|\boldsymbol{x}_i\| \;\leqslant\; \sum_i p_i M \;=\; M \qquad (26)$$

and

$$\|\boldsymbol{m_x}\| \;=\; \left\| \sum_i (1/N)\,\boldsymbol{x}_i \right\| \;\leqslant\; \sum_i (1/N)\,\|\boldsymbol{x}_i\| \;\leqslant\; \sum_i (1/N)\,M \;=\; M \,. \qquad (27)$$

$\square$

### A.1.3 NEW UPDATE RULE

We now introduce an update rule for minimizing the energy function E. The new update rule is

$$\boldsymbol{\xi}^{\mathrm{new}} \;=\; \boldsymbol{X}\boldsymbol{p} \;=\; \boldsymbol{X}\mathrm{softmax}(\beta\boldsymbol{X}^T\boldsymbol{\xi}) \,, \qquad (28)$$

where we used

$$\boldsymbol{p} \;=\; \mathrm{softmax}(\beta\boldsymbol{X}^T\boldsymbol{\xi}) \,. \qquad (29)$$

The new state $\boldsymbol{\xi}^{\mathrm{new}}$ is in the simplex defined by the patterns, no matter what the previous state $\boldsymbol{\xi}$ was. For comparison, the synchronous update rule for the classical Hopfield network with threshold zero is

$$\boldsymbol{\xi}^{\mathrm{new}} \;=\; \mathrm{sgn}\left(\boldsymbol{X}\boldsymbol{X}^T\boldsymbol{\xi}\right) \,. \qquad (30)$$

Therefore, instead of using the vector $\boldsymbol{X}^T\boldsymbol{\xi}$ as in the classical Hopfield network, its softmax version $\mathrm{softmax}(\beta\boldsymbol{X}^T\boldsymbol{\xi})$ is used.

In the next section (Section A.1.4) we show that the update rule Eq. (28) ensures global convergence. We show that all the limit points of any sequence generated by the update rule are the stationary points (local minima or saddle points) of the energy function E. In Section A.1.5 we consider the local convergence of the update rule Eq. (28) and see that patterns are retrieved with one update.

We are interested in the *global convergence*, that is, convergence from each initial point, of the iteration

$$\boldsymbol{\xi}^{\text{new}} = f(\boldsymbol{\xi}) = \boldsymbol{X}\boldsymbol{p} = \boldsymbol{X}\text{softmax}(\beta\boldsymbol{X}^T\boldsymbol{\xi}), \tag{31}$$

where we used

$$\boldsymbol{p} = \text{softmax}(\beta\boldsymbol{X}^T\boldsymbol{\xi}). \tag{32}$$

We defined the energy function

$$\text{E} = -\text{lse}(\beta, \boldsymbol{X}^T\boldsymbol{\xi}) + \frac{1}{2}\boldsymbol{\xi}^T\boldsymbol{\xi} + \beta^{-1}\ln N + \frac{1}{2}M^2 \tag{33}$$

$$= -\beta^{-1}\ln\left(\sum_{i=1}^{N}\exp(\beta\boldsymbol{x}_i^T\boldsymbol{\xi})\right) + \beta^{-1}\ln N + \frac{1}{2}\boldsymbol{\xi}^T\boldsymbol{\xi} + \frac{1}{2}M^2. \tag{34}$$

We will show that the update rule in Eq. (31) is the Concave-Convex Procedure (CCCP) for minimizing the energy E. The CCCP is proven to converge globally.

**Theorem A1** (Global Convergence (Zangwill): Energy). *The update rule Eq. (31) converges globally: For $\boldsymbol{\xi}^{t+1} = f(\boldsymbol{\xi}^t)$, the energy $\text{E}(\boldsymbol{\xi}^t) \to \text{E}(\boldsymbol{\xi}^*)$ for $t \to \infty$ and a fixed point $\boldsymbol{\xi}^*$.*

*Proof.* The Concave-Convex Procedure (CCCP) (Yuille & Rangarajan, 2002; 2003) minimizes a function that is the sum of a concave function and a convex function. CCCP is equivalent to Legendre minimization (Rangarajan et al., 1996; 1999) algorithms (Yuille & Rangarajan, 2003). The Jacobian of the softmax is positive semi-definite according to Lemma A22. The Jacobian of the softmax is the Hessian of the lse, therefore lse is a convex and $-\text{lse}$ a concave function. Therefore, the energy function $\text{E}(\boldsymbol{\xi})$ is the sum of the convex function $\text{E}_1(\boldsymbol{\xi}) = 1/2\boldsymbol{\xi}^T\boldsymbol{\xi} + C_1$ and the concave function $\text{E}_2(\boldsymbol{\xi}) = -\text{lse}$:

$$\text{E}(\boldsymbol{\xi}) = \text{E}_1(\boldsymbol{\xi}) + \text{E}_2(\boldsymbol{\xi}), \tag{35}$$

$$\text{E}_1(\boldsymbol{\xi}) = \frac{1}{2}\boldsymbol{\xi}^T\boldsymbol{\xi} + \beta^{-1}\ln N + \frac{1}{2}M^2 = \frac{1}{2}\boldsymbol{\xi}^T\boldsymbol{\xi} + C_1, \tag{36}$$

$$\text{E}_2(\boldsymbol{\xi}) = -\text{lse}(\beta, \boldsymbol{X}^T\boldsymbol{\xi}), \tag{37}$$

where $C_1$ does not depend on $\boldsymbol{\xi}$.

The Concave-Convex Procedure (CCCP) (Yuille & Rangarajan, 2002; 2003) applied to E is

$$\nabla_\xi\text{E}_1\left(\boldsymbol{\xi}^{t+1}\right) = -\nabla_\xi\text{E}_2\left(\boldsymbol{\xi}^t\right), \tag{38}$$

which is

$$\nabla_\xi\left(\frac{1}{2}\boldsymbol{\xi}^T\boldsymbol{\xi} + C_1\right)\left(\boldsymbol{\xi}^{t+1}\right) = \nabla_\xi\text{lse}(\beta, \boldsymbol{X}^T\boldsymbol{\xi}^t). \tag{39}$$

The resulting update rule is

$$\boldsymbol{\xi}^{t+1} = \boldsymbol{X}\boldsymbol{p}^t = \boldsymbol{X}\text{softmax}(\beta\boldsymbol{X}^T\boldsymbol{\xi}^t) \tag{40}$$

using

$$\boldsymbol{p}^t = \text{softmax}(\beta\boldsymbol{X}^T\boldsymbol{\xi}^t). \tag{41}$$

This is the update rule in Eq. (31).

Theorem 2 in Yuille & Rangarajan (2002) and Theorem 2 in Yuille & Rangarajan (2003) state that the update rule Eq. (31) is guaranteed to monotonically decrease the energy E as a function of time. See also Theorem 2 in Sriperumbudur & Lanckriet (2009). □

Although the objective converges in all cases, it does not necessarily converge to a local minimum (Lipp & Boyd, 2016).

However the convergence proof of CCCP in Yuille & Rangarajan (2002; 2003) was not as rigorous as required. In Sriperumbudur & Lanckriet (2009) a rigorous analysis of the convergence of CCCP is performed using Zangwill's global convergence theory of iterative algorithms.

In Sriperumbudur & Lanckriet (2009) the minimization problem

$$\min_{\boldsymbol{\xi}} \ \mathrm{E}_1 \ + \ \mathrm{E}_2 \tag{42}$$

$$\text{s.t.} \ \ \boldsymbol{c}(\boldsymbol{\xi}) \leqslant \mathbf{0} \, , \quad \boldsymbol{d}(\boldsymbol{\xi}) \ = \ \mathbf{0}$$

is considered with $\mathrm{E}_1$ convex, $-\mathrm{E}_2$ convex, $\boldsymbol{c}$ component-wise convex function, and $\boldsymbol{d}$ an affine function. The CCCP algorithm solves this minimization problem by linearization of the concave part and is defined in Sriperumbudur & Lanckriet (2009) as

$$\boldsymbol{\xi}^{t+1} \ \in \ \arg\min_{\boldsymbol{\xi}} \ \mathrm{E}_1\left(\boldsymbol{\xi}\right) \ + \ \boldsymbol{\xi}^T \nabla_{\xi} \mathrm{E}_2\left(\boldsymbol{\xi}^t\right) \tag{43}$$

$$\text{s.t.} \ \ \boldsymbol{c}(\boldsymbol{\xi}) \leqslant \mathbf{0} \, , \quad \boldsymbol{d}(\boldsymbol{\xi}) \ = \ \mathbf{0} \, .$$

We define the upper bound $\mathrm{E}_{\mathrm{C}}$ on the energy:

$$\mathrm{E}_{\mathrm{C}}\left(\boldsymbol{\xi}, \boldsymbol{\xi}^t\right) \ := \ \mathrm{E}_1\left(\boldsymbol{\xi}\right) \ + \ \mathrm{E}_2\left(\boldsymbol{\xi}^t\right) \ + \ \left(\boldsymbol{\xi} - \boldsymbol{\xi}^t\right)^T \nabla_{\xi} \mathrm{E}_2\left(\boldsymbol{\xi}^t\right) \, . \tag{44}$$

$\mathrm{E}_{\mathrm{C}}$ is equal to the energy $\mathrm{E}\left(\boldsymbol{\xi}^t\right)$ for $\boldsymbol{\xi} = \boldsymbol{\xi}^t$:

$$\mathrm{E}_{\mathrm{C}}\left(\boldsymbol{\xi}^t, \boldsymbol{\xi}^t\right) \ = \ \mathrm{E}_1\left(\boldsymbol{\xi}^t\right) \ + \ \mathrm{E}_2\left(\boldsymbol{\xi}^t\right) \ = \ \mathrm{E}\left(\boldsymbol{\xi}^t\right) \, . \tag{45}$$

Since $-\mathrm{E}_2$ is convex, the first order characterization of convexity holds (Eq. 3.2 in Boyd & Vandenberghe (2009)):

$$- \ \mathrm{E}_2\left(\boldsymbol{\xi}\right) \ \geq \ - \ \mathrm{E}_2\left(\boldsymbol{\xi}^t\right) \ - \ \left(\boldsymbol{\xi} - \boldsymbol{\xi}^t\right)^T \nabla_{\xi} \mathrm{E}_2\left(\boldsymbol{\xi}^t\right) \, , \tag{46}$$

that is

$$\mathrm{E}_2\left(\boldsymbol{\xi}\right) \ \leqslant \ \mathrm{E}_2\left(\boldsymbol{\xi}^t\right) \ + \ \left(\boldsymbol{\xi} - \boldsymbol{\xi}^t\right)^T \nabla_{\xi} \mathrm{E}_2\left(\boldsymbol{\xi}^t\right) \, . \tag{47}$$

Therefore, for $\boldsymbol{\xi} \neq \boldsymbol{\xi}^t$ the function $\mathrm{E}_{\mathrm{C}}$ is an upper bound on the energy:

$$\mathrm{E}\left(\boldsymbol{\xi}\right) \ \leqslant \ \mathrm{E}_{\mathrm{C}}\left(\boldsymbol{\xi}, \boldsymbol{\xi}^t\right) \ = \ \mathrm{E}_1\left(\boldsymbol{\xi}\right) \ + \ \mathrm{E}_2\left(\boldsymbol{\xi}^t\right) \ + \ \left(\boldsymbol{\xi} - \boldsymbol{\xi}^t\right)^T \nabla_{\xi} \mathrm{E}_2\left(\boldsymbol{\xi}^t\right) \tag{48}$$

$$= \ \mathrm{E}_1\left(\boldsymbol{\xi}\right) \ + \ \boldsymbol{\xi}^T \nabla_{\xi} \mathrm{E}_2\left(\boldsymbol{\xi}^t\right) \ + \ C_2 \, ,$$

where $C_2$ does not depend on $\boldsymbol{\xi}$. Since we do not have constraints, $\boldsymbol{\xi}^{t+1}$ is defined as

$$\boldsymbol{\xi}^{t+1} \ \in \ \arg\min_{\boldsymbol{\xi}} \ \mathrm{E}_{\mathrm{C}}\left(\boldsymbol{\xi}, \boldsymbol{\xi}^t\right) \, , \tag{49}$$

hence $\mathrm{E}_{\mathrm{C}}\left(\boldsymbol{\xi}^{t+1}, \boldsymbol{\xi}^t\right) \leqslant \mathrm{E}_{\mathrm{C}}\left(\boldsymbol{\xi}^t, \boldsymbol{\xi}^t\right)$. Combining the inequalities gives:

$$\mathrm{E}\left(\boldsymbol{\xi}^{t+1}\right) \ \leqslant \ \mathrm{E}_{\mathrm{C}}\left(\boldsymbol{\xi}^{t+1}, \boldsymbol{\xi}^t\right) \ \leqslant \ \mathrm{E}_{\mathrm{C}}\left(\boldsymbol{\xi}^t, \boldsymbol{\xi}^t\right) \ = \ \mathrm{E}\left(\boldsymbol{\xi}^t\right) \, . \tag{50}$$

Since we do not have constraints, $\boldsymbol{\xi}^{t+1}$ is the minimum of

$$\mathrm{E}_{\mathrm{C}}\left(\boldsymbol{\xi}, \boldsymbol{\xi}^t\right) \ = \ \mathrm{E}_1\left(\boldsymbol{\xi}\right) \ + \ \boldsymbol{\xi}^T \nabla_{\xi} \mathrm{E}_2\left(\boldsymbol{\xi}^t\right) \ + \ C_2 \tag{51}$$

as a function of $\boldsymbol{\xi}$.

For a minimum not at the border, the derivative has to be the zero vector

$$\frac{\partial \mathrm{E}_{\mathrm{C}}\left(\boldsymbol{\xi}, \boldsymbol{\xi}^t\right)}{\partial \boldsymbol{\xi}} \ = \ \boldsymbol{\xi} \ + \ \nabla_{\xi} \mathrm{E}_2\left(\boldsymbol{\xi}^t\right) \ = \ \boldsymbol{\xi} \ - \ \boldsymbol{X} \mathrm{softmax}(\beta \boldsymbol{X}^T \boldsymbol{\xi}^t) \ = \ \mathbf{0} \tag{52}$$

and the Hessian must be positive semi-definite

$$\frac{\partial^2 \mathrm{E}_{\mathrm{C}}\left(\boldsymbol{\xi}, \boldsymbol{\xi}^t\right)}{\partial \boldsymbol{\xi}^2} \ = \ \boldsymbol{I} \, . \tag{53}$$

The Hessian is strict positive definite everywhere, therefore the optimization problem is strict convex (if the domain is convex) and there exist only one minimum, which is a global minimum. $E_C$ can even be written as a quadratic form:

$$\mathrm{E_C}\left(\boldsymbol{\xi},\boldsymbol{\xi}^t\right) \;=\; \frac{1}{2}\;\left(\boldsymbol{\xi}\;+\;\nabla_\xi \mathrm{E_2}\left(\boldsymbol{\xi}^t\right)\right)^T\left(\boldsymbol{\xi}\;+\;\nabla_\xi \mathrm{E_2}\left(\boldsymbol{\xi}^t\right)\right)\;+\;C_3\;, \tag{54}$$

where $C_3$ does not depend on $\boldsymbol{\xi}$.

Therefore, the minimum is

$$\boldsymbol{\xi}^{t+1} \;=\; -\,\nabla_\xi \mathrm{E_2}\left(\boldsymbol{\xi}^t\right) \;=\; \boldsymbol{X}\mathrm{softmax}(\beta\boldsymbol{X}^T\boldsymbol{\xi}^t) \tag{55}$$

if it is in the domain as we assume.

Using $M = \max_i \|\boldsymbol{x}_i\|$, $\boldsymbol{\xi}^{t+1}$ is in the sphere $\mathrm{S} = \{\boldsymbol{x} \mid \|\boldsymbol{x}\| \leqslant M\}$ which is a convex and compact set. Hence, if $\boldsymbol{\xi}^0 \in \mathrm{S}$, then the iteration is a mapping from S to S. Therefore, the point-set-map defined by the iteration Eq. (55) is uniformly compact on S according to Remark 7 in Sriperumbudur & Lanckriet (2009). Theorem 2 and Theorem 4 in (Sriperumbudur & Lanckriet, 2009) states that all the limit points of the iteration Eq. (55) are stationary points. These theorems follow from Zangwill's global convergence theorem: Convergence Theorem A, page 91 in Zangwill (1969) and page 3 in Wu (1983).

The global convergence theorem only assures that for the sequence $\boldsymbol{\xi}^{t+1} = f(\boldsymbol{\xi}^t)$ and a function $\Phi$ we have $\Phi(\boldsymbol{\xi}^t) \to \Phi(\boldsymbol{\xi}^*)$ for $t \to \infty$ but not $\boldsymbol{\xi}^t \to \boldsymbol{\xi}^*$. However, if $f$ is strictly monotone with respect to $\Phi$, then we can strengthen Zangwill's global convergence theorem (Meyer, 1976). We set $\Phi = \mathrm{E}$ and show $\mathrm{E}(\boldsymbol{\xi}^{t+1}) < \mathrm{E}(\boldsymbol{\xi}^t)$ if $\boldsymbol{\xi}^t$ is not a stationary point of E, that is, $f$ is strictly monotone with respect to E. The following theorem is similar to the convergence results for the expectation maximization (EM) algorithm in Wu (1983) which are given in theorems 1 to 6 in Wu (1983). The following theorem is also very similar to Theorem 8 in Sriperumbudur & Lanckriet (2009).

**Theorem A2** (Global Convergence: Stationary Points). *For the iteration Eq. (55) we have $\mathrm{E}\left(\boldsymbol{\xi}^t\right) \to \mathrm{E}\left(\boldsymbol{\xi}^*\right) = \mathrm{E}^*$ as $t \to \infty$, for some stationary point $\boldsymbol{\xi}^*$. Furthermore $\left\|\boldsymbol{\xi}^{t+1} - \boldsymbol{\xi}^t\right\| \to 0$ and either $\{\boldsymbol{\xi}^t\}_{t=0}^\infty$ converges or, in the other case, the set of limit points of $\{\boldsymbol{\xi}^t\}_{t=0}^\infty$ is a connected and compact subset of $\mathcal{L}(\mathrm{E}^*)$, where $\mathcal{L}(a) = \{\boldsymbol{\xi} \in \mathcal{L} \mid \mathrm{E}(\boldsymbol{\xi}) = a\}$ and $\mathcal{L}$ is the set of stationary points of the iteration Eq. (55). If $\mathcal{L}(\mathrm{E}^*)$ is finite, then any sequence $\{\boldsymbol{\xi}^t\}_{i=0}^\infty$ generated by the iteration Eq. (55) converges to some $\boldsymbol{\xi}^* \in \mathcal{L}(\mathrm{E}^*)$.*

*Proof.* We have $\mathrm{E}\left(\boldsymbol{\xi}^t\right) = \mathrm{E_1}\left(\boldsymbol{\xi}^t\right) + \mathrm{E_2}\left(\boldsymbol{\xi}^t\right)$. The gradient $\nabla_\xi \mathrm{E_2}\left(\boldsymbol{\xi}^t\right) = -\nabla_\xi \mathrm{lse}(\beta, \boldsymbol{X}^T\boldsymbol{\xi})$ is continuous. Therefore, Eq. (51) has minimum in the sphere S, which is a convex and compact set. If $\boldsymbol{\xi}^{t+1} \neq \boldsymbol{\xi}^t$, then $\boldsymbol{\xi}^t$ was not the minimum of Eq. (48) as the derivative at $\boldsymbol{\xi}^t$ is not equal to zero. Eq. (53) shows that the optimization problem Eq. (48) is strict convex, hence it has only one minimum, which is a global minimum. Eq. (54) shows that the optimization problem Eq. (48) is even a quadratic form. Therefore, we have

$$\mathrm{E}\left(\boldsymbol{\xi}^{t+1}\right) \;\leqslant\; \mathrm{E_C}\left(\boldsymbol{\xi}^{t+1}, \boldsymbol{\xi}^t\right) \;<\; \mathrm{E_C}\left(\boldsymbol{\xi}^t, \boldsymbol{\xi}^t\right) \;=\; \mathrm{E}\left(\boldsymbol{\xi}^t\right)\;. \tag{56}$$

Therefore, the point-set-map defined by the iteration Eq. (55) (for definitions see (Sriperumbudur & Lanckriet, 2009)) is strictly monotonic with respect to E. Therefore, we can apply Theorem 3 in Sriperumbudur & Lanckriet (2009) or Theorem 3.1 and Corollary 3.2 in Meyer (1976), which give the statements of the theorem.

$\square$

We showed global convergence of the iteration Eq. (31). We have shown that all the limit points of any sequence generated by the iteration Eq. (31) are the stationary points (critical points; local minima or saddle points) of the energy function E. Local maxima as stationary points are only possible if the iterations exactly hits a local maximum. However, convergence to a local maximum without being there is not possible because Eq. (56) ensures a strict decrease of the energy E. Therefore, almost sure local maxima are not obtained as stationary points. Either the iteration converges or, in the second case, the set of limit points is a connected and compact set. But what happens if $\boldsymbol{\xi}^0$ is in an $\epsilon$-neighborhood around a local minimum $\boldsymbol{\xi}^*$? Will the iteration Eq. (31) converge to $\boldsymbol{\xi}^*$? What is the rate of convergence? These questions are about *local convergence* which will be treated in detail in next section.

For the proof of local convergence to a fixed point we will apply Banach fixed point theorem. For the rate of convergence we will rely on properties of a contraction mapping.

**A.1.5.1   General Bound on the Jacobian of the Iteration.**   We consider the iteration

$$\boldsymbol{\xi}^{\text{new}} \ = \ f(\boldsymbol{\xi}) \ = \ \boldsymbol{X}\boldsymbol{p} \ = \ \boldsymbol{X}\text{softmax}(\beta\boldsymbol{X}^T\boldsymbol{\xi}) \tag{57}$$

using

$$\boldsymbol{p} \ = \ \text{softmax}(\beta\boldsymbol{X}^T\boldsymbol{\xi}) \,. \tag{58}$$

The Jacobian J is symmetric and has the following form:

$$\text{J} \ = \ \frac{\partial f(\boldsymbol{\xi})}{\partial\boldsymbol{\xi}} \ = \ \beta\,\boldsymbol{X}\left(\text{diag}(\boldsymbol{p}) - \boldsymbol{p}\boldsymbol{p}^T\right)\boldsymbol{X}^T \ = \ \boldsymbol{X}\text{J}_s\boldsymbol{X}^T \,, \tag{59}$$

where $\text{J}_s$ is Jacobian of the softmax.

To analyze the local convergence of the iteration, we distinguish between the following three cases (see also Fig. A.1). Here we only provide an informal discussion to give the reader some intuition. A rigorous formulation of the results can be found in the corresponding subsections.

  a) If the patterns $\boldsymbol{x}_i$ are not well separated, the iteration goes to a fixed point close to the arithmetic mean of the vectors. In this case $\boldsymbol{p}$ is close to $p_i = 1/N$.

  b) If the patterns $\boldsymbol{x}_i$ are well separated, then the iteration goes to the pattern to which the initial $\boldsymbol{\xi}$ is similar. If the initial $\boldsymbol{\xi}$ is similar to a vector $\boldsymbol{x}_i$ then it will converge to a vector close to $\boldsymbol{x}_i$ and $\boldsymbol{p}$ will converge to a vector close to $\boldsymbol{e}_i$.

  c) If some vectors are similar to each other but well separated from all other vectors, then a so called metastable state between the similar vectors exists. Iterations that start near the metastable state converge to this metastable state.

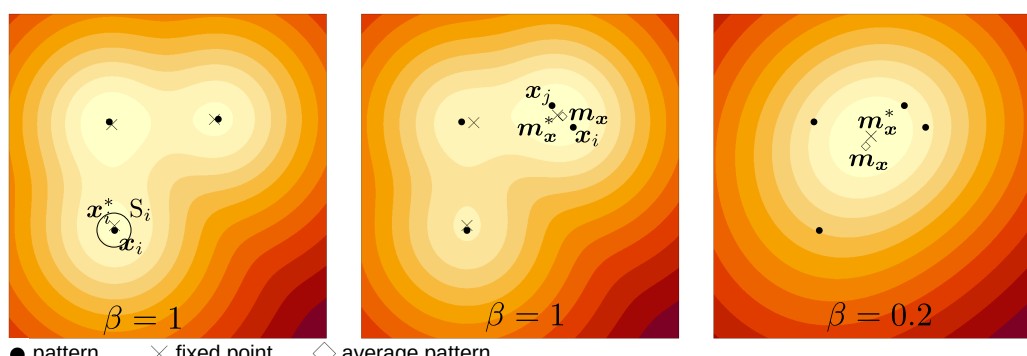

Figure A.1: The three cases of fixed points. **a) Stored patterns (fixed point is single pattern)**: patterns are stored if they are well separated. Each pattern $\boldsymbol{x}_i$ has a single fixed point $\boldsymbol{x}_i^*$ close to it. In the sphere $\text{S}_i$, pattern $\boldsymbol{x}_i$ is the only pattern and $\boldsymbol{x}_i^*$ the only fixed point. **b) Metastable state (fixed point is average of similar patterns)**: $\boldsymbol{x}_i$ and $\boldsymbol{x}_j$ are similar to each other and not well separated. The fixed point $\boldsymbol{m}_x^*$ is a metastable state that is close to the mean $\boldsymbol{m}_x$ of the similar patterns. **c) Global fixed point (fixed point is average of all patterns)**: no pattern is well separated from the others. A single global fixed point $\boldsymbol{m}_x^*$ exists that is close to the arithmetic mean $\boldsymbol{m}_x$ of all patterns. We begin with a bound on the Jacobian of the iteration, thereby heavily relying on the Jacobian of the softmax from Lemma A24.

**Lemma A2.** *For $N$ patterns $\boldsymbol{X} = (\boldsymbol{x}_1, \dots, \boldsymbol{x}_N)$, $\boldsymbol{p} = \text{softmax}(\beta\boldsymbol{X}^T\boldsymbol{\xi})$, $M = \max_i \|\boldsymbol{x}_i\|$, and $m = \max_i p_i(1-p_i)$, the spectral norm of the Jacobian J of the fixed point iteration is bounded:*

$$\|\text{J}\|_2 \ \leqslant \ 2\,\beta\,\|\boldsymbol{X}\|_2^2\,m \ \leqslant \ 2\,\beta\,N\,M^2\,m \,. \tag{60}$$

*If $p_{\max} = \max_i p_i \geq 1 - \epsilon$, then for the spectral norm of the Jacobian holds*

$$\|\text{J}\|_2 \ \leqslant \ 2\,\beta\,N\,M^2\,\epsilon \ - \ 2\,\epsilon^2\,\beta\,N\,M^2 \ < \ 2\,\beta\,N\,M^2\,\epsilon \,. \tag{61}$$

*Proof.* With

$$\boldsymbol{p} = \text{softmax}(\beta \boldsymbol{X}^T \boldsymbol{\xi}) , \tag{62}$$

the symmetric Jacobian J is

$$\text{J} = \frac{\partial f(\boldsymbol{\xi})}{\partial \boldsymbol{\xi}} = \beta \, \boldsymbol{X} \left( \text{diag}(\boldsymbol{p}) - \boldsymbol{p}\boldsymbol{p}^T \right) \boldsymbol{X}^T = \boldsymbol{X} \text{J}_s \boldsymbol{X}^T , \tag{63}$$

where $\text{J}_s$ is Jacobian of the softmax.

With $m = \max_i p_i (1 - p_i)$, Eq. (476) from Lemma A24 is

$$\|\text{J}_s\|_2 = \beta \left\| \text{diag}(\boldsymbol{p}) - \boldsymbol{p}\boldsymbol{p}^T \right\|_2 \leqslant 2 \, m \, \beta . \tag{64}$$

Using this bound on $\|\text{J}_s\|_2$, we obtain

$$\|\text{J}\|_2 \leqslant \beta \left\| \boldsymbol{X}^T \right\|_2 \|\text{J}_s\|_2 \|\boldsymbol{X}\|_2 \leqslant 2 \, m \, \beta \, \|\boldsymbol{X}\|_2^2 . \tag{65}$$

The spectral norm $\|.\|_2$ is bounded by the Frobenius norm $\|.\|_F$ which can be expressed by the norm squared of its column vectors:

$$\|\boldsymbol{X}\|_2 \leqslant \|\boldsymbol{X}\|_F = \sqrt{\sum_i \|\boldsymbol{x}_i\|^2} . \tag{66}$$

Therefore, we obtain the first statement of the lemma:

$$\|\text{J}\|_2 \leqslant 2 \, \beta \, \|\boldsymbol{X}\|_2^2 \, m \leqslant 2 \, \beta \, N \, M^2 \, m . \tag{67}$$

With $p_{\max} = \max_i p_i \geq 1 - \epsilon$ Eq. (480) in Lemma A24 is

$$\|\text{J}_s\|_2 \leqslant 2 \, \beta \, \epsilon - 2 \, \epsilon^2 \, \beta < 2 \, \beta \, \epsilon . \tag{68}$$

Using this inequality, we obtain the second statement of the lemma:

$$\|\text{J}\|_2 \leqslant 2 \, \beta \, N \, M^2 \, \epsilon - 2 \, \epsilon^2 \, \beta \, N \, M^2 < 2 \, \beta \, N \, M^2 \, \epsilon . \tag{69}$$

$\square$

We now define the "separation" $\Delta_i$ of a pattern $\boldsymbol{x}_i$ from data $\boldsymbol{X} = (\boldsymbol{x}_1, \dots, \boldsymbol{x}_N)$ here, since it has an important role for the convergence properties of the iteration.

**Definition 2** (Separation of Patterns). *We define $\Delta_i$, i.e. the separation of pattern $\boldsymbol{x}_i$ from data $\boldsymbol{X} = (\boldsymbol{x}_1, \dots, \boldsymbol{x}_N)$ as:*

$$\Delta_i = \min_{j, j \neq i} \left( \boldsymbol{x}_i^T \boldsymbol{x}_i - \boldsymbol{x}_i^T \boldsymbol{x}_j \right) = \boldsymbol{x}_i^T \boldsymbol{x}_i - \max_{j, j \neq i} \boldsymbol{x}_i^T \boldsymbol{x}_j . \tag{70}$$

*The pattern is separated from the other data if $0 < \Delta_i$. Using the parallelogram identity, $\Delta_i$ can also be expressed as*

$$\Delta_i = \min_{j, j \neq i} \frac{1}{2} \left( \|\boldsymbol{x}_i\|^2 - \|\boldsymbol{x}_j\|^2 + \|\boldsymbol{x}_i - \boldsymbol{x}_j\|^2 \right) \tag{71}$$

$$= \frac{1}{2} \|\boldsymbol{x}_i\|^2 - \frac{1}{2} \max_{j, j \neq i} \left( \|\boldsymbol{x}_j\|^2 - \|\boldsymbol{x}_i - \boldsymbol{x}_j\|^2 \right) .$$

*For $\|\boldsymbol{x}_i\| = \|\boldsymbol{x}_j\|$ we have $\Delta_i = 1/2 \min_{j, j \neq i} \|\boldsymbol{x}_i - \boldsymbol{x}_j\|^2$.*

*Analog we say for a query $\boldsymbol{\xi}$ and data $\boldsymbol{X} = (\boldsymbol{x}_1, \dots, \boldsymbol{x}_N)$, that $\boldsymbol{x}_i$ is least separated from $\boldsymbol{\xi}$ while being separated from other $\boldsymbol{x}_j$ with $j \neq i$ if*

$$i = \arg\max_k \min_{j, j \neq k} \left( \boldsymbol{\xi}^T \boldsymbol{x}_k - \boldsymbol{\xi}^T \boldsymbol{x}_j \right) = \arg\max_k \left( \boldsymbol{\xi}^T \boldsymbol{x}_k - \max_{j, j \neq k} \boldsymbol{\xi}^T \boldsymbol{x}_j \right) \tag{72}$$

$$0 \leqslant c = \max_k \min_{j, j \neq k} \left( \boldsymbol{\xi}^T \boldsymbol{x}_k - \boldsymbol{\xi}^T \boldsymbol{x}_j \right) = \max_k \left( \boldsymbol{\xi}^T \boldsymbol{x}_k - \max_{j, j \neq k} \boldsymbol{\xi}^T \boldsymbol{x}_j \right) . \tag{73}$$

Next we consider the case where the iteration has only one stable fixed point.

**A.1.5.2   One Stable State: Fixed Point Near the Mean of the Patterns.**   We start with the case where no pattern is well separated from the others.

•*Global fixed point near the global mean: Analysis using the data center.*

We revisit the bound on the Jacobian of the iteration by utilizing properties of pattern distributions. We begin with a probabilistic interpretation where we consider $p_i$ as the probability of selecting the vector $\boldsymbol{x}_i$. Consequently, we define expectations as $\mathrm{E}_{\boldsymbol{p}}[f(\boldsymbol{x})] = \sum_{i=1}^{N} p_i f(\boldsymbol{x}_i)$. In this setting the matrix

$$\boldsymbol{X}\left(\mathrm{diag}(\boldsymbol{p}) - \boldsymbol{p}\boldsymbol{p}^T\right)\boldsymbol{X}^T \tag{74}$$

is the covariance matrix of data $\boldsymbol{X}$ when its vectors are selected according to the probability $\boldsymbol{p}$:

$$\boldsymbol{X}\left(\mathrm{diag}(\boldsymbol{p}) - \boldsymbol{p}\boldsymbol{p}^T\right)\boldsymbol{X}^T = \boldsymbol{X}\mathrm{diag}(\boldsymbol{p})\boldsymbol{X}^T - \boldsymbol{X}\boldsymbol{p}\boldsymbol{p}^T\boldsymbol{X}^T \tag{75}$$

$$= \sum_{i=1}^{N} p_i\,\boldsymbol{x}_i\,\boldsymbol{x}_i^T - \left(\sum_{i=1}^{N} p_i\,\boldsymbol{x}_i\right)\left(\sum_{i=1}^{N} p_i\,\boldsymbol{x}_i\right)^T \tag{76}$$

$$= \mathrm{E}_{\boldsymbol{p}}[\boldsymbol{x}\,\boldsymbol{x}^T] - \mathrm{E}_{\boldsymbol{p}}[\boldsymbol{x}]\,\mathrm{E}_{\boldsymbol{p}}[\boldsymbol{x}]^T = \mathrm{Var}_{\boldsymbol{p}}[\boldsymbol{x}]\,, \tag{77}$$

therefore we have

$$\mathrm{J} = \beta\,\mathrm{Var}_{\boldsymbol{p}}[\boldsymbol{x}]\,. \tag{78}$$

The largest eigenvalue of the covariance matrix (equal to the largest singular value) is the variance in the direction of the eigenvector associated with the largest eigenvalue.

We define:

$$\boldsymbol{m_x} = \frac{1}{N}\sum_{i=1}^{N} \boldsymbol{x}_i\,, \tag{79}$$

$$m_{\mathrm{max}} = \max_{1\leqslant i\leqslant N}\|\boldsymbol{x}_i - \boldsymbol{m_x}\|_2\,. \tag{80}$$

$\boldsymbol{m_x}$ is the arithmetic mean (the center) of the patterns. $m_{\mathrm{max}}$ is the maximal distance of the patterns to the center $\boldsymbol{m_x}$ .

The variance of the patterns is

$$\mathrm{Var}_{\boldsymbol{p}}[\boldsymbol{x}] = \sum_{i=1}^{N} p_i\,\boldsymbol{x}_i\,\boldsymbol{x}_i^T - \left(\sum_{i=1}^{N} p_i\,\boldsymbol{x}_i\right)\left(\sum_{i=1}^{N} p_i\,\boldsymbol{x}_i\right)^T \tag{81}$$

$$= \sum_{i=1}^{N} p_i\left(\boldsymbol{x}_i - \sum_{i=1}^{N} p_i\boldsymbol{x}_i\right)\left(\boldsymbol{x}_i - \sum_{i=1}^{N} p_i\boldsymbol{x}_i\right)^T\,.$$

The maximal distance to the center $m_{\mathrm{max}}$ allows the derivation of a bound on the norm of the Jacobian.

Next lemma gives a condition for a global fixed point.

**Lemma A3.** *The following bound on the norm $\|\mathrm{J}\|_2$ of the Jacobian of the fixed point iteration $f$ holds independent of $\boldsymbol{p}$ or the query $\boldsymbol{\xi}$.*

$$\|\mathrm{J}\|_2 \leqslant \beta\,m_{\mathrm{max}}^2\,. \tag{82}$$

*For $\beta\,m_{\mathrm{max}}^2 < 1$ there exists a unique fixed point (global fixed point) of iteration $f$ in each compact set.*

*Proof.*  In order to bound the variance we compute the vector $\boldsymbol{a}$ that minimizes

$$f(\boldsymbol{a}) = \sum_{i=1}^{N} p_i\|\boldsymbol{x}_i - \boldsymbol{a}\|^2 = \sum_{i=1}^{N} p_i(\boldsymbol{x}_i - \boldsymbol{a})^T(\boldsymbol{x}_i - \boldsymbol{a})\,. \tag{83}$$

The solution to

$$\frac{\partial f(\boldsymbol{a})}{\partial \boldsymbol{a}} = 2 \sum_{i=1}^{N} p_i(\boldsymbol{a} - \boldsymbol{x}_i) = 0 \tag{84}$$

is

$$\boldsymbol{a} = \sum_{i=1}^{N} p_i \boldsymbol{x}_i \ . \tag{85}$$

The Hessian of $f$ is positive definite since

$$\frac{\partial^2 f(\boldsymbol{a})}{\partial \boldsymbol{a}^2} = 2 \sum_{i=1}^{N} p_i \, \boldsymbol{I} = 2 \, \boldsymbol{I} \tag{86}$$

and $f$ is a convex function. Hence, the mean

$$\bar{\boldsymbol{x}} := \sum_{i=1}^{N} p_i \, \boldsymbol{x}_i \tag{87}$$

minimizes $\sum_{i=1}^{N} p_i \|\boldsymbol{x}_i - \boldsymbol{a}\|^2$. Therefore, we have

$$\sum_{i=1}^{N} p_i \|\boldsymbol{x}_i - \bar{\boldsymbol{x}}\|^2 \leqslant \sum_{i=1}^{N} p_i \|\boldsymbol{x}_i - \boldsymbol{m}_{\boldsymbol{x}}\|^2 \leqslant m_{\max}^2 \ . \tag{88}$$

Let us quickly recall that the spectral norm of an outer product of two vectors is the product of the Euclidean norms of the vectors:

$$\left\| \boldsymbol{a}\boldsymbol{b}^T \right\|_2 = \sqrt{\lambda_{\max}(\boldsymbol{b}\boldsymbol{a}^T \boldsymbol{a}\boldsymbol{b}^T)} = \|\boldsymbol{a}\| \sqrt{\lambda_{\max}(\boldsymbol{b}\boldsymbol{b}^T)} = \|\boldsymbol{a}\| \, \|\boldsymbol{b}\| \ , \tag{89}$$

since $\boldsymbol{b}\boldsymbol{b}^T$ has eigenvector $\boldsymbol{b}/\|\boldsymbol{b}\|$ with eigenvalue $\|\boldsymbol{b}\|^2$ and otherwise zero eigenvalues.

We now bound the variance of the patterns:

$$\|\mathrm{Var}_{\boldsymbol{p}}[\boldsymbol{x}]\|_2 \leqslant \sum_{i=1}^{N} p_i \left\| (\boldsymbol{x}_i - \bar{\boldsymbol{x}}) \, (\boldsymbol{x}_i - \bar{\boldsymbol{x}})^T \right\|_2 \tag{90}$$

$$= \sum_{i=1}^{N} p_i \|\boldsymbol{x}_i - \bar{\boldsymbol{x}}\|^2 \leqslant \sum_{i=1}^{N} p_i \|\boldsymbol{x}_i - \boldsymbol{m}_{\boldsymbol{x}}\|^2 \leqslant m_{\max}^2 \ .$$

The bound of the lemma on $\|\mathrm{J}\|_2$ follows from Eq. (78).

For $\|\mathrm{J}\|_2 \leqslant \beta \, m_{\max}^2 < 1$ we have a contraction mapping on each compact set. Banach fixed point theorem says there is a unique fixed point in the compact set.

$$\square$$

Now let us further investigate the tightness of the bound on $\|\mathrm{Var}_{\boldsymbol{p}}[\boldsymbol{x}]\|_2$ via $\|\boldsymbol{x}_i - \bar{\boldsymbol{x}}\|^2$: we consider the trace, which is the sum $\sum_{k=1}^{d} e_k$ of the w.l.o.g. ordered nonnegative eigenvalues $e_k$ of $\mathrm{Var}_{\boldsymbol{p}}[\boldsymbol{x}]$ The spectral norm is equal to the largest eigenvalue $e_1$, which is equal to the largest singular value, as we have positive semidefinite matrices. We obtain:

$$\|\mathrm{Var}_{\boldsymbol{p}}[\boldsymbol{x}]\|_2 = \mathrm{Tr}\left( \sum_{i=1}^{N} p_i \, (\boldsymbol{x}_i - \bar{\boldsymbol{x}}) \, (\boldsymbol{x}_i - \bar{\boldsymbol{x}})^T \right) - \sum_{k=2}^{d} e_k \tag{91}$$

$$= \sum_{i=1}^{N} p_i \mathrm{Tr}\left( (\boldsymbol{x}_i - \bar{\boldsymbol{x}}) \, (\boldsymbol{x}_i - \bar{\boldsymbol{x}})^T \right) - \sum_{k=2}^{d} e_k$$

$$= \sum_{i=1}^{N} p_i \|\boldsymbol{x}_i - \bar{\boldsymbol{x}}\|^2 - \sum_{k=2}^{d} e_k \ .$$

Therefore, the tightness of the bound depends on eigenvalues which are not the largest. Hence variations which are not along the largest variation weaken the bound.

Next we investigate the location of fixed points which existence is ensured by the global convergence stated in Theorem A2. For $N$ patterns $\boldsymbol{X} = (\boldsymbol{x}_1, \ldots, \boldsymbol{x}_N)$, we consider the iteration

$$\boldsymbol{\xi}^{\text{new}} \; = \; f(\boldsymbol{\xi}) \; = \; \boldsymbol{X}\boldsymbol{p} \; = \; \boldsymbol{X}\text{softmax}(\beta \boldsymbol{X}^T \boldsymbol{\xi}) \tag{92}$$

using

$$\boldsymbol{p} \; = \; \text{softmax}(\beta \boldsymbol{X}^T \boldsymbol{\xi}) \, . \tag{93}$$

$\boldsymbol{\xi}^{\text{new}}$ is in the simplex of the patterns, that is, $\boldsymbol{\xi}^{\text{new}} = \sum_i p_i \boldsymbol{x}_i$ with $\sum_i p_i = 1$ and $0 \leqslant p_i$. Hence, after one update $\boldsymbol{\xi}$ is in the simplex of the pattern and stays there. If the center $\boldsymbol{m}_{\boldsymbol{x}}$ is the zero vector $\boldsymbol{m}_{\boldsymbol{x}} = \boldsymbol{0}$, that is, the data is centered, then the mean is a fixed point of the iteration. For $\boldsymbol{\xi} = \boldsymbol{m}_{\boldsymbol{x}} = \boldsymbol{0}$ we have

$$\boldsymbol{p} \; = \; 1/N \, \boldsymbol{1} \tag{94}$$

and

$$\boldsymbol{\xi}^{\text{new}} \; = \; 1/N \, \boldsymbol{X} \, \boldsymbol{1} \; = \; \boldsymbol{m}_{\boldsymbol{x}} \; = \; \boldsymbol{\xi} \, . \tag{95}$$

In particular normalization methods like batch normalization would promote the mean as a fixed point.

We consider the differences of dot products for $\boldsymbol{x}_i$: $\boldsymbol{x}_i^T \boldsymbol{x}_i - \boldsymbol{x}_i^T \boldsymbol{x}_j = \boldsymbol{x}_i^T (\boldsymbol{x}_i - \boldsymbol{x}_j)$, for fixed point $\boldsymbol{m}_{\boldsymbol{x}}^*$: $(\boldsymbol{m}_{\boldsymbol{x}}^*)^T \boldsymbol{x}_i - (\boldsymbol{m}_{\boldsymbol{x}}^*)^T \boldsymbol{x}_j = (\boldsymbol{m}_{\boldsymbol{x}}^*)^T (\boldsymbol{x}_i - \boldsymbol{x}_j)$, and for the center $\boldsymbol{m}_{\boldsymbol{x}}$: $\boldsymbol{m}_{\boldsymbol{x}}^T \boldsymbol{x}_i - \boldsymbol{m}_{\boldsymbol{x}}^T \boldsymbol{x}_j = \boldsymbol{m}_{\boldsymbol{x}}^T (\boldsymbol{x}_i - \boldsymbol{x}_j)$. Using the Cauchy-Schwarz inequality, we get

$$\left| \boldsymbol{\xi}^T (\boldsymbol{x}_i - \boldsymbol{x}_j) \right| \; \leqslant \; \|\boldsymbol{\xi}\| \, \|\boldsymbol{x}_i - \boldsymbol{x}_j\| \; \leqslant \; \|\boldsymbol{\xi}\| \, (\|\boldsymbol{x}_i - \boldsymbol{m}_{\boldsymbol{x}}\| + \|\boldsymbol{x}_j - \boldsymbol{m}_{\boldsymbol{x}}\|) \tag{96}$$
$$\leqslant \; 2 \, m_{\max} \, \|\boldsymbol{\xi}\| \, .$$

This inequality gives:

$$\left| \boldsymbol{\xi}^T (\boldsymbol{x}_i - \boldsymbol{x}_j) \right| \; \leqslant \; 2 \, m_{\max} \, (m_{\max} + \|\boldsymbol{m}_{\boldsymbol{x}}\|) \, , \tag{97}$$
$$\left| \boldsymbol{\xi}^T (\boldsymbol{x}_i - \boldsymbol{x}_j) \right| \; \leqslant \; 2 \, m_{\max} \, M \, ,$$

where we used $\|\boldsymbol{\xi} - \boldsymbol{0}\| \leqslant \|\boldsymbol{\xi} - \boldsymbol{m}_{\boldsymbol{x}}\| + \|\boldsymbol{m}_{\boldsymbol{x}} - \boldsymbol{0}\|$, $\|\boldsymbol{\xi} - \boldsymbol{m}_{\boldsymbol{x}}\| = \|\sum_i p_i \boldsymbol{x}_i - \boldsymbol{m}_{\boldsymbol{x}}\| \leqslant \sum_i p_i \|\boldsymbol{x}_i - \boldsymbol{m}_{\boldsymbol{x}}\| \leqslant m_{\max}$, and $M = \max_i \|\boldsymbol{x}_i\|$. In particular

$$\beta \left| \boldsymbol{m}_{\boldsymbol{x}}^T (\boldsymbol{x}_i - \boldsymbol{x}_j) \right| \; \leqslant \; 2 \, \beta \, m_{\max} \, \|\boldsymbol{m}_{\boldsymbol{x}}\| \, , \tag{98}$$
$$\beta \left| (\boldsymbol{m}_{\boldsymbol{x}}^*)^T (\boldsymbol{x}_i - \boldsymbol{x}_j) \right| \; \leqslant \; 2 \, \beta \, m_{\max} \, \|\boldsymbol{m}_{\boldsymbol{x}}^*\| \; \leqslant \; 2 \, \beta \, m_{\max} \, (m_{\max} + \|\boldsymbol{m}_{\boldsymbol{x}}\|) \, , \tag{99}$$
$$\beta \left| \boldsymbol{x}_i^T (\boldsymbol{x}_i - \boldsymbol{x}_j) \right| \; \leqslant \; 2 \, \beta \, m_{\max} \, \|\boldsymbol{x}_i\| \; \leqslant \; 2 \, \beta \, m_{\max} \, (m_{\max} + \|\boldsymbol{m}_{\boldsymbol{x}}\|) \, . \tag{100}$$

Let $i = \arg\max_j \boldsymbol{\xi}^T \boldsymbol{x}_j$, therefore the maximal softmax component is $i$. For the maximal softmax component $i$ we have:

$$[\text{softmax}(\beta \, \boldsymbol{X}^T \boldsymbol{\xi})]_i \; = \; \frac{1}{1 \, + \, \sum_{j \neq i} \exp(- \, \beta \, (\boldsymbol{\xi}^T \boldsymbol{x}_i \, - \, \boldsymbol{\xi}^T \boldsymbol{x}_j))} \tag{101}$$
$$\leqslant \; \frac{1}{1 \, + \, \sum_{j \neq i} \exp(- \, 2 \, \beta \, m_{\max} \, (m_{\max} \, + \, \|\boldsymbol{m}_{\boldsymbol{x}}\|))}$$
$$= \; \frac{1}{1 \, + \, (N - 1) \exp(- \, 2 \, \beta \, m_{\max} \, (m_{\max} \, + \, \|\boldsymbol{m}_{\boldsymbol{x}}\|))}$$
$$= \; \frac{\exp(2 \, \beta \, m_{\max} \, (m_{\max} \, + \, \|\boldsymbol{m}_{\boldsymbol{x}}\|))}{\exp(2 \, \beta \, m_{\max} \, (m_{\max} \, + \, \|\boldsymbol{m}_{\boldsymbol{x}}\|)) \, + \, (N - 1)}$$
$$\leqslant \; 1/N \, \exp(2 \, \beta \, m_{\max} \, (m_{\max} \, + \, \|\boldsymbol{m}_{\boldsymbol{x}}\|)) \, .$$

Analogously we obtain for $i = \arg\max_j \boldsymbol{m}_{\boldsymbol{x}}^T \boldsymbol{x}_j$, a bound on the maximal softmax component $i$ if the center is put into the iteration:

$$[\text{softmax}(\beta \, \boldsymbol{X}^T \boldsymbol{m}_{\boldsymbol{x}})]_i \; \leqslant \; 1/N \; \exp(2 \, \beta \, m_{\max} \, \|\boldsymbol{m}_{\boldsymbol{x}}\|) \, . \tag{102}$$

Analog we obtain a bound for $i = \arg\max_j (\boldsymbol{m}_{\boldsymbol{x}}^*)^T \boldsymbol{x}_j$ on the maximal softmax component $i$ of the fixed point:

$$\begin{aligned}[\text{softmax}(\beta \, \boldsymbol{X}^T \boldsymbol{m}_{\boldsymbol{x}}^*)]_i \; &\leqslant \; 1/N \; \exp(2 \, \beta \, m_{\max} \, \|\boldsymbol{m}_{\boldsymbol{x}}^*\|) \\ &\leqslant \; 1/N \; \exp(2 \, \beta \, m_{\max} \, (m_{\max} \, + \, \|\boldsymbol{m}_{\boldsymbol{x}}\|)) \, .\end{aligned} \tag{103}$$

The two important terms are $m_{\max}$, the variance or spread of the data and $\|\boldsymbol{m}_{\boldsymbol{x}}\|$, which tells how well the data is centered. For a contraction mapping we already required $\beta m_{\max}^2 < 1$, therefore the first term in the exponent is $2\beta m_{\max}^2 < 2$. The second term $2\beta m_{\max}\|\boldsymbol{m}_{\boldsymbol{x}}\|$ is small if the data is centered.

•*Global fixed point near the global mean: Analysis using softmax values.*

If $\boldsymbol{\xi}^T \boldsymbol{x}_i \approx \boldsymbol{\xi}^T \boldsymbol{x}_j$ for all $i$ and $j$, then $p_i \approx 1/N$ and we have $m = \max_i p_i(1 - p_i) < 1/N$. For $M \leqslant 1/\sqrt{2\beta}$ we obtain from Lemma A2:

$$\|\text{J}\|_2 \; < \; 1 \, . \tag{104}$$

The local fixed point is $\boldsymbol{m}_{\boldsymbol{x}}^* \approx \boldsymbol{m}_{\boldsymbol{x}} = (1/N) \sum_{i=1}^N \boldsymbol{x}_i$ with $p_i \approx 1/N$.

We now treat this case more formally. First we discuss conditions that ensure that the iteration is a contraction mapping. We consider the iteration Eq. (57) in the variable $\boldsymbol{p}$:

$$\boldsymbol{p}^{\text{new}} \; = \; g(\boldsymbol{p}) \; = \; \text{softmax}(\beta \boldsymbol{X}^T \boldsymbol{X} \boldsymbol{p}) \, . \tag{105}$$

The Jacobian is

$$\text{J}(\boldsymbol{p}) \; = \; \frac{\partial g(\boldsymbol{p})}{\partial \boldsymbol{p}} \; = \; \boldsymbol{X}^T \boldsymbol{X} \, \text{J}_s \tag{106}$$

with

$$\text{J}_s(\boldsymbol{p}^{\text{new}}) \; = \; \beta \left( \text{diag}(\boldsymbol{p}^{\text{new}}) \, - \, \boldsymbol{p}^{\text{new}}(\boldsymbol{p}^{\text{new}})^T \right) \, . \tag{107}$$

The version of the mean value theorem in Lemma A32 states for $\text{J}^m = \int_0^1 \text{J}(\lambda\boldsymbol{p}) \, \mathrm{d}\lambda = \boldsymbol{X}^T \boldsymbol{X} \text{J}_s^m$ with the symmetric matrix $\text{J}_s^m = \int_0^1 \text{J}_s(\lambda\boldsymbol{p}) \, \mathrm{d}\lambda$:

$$\boldsymbol{p}^{\text{new}} \; = \; g(\boldsymbol{p}) \; = \; g(\boldsymbol{0}) \, + \, (\text{J}^m)^T \boldsymbol{p} \; = \; g(\boldsymbol{0}) \, + \, \text{J}_s^m \, \boldsymbol{X}^T \boldsymbol{X} \, \boldsymbol{p} \; = \; 1/N \, \boldsymbol{1} \, + \, \text{J}_s^m \, \boldsymbol{X}^T \boldsymbol{X} \, \boldsymbol{p} \, . \tag{108}$$

With $m = \max_i p_i(1 - p_i)$, Eq. (476) from Lemma A24 is

$$\|\text{J}_s(\boldsymbol{p})\|_2 \; = \; \beta \left\| \text{diag}(\boldsymbol{p}) - \boldsymbol{p}\boldsymbol{p}^T \right\|_2 \; \leqslant \; 2 \, m \, \beta \, . \tag{109}$$

First observe that $\lambda p_i(1 - \lambda p_i) \leqslant p_i(1 - p_i)$ for $p_i \leqslant 0.5$ and $\lambda \in [0, 1]$, since $p_i(1 - p_i) - \lambda p_i(1 - \lambda p_i) = (1 - \lambda)p_i(1 - (1 + \lambda)p_i) \geq 0$. For $\max_i p_i \leqslant 0.5$ this observation leads to the following bound for $\text{J}_s^m$:

$$\|\text{J}_s^m\|_2 \; \leqslant \; 2 \, m \, \beta \, . \tag{110}$$

Eq. (479) in Lemma A24 states that every $\text{J}_s$ is bounded by $1/2\beta$, therefore also the mean:

$$\|\text{J}_s^m\|_2 \; \leqslant \; 0.5 \, \beta \, . \tag{111}$$

Since $m = \max_i p_i(1 - p_i) < \max_i p_i = p_{\max}$, the previous bounds can be combined as follows:

$$\|\text{J}_s^m\|_2 \; \leqslant \; 2 \, \min\{0.25, p_{\max}\} \, \beta \, . \tag{112}$$

Consequently,

$$\left\|\mathrm{J}^m\right\|_2 \ \leqslant \ N\,M^2\,2\,\min\{0.25, p_{\max}\}\,\beta\;, \tag{113}$$

where we used Eq. (170). $\left\|\boldsymbol{X}^T\boldsymbol{X}\right\|_2 = \left\|\boldsymbol{X}\,\boldsymbol{X}^T\right\|_2$, therefore $\left\|\boldsymbol{X}^T\boldsymbol{X}\right\|_2$ is $N$ times the maximal second moment of the data squared.

Obviously, $g(\boldsymbol{p})$ is a contraction mapping in compact sets, where

$$N\,M^2\,2\,\min\{0.25, p_{\max}\}\,\beta \ < \ 1\;. \tag{114}$$

S is the sphere around the origin $\boldsymbol{0}$ with radius one. For

$$\boldsymbol{p}^{\mathrm{new}} \ = \ g(\boldsymbol{p}) \ = \ 1/N\,\boldsymbol{1} \ + \ \mathrm{J}^m\,\boldsymbol{p}\;, \tag{115}$$

we have $\|\boldsymbol{p}\| \leqslant \|\boldsymbol{p}\|_1 = 1$ and $\|\boldsymbol{p}^{\mathrm{new}}\| \leqslant \|\boldsymbol{p}^{\mathrm{new}}\|_1 = 1$. Therefore, $g$ maps points from S into S. $g$ is a contraction mapping for

$$\left\|\mathrm{J}^m\right\|_2 \ \leqslant \ N\,M^2\,2\,\min\{0.25, p_{\max}\}\,\beta \ = \ c \ < \ 1\;. \tag{116}$$

According to Banach fixed point theorem $g$ has a fixed point in the sphere S.

Hölder's inequality gives:

$$\|\boldsymbol{p}\|^2 \ = \ \boldsymbol{p}^T\boldsymbol{p} \ \leqslant \ \|\boldsymbol{p}\|_1\|\boldsymbol{p}\|_\infty \ = \ \|\boldsymbol{p}\|_\infty \ = \ p_{\max}\;. \tag{117}$$

Alternatively:

$$\|\boldsymbol{p}\|^2 \ = \ \sum_i p_i^2 \ = \ p_{\max}\sum_i \frac{p_i}{p_{\max}}\,p_i \ \leqslant \ p_{\max}\sum_i p_i \ = \ p_{\max}\;. \tag{118}$$

Let now S be the sphere around the origin $\boldsymbol{0}$ with radius $1/\sqrt{N} + \sqrt{p_{\max}}$ and let $\left\|\mathrm{J}^m(\boldsymbol{p})\right\|_2 \leqslant c < 1$ for $\boldsymbol{p} \in$ S. The old $\boldsymbol{p}$ is in the sphere S ($\boldsymbol{p} \in$ S) since $p_{\max} < \sqrt{p_{\max}}$ for $p_{\max} < 1$. We have

$$\|\boldsymbol{p}^{\mathrm{new}}\| \ \leqslant \ 1/\sqrt{N} \ + \ \left\|\mathrm{J}^m\right\|_2\|\boldsymbol{p}\| \ \leqslant \ 1/\sqrt{N} \ + \ \sqrt{p_{\max}}\;. \tag{119}$$

Therefore, $g$ is a mapping from S into S and a contraction mapping. According to Banach fixed point theorem, a fixed point exists in S.

For the 1-norm, we use Lemma A24 and $\|\boldsymbol{p}\|_1 = 1$ to obtain from Eq. (115):

$$\left\|\boldsymbol{p}^{\mathrm{new}} \ - \ 1/N\,\boldsymbol{1}\right\|_1 \ \leqslant \ \left\|\mathrm{J}^m\right\|_1 \ \leqslant \ 2\,\beta\,m\,\|\boldsymbol{X}\|_\infty\,M_1\;, \tag{120}$$

$$\left\|\boldsymbol{p}^{\mathrm{new}} \ - \ 1/N\,\boldsymbol{1}\right\|_1 \ \leqslant \ \left\|\mathrm{J}^m\right\|_1 \ \leqslant \ 2\,\beta\,m\,N\,M_\infty\,M_1\;, \tag{121}$$

$$\left\|\boldsymbol{p}^{\mathrm{new}} \ - \ 1/N\,\boldsymbol{1}\right\|_1 \ \leqslant \ \left\|\mathrm{J}^m\right\|_1 \ \leqslant \ 2\,\beta\,m\,N\,M^2\;, \tag{122}$$

where $m = \max_i p_i(1 - p_i)$, $M_1 = \|\boldsymbol{X}\|_1 = \max_i \|\boldsymbol{x}_i\|_1$, $M = \max_i \|\boldsymbol{x}_i\|$, $\|\boldsymbol{X}\|_\infty = \left\|\boldsymbol{X}^T\right\|_1 = \max_i \left\|[X^T]_i\right\|_1$ (maximal absolute row sum norm), and $M_\infty = \max_i \|\boldsymbol{x}_i\|_\infty$. Let us quickly mention some auxiliary estimates related to $\boldsymbol{X}^T\boldsymbol{X}$:

$$\left\|\boldsymbol{X}^T\boldsymbol{X}\right\|_1 \ = \ \max_i \sum_{j=1}^N \left|\boldsymbol{x}_i^T\boldsymbol{x}_j\right| \ \leqslant \ \max_i \sum_{j=1}^N \|\boldsymbol{x}_i\|_\infty\,\|\boldsymbol{x}_j\|_1 \tag{123}$$

$$\leqslant \ M_\infty \sum_{j=1}^N M_1 \ = \ N\,M_\infty\,M_1\;,$$

where the first inequaltiy is from Hölder's inequality. We used

$$\left\|\boldsymbol{X}^T\boldsymbol{X}\right\|_1 \ = \ \max_i \sum_{j=1}^N \left|\boldsymbol{x}_i^T\boldsymbol{x}_j\right| \ \leqslant \ \max_i \sum_{j=1}^N \|\boldsymbol{x}_i\|\,\|\boldsymbol{x}_j\| \tag{124}$$

$$\leqslant \ M \sum_{j=1}^N M \ = \ N\,M^2\;,$$

where the first inequality is from Hölder's inequality (here the same as the Cauchy-Schwarz inequality). See proof of Lemma A24 for the 1-norm bound on $J_s$. Everything else follows from the fact that the 1-norm is sub-multiplicative as induced matrix norm.

We consider the minimal $\|\boldsymbol{p}\|$.

$$\min_{\boldsymbol{p}} \ \|\boldsymbol{p}\|^2 \tag{125}$$

$$\text{s.t.} \quad \sum_i p_i = 1$$

$$\forall_i : \ p_i \ \geq \ 0 \,.$$

The solution to this minimization problem is $\boldsymbol{p} = (1/N)\mathbf{1}$. Therefore, we have $1/\sqrt{N} \leqslant \|\boldsymbol{p}\|$ and $1/N \leqslant \|\boldsymbol{p}\|^2$ Using Eq. (119) we obtain

$$1/\sqrt{N} \ \leqslant \ \|\boldsymbol{p}^{\text{new}}\| \ \leqslant \ 1/\sqrt{N} \ + \ \sqrt{p_{\max}} \,. \tag{126}$$

Moreover

$$\|\boldsymbol{p}^{\text{new}}\|^2 \ = \ (\boldsymbol{p}^{\text{new}})^T \boldsymbol{p}^{\text{new}} \ = \ 1/N \ + \ (\boldsymbol{p}^{\text{new}})^T \mathrm{J}^m \, \boldsymbol{p} \ \leqslant \ 1/N \ + \ \|\mathrm{J}^m\|_2 \, \|\boldsymbol{p}\| \tag{127}$$
$$\leqslant \ 1/N \ + \ \|\mathrm{J}^m\|_2 \,,$$

since $\boldsymbol{p}^{\text{new}} \in \mathrm{S}$ and $\boldsymbol{p} \in \mathrm{S}$.

For the fixed point, we have

$$\|\boldsymbol{p}^*\|^2 \ = \ (\boldsymbol{p}^*)^T \boldsymbol{p}^* \ = \ 1/N \ + \ (\boldsymbol{p}^*)^T \mathrm{J}^m \, \boldsymbol{p}^* \ \leqslant \ 1/N \ + \ \|\mathrm{J}^m\|_2 \, \|\boldsymbol{p}^*\|^2 \,, \tag{128}$$

and hence

$$1/N \ \leqslant \ \|\boldsymbol{p}^*\|^2 \ \leqslant \ 1/N \frac{1}{1 \ - \ \|\mathrm{J}^m\|_2} \ = \ 1/N \, (1 \ + \ \frac{\|\mathrm{J}^m\|_2}{1 \ - \ \|\mathrm{J}^m\|_2}) \,. \tag{129}$$

Therefore, for small $\|\mathrm{J}^m\|_2$ we have $\boldsymbol{p}^* \approx (1/N)\mathbf{1}$.

**A.1.5.3 Many Stable States: Fixed Points Near Stored Patterns.** We move on to the next case, where the patterns $\boldsymbol{x}_i$ are well separated. In this case the iteration goes to the pattern to which the initial $\boldsymbol{\xi}$ is most similar. If the initial $\boldsymbol{\xi}$ is similar to a vector $\boldsymbol{x}_i$ then it will converge to $\boldsymbol{x}_i$ and $\boldsymbol{p}$ will be $\boldsymbol{e}_i$. The main ingredients are again Banach's Theorem and estimates on the Jacobian norm.

•*Proof of a fixed point by Banach Fixed Point Theorem.*

→ *Mapped Vectors Stay in a Compact Environment.* We show that if $\boldsymbol{x}_i$ is sufficient dissimilar to other $\boldsymbol{x}_j$ then there is an compact environment of $\boldsymbol{x}_i$ (a sphere) where the fixed point iteration maps this environment into itself. The idea of the proof is to define a sphere around $\boldsymbol{x}_i$ for which points from the sphere are mapped by $f$ into the sphere.

We first need following lemma which bounds the distance $\|\boldsymbol{x}_i \ - \ f(\boldsymbol{\xi})\|$, where $\boldsymbol{x}_i$ is the pattern that is least separated from $\boldsymbol{\xi}$ but separated from other patterns.

**Lemma A4.** *For a query $\boldsymbol{\xi}$ and data $\boldsymbol{X} = (\boldsymbol{x}_1, \ldots, \boldsymbol{x}_N)$, there exists a $\boldsymbol{x}_i$ that is least separated from $\boldsymbol{\xi}$ while being separated from other $\boldsymbol{x}_j$ with $j \neq i$:*

$$i \ = \ \arg\max_k \min_{j, j \neq k} \left( \boldsymbol{\xi}^T \boldsymbol{x}_k \ - \ \boldsymbol{\xi}^T \boldsymbol{x}_j \right) \ = \ \arg\max_k \left( \boldsymbol{\xi}^T \boldsymbol{x}_k \ - \ \max_{j, j \neq k} \boldsymbol{\xi}^T \boldsymbol{x}_j \right) \tag{130}$$

$$0 \ \leqslant \ c \ = \ \max_k \min_{j, j \neq k} \left( \boldsymbol{\xi}^T \boldsymbol{x}_k \ - \ \boldsymbol{\xi}^T \boldsymbol{x}_j \right) \ = \ \max_k \left( \boldsymbol{\xi}^T \boldsymbol{x}_k \ - \ \max_{j, j \neq k} \boldsymbol{\xi}^T \boldsymbol{x}_j \right) \,. \tag{131}$$

*For $\boldsymbol{x}_i$, the following holds:*

$$\|\boldsymbol{x}_i \ - \ f(\boldsymbol{\xi})\| \ \leqslant \ 2 \, \epsilon \, M \,, \tag{132}$$

*where*

$$M \ = \ \max_i \|\boldsymbol{x}_i\| \,, \tag{133}$$

$$\epsilon \ = \ (N - 1) \ \exp(- \, \beta \, c) \,. \tag{134}$$

*Proof.* For the softmax component $i$ we have:

$$[\text{softmax}(\beta\,\boldsymbol{X}^T\boldsymbol{\xi})]_i \;=\; \frac{1}{1\,+\,\sum_{j\neq i}\exp(\beta\,(\boldsymbol{\xi}^T\boldsymbol{x}_j\,-\,\boldsymbol{\xi}^T\boldsymbol{x}_i))} \;\geq\; \frac{1}{1\,+\,\sum_{j\neq i}\exp(-\,\beta\,c)} \quad (135)$$

$$=\; \frac{1}{1\,+\,(N-1)\exp(-\,\beta\,c)} \;=\; 1\,-\,\frac{(N-1)\exp(-\,\beta\,c)}{1\,+\,(N-1)\exp(-\,\beta\,c)}$$

$$\geq\; 1\,-\,(N-1)\exp(-\,\beta\,c) \;=\; 1\,-\,\epsilon$$

For softmax components $k\neq i$ we have

$$[\text{softmax}(\beta\boldsymbol{X}^T\boldsymbol{\xi})]_k \;=\; \frac{\exp(\beta\,(\boldsymbol{\xi}^T\boldsymbol{x}_k\,-\,\boldsymbol{\xi}^T\boldsymbol{x}_i))}{1\,+\,\sum_{j\neq i}\exp(\beta\,(\boldsymbol{\xi}^T\boldsymbol{x}_j\,-\,\boldsymbol{\xi}^T\boldsymbol{x}_i))} \;\leqslant\; \exp(-\,\beta\,c) \;=\; \frac{\epsilon}{N-1}\,. \tag{136}$$

The iteration $f$ can be written as

$$f(\boldsymbol{\xi}) \;=\; \boldsymbol{X}\text{softmax}(\beta\boldsymbol{X}^T\boldsymbol{\xi}) \;=\; \sum_{j=1}^{N}\boldsymbol{x}_j\,[\text{softmax}(\beta\boldsymbol{X}^T\boldsymbol{\xi})]_j\,. \tag{137}$$

We now can bound $\|\boldsymbol{x}_i\,-\,f(\boldsymbol{\xi})\|$:

$$\|\boldsymbol{x}_i\,-\,f(\boldsymbol{\xi})\| \;=\; \left\|\boldsymbol{x}_i\,-\,\sum_{j=1}^{N}[\text{softmax}(\beta\boldsymbol{X}^T\boldsymbol{\xi})]_j\,\boldsymbol{x}_j\right\| \tag{138}$$

$$=\; \left\|(1-[\text{softmax}(\beta\boldsymbol{X}^T\boldsymbol{\xi})]_i)\,\boldsymbol{x}_i\,-\,\sum_{j=1,j\neq i}^{N}[\text{softmax}(\beta\boldsymbol{X}^T\boldsymbol{\xi})]_j\,\boldsymbol{x}_j\right\|$$

$$\leqslant\; \epsilon\,\|\boldsymbol{x}_i\|\,+\,\frac{\epsilon}{N-1}\sum_{j=1,j\neq i}^{N}\|\boldsymbol{x}_j\|$$

$$\leqslant\; \epsilon\,M\,+\,\frac{\epsilon}{N-1}\sum_{j=1,j\neq i}^{N}M \;=\; 2\,\epsilon\,M\,.$$

$\square$

We define $\Delta_i$, i.e. the separation of pattern $\boldsymbol{x}_i$ from data $\boldsymbol{X}=(\boldsymbol{x}_1,\ldots,\boldsymbol{x}_N)$ as:

$$\Delta_i \;=\; \min_{j,j\neq i}\left(\boldsymbol{x}_i^T\boldsymbol{x}_i\,-\,\boldsymbol{x}_i^T\boldsymbol{x}_j\right) \;=\; \boldsymbol{x}_i^T\boldsymbol{x}_i\,-\,\max_{j,j\neq i}\boldsymbol{x}_i^T\boldsymbol{x}_j\,. \tag{139}$$

The pattern is separated from the other data if $0<\Delta_i$. Using the parallelogram identity, $\Delta_i$ can also be expressed as

$$\Delta_i \;=\; \min_{j,j\neq i}\frac{1}{2}\left(\|\boldsymbol{x}_i\|^2\,-\,\|\boldsymbol{x}_j\|^2\,+\,\|\boldsymbol{x}_i\,-\,\boldsymbol{x}_j\|^2\right) \tag{140}$$

$$=\; \frac{1}{2}\|\boldsymbol{x}_i\|^2\,-\,\frac{1}{2}\max_{j,j\neq i}\left(\|\boldsymbol{x}_j\|^2\,-\,\|\boldsymbol{x}_i\,-\,\boldsymbol{x}_j\|^2\right)\,.$$

For $\|\boldsymbol{x}_i\|=\|\boldsymbol{x}_j\|$ we have $\Delta_i=1/2\min_{j,j\neq i}\|\boldsymbol{x}_i\,-\,\boldsymbol{x}_j\|^2$.

Next we define the sphere where we want to apply Banach fixed point theorem.

**Definition 3** (Sphere $\text{S}_i$). *The sphere $\text{S}_i$ is defined as*

$$\text{S}_i \;:=\; \left\{\boldsymbol{\xi}\mid\|\boldsymbol{\xi}\,-\,\boldsymbol{x}_i\|\,\leqslant\,\frac{1}{\beta\,N\,M}\right\}\,. \tag{141}$$

**Lemma A5.** *With $\boldsymbol{\xi}$ given, if the assumptions*

*A1:* $\boldsymbol{\xi}$ *is inside sphere:* $\boldsymbol{\xi} \in \mathrm{S}_i$,

*A2:* *data point* $\boldsymbol{x}_i$ *is well separated from the other data:*

$$\Delta_i \; \geq \; \frac{2}{\beta\,N} \; + \; \frac{1}{\beta}\,\ln\left(2\,(N-1)\,N\,\beta\,M^2\right) \tag{142}$$

*hold, then* $f(\boldsymbol{\xi})$ *is inside the sphere:* $f(\boldsymbol{\xi}) \in \mathrm{S}_i$. *Therefore, with assumption (A2),* $f$ *is a mapping from* $\mathrm{S}_i$ *into* $\mathrm{S}_i$.

*Proof.* We need the separation $\tilde{\Delta}_i$ of $\boldsymbol{\xi}$ from the data.

$$\tilde{\Delta}_i \; = \; \min_{j,j\neq i}\left(\boldsymbol{\xi}^T\boldsymbol{x}_i \; - \; \boldsymbol{\xi}^T\boldsymbol{x}_j\right)\;. \tag{143}$$

Using the Cauchy-Schwarz inequality, we obtain for $1 \leqslant j \leqslant N$:

$$\left|\boldsymbol{\xi}^T\boldsymbol{x}_j \; - \; \boldsymbol{x}_i^T\boldsymbol{x}_j\right| \leqslant \|\boldsymbol{\xi} \; - \; \boldsymbol{x}_i\|\,\|\boldsymbol{x}_j\| \leqslant \|\boldsymbol{\xi} \; - \; \boldsymbol{x}_i\|\,M\;. \tag{144}$$

We have the lower bound

$$\tilde{\Delta}_i \; \geq \; \min_{j,j\neq i}\left(\left(\boldsymbol{x}_i^T\boldsymbol{x}_i \; - \; \|\boldsymbol{\xi} \; - \; \boldsymbol{x}_i\|\,M\right) \; - \; \left(\boldsymbol{x}_i^T\boldsymbol{x}_j \; + \; \|\boldsymbol{\xi} \; - \; \boldsymbol{x}_i\|\,M\right)\right) \tag{145}$$

$$= \; - \, 2\,\|\boldsymbol{\xi} \; - \; \boldsymbol{x}_i\|\,M \; + \; \min_{j,j\neq i}\left(\boldsymbol{x}_i^T\boldsymbol{x}_i \; - \; \boldsymbol{x}_i^T\boldsymbol{x}_j\right) \; = \; \Delta_i \; - \; 2\,\|\boldsymbol{\xi} \; - \; \boldsymbol{x}_i\|\,M$$

$$\geq \; \Delta_i \; - \; \frac{2}{\beta\,N}\;,$$

where we used the assumption (A1) of the lemma.

From the proof in Lemma A4 we have

$$p_{\max} \; = \; [\mathrm{softmax}(\beta\boldsymbol{X}^T\boldsymbol{\xi})]_i \; \geq \; 1 \; - \; (N-1)\,\exp(-\,\beta\,\tilde{\Delta}_i) \; = \; 1 \; - \; \tilde{\epsilon}\;. \tag{146}$$

Lemma A4 states that

$$\|\boldsymbol{x}_i \; - \; f(\boldsymbol{\xi})\| \; \leqslant \; 2\,\tilde{\epsilon}\,M \; = \; 2\,(N-1)\,\exp(-\,\beta\,\tilde{\Delta}_i)\,M \tag{147}$$

$$\leqslant \; 2\,(N-1)\,\exp(-\,\beta\,(\Delta_i \; - \; \frac{2}{\beta\,N}))\,M\;.$$

We have

$$\|\boldsymbol{x}_i \; - \; f(\boldsymbol{\xi})\| \tag{148}$$

$$\leqslant \; 2\,(N-1)\,\exp(-\,\beta\,(\frac{2}{\beta\,N} \; + \; \frac{1}{\beta}\,\ln\left(2\,(N-1)\,N\,\beta\,M^2\right) \; - \; \frac{2}{\beta\,N}))\,M$$

$$= \; 2\,(N-1)\,\exp(-\,\ln\left(2\,(N-1)\,N\,\beta\,M^2\right))\,M$$

$$= \; \frac{1}{N\,\beta\,M}\;,$$

where we used assumption (A2) of the lemma. Therefore, $f(\boldsymbol{\xi})$ is a mapping from the sphere $\mathrm{S}_i$ into the sphere $\mathrm{S}_i$: If $\boldsymbol{\xi} \in \mathrm{S}_i$ then $f(\boldsymbol{\xi}) \in \mathrm{S}_i$. $\qquad\square$

•*Contraction mapping.*

For applying Banach fixed point theorem we need to show that $f$ is contraction in the compact environment $\mathrm{S}_i$.

**Lemma A6.** *Assume that*

*A1:*

$$\Delta_i \; \geq \; \frac{2}{\beta\,N} \; + \; \frac{1}{\beta}\,\ln\left(2\,(N-1)\,N\,\beta\,M^2\right)\;, \tag{149}$$

*then* $f$ *is a contraction mapping in* $\mathrm{S}_i$.

*Proof.* The version of the mean value theorem Lemma A32 states for $\mathrm{J}^m = \int_0^1 \mathrm{J}(\lambda \boldsymbol{\xi} + (1-\lambda)\boldsymbol{x}_i) \, \mathrm{d}\lambda$:

$$f(\boldsymbol{\xi}) = f(\boldsymbol{x}_i) + \mathrm{J}^m (\boldsymbol{\xi} - \boldsymbol{x}_i). \tag{150}$$

Therefore

$$\|f(\boldsymbol{\xi}) - f(\boldsymbol{x}_i)\| \leqslant \|\mathrm{J}^m\|_2 \|\boldsymbol{\xi} - \boldsymbol{x}_i\|. \tag{151}$$

We define $\tilde{\boldsymbol{\xi}} = \lambda \boldsymbol{\xi} + (1-\lambda)\boldsymbol{x}_i$ for some $\lambda \in [0, 1]$. From the proof in Lemma A4 we have

$$p_{\max}(\tilde{\boldsymbol{\xi}}) = [\mathrm{softmax}(\beta \, \boldsymbol{X}^T \, \tilde{\boldsymbol{\xi}})]_i \geq 1 - (N-1) \exp(-\beta \, \tilde{\Delta}_i) = 1 - \tilde{\epsilon}, \tag{152}$$

$$\tilde{\epsilon} = (N-1) \exp(-\beta \, \tilde{\Delta}_i), \tag{153}$$

$$\tilde{\Delta}_i = \min_{j,j \neq i} \left( \tilde{\boldsymbol{\xi}}^T \boldsymbol{x}_i - \tilde{\boldsymbol{\xi}}^T \boldsymbol{x}_j \right). \tag{154}$$

First we compute an upper bound on $\tilde{\epsilon}$. We need the separation $\tilde{\Delta}_i$ of $\boldsymbol{\xi}$ from the data. Using the Cauchy-Schwarz inequality, we obtain for $1 \leqslant j \leqslant N$:

$$\left| \tilde{\boldsymbol{\xi}}^T \boldsymbol{x}_j - \boldsymbol{x}_i^T \boldsymbol{x}_j \right| \leqslant \left\| \tilde{\boldsymbol{\xi}} - \boldsymbol{x}_i \right\| \|\boldsymbol{x}_j\| \leqslant \left\| \tilde{\boldsymbol{\xi}} - \boldsymbol{x}_i \right\| M. \tag{155}$$

We have the lower bound on $\tilde{\Delta}_i$:

$$\begin{aligned}
\tilde{\Delta}_i &\geq \min_{j,j \neq i} \left( \left( \boldsymbol{x}_i^T \boldsymbol{x}_i - \left\| \tilde{\boldsymbol{\xi}} - \boldsymbol{x}_i \right\| M \right) - \left( \boldsymbol{x}_i^T \boldsymbol{x}_j + \left\| \tilde{\boldsymbol{\xi}} - \boldsymbol{x}_i \right\| M \right) \right) \\
&= -2 \left\| \tilde{\boldsymbol{\xi}} - \boldsymbol{x}_i \right\| M + \min_{j,j \neq i} \left( \boldsymbol{x}_i^T \boldsymbol{x}_i - \boldsymbol{x}_i^T \boldsymbol{x}_j \right) = \Delta_i - 2 \left\| \tilde{\boldsymbol{\xi}} - \boldsymbol{x}_i \right\| M \\
&\geq \Delta_i - 2 \|\boldsymbol{\xi} - \boldsymbol{x}_i\| M,
\end{aligned} \tag{156}$$

where we used $\left\| \tilde{\boldsymbol{\xi}} - \boldsymbol{x}_i \right\| = \lambda \|\boldsymbol{\xi} - \boldsymbol{x}_i\| \leqslant \|\boldsymbol{\xi} - \boldsymbol{x}_i\|$. From the definition of $\tilde{\epsilon}$ in Eq. (152) we have

$$\begin{aligned}
\tilde{\epsilon} &= (N-1) \exp(-\beta \, \tilde{\Delta}_i) \\
&\leqslant (N-1) \exp(-\beta (\Delta_i - 2 \|\boldsymbol{\xi} - \boldsymbol{x}_i\| M)) \\
&\leqslant (N-1) \exp\left( -\beta \left( \Delta_i - \frac{2}{\beta \, N} \right) \right),
\end{aligned} \tag{157}$$

where we used $\boldsymbol{\xi} \in \mathrm{S}_i$, therefore $\|\boldsymbol{\xi} - \boldsymbol{x}_i\| \leqslant \frac{1}{\beta \, N \, M}$.

Next we compute an lower bound on $\tilde{\epsilon}$. We start with an upper on $\tilde{\Delta}_i$:

$$\begin{aligned}
\tilde{\Delta}_i &\leqslant \min_{j,j \neq i} \left( \left( \boldsymbol{x}_i^T \boldsymbol{x}_i + \left\| \tilde{\boldsymbol{\xi}} - \boldsymbol{x}_i \right\| M \right) - \left( \boldsymbol{x}_i^T \boldsymbol{x}_j - \left\| \tilde{\boldsymbol{\xi}} - \boldsymbol{x}_i \right\| M \right) \right) \\
&= 2 \left\| \tilde{\boldsymbol{\xi}} - \boldsymbol{x}_i \right\| M + \min_{j,j \neq i} \left( \boldsymbol{x}_i^T \boldsymbol{x}_i - \boldsymbol{x}_i^T \boldsymbol{x}_j \right) = \Delta_i + 2 \left\| \tilde{\boldsymbol{\xi}} - \boldsymbol{x}_i \right\| M \\
&\leqslant \Delta_i + 2 \|\boldsymbol{\xi} - \boldsymbol{x}_i\| M,
\end{aligned} \tag{158}$$

where we used $\left\| \tilde{\boldsymbol{\xi}} - \boldsymbol{x}_i \right\| = \lambda \|\boldsymbol{\xi} - \boldsymbol{x}_i\| \leqslant \|\boldsymbol{\xi} - \boldsymbol{x}_i\|$. From the definition of $\tilde{\epsilon}$ in Eq. (152) we have

$$\begin{aligned}
\tilde{\epsilon} &= (N-1) \exp(-\beta \, \tilde{\Delta}_i) \\
&\geq (N-1) \exp(-\beta (\Delta_i + 2 \|\boldsymbol{\xi} - \boldsymbol{x}_i\| M)) \\
&\geq (N-1) \exp\left( -\beta \left( \Delta_i + \frac{2}{\beta \, N} \right) \right),
\end{aligned} \tag{159}$$

where we used $\boldsymbol{\xi} \in \mathrm{S}_i$, therefore $\|\boldsymbol{\xi} - \boldsymbol{x}_i\| \leqslant \frac{1}{\beta \, N \, M}$.

Now we bound the Jacobian. We can assume $\tilde{\epsilon} \leqslant 0.5$ otherwise $(1 - \tilde{\epsilon}) \leqslant 0.5$ in the following. From the proof of Lemma A24 we know for $p_{\max}(\tilde{\boldsymbol{\xi}}) \geq 1 - \tilde{\epsilon}$, then $p_i(\tilde{\boldsymbol{\xi}}) \leqslant \tilde{\epsilon}$ for $p_i(\tilde{\boldsymbol{\xi}}) \neq p_{\max}(\tilde{\boldsymbol{\xi}})$.

Therefore, $p_i(\tilde{\boldsymbol{\xi}})(1 - p_i(\tilde{\boldsymbol{\xi}})) \leqslant m \leqslant \tilde{\epsilon}(1 - \tilde{\epsilon})$ for all $i$. Next we use the derived upper and lower bound on $\tilde{\epsilon}$ in previous Eq. (61) in Lemma A2:

$$\left\| \mathrm{J}(\tilde{\boldsymbol{\xi}}) \right\|_2 \leqslant 2\,\beta\,N\,M^2\,\tilde{\epsilon} \,-\, 2\,\tilde{\epsilon}^2\,\beta\,N\,M^2 \tag{160}$$

$$\leqslant 2\,\beta\,N\,M^2\,(N-1)\,\exp\left(-\beta\left(\Delta_i - \frac{2}{\beta\,N}\right)\right) \,-$$

$$2\,(N-1)^2\,\exp\left(-2\,\beta\left(\Delta_i + \frac{2}{\beta\,N}\right)\right)\,\beta\,N\,M^2\,.$$

The bound Eq. (160) holds for the mean $\mathrm{J}^m$, too, since it averages over $\mathrm{J}(\tilde{\boldsymbol{\xi}})$:

$$\|\mathrm{J}^m\|_2 \leqslant 2\,\beta\,N\,M^2\,(N-1)\,\exp\left(-\beta\left(\Delta_i - \frac{2}{\beta\,N}\right)\right) \,- \tag{161}$$

$$2\,(N-1)^2\,\exp\left(-2\,\beta\left(\Delta_i + \frac{2}{\beta\,N}\right)\right)\,\beta\,N\,M^2\,.$$

The assumption of the lemma is

$$\Delta_i \,\geq\, \frac{2}{\beta\,N} \,+\, \frac{1}{\beta}\,\ln\left(2\,(N-1)\,N\,\beta\,M^2\right)\,, \tag{162}$$

This is

$$\Delta_i \,-\, \frac{2}{\beta\,N} \,\geq\, \frac{1}{\beta}\,\ln\left(2\,(N-1)\,N\,\beta\,M^2\right)\,, \tag{163}$$

Therefore, the spectral norm $\|\mathrm{J}\|_2$ can be bounded by:

$$\|\mathrm{J}^m\|_2 \,\leqslant\, 2\,\beta\,(N-1)\,\exp\left(-\beta\,\frac{1}{\beta}\,\ln\left(2\,(N-1)\,N\,\beta\,M^2\right)\right)\,N\,M^2\,- \tag{164}$$

$$2\,(N-1)^2\,\exp\left(-2\,\beta\left(\Delta_i + \frac{2}{\beta\,N}\right)\right)\,\beta\,N\,M^2$$

$$=\, 2\,\beta\,(N-1)\,\frac{1}{2\,(N-1)\,N\,\beta\,M^2}\,N\,M^2\,-$$

$$2\,(N-1)^2\,\exp\left(-2\,\beta\left(\Delta_i + \frac{2}{\beta\,N}\right)\right)\,\beta\,N\,M^2$$

$$=\, 1 \,-\, 2\,(N-1)^2\,\exp\left(-2\,\beta\left(\Delta_i + \frac{2}{\beta\,N}\right)\right)\,\beta\,N\,M^2 \,<\, 1\,.$$

Therefore, $f$ is a contraction mapping in $\mathrm{S}_i$. $\qquad\square$

•*Banach Fixed Point Theorem.* Now we have all ingredients to apply Banach fixed point theorem.

**Lemma A7.** *Assume that*

   *A1:*

$$\Delta_i \,\geq\, \frac{2}{\beta\,N} \,+\, \frac{1}{\beta}\,\ln\left(2\,(N-1)\,N\,\beta\,M^2\right)\,, \tag{165}$$

*then $f$ has a fixed point in $\mathrm{S}_i$.*

*Proof.* We use Banach fixed point theorem: Lemma A5 says that $f$ maps from $\mathrm{S}_i$ into $\mathrm{S}_i$. Lemma A6 says that $f$ is a contraction mapping in $\mathrm{S}_i$. $\qquad\square$

*•Contraction mapping with a fixed point.*

We have shown that a fixed point exists. We want to know how fast the iteration converges to the fixed point. Let $\boldsymbol{x}_i^*$ be the fixed point of the iteration $f$ in the sphere $\mathrm{S}_i$. Using the mean value theorem Lemma A32, we have with $\mathrm{J}^m = \int_0^1 \mathrm{J}(\lambda\boldsymbol{\xi} + (1-\lambda)\boldsymbol{x}_i^*)\,\mathrm{d}\lambda$:

$$\|f(\boldsymbol{\xi}) - \boldsymbol{x}_i^*\| = \|f(\boldsymbol{\xi}) - f(\boldsymbol{x}_i^*)\| \leqslant \|\mathrm{J}^m\|_2 \|\boldsymbol{\xi} - \boldsymbol{x}_i^*\| \tag{166}$$

According to Lemma A24, if $p_{\max} = \max_i p_i \geq 1 - \epsilon$ for all $\tilde{\boldsymbol{x}} = \lambda\boldsymbol{\xi} + (1-\lambda)\boldsymbol{x}_i^*$, then the spectral norm of the Jacobian is bounded by

$$\|\mathrm{J}_s(\tilde{\boldsymbol{x}})\|_2 < 2\,\epsilon\,\beta\,. \tag{167}$$

The norm of Jacobian at $\tilde{\boldsymbol{x}}$ is bounded

$$\|\mathrm{J}(\tilde{\boldsymbol{x}})\|_2 \leqslant 2\,\beta\,\|\boldsymbol{X}\|_2^2\,\epsilon \leqslant 2\,\beta\,NM^2\,\epsilon\,. \tag{168}$$

We used that the spectral norm $\|.\|_2$ is bounded by the Frobenius norm $\|.\|_F$ which can be expressed by the norm squared of its column vectors:

$$\|\boldsymbol{X}\|_2 \leqslant \|\boldsymbol{X}\|_F = \sqrt{\sum_i \|\boldsymbol{x}_i\|^2}\,. \tag{169}$$

Therefore

$$\|\boldsymbol{X}\|_2^2 \leqslant N\,M^2\,. \tag{170}$$

The norm of Jacobian of the fixed point iteration is bounded

$$\|\mathrm{J}^m\|_2 \leqslant 2\,\beta\,\|\boldsymbol{X}\|_2^2\,\epsilon \leqslant 2\,\beta\,NM^2\,\epsilon\,. \tag{171}$$

The separation of pattern $\boldsymbol{x}_i$ from data $\boldsymbol{X} = (\boldsymbol{x}_1, \ldots, \boldsymbol{x}_N)$ is

$$\Delta_i = \min_{j, j \neq i}\left(\boldsymbol{x}_i^T\boldsymbol{x}_i - \boldsymbol{x}_i^T\boldsymbol{x}_j\right) = \boldsymbol{x}_i^T\boldsymbol{x}_i - \max_{j, j \neq i}\boldsymbol{x}_i^T\boldsymbol{x}_j\,. \tag{172}$$

We need the separation $\tilde{\Delta}_i$ of $\tilde{\boldsymbol{x}} = \lambda\boldsymbol{\xi} + (1-\lambda)\boldsymbol{x}_i^*$ from the data:

$$\tilde{\Delta}_i = \min_{j, j \neq i}\left(\tilde{\boldsymbol{x}}^T\boldsymbol{x}_i - \tilde{\boldsymbol{x}}^T\boldsymbol{x}_j\right)\,. \tag{173}$$

We compute a lower bound on $\tilde{\Delta}_i$. Using the Cauchy-Schwarz inequality, we obtain for $1 \leqslant j \leqslant N$:

$$\left|\tilde{\boldsymbol{x}}^T\boldsymbol{x}_j - \boldsymbol{x}_i^T\boldsymbol{x}_j\right| \leqslant \|\tilde{\boldsymbol{x}} - \boldsymbol{x}_i\|\,\|\boldsymbol{x}_j\| \leqslant \|\tilde{\boldsymbol{x}} - \boldsymbol{x}_i\|\,M\,. \tag{174}$$

We have the lower bound

$$\tilde{\Delta}_i \geq \min_{j, j \neq i}\left(\left(\boldsymbol{x}_i^T\boldsymbol{x}_i - \|\tilde{\boldsymbol{x}} - \boldsymbol{x}_i\|\,M\right) - \left(\boldsymbol{x}_i^T\boldsymbol{x}_j + \|\tilde{\boldsymbol{x}} - \boldsymbol{x}_i\|\,M\right)\right) \tag{175}$$

$$= -2\,\|\tilde{\boldsymbol{x}} - \boldsymbol{x}_i\|\,M + \min_{j, j \neq i}\left(\boldsymbol{x}_i^T\boldsymbol{x}_i - \boldsymbol{x}_i^T\boldsymbol{x}_j\right) = \Delta_i - 2\,\|\tilde{\boldsymbol{x}} - \boldsymbol{x}_i\|\,M\,.$$

Since

$$\|\tilde{\boldsymbol{x}} - \boldsymbol{x}_i\| = \|\lambda\boldsymbol{\xi} + (1-\lambda)\boldsymbol{x}_i^* - \boldsymbol{x}_i\| \tag{176}$$
$$\leqslant \lambda\,\|\boldsymbol{\xi} - \boldsymbol{x}_i\| + (1-\lambda)\,\|\boldsymbol{x}_i^* - \boldsymbol{x}_i\|$$
$$\leqslant \max\{\|\boldsymbol{\xi} - \boldsymbol{x}_i\|, \|\boldsymbol{x}_i^* - \boldsymbol{x}_i\|\}\,,$$

we have

$$\tilde{\Delta}_i \geq \Delta_i - 2\,\max\{\|\boldsymbol{\xi} - \boldsymbol{x}_i\|, \|\boldsymbol{x}_i^* - \boldsymbol{x}_i\|\}\,M\,. \tag{177}$$

For the softmax component $i$ we have:

$$[\mathrm{softmax}(\beta\,\boldsymbol{X}^T\tilde{\boldsymbol{\xi}})]_i = \frac{1}{1 + \sum_{j \neq i}\exp(\beta\,(\tilde{\boldsymbol{\xi}}^T\boldsymbol{x}_j - \tilde{\boldsymbol{\xi}}^T\boldsymbol{x}_i))} \tag{178}$$

$$\geq \frac{1}{1 + \sum_{j \neq i}\exp(-\beta\,(\Delta_i - 2\,\max\{\|\boldsymbol{\xi} - \boldsymbol{x}_i\|, \|\boldsymbol{x}_i^* - \boldsymbol{x}_i\|\}\,M))}$$

$$= \frac{1}{1 + (N-1)\exp(-\beta\,(\Delta_i - 2\,\max\{\|\boldsymbol{\xi} - \boldsymbol{x}_i\|, \|\boldsymbol{x}_i^* - \boldsymbol{x}_i\|\}\,M))}$$

$$= 1 - \frac{(N-1)\exp(-\beta\,(\Delta_i - 2\,\max\{\|\boldsymbol{\xi} - \boldsymbol{x}_i\|, \|\boldsymbol{x}_i^* - \boldsymbol{x}_i\|\}\,M))}{1 + (N-1)\exp(-\beta\,(\Delta_i - 2\,\max\{\|\boldsymbol{\xi} - \boldsymbol{x}_i\|, \|\boldsymbol{x}_i^* - \boldsymbol{x}_i\|\}\,M))}$$

$$\geq 1 - (N-1)\exp(-\beta\,(\Delta_i - 2\,\max\{\|\boldsymbol{\xi} - \boldsymbol{x}_i\|, \|\boldsymbol{x}_i^* - \boldsymbol{x}_i\|\}\,M))$$

$$= 1 - \epsilon\,.$$

Therefore

$$\epsilon \;=\; (N-1)\exp(-\,\beta\,(\Delta_i \;-\; 2\,\max\{\|\boldsymbol{\xi}\,-\,\boldsymbol{x}_i\|, \|\boldsymbol{x}_i^*\,-\,\boldsymbol{x}_i\|\}\,M))\,. \tag{179}$$

We can bound the spectral norm of the Jacobian, which upper bounds the Lipschitz constant:

$$\|\mathrm{J}^m\|_2 \;\leqslant\; 2\,\beta\,N\,M^2\,(N-1)\exp(-\,\beta\,(\Delta_i \;-\; 2\,\max\{\|\boldsymbol{\xi}\,-\,\boldsymbol{x}_i\|, \|\boldsymbol{x}_i^*\,-\,\boldsymbol{x}_i\|\}\,M))\,. \tag{180}$$

For a contraction mapping we require

$$\|\mathrm{J}^m\|_2 \;<\; 1\,, \tag{181}$$

which can be ensured by

$$2\,\beta\,NM^2\,(N-1)\exp(-\,\beta\,(\Delta_i \;-\; 2\,\max\{\|\boldsymbol{\xi}\,-\,\boldsymbol{x}_i\|, \|\boldsymbol{x}_i^*\,-\,\boldsymbol{x}_i\|\}\,M)) \;<\; 1\,. \tag{182}$$

Solving this inequality for $\Delta_i$ gives

$$\Delta_i \;>\; 2\,\max\{\|\boldsymbol{\xi}\,-\,\boldsymbol{x}_i\|, \|\boldsymbol{x}_i^*\,-\,\boldsymbol{x}_i\|\}\,M \;+\; \frac{1}{\beta}\,\ln\left(2\,(N-1)\,N\,\beta\,M^2\right)\,. \tag{183}$$

In an environment around $\boldsymbol{x}_i^*$ in which Eq. (183) holds, $f$ is a contraction mapping and every point converges under the iteration $f$ to $\boldsymbol{x}_i^*$ when the iteration stays in the environment. After every iteration the mapped point $f(\boldsymbol{\xi})$ is closer to the fixed point $\boldsymbol{x}_i^*$ than the original point $\boldsymbol{x}_i$:

$$\|f(\boldsymbol{\xi}) \,-\, \boldsymbol{x}_i^*\| \;\leqslant\; \|\mathrm{J}^m\|_2\,\|\boldsymbol{\xi}\,-\,\boldsymbol{x}_i^*\| \;<\; \|\boldsymbol{\xi}\,-\,\boldsymbol{x}_i^*\|\,. \tag{184}$$

Using

$$\|f(\boldsymbol{\xi}) \,-\, \boldsymbol{x}_i^*\| \;\leqslant\; \|\mathrm{J}^m\|_2\,\|\boldsymbol{\xi}\,-\,\boldsymbol{x}_i^*\| \;\leqslant\; \|\mathrm{J}^m\|_2\,\|\boldsymbol{\xi}\,-\,f(\boldsymbol{\xi})\| \;+\; \|\mathrm{J}^m\|_2\,\|f(\boldsymbol{\xi})\,-\,\boldsymbol{x}_i^*\|\,, \tag{185}$$

we obtain

$$\|f(\boldsymbol{\xi}) \,-\, \boldsymbol{x}_i^*\| \;\leqslant\; \frac{\|\mathrm{J}^m\|_2}{1\,-\,\|\mathrm{J}^m\|_2}\,\|\boldsymbol{\xi}\,-\,f(\boldsymbol{\xi})\|\,. \tag{186}$$

For large $\Delta_i$ the iteration is close to the fixed point even after one update. This has been confirmed in several experiments.

### A.1.5.4 Metastable States: Fixed Points Near Mean of Similar Patterns. The proof concept is the same as for a single pattern but now for the arithmetic mean of similar patterns.

•*Bound on the Jacobian.*

The Jacobian of the fixed point iteration is

$$\mathrm{J} \;=\; \beta\,\boldsymbol{X}\left(\mathrm{diag}(\boldsymbol{p})-\boldsymbol{p}\boldsymbol{p}^T\right)\boldsymbol{X}^T \;=\; \boldsymbol{X}\mathrm{J}_s\boldsymbol{X}^T\,. \tag{187}$$

If we consider $p_i$ as the probability of selecting the vector $\boldsymbol{x}_i$, then we can define expectations as $\mathrm{E}_{\boldsymbol{p}}[f(\boldsymbol{x})] = \sum_{i=1}^N p_i f(\boldsymbol{x}_i)$. In this setting the matrix

$$\boldsymbol{X}\left(\mathrm{diag}(\boldsymbol{p})-\boldsymbol{p}\boldsymbol{p}^T\right)\boldsymbol{X}^T \tag{188}$$

is the covariance matrix of data $\boldsymbol{X}$ when its vectors are selected according to the probability $\boldsymbol{p}$:

$$\boldsymbol{X}\left(\mathrm{diag}(\boldsymbol{p})\,-\,\boldsymbol{p}\boldsymbol{p}^T\right)\boldsymbol{X}^T \;=\; \boldsymbol{X}\mathrm{diag}(\boldsymbol{p})\boldsymbol{X}^T \;-\; \boldsymbol{X}\boldsymbol{p}\boldsymbol{p}^T\boldsymbol{X}^T \tag{189}$$

$$=\; \sum_{i=1}^N p_i\,\boldsymbol{x}_i\,\boldsymbol{x}_i^T \;-\; \left(\sum_{i=1}^N p_i\,\boldsymbol{x}_i\right)\left(\sum_{i=1}^N p_i\,\boldsymbol{x}_i\right)^T \tag{190}$$

$$=\; \mathrm{E}_{\boldsymbol{p}}[\boldsymbol{x}\,\boldsymbol{x}^T] \;-\; \mathrm{E}_{\boldsymbol{p}}[\boldsymbol{x}]\,\mathrm{E}_{\boldsymbol{p}}[\boldsymbol{x}]^T \;=\; \mathrm{Var}_{\boldsymbol{p}}[\boldsymbol{x}]\,, \tag{191}$$

therefore we have

$$\mathrm{J} \;=\; \beta\,\mathrm{Var}_{\boldsymbol{p}}[\boldsymbol{x}]\,. \tag{192}$$

We now elaborate more on this interpretation as variance. Specifically the singular values of J (or in other words: the covariance) should be reasonably small. The singular values are the key to ensure convergence of the iteration Eq. (57). Next we present some thoughts.

1. It's clear that the largest eigenvalue of the covariance matrix (equal to the largest singular value) is the variance in the direction of the eigenvector associated with the largest eigenvalue.

2. Furthermore the variance goes to zero as one $p_i$ goes to one, since only one pattern is chosen and there is no variance.

3. The variance is reasonable small if all patterns are chosen with equal probability.

4. The variance is small if few similar patterns are chosen with high probability. If the patterns are sufficient similar, then the spectral norm of the covariance matrix is smaller than one.

The first three issues have already been adressed. Now we focus on the last one in greater detail. We assume that the first $l$ patterns are much more probable (and similar to one another) than the other patterns. Therefore, we define:

$$M := \max_i \|\boldsymbol{x}_i\| ,\tag{193}$$

$$\gamma = \sum_{i=l+1}^{N} p_i \leqslant \epsilon ,\tag{194}$$

$$1 - \gamma = \sum_{i=1}^{l} p_i \geq 1 - \epsilon ,\tag{195}$$

$$\tilde{p}_i := \frac{p_i}{1 - \gamma} \leqslant p_i/(1 - \epsilon) ,\tag{196}$$

$$\sum_{i=1}^{l} \tilde{p}_i = 1 ,\tag{197}$$

$$\boldsymbol{m_x} = \frac{1}{l} \sum_{i=1}^{l} \boldsymbol{x}_i ,\tag{198}$$

$$m_{\max} = \max_{1 \leqslant i \leqslant l} \|\boldsymbol{x}_i - \boldsymbol{m_x}\| .\tag{199}$$

$M$ is an upper bound on the Euclidean norm of the patterns, which are vectors. $\epsilon$ is an upper bound on the probability $\gamma$ of not choosing one of the first $l$ patterns, while $1 - \epsilon$ is a lower bound the probability $(1 - \gamma)$ of choosing one of the first $l$ patterns. $\boldsymbol{m_x}$ is the arithmetic mean (the center) of the first $l$ patterns. $m_{\max}$ is the maximal distance of the patterns to the center $\boldsymbol{m_x}$. $\tilde{\boldsymbol{p}}$ is the probability $\boldsymbol{p}$ normalized for the first $l$ patterns.

The variance of the first $l$ patterns is

$$\begin{aligned}
\mathrm{Var}_{\tilde{p}}[\boldsymbol{x}_{1:l}] &= \sum_{i=1}^{l} \tilde{p}_i \, \boldsymbol{x}_i \, \boldsymbol{x}_i^T - \left( \sum_{i=1}^{l} \tilde{p}_i \, \boldsymbol{x}_i \right) \left( \sum_{i=1}^{l} \tilde{p}_i \, \boldsymbol{x}_i \right)^T \\
&= \sum_{i=1}^{l} \tilde{p}_i \left( \boldsymbol{x}_i - \sum_{i=1}^{l} \tilde{p}_i \boldsymbol{x}_i \right) \left( \boldsymbol{x}_i - \sum_{i=1}^{l} \tilde{p}_i \boldsymbol{x}_i \right)^T .
\end{aligned}\tag{200}$$

**Lemma A8.** *With the definitions in Eq. (193) to Eq. (200), the following bounds on the norm $\|\mathrm{J}\|_2$ of the Jacobian of the fixed point iteration hold. The $\gamma$-bound for $\|\mathrm{J}\|_2$ is*

$$\|\mathrm{J}\|_2 \leqslant \beta \left( (1 - \gamma) \, m_{\max}^2 + \gamma \, 2 \, (2 - \gamma) \, M^2 \right)\tag{201}$$

*and the $\epsilon$-bound for $\|\mathrm{J}\|_2$ is:*

$$\|\mathrm{J}\|_2 \leqslant \beta \left( m_{\max}^2 + \epsilon \, 2 \, (2 - \epsilon) \, M^2 \right) .\tag{202}$$

*Proof.* The variance $\text{Var}_{\tilde{p}}[\boldsymbol{x}_{1:l}]$ can be expressed as:

$$(1-\gamma)\,\text{Var}_{\tilde{p}}[\boldsymbol{x}_{1:l}] = \sum_{i=1}^{l} p_i \left(\boldsymbol{x}_i - \frac{1}{1-\gamma}\sum_{i=1}^{l} p_i\,\boldsymbol{x}_i\right)\left(\boldsymbol{x}_i - \frac{1}{1-\gamma}\sum_{i=1}^{l} p_i\,\boldsymbol{x}_i\right)^T \qquad (203)$$

$$= \sum_{i=1}^{l} p_i\,\boldsymbol{x}_i\,\boldsymbol{x}_i^T - \left(\sum_{i=1}^{l} p_i\,\boldsymbol{x}_i\right)\frac{1}{1-\gamma}\left(\sum_{i=1}^{l} p_i\,\boldsymbol{x}_i\right)^T - \frac{1}{1-\gamma}\left(\sum_{i=1}^{l} p_i\,\boldsymbol{x}_i\right)\left(\sum_{i=1}^{l} p_i\,\boldsymbol{x}_i\right)^T$$

$$+ \frac{\sum_{i=1}^{l} p_i}{(1-\gamma)^2}\left(\sum_{i=1}^{l} p_i\,\boldsymbol{x}_i\right)\left(\sum_{i=1}^{l} p_i\,\boldsymbol{x}_i\right)^T = \sum_{i=1}^{l} p_i\,\boldsymbol{x}_i\,\boldsymbol{x}_i^T - \frac{1}{1-\gamma}\left(\sum_{i=1}^{l} p_i\,\boldsymbol{x}_i\right)\left(\sum_{i=1}^{l} p_i\,\boldsymbol{x}_i\right)^T$$

$$= \sum_{i=1}^{l} p_i\,\boldsymbol{x}_i\,\boldsymbol{x}_i^T - \left(\sum_{i=1}^{l} p_i\,\boldsymbol{x}_i\right)\left(\sum_{i=1}^{l} p_i\,\boldsymbol{x}_i\right)^T + \left(1 - \frac{1}{1-\gamma}\right)\left(\sum_{i=1}^{l} p_i\,\boldsymbol{x}_i\right)\left(\sum_{i=1}^{l} p_i\,\boldsymbol{x}_i\right)^T$$

$$= \sum_{i=1}^{l} p_i\,\boldsymbol{x}_i\,\boldsymbol{x}_i^T - \left(\sum_{i=1}^{l} p_i\,\boldsymbol{x}_i\right)\left(\sum_{i=1}^{l} p_i\,\boldsymbol{x}_i\right)^T - \frac{\gamma}{1-\gamma}\left(\sum_{i=1}^{l} p_i\,\boldsymbol{x}_i\right)\left(\sum_{i=1}^{l} p_i\,\boldsymbol{x}_i\right)^T .$$

Therefore, we have

$$\sum_{i=1}^{l} p_i\,\boldsymbol{x}_i\,\boldsymbol{x}_i^T - \left(\sum_{i=1}^{l} p_i\,\boldsymbol{x}_i\right)\left(\sum_{i=1}^{l} p_i\,\boldsymbol{x}_i\right)^T \qquad (204)$$

$$= (1-\gamma)\,\text{Var}_{\tilde{p}}[\boldsymbol{x}_{1:l}] + \frac{\gamma}{1-\gamma}\left(\sum_{i=1}^{l} p_i\,\boldsymbol{x}_i\right)\left(\sum_{i=1}^{l} p_i\,\boldsymbol{x}_i\right)^T .$$

We now can reformulate the Jacobian J:

$$\text{J} = \beta\left(\sum_{i=1}^{l} p_i\,\boldsymbol{x}_i\,\boldsymbol{x}_i^T + \sum_{i=l+1}^{N} p_i\,\boldsymbol{x}_i\,\boldsymbol{x}_i^T \right. \qquad (205)$$

$$\left. - \left(\sum_{i=1}^{l} p_i\,\boldsymbol{x}_i + \sum_{i=l+1}^{N} p_i\,\boldsymbol{x}_i\right)\left(\sum_{i=1}^{l} p_i\,\boldsymbol{x}_i + \sum_{i=l+1}^{N} p_i\,\boldsymbol{x}_i\right)^T\right)$$

$$= \beta\left(\sum_{i=1}^{l} p_i\,\boldsymbol{x}_i\,\boldsymbol{x}_i^T - \left(\sum_{i=1}^{l} p_i\,\boldsymbol{x}_i\right)\left(\sum_{i=1}^{l} p_i\,\boldsymbol{x}_i\right)^T\right.$$

$$+ \sum_{i=l+1}^{N} p_i\,\boldsymbol{x}_i\,\boldsymbol{x}_i^T - \left(\sum_{i=l+1}^{N} p_i\,\boldsymbol{x}_i\right)\left(\sum_{i=l+1}^{N} p_i\,\boldsymbol{x}_i\right)^T$$

$$\left. - \left(\sum_{i=1}^{l} p_i\,\boldsymbol{x}_i\right)\left(\sum_{i=l+1}^{N} p_i\,\boldsymbol{x}_i\right)^T - \left(\sum_{i=l+1}^{N} p_i\,\boldsymbol{x}_i\right)\left(\sum_{i=1}^{l} p_i\,\boldsymbol{x}_i\right)^T\right)$$

$$= \beta\left((1-\gamma)\,\text{Var}_{\tilde{p}}[\boldsymbol{x}_{1:l}] + \frac{\gamma}{1-\gamma}\left(\sum_{i=1}^{l} p_i\,\boldsymbol{x}_i\right)\left(\sum_{i=1}^{l} p_i\,\boldsymbol{x}_i\right)^T\right.$$

$$+ \sum_{i=l+1}^{N} p_i\,\boldsymbol{x}_i\,\boldsymbol{x}_i^T - \left(\sum_{i=l+1}^{N} p_i\,\boldsymbol{x}_i\right)\left(\sum_{i=l+1}^{N} p_i\,\boldsymbol{x}_i\right)^T$$

$$\left. - \left(\sum_{i=1}^{l} p_i\,\boldsymbol{x}_i\right)\left(\sum_{i=l+1}^{N} p_i\,\boldsymbol{x}_i\right)^T - \left(\sum_{i=l+1}^{N} p_i\,\boldsymbol{x}_i\right)\left(\sum_{i=1}^{l} p_i\,\boldsymbol{x}_i\right)^T\right) .$$

The spectral norm of an outer product of two vectors is the product of the Euclidean norms of the vectors:

$$\left\| \boldsymbol{a}\boldsymbol{b}^T \right\|_2 \;=\; \sqrt{\lambda_{\max}(\boldsymbol{b}\boldsymbol{a}^T\boldsymbol{a}\boldsymbol{b}^T)} \;=\; \|\boldsymbol{a}\|\,\sqrt{\lambda_{\max}(\boldsymbol{b}\boldsymbol{b}^T)} \;=\; \|\boldsymbol{a}\|\,\|\boldsymbol{b}\|\,, \tag{206}$$

since $\boldsymbol{b}\boldsymbol{b}^T$ has eigenvector $\boldsymbol{b}/\|\boldsymbol{b}\|$ with eigenvalue $\|\boldsymbol{b}\|^2$ and otherwise zero eigenvalues.

We now bound the norms of some matrices and vectors:

$$\left\| \sum_{i=1}^{l} p_i\,\boldsymbol{x}_i \right\| \;\leqslant\; \sum_{i=1}^{l} p_i\,\|\boldsymbol{x}_i\| \;\leqslant\; (1-\gamma)\,M\,, \tag{207}$$

$$\left\| \sum_{i=l+1}^{N} p_i\,\boldsymbol{x}_i \right\| \;\leqslant\; \sum_{i=l+1}^{N} p_i\,\|\boldsymbol{x}_i\| \;\leqslant\; \gamma\,M\,, \tag{208}$$

$$\left\| \sum_{i=l+1}^{N} p_i\,\boldsymbol{x}_i\,\boldsymbol{x}_i^T \right\|_2 \;\leqslant\; \sum_{i=l+1}^{N} p_i\,\left\| \boldsymbol{x}_i\,\boldsymbol{x}_i^T \right\|_2 \;=\; \sum_{i=l+1}^{N} p_i\,\|\boldsymbol{x}_i\|^2 \;\leqslant\; \sum_{i=l+1}^{N} p_i\,M^2 \;=\; \gamma\,M^2\,. \tag{209}$$

In order to bound the variance of the first $l$ patterns, we compute the vector $\boldsymbol{a}$ that minimizes

$$f(\boldsymbol{a}) \;=\; \sum_{i=1}^{l} p_i\|\boldsymbol{x}_i\,-\,\boldsymbol{a}\|^2 \;=\; \sum_{i=1}^{l} p_i(\boldsymbol{x}_i\,-\,\boldsymbol{a})^T(\boldsymbol{x}_i\,-\,\boldsymbol{a})\,. \tag{210}$$

The solution to

$$\frac{\partial f(\boldsymbol{a})}{\partial \boldsymbol{a}} \;=\; 2\sum_{i=1}^{N} p_i(\boldsymbol{a}\,-\,\boldsymbol{x}_i) \;=\; 0 \tag{211}$$

is

$$\boldsymbol{a} \;=\; \sum_{i=1}^{N} p_i\boldsymbol{x}_i\,. \tag{212}$$

The Hessian of $f$ is positive definite since

$$\frac{\partial^2 f(\boldsymbol{a})}{\partial \boldsymbol{a}^2} \;=\; 2\sum_{i=1}^{N} p_i\,\boldsymbol{I} \;=\; 2\,\boldsymbol{I} \tag{213}$$

and $f$ is a convex function. Hence, the mean

$$\bar{\boldsymbol{x}} \;:=\; \sum_{i=1}^{N} p_i\,\boldsymbol{x}_i \tag{214}$$

minimizes $\sum_{i=1}^{N} p_i\|\boldsymbol{x}_i - \boldsymbol{a}\|^2$. Therefore, we have

$$\sum_{i=1}^{l} p_i\|\boldsymbol{x}_i\,-\,\bar{\boldsymbol{x}}\|^2 \;\leqslant\; \sum_{i=1}^{l} p_i\|\boldsymbol{x}_i\,-\,\boldsymbol{m}_{\boldsymbol{x}}\|^2 \;\leqslant\; (1\,-\,\gamma)\,m_{\max}^2\,. \tag{215}$$

We now bound the variance on the first $l$ patterns:

$$(1-\gamma)\,\left\| \mathrm{Var}_{\tilde{p}}[\boldsymbol{x}_{1:l}] \right\|_2 \;\leqslant\; \sum_{i=1}^{l} p_i\left\| (\boldsymbol{x}_i\,-\,\bar{\boldsymbol{x}})\,(\boldsymbol{x}_i\,-\,\bar{\boldsymbol{x}})^T \right\|_2 \tag{216}$$

$$=\; \sum_{i=1}^{l} p_i\|\boldsymbol{x}_i\,-\,\bar{\boldsymbol{x}}\|^2 \;\leqslant\; \sum_{i=1}^{l} p_i\|\boldsymbol{x}_i\,-\,\boldsymbol{m}_{\boldsymbol{x}}\|^2 \;\leqslant\; (1\,-\,\gamma)\,m_{\max}^2\,.$$

We obtain for the spectral norm of J:

$$
\begin{aligned}
\|\mathrm{J}\|_2 \;\leqslant\; & \beta\Big((1-\gamma)\,\|\mathrm{Var}_{\tilde{p}}[\boldsymbol{x}_{1:l}]\|_2 \\
& + \frac{\gamma}{1-\gamma}\left\|\left(\sum_{i=1}^{l} p_i\,\boldsymbol{x}_i\right)\left(\sum_{i=1}^{l} p_i\,\boldsymbol{x}_i\right)^{T}\right\|_2 \\
& + \left\|\sum_{i=l+1}^{N} p_i\,\boldsymbol{x}_i\,\boldsymbol{x}_i^{T}\right\|_2 + \left\|\left(\sum_{i=l+1}^{N} p_i\,\boldsymbol{x}_i\right)\left(\sum_{i=l+1}^{N} p_i\,\boldsymbol{x}_i\right)^{T}\right\|_2 \\
& + \left\|\left(\sum_{i=1}^{l} p_i\,\boldsymbol{x}_i\right)\left(\sum_{i=l+1}^{N} p_i\,\boldsymbol{x}_i\right)^{T}\right\|_2 + \left\|\left(\sum_{i=l+1}^{N} p_i\,\boldsymbol{x}_i\right)\left(\sum_{i=1}^{l} p_i\,\boldsymbol{x}_i\right)^{T}\right\|_2\Big) \\
\leqslant\; & \beta\big((1-\gamma)\,\|\mathrm{Var}_{\tilde{p}}[\boldsymbol{x}_{1:l}]\|_2 + \gamma\,(1-\gamma)\,M^2 + \gamma\,M^2 + \gamma^2\,M^2 + \\
& \gamma\,(1-\gamma)\,M^2 + \gamma\,(1-\gamma)\,M^2\big) \\
=\; & \beta\big((1-\gamma)\,\|\mathrm{Var}_{\tilde{p}}[\boldsymbol{x}_{1:l}]\|_2 + \gamma\,2\,(2-\gamma)\,M^2\big)\;.
\end{aligned}
\tag{217}
$$

Combining the previous two estimates immediately leads to Eq. (201).

The function $h(x) = x2(2-x)$ has the derivative $h'(x) = 4(1-x)$. Therefore, $h(x)$ is monotone increasing for $x < 1$. For $0 \leqslant \gamma \leqslant \epsilon < 1$, we can immediately deduce that $\gamma 2(2-\gamma) \leqslant \epsilon 2(2-\epsilon)$. Since $\epsilon$ is larger than $\gamma$, we obtain the following $\epsilon$-bound for $\|\mathrm{J}\|_2$:

$$
\|\mathrm{J}\|_2 \;\leqslant\; \beta\left(m_{\max}^2 + \epsilon\,2\,(2-\epsilon)\,M^2\right)\;.
\tag{218}
$$

$\square$

We revisit the bound on $(1-\gamma)\,\mathrm{Var}_{\tilde{p}}[\boldsymbol{x}_{1:l}]$. The trace $\sum_{k=1}^{d} e_k$ is the sum of the eigenvalues $e_k$. The spectral norm is equal to the largest eigenvalue $e_1$, that is, the largest singular value. We obtain:

$$
\begin{aligned}
\|\mathrm{Var}_{\tilde{p}}[\boldsymbol{x}_{1:l}]\|_2 \;=\; & \mathrm{Tr}\left(\sum_{i=1}^{l} p_i\,(\boldsymbol{x}_i - \bar{\boldsymbol{x}})\,(\boldsymbol{x}_i - \bar{\boldsymbol{x}})^{T}\right) - \sum_{k=2}^{d} e_k \\
=\; & \sum_{i=1}^{l} p_i\,\mathrm{Tr}\left((\boldsymbol{x}_i - \bar{\boldsymbol{x}})\,(\boldsymbol{x}_i - \bar{\boldsymbol{x}})^{T}\right) - \sum_{k=2}^{d} e_k \\
=\; & \sum_{i=1}^{l} p_i\,\|\boldsymbol{x}_i - \bar{\boldsymbol{x}}\|^2 - \sum_{k=2}^{d} e_k\;.
\end{aligned}
\tag{219}
$$

Therefore, the tightness of the bound depends on eigenvalues which are not the largest. That is variations which are not along the strongest variation weaken the bound.

•*Proof of a fixed point by Banach Fixed Point Theorem.*

Without restricting the generality, we assume that the first $l$ patterns are much more probable (and similar to one another) than the other patterns. Therefore, we define:

$$M := \max_i \|\boldsymbol{x}_i\| \,, \tag{220}$$

$$\gamma = \sum_{i=l+1}^{N} p_i \leqslant \epsilon \,, \tag{221}$$

$$1 - \gamma = \sum_{i=1}^{l} p_i \geq 1 - \epsilon \,, \tag{222}$$

$$\tilde{p}_i := \frac{p_i}{1 - \gamma} \leqslant p_i/(1 - \epsilon) \,, \tag{223}$$

$$\sum_{i=1}^{l} \tilde{p}_i = 1 \,, \tag{224}$$

$$\boldsymbol{m_x} = \frac{1}{l} \sum_{i=1}^{l} \boldsymbol{x}_i \,, \tag{225}$$

$$m_{\max} = \max_{1 \leqslant i \leqslant l} \|\boldsymbol{x}_i - \boldsymbol{m_x}\| \,. \tag{226}$$

$M$ is an upper bound on the Euclidean norm of the patterns, which are vectors. $\epsilon$ is an upper bound on the probability $\gamma$ of not choosing one of the first $l$ patterns, while $1 - \epsilon$ is a lower bound the probability $(1 - \gamma)$ of choosing one of the first $l$ patterns. $\boldsymbol{m_x}$ is the arithmetic mean (the center) of the first $l$ patterns. $m_{\max}$ is the maximal distance of the patterns to the center $\boldsymbol{m_x}$. $\tilde{\boldsymbol{p}}$ is the probability $\boldsymbol{p}$ normalized for the first $l$ patterns.

•*Mapped vectors stay in a compact environment.* We show that if $\boldsymbol{m_x}$ is sufficient dissimilar to other $\boldsymbol{x}_j$ with $l < j$ then there is an compact environment of $\boldsymbol{m_x}$ (a sphere) where the fixed point iteration maps this environment into itself. The idea of the proof is to define a sphere around $\boldsymbol{m_x}$ for which the points from the sphere are mapped by $f$ into the sphere.

We first need following lemma which bounds the distance $\|\boldsymbol{m_x} - f(\boldsymbol{\xi})\|$ of a $\boldsymbol{\xi}$ which is close to $\boldsymbol{m_x}$.

**Lemma A9.** *For a query $\boldsymbol{\xi}$ and data $\boldsymbol{X} = (\boldsymbol{x}_1, \ldots, \boldsymbol{x}_N)$, we define*

$$0 \leqslant c = \min_{j, l < j} \left( \boldsymbol{\xi}^T \boldsymbol{m_x} - \boldsymbol{\xi}^T \boldsymbol{x}_j \right) = \boldsymbol{\xi}^T \boldsymbol{m_x} - \max_{j, l < j} \boldsymbol{\xi}^T \boldsymbol{x}_j \,. \tag{227}$$

*The following holds:*

$$\|\boldsymbol{m_x} - f(\boldsymbol{\xi})\| \leqslant m_{\max} + 2\gamma M \leqslant m_{\max} + 2\epsilon M \,, \tag{228}$$

*where*

$$M = \max_i \|\boldsymbol{x}_i\| \,, \tag{229}$$

$$\epsilon = (N - l) \exp(-\beta c) \,. \tag{230}$$

*Proof.* Let $s = \arg\max_{j, j \leqslant l} \boldsymbol{\xi}^T \boldsymbol{x}_j$, therefore $\boldsymbol{\xi}^T \boldsymbol{m_x} = \frac{1}{l} \sum_{i=1}^{l} \boldsymbol{\xi}^T \boldsymbol{x}_i \leqslant \frac{1}{l} \sum_{i=1}^{l} \boldsymbol{\xi}^T \boldsymbol{x}_s = \boldsymbol{\xi}^T \boldsymbol{x}_s$. For softmax components $j$ with $l < j$ we have

$$[\text{softmax}(\beta \boldsymbol{X}^T \boldsymbol{\xi})]_j = \frac{\exp(\beta (\boldsymbol{\xi}^T \boldsymbol{x}_j - \boldsymbol{\xi}^T \boldsymbol{x}_s))}{1 + \sum_{k, k \neq s} \exp(\beta (\boldsymbol{\xi}^T \boldsymbol{x}_k - \boldsymbol{\xi}^T \boldsymbol{x}_s))} \leqslant \exp(-\beta c) = \frac{\epsilon}{N - l} \,, \tag{231}$$

since $\boldsymbol{\xi}^T \boldsymbol{x}_s - \boldsymbol{\xi}^T \boldsymbol{x}_j \geq \boldsymbol{\xi}^T \boldsymbol{m_x} - \boldsymbol{\xi}^T \boldsymbol{x}_j$ for each $j$ with $l < j$, therefore $\boldsymbol{\xi}^T \boldsymbol{x}_s - \boldsymbol{\xi}^T \boldsymbol{x}_j \geq c$

The iteration $f$ can be written as

$$f(\boldsymbol{\xi}) = \boldsymbol{X} \text{softmax}(\beta \boldsymbol{X}^T \boldsymbol{\xi}) = \sum_{j=1}^{N} \boldsymbol{x}_j [\text{softmax}(\beta \boldsymbol{X}^T \boldsymbol{\xi})]_j \,. \tag{232}$$

We set $p_i = [\text{softmax}(\beta \boldsymbol{X}^T \boldsymbol{\xi})]_i$, therefore $\sum_{i=1}^{l} p_i = 1 - \gamma \geq 1 - \epsilon$ and $\sum_{i=l+1}^{N} p_i = \gamma \leqslant \epsilon$.
Therefore

$$
\left\| \boldsymbol{m_x} - \sum_{j=1}^{l} \frac{p_j}{1-\gamma} \, \boldsymbol{x}_j \right\|^2 = \left\| \sum_{j=1}^{l} \frac{p_j}{1-\gamma} \, (\boldsymbol{m_x} - \boldsymbol{x}_j) \right\|^2 \tag{233}
$$

$$
= \sum_{j=1,k=1}^{l} \frac{p_j}{1-\gamma} \frac{p_k}{1-\gamma} \, (\boldsymbol{m_x} - \boldsymbol{x}_j)^T (\boldsymbol{m_x} - \boldsymbol{x}_k)
$$

$$
= \frac{1}{2} \sum_{j=1,k=1}^{l} \frac{p_j}{1-\gamma} \frac{p_k}{1-\gamma} \left( \| \boldsymbol{m_x} - \boldsymbol{x}_j \|^2 + \| \boldsymbol{m_x} - \boldsymbol{x}_k \|^2 - \| \boldsymbol{x}_j - \boldsymbol{x}_k \|^2 \right)
$$

$$
= \sum_{j=1}^{l} \frac{p_j}{1-\gamma} \| \boldsymbol{m_x} - \boldsymbol{x}_j \|^2 - \frac{1}{2} \sum_{j=1,k=1}^{l} \frac{p_j}{1-\gamma} \frac{p_k}{1-\gamma} \| \boldsymbol{x}_j - \boldsymbol{x}_k \|^2
$$

$$
\leqslant \sum_{j=1}^{l} \frac{p_j}{1-\gamma} \| \boldsymbol{m_x} - \boldsymbol{x}_j \|^2 \leqslant m_{\max}^2 \, .
$$

It follows that

$$
\left\| \boldsymbol{m_x} - \sum_{j=1}^{l} \frac{p_j}{1-\gamma} \, \boldsymbol{x}_j \right\| \leqslant m_{\max} \tag{234}
$$

We now can bound $\| \boldsymbol{m_x} - f(\boldsymbol{\xi}) \|$:

$$
\| \boldsymbol{m_x} - f(\boldsymbol{\xi}) \| = \left\| \boldsymbol{m_x} - \sum_{j=1}^{N} p_j \, \boldsymbol{x}_j \right\| \tag{235}
$$

$$
= \left\| \boldsymbol{m_x} - \sum_{j=1}^{l} p_j \, \boldsymbol{x}_j - \sum_{j=l+1}^{N} p_j \, \boldsymbol{x}_j \right\|
$$

$$
= \left\| \boldsymbol{m_x} - \sum_{j=1}^{l} \frac{p_j}{1-\gamma} \, \boldsymbol{x}_j + \frac{\gamma}{1-\gamma} \sum_{j=1}^{l} p_j \, \boldsymbol{x}_j - \sum_{j=l+1}^{N} p_j \, \boldsymbol{x}_j \right\|
$$

$$
\leqslant \left\| \boldsymbol{m_x} - \sum_{j=1}^{l} \frac{p_j}{1-\gamma} \, \boldsymbol{x}_j \right\| + \frac{\gamma}{1-\gamma} \left\| \sum_{j=1}^{l} p_j \, \boldsymbol{x}_j \right\| + \left\| \sum_{j=l+1}^{N} p_j \, \boldsymbol{x}_j \right\|
$$

$$
\leqslant \left\| \boldsymbol{m_x} - \sum_{j=1}^{l} \frac{p_j}{1-\gamma} \, \boldsymbol{x}_j \right\| + \frac{\gamma}{1-\gamma} \sum_{j=1}^{l} p_j \, M + \sum_{j=l+1}^{N} p_j \, M
$$

$$
\leqslant \left\| \boldsymbol{m_x} - \sum_{j=1}^{l} \frac{p_j}{1-\gamma} \, \boldsymbol{x}_j \right\| + 2 \, \gamma \, M
$$

$$
\leqslant m_{\max} + 2 \, \gamma \, M \leqslant m_{\max} + 2 \, \epsilon \, M \, ,
$$

where we applied Eq. (233) in the penultimate inequality. This is the statement of the lemma. $\qquad\square$

The separation of the center (the arithmetic mean) $\boldsymbol{m_x}$ of the first $l$ from data $\boldsymbol{X} = (\boldsymbol{x}_{l+1}, \ldots, \boldsymbol{x}_N)$ is $\Delta_m$, defined as

$$
\Delta_m = \min_{j, l<j} \left( \boldsymbol{m_x}^T \boldsymbol{m_x} - \boldsymbol{m_x}^T \boldsymbol{x}_j \right) = \boldsymbol{m_x}^T \boldsymbol{m_x} - \max_{j, l<j} \boldsymbol{m_x}^T \boldsymbol{x}_j \, . \tag{236}
$$

The center is separated from the other data $\boldsymbol{x}_j$ with $l < j$ if $0 < \Delta_m$. By the same arguments as in Eq. (140), $\Delta_m$ can also be expressed as

$$\Delta_m = \min_{j,l<j} \frac{1}{2} \left( \|\boldsymbol{m_x}\|^2 - \|\boldsymbol{x}_j\|^2 + \|\boldsymbol{m_x} - \boldsymbol{x}_j\|^2 \right) \tag{237}$$

$$= \frac{1}{2} \|\boldsymbol{m_x}\|^2 - \frac{1}{2} \max_{j,l<j} \left( \|\boldsymbol{x}_j\|^2 - \|\boldsymbol{m_x} - \boldsymbol{x}_j\|^2 \right) .$$

For $\|\boldsymbol{m_x}\| = \|\boldsymbol{x}_j\|$ we have $\Delta_m = 1/2 \min_{j,l<j} \|\boldsymbol{m_x} - \boldsymbol{x}_j\|^2$.

Next we define the sphere where we want to apply Banach fixed point theorem.

**Definition 4** (Sphere $\mathrm{S}_m$). *The sphere $\mathrm{S}_m$ is defined as*

$$\mathrm{S}_m := \left\{ \boldsymbol{\xi} \mid \|\boldsymbol{\xi} - \boldsymbol{m_x}\| \leqslant \frac{1}{\beta \, m_{\max}} \right\} . \tag{238}$$

**Lemma A10.** *With $\boldsymbol{\xi}$ given, if the assumptions*

   *A1: $\boldsymbol{\xi}$ is inside sphere: $\boldsymbol{\xi} \in \mathrm{S}_m$,*

   *A2: the center $\boldsymbol{m_x}$ is well separated from other data $\boldsymbol{x}_j$ with $l < j$:*

$$\Delta_m \geq \frac{2 \, M}{\beta \, m_{\max}} - \frac{1}{\beta} \, \ln \left( \frac{1 - \beta \, m_{\max}^2}{2 \, \beta \, (N - l) \, M \, \max\{m_{\max} \,, \, 2 \, M\}} \right) , \tag{239}$$

   *A3: the distance $m_{\max}$ of similar patterns to the center is sufficient small:*

$$\beta \, m_{\max}^2 \leqslant 1 \tag{240}$$

*hold, then $f(\boldsymbol{\xi}) \in \mathrm{S}_m$. Therefore, under conditions (A2) and (A3), $f$ is a mapping from $\mathrm{S}_m$ into $\mathrm{S}_m$.*

*Proof.* We need the separation $\tilde{\Delta}_m$ of $\boldsymbol{\xi}$ from the rest of the data, which is the last $N - l$ data points $\boldsymbol{X} = (\boldsymbol{x}_{l+1}, \ldots, \boldsymbol{x}_N)$.

$$\tilde{\Delta}_m = \min_{j,l<j} \left( \boldsymbol{\xi}^T \boldsymbol{m_x} - \boldsymbol{\xi}^T \boldsymbol{x}_j \right) . \tag{241}$$

Using the Cauchy-Schwarz inequality, we obtain for $l + 1 \leqslant j \leqslant N$:

$$\left| \boldsymbol{\xi}^T \boldsymbol{x}_j - \boldsymbol{m_x}^T \boldsymbol{x}_j \right| \leqslant \|\boldsymbol{\xi} - \boldsymbol{m_x}\| \, \|\boldsymbol{x}_j\| \leqslant \|\boldsymbol{\xi} - \boldsymbol{m_x}\| \, M . \tag{242}$$

We have the lower bound

$$\tilde{\Delta}_m \geq \min_{j,l<j} \left( \left( \boldsymbol{m_x}^T \boldsymbol{m_x} - \|\boldsymbol{\xi} - \boldsymbol{m_x}\| \, M \right) - \left( \boldsymbol{m_x}^T \boldsymbol{x}_j + \|\boldsymbol{\xi} - \boldsymbol{m_x}\| \, M \right) \right) \tag{243}$$

$$= - 2 \, \|\boldsymbol{\xi} - \boldsymbol{m_x}\| \, M + \min_{j,l<j} \left( \boldsymbol{m_x}^T \boldsymbol{m_x} - \boldsymbol{m_x}^T \boldsymbol{x}_j \right) = \Delta_m - 2 \, \|\boldsymbol{\xi} - \boldsymbol{m_x}\| \, M$$

$$\geq \Delta_m - 2 \, \frac{M}{\beta \, m_{\max}} ,$$

where we used the assumption (A1) of the lemma.

From the proof in Lemma A9 we have

$$\sum_{i=1}^{l} p_i \geq 1 - (N - l) \, \exp(- \beta \, \tilde{\Delta}_m) = 1 - \tilde{\epsilon} , \tag{244}$$

$$\sum_{i=l+1}^{N} p_i \leqslant (N - l) \, \exp(- \beta \, \tilde{\Delta}_m) = \tilde{\epsilon} . \tag{245}$$

Lemma A9 states that

$$\|\boldsymbol{m_x} - f(\boldsymbol{\xi})\| \leqslant m_{\max} + 2 \, \tilde{\epsilon} \, M \tag{246}$$

$$\leqslant m_{\max} + 2 \, (N - l) \, \exp(- \beta \, \tilde{\Delta}_m) \, M .$$

$$\leqslant m_{\max} + 2 \, (N - l) \, \exp(- \beta \, (\Delta_m - 2 \, \frac{M}{\beta \, m_{\max}})) \, M .$$

Therefore, we have

$$\|\boldsymbol{m_x} - f(\boldsymbol{\xi})\| \leqslant m_{\max} + 2(N-l) \exp\left(-\beta\left(\Delta_m - 2\frac{M}{\beta\, m_{\max}}\right)\right) M \qquad (247)$$

$$\leqslant m_{\max} + 2(N-l) \exp\left(-\beta\left(\frac{2M}{\beta\, m_{\max}} - \right.\right.$$

$$\frac{1}{\beta}\ln\left(\frac{1 - \beta\, m_{\max}^2}{2\beta\,(N-l)\,M\,\max\{m_{\max}\,,\,2\,M\}}\right) - 2\frac{M}{\beta\, m_{\max}}\left.\left.\right)\right) M$$

$$= m_{\max} + 2(N-l)\frac{1 - \beta\, m_{\max}^2}{2\beta\,(N-l)\,M\,\max\{m_{\max}\,,\,2\,M\}} M$$

$$\leqslant m_{\max} + \frac{1 - \beta\, m_{\max}^2}{\beta\, m_{\max}} = \frac{1}{\beta\, m_{\max}}\,,$$

where we used assumption (A2) of the lemma. Therefore, $f(\boldsymbol{\xi})$ is a mapping from the sphere $\mathrm{S}_m$ into the sphere $\mathrm{S}_m$.

$$m_{\max} = \max_{1\leqslant i\leqslant l} \|\boldsymbol{x}_i - \boldsymbol{m_x}\| \qquad (248)$$

$$= \max_{1\leqslant i\leqslant l} \left\|\boldsymbol{x}_i - 1/l\sum_{j=1}^{l}\boldsymbol{x}_j\right\| \qquad (249)$$

$$= \max_{1\leqslant i\leqslant l} \left\|1/l\sum_{j=1}^{l}(\boldsymbol{x}_i - \boldsymbol{x}_j)\right\| \qquad (250)$$

$$\leqslant \max_{1\leqslant i,j\leqslant l} \|\boldsymbol{x}_i - \boldsymbol{x}_j\| \qquad (251)$$

$$\leqslant \max_{1\leqslant i\leqslant l} \|\boldsymbol{x}_i\| + \max_{1\leqslant j\leqslant l} \|\boldsymbol{x}_i\| \qquad (252)$$

$$\leqslant 2M \qquad (253)$$

$$\square$$

•*Contraction mapping.*

For applying Banach fixed point theorem we need to show that $f$ is contraction in the compact environment $\mathrm{S}_m$.

**Lemma A11.** *Assume that*

*A1:*

$$\Delta_m \geqslant \frac{2M}{\beta\, m_{\max}} - \frac{1}{\beta}\ln\left(\frac{1 - \beta\, m_{\max}^2}{2\beta\,(N-l)\,M\,\max\{m_{\max}\,,\,2\,M\}}\right)\,, \qquad (254)$$

*and*

*A2:*

$$\beta\, m_{\max}^2 \leqslant 1\,, \qquad (255)$$

*then $f$ is a contraction mapping in $\mathrm{S}_m$.*

*Proof.* The version of the mean value theorem Lemma A32 states for the symmetric $\mathrm{J}^m = \int_0^1 \mathrm{J}(\lambda\boldsymbol{\xi} + (1-\lambda)\boldsymbol{m_x})\,\mathrm{d}\lambda$:

$$f(\boldsymbol{\xi}) = f(\boldsymbol{m_x}) + \mathrm{J}^m\,(\boldsymbol{\xi} - \boldsymbol{m_x})\,. \qquad (256)$$

In complete analogy to Lemma A6, we get:

$$\|f(\boldsymbol{\xi}) - f(\boldsymbol{m_x})\| \leqslant \|\mathrm{J}^m\|_2\,\|\boldsymbol{\xi} - \boldsymbol{m_x}\|\,. \qquad (257)$$

We define $\tilde{\boldsymbol{\xi}} = \lambda \boldsymbol{\xi} + (1 - \lambda) \boldsymbol{m_x}$ for some $\lambda \in [0, 1]$. We need the separation $\tilde{\Delta}_m$ of $\tilde{\boldsymbol{\xi}}$ from the rest of the data, which is the last $N - l$ data points $\boldsymbol{X} = (\boldsymbol{x}_{l+1}, \dots, \boldsymbol{x}_N)$.

$$\tilde{\Delta}_m = \min_{j, l < j} \left( \tilde{\boldsymbol{\xi}}^T \boldsymbol{m_x} - \tilde{\boldsymbol{\xi}}^T \boldsymbol{x}_j \right) . \tag{258}$$

From the proof in Lemma A9 we have

$$\tilde{\epsilon} = (N - l) \exp(- \beta \tilde{\Delta}_m) , \tag{259}$$

$$\sum_{i=1}^{l} p_i(\tilde{\boldsymbol{\xi}}) \geq 1 - (N - l) \exp(- \beta \tilde{\Delta}_m) = 1 - \tilde{\epsilon} , \tag{260}$$

$$\sum_{i=l+1}^{N} p_i(\tilde{\boldsymbol{\xi}}) \leqslant (N - l) \exp(- \beta \tilde{\Delta}_m) = \tilde{\epsilon} . \tag{261}$$

We first compute an upper bound on $\tilde{\epsilon}$. Using the Cauchy-Schwarz inequality, we obtain for $l + 1 \leqslant j \leqslant N$:

$$\left| \tilde{\boldsymbol{\xi}}^T \boldsymbol{x}_j - \boldsymbol{m_x}^T \boldsymbol{x}_j \right| \leqslant \left\| \tilde{\boldsymbol{\xi}} - \boldsymbol{m_x} \right\| \|\boldsymbol{x}_j\| \leqslant \left\| \tilde{\boldsymbol{\xi}} - \boldsymbol{m_x} \right\| M . \tag{262}$$

We have the lower bound on $\tilde{\Delta}_m$:

$$\tilde{\Delta}_m \geq \min_{j, l < j} \left( \left( \boldsymbol{m_x}^T \boldsymbol{m_x} - \left\| \tilde{\boldsymbol{\xi}} - \boldsymbol{m_x} \right\| M \right) - \left( \boldsymbol{m_x}^T \boldsymbol{x}_j + \left\| \tilde{\boldsymbol{\xi}} - \boldsymbol{m_x} \right\| M \right) \right) \tag{263}$$

$$= - 2 \left\| \tilde{\boldsymbol{\xi}} - \boldsymbol{m_x} \right\| M + \min_{j, l < j} \left( \boldsymbol{m_x}^T \boldsymbol{m_x} - \boldsymbol{m_x}^T \boldsymbol{x}_j \right) = \Delta_m - 2 \left\| \tilde{\boldsymbol{\xi}} - \boldsymbol{m_x} \right\| M$$

$$\geq \Delta_m - 2 \left\| \boldsymbol{\xi} - \boldsymbol{m_x} \right\| M .$$

where we used $\left\| \tilde{\boldsymbol{\xi}} - \boldsymbol{m_x} \right\| = \lambda \| \boldsymbol{\xi} - \boldsymbol{m_x} \| \leqslant \| \boldsymbol{\xi} - \boldsymbol{m_x} \|$. We obtain the upper bound on $\tilde{\epsilon}$:

$$\tilde{\epsilon} \leqslant (N - l) \exp\left( - \beta \left( \Delta_m - 2 \left\| \boldsymbol{\xi} - \boldsymbol{m_x} \right\| M \right) \right) \tag{264}$$

$$\leqslant (N - l) \exp\left( - \beta \left( \Delta_m - \frac{2 M}{\beta m_{\max}} \right) \right) .$$

where we used that in the sphere $S_i$ holds:

$$\| \boldsymbol{\xi} - \boldsymbol{m_x} \| \leqslant \frac{1}{\beta m_{\max}} , \tag{265}$$

therefore

$$2 \| \boldsymbol{\xi} - \boldsymbol{m_x} \| M \leqslant \frac{2 M}{\beta m_{\max}} . \tag{266}$$

Next we compute a lower bound on $\tilde{\epsilon}$ and to this end start with the upper bound on $\tilde{\Delta}_m$ using the same arguments as in Eq. (158) in combination with Eq. (266).

$$\tilde{\Delta}_m \geq \min_{j, l < j} \left( \left( \boldsymbol{m_x}^T \boldsymbol{m_x} + \left\| \tilde{\boldsymbol{\xi}} - \boldsymbol{m_x} \right\| M \right) - \left( \boldsymbol{m_x}^T \boldsymbol{x}_j - \left\| \tilde{\boldsymbol{\xi}} - \boldsymbol{m_x} \right\| M \right) \right) \tag{267}$$

$$= 2 \left\| \tilde{\boldsymbol{\xi}} - \boldsymbol{m_x} \right\| M + \min_{j, l < j} \left( \boldsymbol{m_x}^T \boldsymbol{m_x} - \boldsymbol{m_x}^T \boldsymbol{x}_j \right) = \Delta_m + 2 \left\| \tilde{\boldsymbol{\xi}} - \boldsymbol{m_x} \right\| M$$

$$\geq \Delta_m + 2 \| \boldsymbol{\xi} - \boldsymbol{m_x} \| M .$$

where we used $\left\| \tilde{\boldsymbol{\xi}} - \boldsymbol{m_x} \right\| = \lambda \| \boldsymbol{\xi} - \boldsymbol{m_x} \| \leqslant \| \boldsymbol{\xi} - \boldsymbol{m_x} \|$. We obtain the lower bound on $\tilde{\epsilon}$:

$$\tilde{\epsilon} \geq (N - l) \exp\left( - \beta \left( \Delta_m + \frac{2 M}{\beta m_{\max}} \right) \right) , \tag{268}$$

where we used that in the sphere $S_i$ holds:

$$\| \boldsymbol{\xi} - \boldsymbol{m_x} \| \leqslant \frac{1}{\beta m_{\max}} , \tag{269}$$

therefore

$$2 \left\| \boldsymbol{\xi} \, - \, \boldsymbol{m_x} \right\| M \; \leqslant \; \frac{2 \, M}{\beta \, m_{\max}} \; . \tag{270}$$

From Lemma A8 we have

$$\left\| \mathrm{J}(\tilde{\boldsymbol{\xi}}) \right\|_2 \; \leqslant \; \beta \left( \, m_{\max}^2 \, + \, \tilde{\epsilon} \, 2 \, (2 \, - \, \tilde{\epsilon}) \, M^2 \right) \tag{271}$$

$$= \; \beta \left( m_{\max}^2 \, + \, \tilde{\epsilon} 4 \, M^2 \, - \, 2 \, \tilde{\epsilon}^2 \, M^2 \right)$$

$$\leqslant \; \beta \left( m_{\max}^2 \, + \, (N - l) \, \exp \left( - \, \beta \, \left( \Delta_m \, - \, \frac{2 \, M}{\beta \, m_{\max}} \right) \right) 4 \, M^2 \, - \right.$$

$$\left. 2 \, (N - l)^2 \, \exp \left( - \, 2 \, \beta \, \left( \Delta_m \, + \, \frac{2 \, M}{\beta \, m_{\max}} \right) \right) \, M^2 \right) \; .$$

The bound Eq. (271) holds for the mean $\mathrm{J}^m$, too, since it averages over $\mathrm{J}(\tilde{\boldsymbol{\xi}})$:

$$\left\| \mathrm{J}^m \right\|_2 \; \leqslant \; \beta \left( m_{\max}^2 \, + \, (N - l) \, \exp \left( - \, \beta \, \left( \Delta_m \, - \, \frac{2 \, M}{\beta \, m_{\max}} \right) \right) 4 \, M^2 \, - \tag{272}$$

$$2 \, (N - l)^2 \, \exp \left( - \, 2 \, \beta \, \left( \Delta_m \, + \, \frac{2 \, M}{\beta \, m_{\max}} \right) \right) \, M^2 \right) \; .$$

The assumption of the lemma is

$$\Delta_m \; \geq \; \frac{2 \, M}{\beta \, m_{\max}} \, - \, \frac{1}{\beta} \, \ln \left( \frac{1 \, - \, \beta \, m_{\max}^2}{2 \, \beta \, (N - l) \, M \, \max\{m_{\max} \, , \, 2 \, M\}} \right) \; , \tag{273}$$

Therefore, we have

$$\Delta_m \, - \, \frac{2 \, M}{\beta \, m_{\max}} \; \geq \; - \, \frac{1}{\beta} \, \ln \left( \frac{1 \, - \, \beta \, m_{\max}^2}{2 \, \beta \, (N - l) \, M \, \max\{m_{\max} \, , \, 2 \, M\}} \right) \; . \tag{274}$$

Therefore, the spectral norm $\left\| \mathrm{J}^m \right\|_2$ can be bounded by:

$$\left\| \mathrm{J}^m \right\|_2 \; \leqslant \tag{275}$$

$$\beta \left( m_{\max}^2 \, + \, (N - l) \, \exp \left( - \, \beta \, \left( - \, \frac{1}{\beta} \, \ln \left( \frac{1 \, - \, \beta \, m_{\max}^2}{2 \, \beta \, (N - l) \, M \, \max\{m_{\max} \, , \, 2 \, M\}} \right) \right) \right) \right)$$

$$4 \, M^2 \, - \, 2 \, (N - l)^2 \, \exp \left( - \, 2 \, \beta \, \left( \Delta_m \, + \, \frac{2 \, M}{\beta \, m_{\max}} \right) \right) \, M^2 \right)$$

$$= \; \beta \left( m_{\max}^2 \, + \, (N - l) \, \exp \left( \ln \left( \frac{1 \, - \, \beta \, m_{\max}^2}{2 \, \beta \, (N - l) \, M \, \max\{m_{\max} \, , \, 2 \, M\}} \right) \right) \right)$$

$$4 \, M^2 \, - \, 2 \, (N - l)^2 \, \exp \left( - \, 2 \, \beta \, \left( \Delta_m \, + \, \frac{2 \, M}{\beta \, m_{\max}} \right) \right) \, M^2 \right)$$

$$= \; \beta \left( m_{\max}^2 \, + \, (N - l) \, \frac{1 \, - \, \beta \, m_{\max}^2}{2 \, \beta \, (N - l) \, M \, \max\{m_{\max} \, , \, 2 \, M\}} \, 4 \, M^2 \, - \right.$$

$$\left. 2 \, (N - l)^2 \, \exp \left( - \, 2 \, \beta \, \left( \Delta_m \, + \, \frac{2 \, M}{\beta \, m_{\max}} \right) \right) \, M^2 \right)$$

$$= \; \beta m_{\max}^2 \, + \, \frac{1 \, - \, \beta \, m_{\max}^2}{\max\{m_{\max} \, , \, 2 \, M\}} \, 2 \, M \, -$$

$$\beta \, 2 \, (N - l)^2 \, \exp \left( - \, 2 \, \beta \, \left( \Delta_m \, + \, \frac{2 \, M}{\beta \, m_{\max}} \right) \right) \, M^2$$

$$\leqslant \; \beta m_{\max}^2 \, + \, 1 \, - \, \beta \, m_{\max}^2 \, - \, \beta \, 2 \, (N - l)^2 \, \exp \left( - \, 2 \, \beta \, \left( \Delta_m \, + \, \frac{2 \, M}{\beta \, m_{\max}} \right) \right) \, M^2$$

$$= \; 1 \, - \, \beta \, 2 \, (N - l)^2 \, \exp \left( - \, 2 \, \beta \, \left( \Delta_m \, + \, \frac{2 \, M}{\beta \, m_{\max}} \right) \right) \, M^2 \; < \, 1 \; .$$

For the last but one inequality we used $2M \leqslant \max\{m_{\max}, 2M\}$.

Therefore, $f$ is a contraction mapping in $\mathrm{S}_m$. □

•*Banach Fixed Point Theorem.* Now we have all ingredients to apply Banach fixed point theorem.

**Lemma A12.** *Assume that*

*A1:*

$$\Delta_m \; \geq \; \frac{2\,M}{\beta\,m_{\max}} \; - \; \frac{1}{\beta}\,\ln\left(\frac{1\;-\;\beta\,m_{\max}^2}{2\,\beta\,(N-l)\,M\;\max\{m_{\max}\,,\;2\,M\}}\right)\,, \tag{276}$$

*and*

*A2:*

$$\beta\,m_{\max}^2 \; \leqslant \; 1\,, \tag{277}$$

*then $f$ has a fixed point in $\mathrm{S}_m$.*

*Proof.* We use Banach fixed point theorem: Lemma A10 says that $f$ maps from the compact set $\mathrm{S}_m$ into the same compact set $\mathrm{S}_m$. Lemma A11 says that $f$ is a contraction mapping in $\mathrm{S}_m$. □

•*Contraction mapping with a fixed point.*

We assume that the first $l$ patterns are much more probable (and similar to one another) than the other patterns. Therefore, we define:

$$M \; := \; \max_i \|\boldsymbol{x}_i\|\,, \tag{278}$$

$$\gamma \; = \; \sum_{i=l+1}^{N} p_i \; \leqslant \; \epsilon\,, \tag{279}$$

$$1 - \gamma \; = \; \sum_{i=1}^{l} p_i \; \geq \; 1 \; - \; \epsilon\,, \tag{280}$$

$$\tilde{p}_i \; := \; \frac{p_i}{1-\gamma} \; \leqslant \; p_i/(1-\epsilon)\,, \tag{281}$$

$$\sum_{i=1}^{l} \tilde{p}_i \; = \; 1\,, \tag{282}$$

$$\boldsymbol{m_x} \; = \; \frac{1}{l}\,\sum_{i=1}^{l} \boldsymbol{x}_i\,, \tag{283}$$

$$m_{\max} \; = \; \max_{1 \leqslant i \leqslant l} \|\boldsymbol{x}_i \; - \; \boldsymbol{m_x}\|\,. \tag{284}$$

$M$ is an upper bound on the Euclidean norm of the patterns, which are vectors. $\epsilon$ is an upper bound on the probability $\gamma$ of not choosing one of the first $l$ patterns, while $1 - \epsilon$ is a lower bound the probability $(1 - \gamma)$ of choosing one of the first $l$ patterns. $\boldsymbol{m_x}$ is the arithmetic mean (the center) of the first $l$ patterns. $m_{\max}$ is the maximal distance of the patterns to the center $\boldsymbol{m_x}$. $\tilde{\boldsymbol{p}}$ is the probability $\boldsymbol{p}$ normalized for the first $l$ patterns.

The variance of the first $l$ patterns is

$$\begin{aligned} \mathrm{Var}_{\tilde{p}}[\boldsymbol{x}_{1:l}] \; &= \; \sum_{i=1}^{l} \tilde{p}_i\,\boldsymbol{x}_i\,\boldsymbol{x}_i^T \; - \; \left(\sum_{i=1}^{l} \tilde{p}_i\,\boldsymbol{x}_i\right)\left(\sum_{i=1}^{l} \tilde{p}_i\,\boldsymbol{x}_i\right)^T \\ &= \; \sum_{i=1}^{l} \tilde{p}_i\,\left(\boldsymbol{x}_i \; - \; \sum_{i=1}^{l} \tilde{p}_i\boldsymbol{x}_i\right)\left(\boldsymbol{x}_i \; - \; \sum_{i=1}^{l} \tilde{p}_i\boldsymbol{x}_i\right)^T\,. \end{aligned} \tag{285}$$

We have shown that a fixed point exists. We want to know how fast the iteration converges to the fixed point. Let $\boldsymbol{m}_{\boldsymbol{x}}^*$ be the fixed point of the iteration $f$ in the sphere $S_m$. Using the mean value theorem Lemma A32, we have with $J^m = \int_0^1 J(\lambda\boldsymbol{\xi} + (1-\lambda)\boldsymbol{m}_{\boldsymbol{x}}^*)\,\mathrm{d}\lambda$:

$$\|f(\boldsymbol{\xi}) \, - \, \boldsymbol{m}_{\boldsymbol{x}}^*\| \, = \, \|f(\boldsymbol{\xi}) \, - \, f(\boldsymbol{m}_{\boldsymbol{x}}^*)\| \, \leqslant \, \|J^m\|_2 \, \|\boldsymbol{\xi} \, - \, \boldsymbol{m}_{\boldsymbol{x}}^*\| \tag{286}$$

According to Lemma A8 the following bounds on the norm $\|J\|_2$ of the Jacobian of the fixed point iteration hold. The $\gamma$-bound for $\|J\|_2$ is

$$\|J\|_2 \, \leqslant \, \beta \left((1-\gamma)\,m_{\max}^2 \, + \, \gamma\,2\,(2-\gamma)\,M^2\right) \, , \tag{287}$$

while the $\epsilon$-bound for $\|J\|_2$ is:

$$\|J\|_2 \, \leqslant \, \beta \left(m_{\max}^2 \, + \, \epsilon\,2\,(2-\epsilon)\,M^2\right) \, . \tag{288}$$

From the last condition we require for a contraction mapping:

$$\beta\,m_{\max}^2 \, < \, 1 \, . \tag{289}$$

We want to see how large $\epsilon$ is. The separation of center $\boldsymbol{m}_{\boldsymbol{x}}$ from data $\boldsymbol{X} = (\boldsymbol{x}_{l+1}, \ldots, \boldsymbol{x}_N)$ is

$$\Delta_m \, = \, \min_{j,l<j}\left(\boldsymbol{m}_{\boldsymbol{x}}^T\boldsymbol{m}_{\boldsymbol{x}} \, - \, \boldsymbol{m}_{\boldsymbol{x}}^T\boldsymbol{x}_j\right) \, = \, \boldsymbol{m}_{\boldsymbol{x}}^T\boldsymbol{m}_{\boldsymbol{x}} \, - \, \max_{j,l<j}\boldsymbol{m}_{\boldsymbol{x}}^T\boldsymbol{x}_j \, . \tag{290}$$

We need the separation $\tilde{\Delta}_m$ of $\tilde{\boldsymbol{x}} = \lambda\boldsymbol{\xi} + (1-\lambda)\boldsymbol{m}_{\boldsymbol{x}}^*$ from the data.

$$\tilde{\Delta}_m \, = \, \min_{j,l<j}\left(\tilde{\boldsymbol{x}}^T\boldsymbol{m}_{\boldsymbol{x}} \, - \, \tilde{\boldsymbol{x}}^T\boldsymbol{x}_j\right) \, . \tag{291}$$

We compute a lower bound on $\tilde{\Delta}_m$. Using the Cauchy-Schwarz inequality, we obtain for $1 \leqslant j \leqslant N$:

$$\left|\tilde{\boldsymbol{x}}^T\boldsymbol{x}_j \, - \, \boldsymbol{m}_{\boldsymbol{x}}^T\boldsymbol{x}_j\right| \, \leqslant \, \|\tilde{\boldsymbol{x}} - \boldsymbol{m}_{\boldsymbol{x}}\|\,\|\boldsymbol{x}_j\| \, \leqslant \, \|\tilde{\boldsymbol{x}} \, - \, \boldsymbol{m}_{\boldsymbol{x}}\|\,M \, . \tag{292}$$

We have the lower bound

$$\tilde{\Delta}_m \, \geq \, \min_{j,l<j}\left(\left(\boldsymbol{m}_{\boldsymbol{x}}^T\boldsymbol{m}_{\boldsymbol{x}} \, - \, \|\tilde{\boldsymbol{x}} - \boldsymbol{m}_{\boldsymbol{x}}\|\,M\right) \, - \, \left(\boldsymbol{m}_{\boldsymbol{x}}^T\boldsymbol{x}_j \, + \, \|\tilde{\boldsymbol{x}} - \boldsymbol{m}_{\boldsymbol{x}}\|\,M\right)\right) \tag{293}$$

$$= \, - 2\,\|\tilde{\boldsymbol{x}} \, - \, \boldsymbol{m}_{\boldsymbol{x}}\|\,M \, + \, \min_{j,l<j}\left(\boldsymbol{m}_{\boldsymbol{x}}^T\boldsymbol{m}_{\boldsymbol{x}} \, - \, \boldsymbol{m}_{\boldsymbol{x}}^T\boldsymbol{x}_j\right) \, = \, \Delta_m \, - \, 2\,\|\tilde{\boldsymbol{x}} \, - \, \boldsymbol{m}_{\boldsymbol{x}}\|\,M \, .$$

Since

$$\begin{aligned}\|\tilde{\boldsymbol{x}} \, - \, \boldsymbol{m}_{\boldsymbol{x}}\| \, &= \, \|\lambda\boldsymbol{\xi} + (1-\lambda)\boldsymbol{m}_{\boldsymbol{x}}^* \, - \, \boldsymbol{m}_{\boldsymbol{x}}\| \\ &\leqslant \, \lambda\,\|\boldsymbol{\xi} \, - \, \boldsymbol{m}_{\boldsymbol{x}}\| \, + \, (1-\lambda)\,\|\boldsymbol{m}_{\boldsymbol{x}}^* \, - \, \boldsymbol{m}_{\boldsymbol{x}}\| \\ &\leqslant \, \max\{\|\boldsymbol{\xi} \, - \, \boldsymbol{m}_{\boldsymbol{x}}\|, \|\boldsymbol{m}_{\boldsymbol{x}}^* \, - \, \boldsymbol{m}_{\boldsymbol{x}}\|\} \, ,\end{aligned} \tag{294}$$

we have

$$\tilde{\Delta}_m \, \geq \, \Delta_m \, - \, 2\,\max\{\|\boldsymbol{\xi} \, - \, \boldsymbol{m}_{\boldsymbol{x}}\|, \|\boldsymbol{m}_{\boldsymbol{x}}^* \, - \, \boldsymbol{m}_{\boldsymbol{x}}\|\}\,M \, . \tag{295}$$

$$\epsilon \, = \, (N-l)\exp(-\,\beta\,(\Delta_m \, - \, 2\,\max\{\|\boldsymbol{\xi} \, - \, \boldsymbol{m}_{\boldsymbol{x}}\|, \|\boldsymbol{m}_{\boldsymbol{x}}^* \, - \, \boldsymbol{m}_{\boldsymbol{x}}\|\}\,M)) \, . \tag{296}$$

### A.1.6  PROPERTIES OF FIXED POINTS NEAR STORED PATTERN

In Subsection A.1.5.3 many stable states that are fixed points near the stored patterns are considered. We now consider this case. In the fist subsection we investigate the storage capacity if all patterns are sufficiently separated so that metastable states do not appear. In the next subsection we look into the updates required and error when retrieving the stored patterns. For metastable states we can do the same analyses if each metastable state is treated as one state like one pattern.

We see a trade-off that is known from classical Hopfield networks and for modern Hopfield networks. Small separation $\Delta_i$ of the pattern $\boldsymbol{x}_i$ from the other patterns gives high storage capacity. However the convergence speed is lower and the retrieval error higher. In contrast, large separation $\Delta_i$ of the pattern $\boldsymbol{x}_i$ from the other pattern allows the retrieval of patterns with one update step and exponentially low error.

**A.1.6.1 Exponentially Many Patterns can be Stored.** From Subsection A.1.5.3 need some definitions. We assume to have $N$ patterns, the separation of pattern $\boldsymbol{x}_i$ from the other patterns $\{\boldsymbol{x}_1, \ldots, \boldsymbol{x}_{i-1}, \boldsymbol{x}_{i+1}, \ldots, \boldsymbol{x}_N\}$ is $\Delta_i$, defined as

$$\Delta_i = \min_{j,j \neq i} \left( \boldsymbol{x}_i^T \boldsymbol{x}_i - \boldsymbol{x}_i^T \boldsymbol{x}_j \right) = \boldsymbol{x}_i^T \boldsymbol{x}_i - \max_{j,j \neq i} \boldsymbol{x}_i^T \boldsymbol{x}_j . \tag{297}$$

The pattern is separated from the other data if $0 < \Delta_i$. The separation $\Delta_i$ can also be expressed as

$$\Delta_i = \min_{j,j \neq i} \frac{1}{2} \left( \|\boldsymbol{x}_i\|^2 - \|\boldsymbol{x}_j\|^2 + \|\boldsymbol{x}_i - \boldsymbol{x}_j\|^2 \right) \tag{298}$$

$$= \frac{1}{2}\|\boldsymbol{x}_i\|^2 - \frac{1}{2} \max_{j,j \neq i} \left( \|\boldsymbol{x}_j\|^2 - \|\boldsymbol{x}_i - \boldsymbol{x}_j\|^2 \right) .$$

For $\|\boldsymbol{x}_i\| = \|\boldsymbol{x}_j\|$ we have $\Delta_i = 1/2 \min_{j,j \neq i} \|\boldsymbol{x}_i - \boldsymbol{x}_j\|^2$. The sphere $\mathrm{S}_i$ with center $\boldsymbol{x}_i$ is defined as

$$\mathrm{S}_i = \left\{ \boldsymbol{\xi} \mid \|\boldsymbol{\xi} - \boldsymbol{x}_i\| \leqslant \frac{1}{\beta N M} \right\} . \tag{299}$$

The maximal length of a pattern is $M = \max_i \|\boldsymbol{x}_i\|$.

We next define what we mean with storing and retrieving a pattern.

**Definition 5** (Pattern Stored and Retrieved). *We assume that around every pattern $\boldsymbol{x}_i$ a sphere $\mathrm{S}_i$ is given. We say $\boldsymbol{x}_i$ is stored if there is a single fixed point $\boldsymbol{x}_i^* \in \mathrm{S}_i$ to which all points $\boldsymbol{\xi} \in \mathrm{S}_i$ converge, and $\mathrm{S}_i \cap \mathrm{S}_j = \emptyset$ for $i \neq j$. We say $\boldsymbol{x}_i$ is retrieved for a given $\epsilon$ if iteration (update rule) Eq. (92) gives a point $\tilde{\boldsymbol{x}}_i$ that is at least $\epsilon$-close to the single fixed point $\boldsymbol{x}_i^* \in \mathrm{S}_i$. The retrieval error is $\|\tilde{\boldsymbol{x}}_i - \boldsymbol{x}_i\|$.*

The sphere $\mathrm{S}_i$ around pattern $\boldsymbol{x}_i$ can be any a sphere and do not have the specific sphere defined in Def. 3.

For a query $\boldsymbol{\xi} \in \mathrm{S}_i$ to converge to a fixed point $\boldsymbol{x}_i^* \in \mathrm{S}_i$ we required for the application of Banach fixed point theorem and for ensuring a contraction mapping the following inequality:

$$\Delta_i \geqslant \frac{2}{\beta N} + \frac{1}{\beta} \ln \left( 2 (N-1) N \beta M^2 \right) . \tag{300}$$

This is the assumption in Lemma A7 to ensure a fixed point in sphere $\mathrm{S}_i$. Since replacing $(N-1)N$ by $N^2$ gives

$$\frac{2}{\beta N} + \frac{1}{\beta} \ln \left( 2 N^2 \beta M^2 \right) > \frac{2}{\beta N} + \frac{1}{\beta} \ln \left( 2 (N-1) N \beta M^2 \right) , \tag{301}$$

the inequality follows from following master inequality

$$\Delta_i \geqslant \frac{2}{\beta N} + \frac{1}{\beta} \ln \left( 2 N^2 \beta M^2 \right) , \tag{302}$$

If we assume that $\mathrm{S}_i \cap \mathrm{S}_j \neq \emptyset$ with $i \neq j$, then the triangle inequality with a point from the intersection gives

$$\|\boldsymbol{x}_i - \boldsymbol{x}_j\| \leqslant \frac{2}{\beta N M} . \tag{303}$$

Therefore, we have using the Cauchy-Schwarz inequality:

$$\Delta_i \leqslant \boldsymbol{x}_i^T (\boldsymbol{x}_i - \boldsymbol{x}_j) \leqslant \|\boldsymbol{x}_i\| \|\boldsymbol{x}_i - \boldsymbol{x}_j\| \leqslant M \frac{2}{\beta N M} = \frac{2}{\beta N} . \tag{304}$$

The last inequality is a contraction to Eq. (302) if we assume that

$$1 < 2 (N-1) N \beta M^2 . \tag{305}$$

With this assumption, the spheres $\mathrm{S}_i$ and $\mathrm{S}_j$ do not intersect. Therefore, each $\boldsymbol{x}_i$ has its separate fixed point in $\mathrm{S}_i$. We define

$$\Delta_{\min} = \min_{1 \leqslant i \leqslant N} \Delta_i \tag{306}$$

to obtain the master inequality

$$\Delta_{\min} \geq \frac{2}{\beta\,N} + \frac{1}{\beta}\,\ln\left(2\,N^2\,\beta\,M^2\right) . \tag{307}$$

•*Patterns on a sphere.*

For simplicity and in accordance with the results of the classical Hopfield network, we assume all *patterns being on a sphere* with radius $M$:

$$\forall_i : \ \|\boldsymbol{x}_i\| = M . \tag{308}$$

Under assumption Eq. (305) we have only to show that the master inequality Eq. (307) is fulfilled for each $\boldsymbol{x}_i$ to have a separate fixed point near each $\boldsymbol{x}_i$.

We defined $\alpha_{ij}$ as the angle between $\boldsymbol{x}_i$ and $\boldsymbol{x}_j$. The minimal angle $\alpha_{\min}$ between two data points is

$$\alpha_{\min} = \min_{1 \leqslant i < j \leqslant N} \alpha_{ij} . \tag{309}$$

On the sphere with radius $M$ we have

$$\Delta_{\min} = \min_{1 \leqslant i < j \leqslant N} M^2(1 - \cos(\alpha_{ij})) = M^2(1 - \cos(\alpha_{\min})) , \tag{310}$$

therefore it is sufficient to show the master inequality on the sphere:

$$M^2(1 - \cos(\alpha_{\min})) \geq \frac{2}{\beta\,N} + \frac{1}{\beta}\,\ln\left(2\,N^2\,\beta\,M^2\right) . \tag{311}$$

Under assumption Eq. (305) we have only to show that the master inequality Eq. (307) is fulfilled for $\Delta_{\min}$. We consider patterns on the sphere, therefore the master inequality Eq. (307) becomes Eq. (311). First we show results when pattern positions on the sphere are constructed and $\Delta_{\min}$ is ensured. Then we move on to random patterns on a sphere, where $\Delta_{\min}$ becomes a random variable.

•*Storage capacity for patterns placed on the sphere.*

Next theorem says how many patterns we can stored (fixed point with attraction basin near pattern) if we are allowed to place them on the sphere.

**Theorem A3** (Storage Capacity (M=2): Placed Patterns). *We assume $\beta = 1$ and patterns on the sphere with radius $M$. If $M = 2\sqrt{d-1}$ and the dimension $d$ of the space is $d \geq 4$ or if $M = 1.7\sqrt{d-1}$ and the dimension $d$ of the space is $d \geq 50$, then the number of patterns $N$ that can be stored (fixed point with attraction basin near pattern) is at least*

$$N = 2^{2(d-1)} . \tag{312}$$

*Proof.* For random patterns on the sphere, we have to show that the master inequality Eq. (311) holds:

$$M^2(1 - \cos(\alpha_{\min})) \geq \frac{2}{\beta\,N} + \frac{1}{\beta}\,\ln\left(2\,N^2\,\beta\,M^2\right) . \tag{313}$$

We now place the patterns equidistant on the sphere where the pattern are separated by an angle $\alpha_{\min}$:

$$\forall_i : \ \min_{j, j \neq i} \alpha_{ij} = \alpha_{\min} , \tag{314}$$

In a $d$-dimensional space we can place

$$N = \left(\frac{2\pi}{\alpha_{\min}}\right)^{d-1} \tag{315}$$

points on the sphere. In a spherical coordinate system a pattern differs from its most closest patterns by an angle $\alpha_{\min}$ and there are $d-1$ angles. Solving for $\alpha_{\min}$ gives

$$\alpha_{\min} = \frac{2\pi}{N^{1/(d-1)}} . \tag{316}$$

The number of patterns that can be stored is determined by the largest $N$ that fulfils

$$M^2 \left( 1 - \cos \left( \frac{2\pi}{N^{1/(d-1)}} \right) \right) \geq \frac{2}{\beta N} + \frac{1}{\beta} \ln \left( 2 N^2 \beta M^2 \right) . \tag{317}$$

We set $N = 2^{2(d-1)}$ and obtain for Eq. (317):

$$M^2 \left( 1 - \cos \left( \frac{\pi}{2} \right) \right) \geq \frac{2}{\beta \, 2^{3(d-1)}} + \frac{1}{\beta} \ln \left( 2 \beta M^2 \right) + \frac{1}{\beta} 4 \, (d-1) \ln 2 . \tag{318}$$

This inequality is equivalent to

$$\beta M^2 \geq \frac{1}{2^{2(d-1)-1}} + \ln \left( 2 \beta M^2 \right) + 4 \, (d-1) \ln 2 . \tag{319}$$

The last inequality can be fulfilled with $M = K\sqrt{d-1}$ and proper $K$. For $\beta = 1$, $d = 4$ and $K = 2$ the inequality is fulfilled. The left hand side minus the right hand side is $4(d-1) - 1/2^{2(d-1)-1} - \ln(8(d-1)) - 4(d-1)\ln 2$. Its derivative with respect to $d$ is strict positive. Therefore, the inequality holds for $d \geq 4$.

For $\beta = 1$, $d = 50$ and $K = 1.7$ the inequality is fulfilled. The left hand side minus the right hand side is $2.89(d-1) - 1/2^{2(d-1)-1} - \ln(5.78(d-1)) - 4(d-1)\ln 2$. Its derivative with respect to $d$ is strict positive. Therefore, the inequality holds for $d \geq 50$.

$\square$

If we want to store considerably more patterns, then we have to increase the length of the vectors or the dimension of the space where the vectors live. The next theorem shows results for the number of patterns $N$ with $N = 2^{3(d-1)}$.

**Theorem A4** (Storage Capacity (M=5): Placed Patterns). *We assume $\beta = 1$ and patterns on the sphere with radius $M$. If $M = 5\sqrt{d-1}$ and the dimension $d$ of the space is $d \geq 3$ or if $M = 4\sqrt{d-1}$ and the dimension $d$ of the space is $d \geq 13$, then the number of patterns $N$ that can be stored (fixed point with attraction basin near pattern) is at least*

$$N = 2^{3(d-1)} . \tag{320}$$

*Proof.* We set $N = 2^{3(d-1)}$ and obtain for Eq. (317):

$$M^2 \left( 1 - \cos \left( \frac{\pi}{4} \right) \right) \geq \frac{2}{\beta \, 2^{3(d-1)}} + \frac{1}{\beta} \ln \left( 2 \beta M^2 \right) + \frac{1}{\beta} 6 \, (d-1) \ln 2 . \tag{321}$$

This inequality is equivalent to

$$\beta M^2 \left( 1 - \frac{\sqrt{2}}{2} \right) \geq \frac{1}{2^{3(d-1)-1}} + \ln \left( 2 \beta M^2 \right) + 6 \, (d-1) \ln 2 . \tag{322}$$

The last inequality can be fulfilled with $M = K\sqrt{d-1}$ and proper $K$. For $\beta = 1$, $d = 13$ and $K = 4$ the inequality is fulfilled. The left hand side minus the right hand side is $4.686292(d-1) - 1/2^{3(d-1)-1} - \ln(32(d-1)) - 6(d-1)\ln 2$. Its derivative with respect to $d$ is strict positive. Therefore, the inequality holds for $d \geq 13$.

For $\beta = 1$, $d = 3$ and $K = 5$ the inequality is fulfilled. The left hand side minus the right hand side is $7.32233(d-1) - 1/2^{3(d-1)-1} - \ln(50(d-1)) - 6(d-1)\ln 2$. Its derivative with respect to $d$ is strict positive. Therefore, the inequality holds for $d \geq 3$.

$\square$

•*Storage capacity for random patterns on the sphere.*

Next we investigate random points on the sphere. Under assumption Eq. (305) we have to show that the master inequality Eq. (311) is fulfilled for $\alpha_{\min}$, where now $\alpha_{\min}$ is now a random variable. We use results on the distribution of the minimal angles between random patterns on a sphere according to Cai et al. (2013) and Brauchart et al. (2018). Theorem 2 in Cai et al. (2013) gives the distribution of the minimal angle for random patterns on the unit sphere. Proposition 3.5 in Brauchart et al. (2018) gives a lower bound on the probability of the minimal angle being larger than a given constant. We require this proposition to derive the probability of pattern having a minimal angle $\alpha_{\min}$. Proposition 3.6 in Brauchart et al. (2018) gives the expectation of the minimal angle.

We will prove high probability bounds for the expected storage capacity. We need the following tail-bound on $\alpha_{\min}$ (the minimal angle of random patterns on a sphere):

**Lemma A13** ((Brauchart et al., 2018)). *Let $d$ be the dimension of the pattern space,*

$$\kappa_d \ := \ \frac{1}{d\sqrt{\pi}} \, \frac{\Gamma((d+1)/2)}{\Gamma(d/2)} \ . \tag{323}$$

*and $\delta > 0$ such that $\frac{\kappa_{d-1}}{2}\delta^{(d-1)} \leqslant 1$. Then*

$$\Pr(N^{\frac{2}{d-1}}\alpha_{\min} \ \geq \ \delta) \ \geq \ 1 \ - \ \frac{\kappa_{d-1}}{2}\,\delta^{d-1} \ . \tag{324}$$

*Proof.* The statement of the lemma is Eq. (3-6) from Proposition 3.5 in Brauchart et al. (2018). $\square$

Next we derive upper and lower bounds on the constant $\kappa_d$ since we require them later for proving storage capacity bounds.

**Lemma A14.** *For $\kappa_d$ defined in Eq. (323) we have the following bounds for every $d \geq 1$:*

$$\frac{1}{\exp(1/6)\sqrt{e\,\pi\,d}} \ \leqslant \ \kappa_d \ \leqslant \ \frac{\exp(1/12)}{\sqrt{2\,\pi\,d}} \ < \ 1 \ . \tag{325}$$

*Proof.* We use for $x > 0$ the following bound related to Stirling's approximation formula for the gamma function, c.f. (Olver et al., 2010, (5.6.1)):

$$1 \ < \ \Gamma(x)\,(2\,\pi)^{-\frac{1}{2}}x^{\frac{1}{2}-x}\exp(x) \ < \ \exp\left(\frac{1}{12\,x}\right) \ . \tag{326}$$

Using Stirling's formula Eq. (326), we upper bound $\kappa_d$:

$$\kappa_d \ = \ \frac{1}{d\sqrt{\pi}} \, \frac{\Gamma((d+1)/2)}{\Gamma(d/2)} \ < \ \frac{1}{d\sqrt{\pi}} \, \frac{\exp\left(\frac{1}{6(d+1)}\right)\exp\left(-\frac{d+1}{2}\right)\left(\frac{d+1}{2}\right)^{\frac{d}{2}}}{\exp\left(-\frac{d}{2}\right)\left(\frac{d}{2}\right)^{\frac{d}{2}-\frac{1}{2}}} \tag{327}$$

$$= \ \frac{1}{d\sqrt{\pi\,e}}\exp\left(\frac{1}{6(d+1)}\right)\left(1+\frac{1}{d}\right)^{\frac{d}{2}}\sqrt{\frac{d}{2}} \ \leqslant \ \frac{\exp\left(\frac{1}{12}\right)}{\sqrt{2\,\pi}\,\sqrt{d}} \ .$$

For the first inequality, we applied Eq. (326), while for the second we used $(1+\frac{1}{d})^d < e$ for $d \geq 1$.

Next, we lower bound $\kappa_d$ by again applying Stirling's formula Eq. (326):

$$\kappa_d \ = \ \frac{1}{d\sqrt{\pi}} \, \frac{\Gamma((d+1)/2)}{\Gamma(d/2)} \ > \ \frac{1}{d\sqrt{\pi}} \, \frac{\exp\left(-\frac{d+1}{2}\right)\left(\frac{d+1}{2}\right)^{\frac{d}{2}}}{\exp\left(\frac{1}{6\,d}\right)\exp\left(-\frac{d}{2}\right)\left(\frac{d}{2}\right)^{\frac{d}{2}-\frac{1}{2}}} \tag{328}$$

$$= \ \frac{1}{d\sqrt{\pi\,e}\,\exp\left(\frac{1}{6\,d}\right)}\left(1+\frac{1}{d}\right)^{\frac{d}{2}}\sqrt{\frac{d}{2}} \ \geq \ \frac{1}{\exp\left(\frac{1}{6}\right)\,\sqrt{e\,\pi\,d}} \ ,$$

where the last inequality holds because of monotonicity of $(1+\frac{1}{d})^d$ and using the fact that for $d=1$ it takes on the value 2. $\square$

We require a bound on $\cos$ to bound the master inequality Eq. (311).

**Lemma A15.** *For $0 \leqslant x \leqslant \pi$ the function $\cos$ can be upper bounded by:*

$$\cos(x) \ \leqslant \ 1 \ - \ \frac{x^2}{5} \ . \tag{329}$$

*Proof.* We use the infinite product representation of $\cos$, c.f. (Olver et al., 2010, (4.22.2)):

$$\cos(x) \ = \ \prod_{n=1}^{\infty} \left( 1 - \frac{4\,x^2}{(2n-1)^2\,\pi^2} \right) \ . \tag{330}$$

Since it holds that

$$1 \ - \ \frac{4\,x^2}{(2n-1)^2\,\pi^2} \ \leqslant \ 1 \tag{331}$$

for $|x| \leqslant \pi$ and $n \geq 2$, we can get the following upper bound on Eq. (330):

$$\cos(x) \ \leqslant \ \prod_{n=1}^{2} \left( 1 - \frac{4\,x^2}{(2n-1)^2\pi^2} \right) \ = \ \left( 1 - \frac{4\,x^2}{\pi^2} \right) \left( 1 - \frac{4\,x^2}{9\,\pi^2} \right) \tag{332}$$

$$= \ 1 \ - \ \frac{40\,x^2}{9\,\pi^2} \ + \ \frac{16\,x^4}{9\,\pi^4} \ \leqslant \ 1 \ - \ \frac{40\,x^2}{9\,\pi^2} \ + \ \frac{16\,x^2}{9\,\pi^2}$$

$$= \ 1 \ - \ \frac{24\,x^2}{9\,\pi^2} \ \leqslant \ 1 \ - \ \frac{x^2}{5} \ .$$

The last but one inequality uses $x \leqslant \pi$, which implies $x/\pi \leqslant 1$. Thus Eq. (329) is proven.

$\square$

• *Exponential storage capacity: the base $c$ as a function of the parameter $\beta$, the radius of the sphere $M$, the probability $p$, and the dimension $d$ of the space.*

We express the number $N$ of stored patterns by an exponential function with base $c > 1$ and an exponent linear in $d$. We derive constraints on he base $c$ as a function of $\beta$, the radius of the sphere $M$, the probability $p$ that all patterns can be stored, and the dimension $d$ of the space. With $\beta > 0$, $K > 0$, and $d \geq 2$ (to ensure a sphere), the following theorem gives our main result.

**Theorem A5** (Storage Capacity (Main): Random Patterns). *We assume a failure probability $0 < p \leqslant 1$ and randomly chosen patterns on the sphere with radius $M := K\sqrt{d-1}$. We define*

$$a \ := \ \frac{2}{d-1} \left( 1 \ + \ \ln(2\,\beta\,K^2\,p\,(d-1)) \right), \quad b \ := \ \frac{2\,K^2\,\beta}{5} \ ,$$

$$c \ := \ \frac{b}{W_0(\exp(a \ + \ \ln(b)))} \ , \tag{333}$$

*where $W_0$ is the upper branch of the Lambert $W$ function (Olver et al., 2010, (4.13)) and ensure*

$$c \ \geq \ \left( \frac{2}{\sqrt{p}} \right)^{\frac{4}{d-1}} \ . \tag{334}$$

*Then with probability $1 - p$, the number of random patterns that can be stored is*

$$N \ \geq \ \sqrt{p}\,c^{\frac{d-1}{4}} \ . \tag{335}$$

*Therefore it is proven for $c \geq 3.1546$ with $\beta = 1$, $K = 3$, $d = 20$ and $p = 0.001$ ($a + \ln(b) > 1.27$) and proven for $c \geq 1.3718$ with $\beta = 1$, $K = 1$, $d = 75$, and $p = 0.001$ ($a + \ln(b) < -0.94$).*

*Proof.* We consider the probability that the master inequality Eq. (311) is fulfilled:

$$\Pr \left( M^2(1 \ - \ \cos(\alpha_{\min})) \ \geq \ \frac{2}{\beta\,N} \ + \ \frac{1}{\beta} \ \ln\left(2\,N^2\,\beta\,M^2\right) \right) \ \geq \ 1 \ - \ p \ . \tag{336}$$

Using Eq. (329), we have:

$$1 - \cos(\alpha_{\min}) \geq \frac{1}{5} \alpha_{\min}^2 \, . \tag{337}$$

Therefore, with probability $1 - p$ the storage capacity is largest $N$ that fulfills

$$\Pr\left(M^2 \frac{\alpha_{min}^2}{5} \geq \frac{2}{\beta N} + \frac{1}{\beta} \ln\left(2 N^2 \beta M^2\right)\right) \geq 1 - p \, . \tag{338}$$

This inequality is equivalent to

$$\Pr\left(N^{\frac{2}{d-1}} \alpha_{min} \geq \frac{\sqrt{5} N^{\frac{2}{d-1}}}{M} \left(\frac{2}{\beta N} + \frac{1}{\beta} \ln\left(2 N^2 \beta M^2\right)\right)^{\frac{1}{2}}\right) \geq 1 - p \, . \tag{339}$$

We use Eq. (324) to obtain:

$$\Pr\left(N^{\frac{2}{d-1}} \alpha_{min} \geq \frac{\sqrt{5} N^{\frac{2}{d-1}}}{M} \left(\frac{2}{\beta N} + \frac{1}{\beta} \ln\left(2 N^2 \beta M^2\right)\right)^{\frac{1}{2}}\right) \tag{340}$$

$$\geq 1 - \frac{\kappa_{d-1}}{2} 5^{\frac{d-1}{2}} N^2 M^{-(d-1)} \left(\frac{2}{\beta N} + \frac{1}{\beta} \ln\left(2 N^2 \beta M^2\right)\right)^{\frac{d-1}{2}} \, .$$

For Eq. (339) to be fulfilled, it is sufficient that

$$\frac{\kappa_{d-1}}{2} 5^{\frac{d-1}{2}} N^2 M^{-(d-1)} \left(\frac{2}{\beta N} + \frac{1}{\beta} \ln\left(2 N^2 \beta M^2\right)\right)^{\frac{d-1}{2}} - p \leqslant 0 \, . \tag{341}$$

If we insert the assumption Eq. (334) of the theorem into Eq. (335), then we obtain $N \geq 2$. We now apply the upper bound $\kappa_{d-1}/2 < \kappa_{d-1} < 1$ from Eq. (325) and the upper bound $\frac{2}{\beta N} \leqslant \frac{1}{\beta}$ from $N \geq 2$ to inequality Eq. (341). In the resulting inequality we insert $N = \sqrt{p} c^{\frac{d-1}{4}}$ to check whether it is fulfilled with this special value of $N$ and obtain:

$$5^{\frac{d-1}{2}} p\, c^{\frac{d-1}{2}} M^{-(d-1)} \left(\frac{1}{\beta} + \frac{1}{\beta} \ln\left(2 p\, c^{\frac{d-1}{2}} \beta M^2\right)\right)^{\frac{d-1}{2}} \leqslant p \, . \tag{342}$$

Dividing by $p$, inserting $M = K\sqrt{d-1}$, and exponentiation of the left and right side by $\frac{2}{d-1}$ gives:

$$\frac{5\, c}{K^2\, (d-1)} \left(\frac{1}{\beta} + \frac{1}{\beta} \ln\left(2\, \beta\, c^{\frac{d-1}{2}} p\, K^2\, (d-1)\right)\right) - 1 \leqslant 0 \, . \tag{343}$$

After some algebraic manipulation, this inequality can be written as

$$a\, c + c\, \ln(c) - b \leqslant 0 \, , \tag{344}$$

where we used

$$a := \frac{2}{d-1} \left(1 + \ln(2\, \beta\, K^2\, p\, (d-1))\right), \quad b := \frac{2\, K^2\, \beta}{5} \, .$$

We determine the value $\hat{c}$ of $c$ which makes the inequality Eq. (344) equal to zero. We solve

$$a\, \hat{c} + \hat{c}\, \ln(\hat{c}) - b = 0 \tag{345}$$

for $\hat{c}$:

$$\begin{aligned}
& a\, \hat{c} + \hat{c}\, \ln(\hat{c}) - b = 0 && \tag{346} \\
\Leftrightarrow\ & a + \ln(\hat{c}) = b/\hat{c} \\
\Leftrightarrow\ & a + \ln(b) + \ln(\hat{c}/b) = b/\hat{c} \\
\Leftrightarrow\ & b/\hat{c} + \ln(b/\hat{c}) = a + \ln(b) \\
\Leftrightarrow\ & b/\hat{c} \exp(b/\hat{c}) = \exp(a + \ln(b)) \\
\Leftrightarrow\ & b/\hat{c} = W_0(\exp(a + \ln(b))) \\
\Leftrightarrow\ & \hat{c} = \frac{b}{W_0(\exp(a + \ln(b)))} \, ,
\end{aligned}$$

where $W_0$ is the upper branch of the Lambert $W$ function (see Def. A6). Hence, the solution is

$$\hat{c} \; = \; \frac{b}{W_0(\exp(a \; + \; \ln(b))} \; . \tag{347}$$

The solution exist, since the Lambert function $W_0(x)$ (Olver et al., 2010, (4.13)) is defined for $-1/e < x$ and we have $0 < \exp(a + \ln(b))$.

Since $\hat{c}$ fulfills inequality Eq. (344) and therefore also Eq. (342), we have a lower bound on the storage capacity $N$:

$$N \; \geq \; \sqrt{p} \, \hat{c}^{\frac{d-1}{4}} \; . \tag{348}$$

$\square$

Next we aim at a lower bound on $c$ which does not use the Lambert $W$ function (Olver et al., 2010, (4.13)). Therefore, we upper bound $W_0(\exp(a + \ln(b))$ to obtain a lower bound on $c$, therefore, also a lower bound on the storage capacity $N$. The lower bound is given in the next corollary.

**Corollary A1.** *We assume a failure probability $0 < p \leqslant 1$ and randomly chosen patterns on the sphere with radius $M = K\sqrt{d-1}$. We define*

$$a \; := \; \frac{2}{d-1} \left( 1 \; + \; \ln(2 \, \beta \, K^2 \, p \, (d-1)) \right), \quad b \; := \; \frac{2 \, K^2 \, \beta}{5} \; .$$

*Using the omega constant $\Omega \approx 0.56714329$ we set*

$$c \; = \; \begin{cases} b \, \ln \left( \frac{\Omega \, \exp(a \, + \, \ln(b)) \, + \, 1}{\Omega \, (1 \, + \, \Omega)} \right)^{-1} & \text{for } a \, + \, \ln(b) \, \leqslant \, 0 \, , \\ b \, (a \; + \; \ln(b))^{-\frac{a \, + \, \ln(b)}{a \, + \, \ln(b) \, + \, 1}} & \text{for } a \; + \; \ln(b) \; > \; 0 \end{cases} \tag{349}$$

*and ensure*

$$c \; \geq \; \left( \frac{2}{\sqrt{p}} \right)^{\frac{4}{d-1}} \; . \tag{350}$$

*Then with probability $1 - p$, the number of random patterns that can be stored is*

$$N \; \geq \; \sqrt{p} \, c^{\frac{d-1}{4}} \; . \tag{351}$$

*Examples are $c \geq 3.1444$ for $\beta = 1$, $K = 3$, $d = 20$ and $p = 0.001$ $(a + \ln(b) > 1.27)$ and $c \geq 1.2585$ for $\beta = 1$ $K = 1$, $d = 75$, and $p = 0.001$ $(a + \ln(b) < -0.94)$.*

*Proof.* We lower bound the $c$ defined in Theorem A5. According to (Hoorfar & Hassani, 2008, Theorem 2.3) we have for any real $u$ and $y > \frac{1}{e}$:

$$W_0(\exp(u)) \; \leqslant \; \ln \left( \frac{\exp(u) \; + \; y}{1 \; + \; \ln(y)} \right) \; . \tag{352}$$

To upper bound $W_0(x)$ for $x \in [0, 1]$, we set

$$y \; = \; 1/W_0(1) \; = \; 1/\Omega \; = \; \exp \Omega \; = \; -1/\ln \Omega \; \approx \; 1.76322 \, , \tag{353}$$

where the Omega constant $\Omega$ is

$$\Omega \; = \; \left( \int_{-\infty}^{\infty} \frac{\mathrm{d}t}{(e^t \; - \; t)^2 \; + \; \pi^2} \right)^{-1} \; - \; 1 \; \approx \; 0.56714329 \, . \tag{354}$$

See for these equations the special values of the Lambert $W$ function in Lemma A31. We have the upper bound on $W_0$:

$$W_0(\exp(u)) \; \leqslant \; \ln \left( \frac{\exp(u) \; + \; 1/\Omega}{1 \; + \; \ln(1/\Omega)} \right) \; = \; \ln \left( \frac{\Omega \, \exp(u) \; + \; 1}{\Omega(1 \; + \; \Omega)} \right) \; . \tag{355}$$

At the right hand side of interval $[0, 1]$, we have $u = 0$ and $\exp(u) = 1$ and get:

$$\ln\left(\frac{\Omega\, 1 + 1}{\Omega(1 + \Omega)}\right) = \ln\left(\frac{1}{\Omega}\right) = -\ln(\Omega) = \Omega = W_0(1).\tag{356}$$

Therefore, the bound is tight at the right hand side of of interval $[0, 1]$, that is for $\exp(u) = 1$, i.e. $u = 0$. We have derived an bound for $W_0(\exp(u))$ with $\exp(u) \in [0, 1]$ or, equivalently, $u \in [-\infty, 0]$. We obtain from Hoorfar & Hassani (2008, Corollary 2.6) the following bound on $W_0(\exp(u))$ for $1 < \exp(u)$, or, equivalently $0 < u$:

$$W_0(\exp(u)) \leqslant u^{\frac{u}{1+u}}.\tag{357}$$

A lower bound on $\hat{c}$ is obtained via the upper bounds Eq. (357) and Eq. (355) on $W_0$ as $W_0 > 0$. We set $u = a + \ln(b)$ and obtain

$$W_0(\exp(a + \ln(b))) \leqslant \begin{cases} \ln\left(\frac{\Omega\,\exp(a + \ln(b)) + 1}{\Omega\,(1 + \Omega)}\right)^{-1} & \text{for } a + \ln(b) \leqslant 0, \\ (a + \ln(b))^{-\frac{a + \ln(b)}{a + \ln(b) + 1}} & \text{for } a + \ln(b) > 0 \end{cases}\tag{358}$$

We insert this bound into Eq. (347), the solution for $\hat{c}$, to obtain the statement of the theorem.

$\square$

•*Exponential storage capacity: the dimension $d$ of the space as a function of the parameter $\beta$, the radius of the sphere $M$, and the probability $p$.*

We express the number $N$ of stored patterns by an exponential function with base $c > 1$ and an exponent linear in $d$. We derive constraints on the dimension $d$ of the space as a function of $\beta$, the radius of the sphere $M$, the probability $p$ that all patterns can be stored, and the base of the exponential storage capacity. The following theorem gives this result.

**Theorem A6** (Storage Capacity (d computed): Random Patterns). *We assume a failure probability $0 < p \leqslant 1$ and randomly chosen patterns on the sphere with radius $M = K\sqrt{d-1}$. We define*

$$a := \frac{\ln(c)}{2} - \frac{K^2\,\beta}{5\,c}, \quad b := 1 + \ln\left(2\,p\,\beta\,K^2\right),$$
$$d = \begin{cases} 1 + \frac{1}{a}\,W(a\,\exp(-b)) & \text{for } a \neq 0, \\ 1 + \exp(-b) & \text{for } a = 0, \end{cases}\tag{359}$$

*where $W$ is the Lambert W function (Olver et al., 2010, (4.13)). For $0 < a$ the function $W$ is the upper branch $W_0$ and for $a < 0$ we use the lower branch $W_{-1}$. If we ensure that*

$$c \geq \left(\frac{2}{\sqrt{p}}\right)^{\frac{4}{d-1}}, \quad -\frac{1}{e} \leqslant a\,\exp(-b),\tag{360}$$

*then with probability $1 - p$, the number of random patterns that can be stored is*

$$N \geq \sqrt{p}\,c^{\frac{d-1}{4}}.\tag{361}$$

*Proof.* We consider the probability that the master inequality Eq. (311) is fulfilled:

$$\Pr\left(M^2(1 - \cos(\alpha_{\min}))) \geq \frac{2}{\beta\,N} + \frac{1}{\beta}\,\ln\left(2\,N^2\,\beta\,M^2\right)\right) \geq 1 - p.\tag{362}$$

Using Eq. (329), we have:

$$1 - \cos(\alpha_{\min}) \geq \frac{1}{5}\,\alpha_{\min}^2.\tag{363}$$

Therefore, with probability $1 - p$ the storage capacity is largest $N$ that fulfills

$$\Pr\left(M^2\frac{\alpha_{min}^2}{5} \geq \frac{2}{\beta\,N} + \frac{1}{\beta}\,\ln\left(2\,N^2\,\beta\,M^2\right)\right) \geq 1 - p.\tag{364}$$

This inequality is equivalent to

$$\Pr\left(N^{\frac{2}{d-1}}\,\alpha_{min}\;\geq\;\frac{\sqrt{5}\,N^{\frac{2}{d-1}}}{M}\,\left(\frac{2}{\beta\,N}\,+\,\frac{1}{\beta}\,\ln\left(2\,N^2\,\beta\,M^2\right)\right)^{\frac{1}{2}}\right)\;\geq\;1\,-\,p\,. \tag{365}$$

We use Eq. (324) to obtain:

$$\Pr\left(N^{\frac{2}{d-1}}\,\alpha_{min}\;\geq\;\frac{\sqrt{5}\,N^{\frac{2}{d-1}}}{M}\,\left(\frac{2}{\beta\,N}\,+\,\frac{1}{\beta}\,\ln\left(2\,N^2\,\beta\,M^2\right)\right)^{\frac{1}{2}}\right) \tag{366}$$

$$\geq\;1\,-\,\frac{\kappa_{d-1}}{2}\,5^{\frac{d-1}{2}}\,N^2\,M^{-(d-1)}\left(\frac{2}{\beta\,N}\,+\,\frac{1}{\beta}\,\ln\left(2\,N^2\,\beta\,M^2\right)\right)^{\frac{d-1}{2}}\,.$$

For Eq. (365) to be fulfilled, it is sufficient that

$$\frac{\kappa_{d-1}}{2}\,5^{\frac{d-1}{2}}\,N^2\,M^{-(d-1)}\left(\frac{2}{\beta\,N}\,+\,\frac{1}{\beta}\,\ln\left(2\,N^2\,\beta M^2\right)\right)^{\frac{d-1}{2}}\,-\,p\,\leqslant\,0\,. \tag{367}$$

If we insert the assumption Eq. (360) of the theorem into Eq. (361), then we obtain $N \geq 2$. We now apply the upper bound $\kappa_{d-1}/2 < \kappa_{d-1} < 1$ from Eq. (325) and the upper bound $\frac{2}{\beta N} \leqslant \frac{1}{\beta}$ from $N \geq 2$ to inequality Eq. (367). In the resulting inequality we insert $N = \sqrt{p}c^{\frac{d-1}{4}}$ to check whether it is fulfilled with this special value of $N$ and obtain:

$$5^{\frac{d-1}{2}}\,p\,c^{\frac{d-1}{2}}\,M^{-(d-1)}\left(\frac{1}{\beta}\,+\,\frac{1}{\beta}\,\ln\left(2\,p\,c^{\frac{d-1}{2}}\,\beta M^2\right)\right)^{\frac{d-1}{2}}\,\leqslant\,p\,. \tag{368}$$

Dividing by $p$, inserting $M = K\sqrt{d-1}$, and exponentiation of the left and right side by $\frac{2}{d-1}$ gives:

$$\frac{5\,c}{K^2\,(d-1)}\,\left(\frac{1}{\beta}\,+\,\frac{1}{\beta}\,\ln\left(2\,\beta\,c^{\frac{d-1}{2}}\,p\,K^2\,(d-1)\right)\right)\,-\,1\,\leqslant\,0\,. \tag{369}$$

This inequality Eq. (369) can be reformulated as:

$$1\,+\,\ln\left(2\,p\,\beta\,c^{\frac{d-1}{2}}\,K^2\,(d-1)\right)\,-\,\frac{(d-1)\,K^2\,\beta}{5\,c}\,\leqslant\,0\,. \tag{370}$$

Using

$$a\;:=\;\frac{\ln(c)}{2}\,-\,\frac{K^2\,\beta}{5\,c}\,,\quad b\;:=\;1\,+\,\ln\left(2\,p\,\beta\,K^2\right)\,, \tag{371}$$

we write inequality Eq. (370) as

$$\ln(d-1)\,+\,a\,(d-1)\,+\,b\,\leqslant\,0\,. \tag{372}$$

We determine the value $\hat{d}$ of $d$ which makes the inequality Eq. (372) equal to zero. We solve

$$\ln(\hat{d}-1)\,+\,a\,(\hat{d}-1)\,+\,b\,=\,0\,. \tag{373}$$

for $\hat{d}$

For $a \neq 0$ we have

$$\ln(\hat{d}-1)\,+\,a\,(\hat{d}-1)\,+\,b\,=\,0 \tag{374}$$
$$\Leftrightarrow\,a\,(\hat{d}-1)\,+\,\ln(\hat{d}-1)\,=\,-b$$
$$\Leftrightarrow\,(\hat{d}-1)\exp(a\,(\hat{d}-1))\,=\,\exp(-b)$$
$$\Leftrightarrow\,a\,(\hat{d}-1)\exp(a\,(\hat{d}-1))\,=\,a\,\exp(-b)$$
$$\Leftrightarrow\,a\,(\hat{d}-1)\,=\,W(a\,\exp(-b))$$
$$\Leftrightarrow\,\hat{d}\,-\,1\,=\,\frac{1}{a}\,W(a\,\exp(-b))$$
$$\Leftrightarrow\,\hat{d}\,=\,1\,+\,\frac{1}{a}\,W(a\,\exp(-b))\,,$$

where $W$ is the Lambert $W$ function (see Def. A6). For $a > 0$ we have to use the upper branch $W_0$ of the Lambert $W$ function and for $a < 0$ we use the lower branch $W_{-1}$ of the Lambert $W$ function (Olver et al., 2010, (4.13)). We have to ensure that $-1/e \leqslant a \exp(-b)$ for a solution to exist. For $a = 0$ we have $\hat{d} = 1 + \exp(-b)$.

Hence, the solution is

$$\hat{d} \; = \; 1 \; + \; \frac{1}{a} \, W(a \exp(-b)) \,. \tag{375}$$

Since $\hat{d}$ fulfills inequality Eq. (369) and therefore also Eq. (368), we have a lower bound on the storage capacity $N$:

$$N \; \geq \; \sqrt{p} \, \hat{c}^{\frac{d-1}{4}} \,. \tag{376}$$

$\square$

**Corollary A2.** *We assume a failure probability $0 < p \leqslant 1$ and randomly chosen patterns on the sphere with radius $M = K\sqrt{d-1}$. We define*

$$a \; := \; \frac{\ln(c)}{2} \; - \; \frac{K^2 \, \beta}{5 \, c} \,, \quad b \; := \; 1 \; + \; \ln\left(2 \, p \, \beta \, K^2\right) \,,$$
$$d \; = \; 1 \; + \; \frac{1}{a} \, \left(-\ln(-a) \; + \; b\right) \,, \tag{377}$$

*and ensure*

$$c \; \geq \; \left(\frac{2}{\sqrt{p}}\right)^{\frac{4}{d-1}} \,, \quad -\frac{1}{e} \; \leqslant \; a \, \exp(-b) \,, \quad a \; < \; 0 \,, \tag{378}$$

*then with probability $1 - p$, the number of random patterns that can be stored is*

$$N \; \geq \; \sqrt{p} \, c^{\frac{d-1}{4}} \,. \tag{379}$$

*Setting $\beta = 1$, $K = 3$, $c = 2$ and $p = 0.001$ yields $d < 24$.*

*Proof.* For $a < 0$ the Eq. (359) from Theorem (A6) can be written as

$$d \; = \; 1 \; + \; \frac{W_{-1}(a \exp(-b))}{a} \; = \; 1 \; + \; \frac{W_{-1}(-\exp\left(-(-\ln(-a) + b - 1) - 1\right))}{a} \tag{380}$$

From Alzahrani & Salem (2018, Theorem 3.1) we get the following bound on $W_{-1}$:

$$-\frac{e}{e-1} \, (u + 1) \; < \; W_{-1}(-\exp(-u - 1)) \; < \; -(u + 1) \,. \tag{381}$$

for $u > 0$. We apply Eq. (381) to Eq. (380) with $u = -\ln(-a) + b - 1$.

Since $a < 0$ we get

$$d \; > \; 1 \; + \; \frac{-\ln(-a) + b}{a} \,. \tag{382}$$

$\square$

• *Storage capacity for the expected minimal separation instead of the probability that all patterns can be stored.* In contrast to the previous paragraph, we want to argue about the storage capacity for the expected minimal separation. Therefore, we will use the following bound on the expectation of $\alpha_{\min}$ (minimal angle), which gives also a bound on the expected of $\Delta_{\min}$ (minimal separation):

**Lemma A16** (Proposition 3.6 in Brauchart et al. (2018)). *We have the following lower bound on the expectation of $\alpha_{\min}$:*

$$\mathrm{E}\left[N^{\frac{2}{d-1}} \, \alpha_{\min}\right] \; \geq \; \left(\frac{\Gamma(\frac{d}{2})}{2(d-1) \, \sqrt{\pi} \, \Gamma(\frac{d-1}{2})}\right)^{-\frac{1}{d-1}} \Gamma(1 + \frac{1}{d-1}) \, \frac{d^{-\frac{1}{d-1}}}{\Gamma(2 + \frac{1}{d-1})} \; := \; C_{d-1}. \tag{383}$$

*The bound is valid for all $N \geq 2$ and $d \geq 2$.*

Let us start with some preliminary estimates. First of all we need some asymptotics for the constant $C_{d-1}$ in Eq. (383):

**Lemma A17.** *The following estimate holds for $d \geq 2$:*

$$C_d \geq 1 - \frac{\ln(d+1)}{d} \ . \tag{384}$$

*Proof.* The recursion formula for the Gamma function is (Olver et al., 2010, (5.5.1)):

$$\Gamma(x+1) = x \, \Gamma(x) \ . \tag{385}$$

We use Eq. (325) and the fact that $d^{\frac{1}{d}} \geq 1$ for $d \geq 1$ to obtain:

$$C_d \geq (2\sqrt{d})^{\frac{1}{d}} \Gamma(1 + \frac{1}{d}) \frac{(d+1)^{-\frac{1}{d}}}{\Gamma(2 + \frac{1}{d})} = (2\sqrt{d})^{\frac{1}{d}} \frac{(d+1)^{-\frac{1}{d}}}{1 - \frac{1}{d}} > (d+1)^{\frac{1}{d}} \tag{386}$$

$$= \exp(-\frac{1}{d} \ln(d+1)) \geq 1 - \frac{1}{d} \ln(d+1) \ ,$$

where in the last step we used the elementary inequality $\exp(x) \geq 1 + x$, which follows from the mean value theorem. $\qquad\square$

The next theorem states the number of stored patterns for the expected minimal separation.

**Theorem A7** (Storage Capacity (expected separation): Random Patterns)**.** *We assume patterns on the sphere with radius $M = K\sqrt{d-1}$ that are randomly chosen. Then for all values $c \geq 1$ for which*

$$\frac{1}{5}(d-1) K^2 c^{-1}(1 - \frac{\ln(d-1)}{(d-1)})^2 \geq \frac{2}{\beta \, c^{\frac{d-1}{4}}} + \frac{1}{\beta} \ln\left(2 \, c^{\frac{d-1}{2}} \, \beta \, (d-1) \, K^2\right) \tag{387}$$

*holds, the number of stored patterns for the expected minimal separation is at least*

$$N = c^{\frac{d-1}{4}} \ . \tag{388}$$

*The inequality Eq. (387) is e.g. fulfilled with $\beta = 1$, $K = 3$, $c = 2$ and $d \geq 17$.*

*Proof.* Instead of considering the probability that the master inequality Eq. (311) is fulfilled we now consider whether this inequality is fulfilled for the expected minimal distance. We consider the expectation of the minimal distance $\Delta_{\min}$:

$$\mathrm{E}[\Delta_{\min}] = \mathrm{E}[M^2(1 - \cos(\alpha_{\min})))] = M^2(1 - \mathrm{E}[\cos(\alpha_{\min})]) \ . \tag{389}$$

For this expectation, the master inequality Eq. (311) becomes

$$M^2(1 - \mathrm{E}[\cos(\alpha_{\min})]) \geq \frac{2}{\beta \, N} + \frac{1}{\beta} \ln\left(2 \, N^2 \, \beta \, M^2\right) \ . \tag{390}$$

We want to find the largest $N$ that fulfills this inequality.

We apply Eq. (329) and Jensen's inequality to deduce the following lower bound:

$$1 - \mathrm{E}[\cos(\alpha_{\min})] \geq \frac{1}{5} \mathrm{E}\left[\alpha_{\min}^2\right] \geq \frac{1}{5} \mathrm{E}[\alpha_{\min}]^2 \ . \tag{391}$$

Now we use Eq. (383) and Eq. (384) to arrive at

$$\mathrm{E}[\alpha_{\min}]^2 \geq N^{-\frac{4}{d-1}} \mathrm{E}[N^{\frac{2}{d-1}} \alpha_{\min}]^2 \geq N^{-\frac{4}{d-1}} C_{d-1}^2 \geq N^{-\frac{4}{d-1}} (1 - \frac{\ln(d-1)}{(d-1)})^2 \ , \tag{392}$$

for sufficiently large $d$. Thus in order to fulfill Eq. (390), it is enough to find values that satisfy Eq. (387).

$\qquad\square$

**A.1.6.2 Retrieval of Patterns with One Update and Small Retrieval Error.** Retrieval of a pattern $\boldsymbol{x}_i$ for fixed point $\boldsymbol{x}_i^*$ and query $\boldsymbol{\xi}$ is defined via an $\epsilon$ by $\|f(\boldsymbol{\xi}) - \boldsymbol{x}_i^*\| < \epsilon$, that is, the update is $\epsilon$-close to the fixed point. The update rule retrieves a pattern with one update for well separated patterns, that is, $\Delta_i$ is large.

**Theorem A8** (Pattern Retrieval with One Update). *With query $\boldsymbol{\xi}$, after one update the distance of the new point $f(\boldsymbol{\xi})$ to the fixed point $\boldsymbol{x}_i^*$ is exponentially small in the separation $\Delta_i$. The precise bounds using the Jacobian $\mathrm{J} = \frac{\partial f(\boldsymbol{\xi})}{\partial \boldsymbol{\xi}}$ and its value $\mathrm{J}^m$ in the mean value theorem are:*

$$\|f(\boldsymbol{\xi}) - \boldsymbol{x}_i^*\| \leqslant \|\mathrm{J}^m\|_2 \|\boldsymbol{\xi} - \boldsymbol{x}_i^*\|, \tag{393}$$
$$\|\mathrm{J}^m\|_2 \leqslant 2\,\beta\,N\,M^2\,(N-1)\exp(-\,\beta\,(\Delta_i\,-\,2\,\max\{\|\boldsymbol{\xi}\,-\,\boldsymbol{x}_i\|, \|\boldsymbol{x}_i^*\,-\,\boldsymbol{x}_i\|\}\,M))\,. \tag{394}$$

*For given $\epsilon$ and sufficient large $\Delta_i$, we have $\|f(\boldsymbol{\xi}) - \boldsymbol{x}_i^*\| < \epsilon$, that is, retrieval with one update.*

*Proof.* From Eq. (180) we have

$$\|\mathrm{J}^m\|_2 \leqslant 2\,\beta\,N\,M^2\,(N-1)\exp(-\,\beta\,(\Delta_i\,-\,2\,\max\{\|\boldsymbol{\xi}\,-\,\boldsymbol{x}_i\|, \|\boldsymbol{x}_i^*\,-\,\boldsymbol{x}_i\|\}\,M))\,. \tag{395}$$

After every iteration the mapped point $f(\boldsymbol{\xi})$ is closer to the fixed point $\boldsymbol{x}_i^*$ than the original point $\boldsymbol{x}_i$:

$$\|f(\boldsymbol{\xi}) - \boldsymbol{x}_i^*\| \leqslant \|\mathrm{J}^m\|_2 \|\boldsymbol{\xi} - \boldsymbol{x}_i^*\|\,. \tag{396}$$

For given $\epsilon$ and sufficient large $\Delta_i$, we have $\|f(\boldsymbol{\xi}) - \boldsymbol{x}_i^*\| < \epsilon$, since $\|\mathrm{J}^m\|_2$ foes exponentially fast to zero with increasing $\Delta_i$. $\qquad\square$

We want to estimate how large $\Delta_i$ is. For $\boldsymbol{x}_i$ we have:

$$\Delta_i \;=\; \min_{j,j\neq i}\left(\boldsymbol{x}_i^T\boldsymbol{x}_i \,-\, \boldsymbol{x}_i^T\boldsymbol{x}_j\right) \;=\; \boldsymbol{x}_i^T\boldsymbol{x}_i \,-\, \max_{j,j\neq i}\boldsymbol{x}_i^T\boldsymbol{x}_j\,. \tag{397}$$

To estimate how large $\Delta_i$ is, assume vectors $\boldsymbol{x} \in \mathbb{R}^d$ and $\boldsymbol{y} \in \mathbb{R}^d$ that have as components standard normally distributed values. The expected value of the separation of two points with normally distributed components is

$$\mathrm{E}\left[\boldsymbol{x}^T\boldsymbol{x} \,-\, \boldsymbol{x}^T\boldsymbol{y}\right] \;=\; \sum_{j=1}^{d}\mathrm{E}\left[x_j^2\right] \,+\, \sum_{j=1}^{d}\mathrm{E}\left[x_j\right]\sum_{j=1}^{d}\mathrm{E}\left[y_j\right] \;=\; d\,. \tag{398}$$

The variance of the separation of two points with normally distributed components is

$$\mathrm{Var}\left[\boldsymbol{x}^T\boldsymbol{x} \,-\, \boldsymbol{x}^T\boldsymbol{y}\right] \;=\; \mathrm{E}\left[\left(\boldsymbol{x}^T\boldsymbol{x} \,-\, \boldsymbol{x}^T\boldsymbol{y}\right)^2\right] \,-\, d^2 \tag{399}$$

$$= \sum_{j=1}^{d}\mathrm{E}\left[x_j^4\right] \,+\, \sum_{j=1,k=1,k\neq j}^{d}\mathrm{E}\left[x_j^2\right]\mathrm{E}\left[x_k^2\right] \,-\, 2\sum_{j=1}^{d}\mathrm{E}\left[x_j^3\right]\mathrm{E}\left[y_j\right] -$$

$$2\sum_{j=1,k=1,k\neq j}^{d}\mathrm{E}\left[x_j^2\right]\mathrm{E}\left[x_k\right]\mathrm{E}\left[y_k\right] \,+\, \sum_{j=1}^{d}\mathrm{E}\left[x_j^2\right]\mathrm{E}\left[y_j^2\right] +$$

$$\sum_{j=1,k=1,k\neq j}^{d}\mathrm{E}\left[x_j\right]\mathrm{E}\left[y_j\right]\mathrm{E}\left[x_k\right]\mathrm{E}\left[y_k\right] \,-\, d^2$$

$$= 3\,d \,+\, d\,(d-1) \,+\, d \,-\, d^2 \;=\; 3\,d\,.$$

The expected value for the separation of two random vectors gives:

$$\|\mathrm{J}^m\|_2 \leqslant 2\,\beta\,N\,M^2\,(N-1)\exp(-\,\beta\,(d\,-\,2\,\max\{\|\boldsymbol{\xi}\,-\,\boldsymbol{x}_i\|, \|\boldsymbol{x}_i^*\,-\,\boldsymbol{x}_i\|\}\,M))\,. \tag{400}$$

For the exponential storage we set $M = 2\sqrt{d-1}$. We see the Lipschitz constant $\|\mathrm{J}^m\|_2$ decreases exponentially with the dimension. Therefore, $\|f(\boldsymbol{\xi}) - \boldsymbol{x}_i^*\|$ is exponentially small after just one update. Therefore, the fixed point is well retrieved after one update.

The retrieval error decreases exponentially with the separation $\Delta_i$.

**Theorem A9** (Exponentially Small Retrieval Error). *The retrieval error $\|f(\boldsymbol{\xi}) - \boldsymbol{x}_i\|$ of pattern $\boldsymbol{x}_i$ is bounded by*

$$\|f(\boldsymbol{\xi}) - \boldsymbol{x}_i\| \leqslant 2\,(N-1)\,\exp(-\,\beta\,(\Delta_i\,-\,2\,\max\{\|\boldsymbol{\xi}\,-\,\boldsymbol{x}_i\|, \|\boldsymbol{x}_i^*\,-\,\boldsymbol{x}_i\|\}\,M))\,M \quad (401)$$

*and for $\|\boldsymbol{x}_i - \boldsymbol{x}_i^*\| \leqslant \frac{1}{2\,\beta\,M}$ together with $\|\boldsymbol{x}_i - \boldsymbol{\xi}\| \leqslant \frac{1}{2\,\beta\,M}$ by*

$$\|\boldsymbol{x}_i\,-\,\boldsymbol{x}_i^*\|\,\leqslant\,2\,e\,(N-1)\,M\,\exp(-\,\beta\,\Delta_i)\,. \quad (402)$$

*Proof.* We compute the retrieval error which is just $\|f(\boldsymbol{\xi}) - \boldsymbol{x}_i\|$. From Lemma A4 we have

$$\|\boldsymbol{x}_i\,-\,f(\boldsymbol{\xi})\|\,\leqslant\,2\,\epsilon\,M\,, \quad (403)$$

From Eq. (179) we have

$$\epsilon\,=\,(N-1)\exp(-\,\beta\,(\Delta_i\,-\,2\,\max\{\|\boldsymbol{\xi}\,-\,\boldsymbol{x}_i\|, \|\boldsymbol{x}_i^*\,-\,\boldsymbol{x}_i\|\}\,M))\,. \quad (404)$$

For $\|\boldsymbol{x}_i - \boldsymbol{x}_i^*\| \leqslant \frac{1}{2\,\beta\,M}$ and $\|\boldsymbol{x}_i - \boldsymbol{\xi}\| \leqslant \frac{1}{2\,\beta\,M}$ Eq. (404) gives

$$\epsilon\,\leqslant\,e\,(N-1)\,M\,\exp(-\,\beta\,\Delta_i)\,. \quad (405)$$

$\square$

### A.1.7 LEARNING ASSOCIATIONS

We consider three cases of learning associations, i.e. three cases of how sets are associated. (i) Non of the sets is mapped in an associative space. The raw state pattern $\boldsymbol{r}_n$ is the state (query) pattern $\boldsymbol{\xi}_n$, i.e. $\boldsymbol{\xi}_n = \boldsymbol{r}_n$, and the raw stored pattern $\boldsymbol{y}_s$ is the stored pattern (key), i.e. $\boldsymbol{x}_s = \boldsymbol{y}_s$. (ii) Either one of the sets is mapped to the space of the other set or an association matrix is learned. (iia) The state patterns are equal to the raw patterns, i.e. $\boldsymbol{\xi}_n = \boldsymbol{r}_n$, and raw stored patterns are mapped via $\boldsymbol{W}$ to the space of the state patterns, i.e. $\boldsymbol{x}_s = \boldsymbol{W}\boldsymbol{y}_s$. (iib) The stored patterns are equal to the raw patterns, i.e. $\boldsymbol{x}_s = \boldsymbol{y}_s$, and raw state patterns are mapped via $\boldsymbol{W}$ to the space of the stored patterns, i.e. $\boldsymbol{\xi}_n = \boldsymbol{W}^T\boldsymbol{r}_n$. (iic) The matrix $\boldsymbol{W}$ is an association matrix. We will compute the derivative of the new state pattern with respect to $\boldsymbol{W}$, which is valid for all sub-cases (iib)–(iic). (iii) Both set of patterns are mapped in a common associative space. A raw state pattern $\boldsymbol{r}_n$ is mapped by $\boldsymbol{W}_Q$ to a state pattern (query) $\boldsymbol{\xi}_n$, that is $\boldsymbol{\xi}_n = \boldsymbol{W}_Q\boldsymbol{r}_n$. A raw stored pattern $\boldsymbol{y}_s$ is mapped via $\boldsymbol{W}_K$ to stored pattern (key) $\boldsymbol{x}_s$, that is $\boldsymbol{x}_s = \boldsymbol{W}_K\boldsymbol{y}_s$. We will compute the derivative of the new state pattern with respect to both $\boldsymbol{W}_Q$ and $\boldsymbol{W}_K$.

**A.1.7.1 Association of Raw Patterns – No Mapping in an Associative Space.** The sets are associated via their raw patterns, i.e. the raw state pattern $\boldsymbol{r}_n$ is the state (query) pattern $\boldsymbol{\xi}_n$, i.e. $\boldsymbol{\xi}_n = \boldsymbol{r}_n$, and raw stored pattern $\boldsymbol{y}_s$ is the stored pattern (key), i.e. $\boldsymbol{x}_s = \boldsymbol{y}_s$. There is no mapping in an associative space.

The update rule is

$$\boldsymbol{\xi}^{\text{new}}\,=\,\boldsymbol{X}\,\boldsymbol{p}\,, \quad (406)$$

where we used

$$\boldsymbol{p}\,=\,\text{softmax}(\beta\,\boldsymbol{X}^T\boldsymbol{\xi})\,. \quad (407)$$

The derivative with respect to $\boldsymbol{\xi}$ is

$$\frac{\partial\boldsymbol{\xi}^{\text{new}}}{\partial\boldsymbol{\xi}}\,=\,\beta\,\boldsymbol{X}\,\left(\text{diag}(\boldsymbol{p})-\boldsymbol{p}\boldsymbol{p}^T\right)\,\boldsymbol{X}^T \quad (408)$$

The derivative with respect to $\boldsymbol{X}$ is

$$\frac{\partial\boldsymbol{a}^T\boldsymbol{\xi}^{\text{new}}}{\partial\boldsymbol{X}}\,=\,\boldsymbol{a}\,\boldsymbol{p}^T\,+\,\beta\,\boldsymbol{X}\,\left(\text{diag}(\boldsymbol{p})-\boldsymbol{p}\boldsymbol{p}^T\right)\,(\boldsymbol{\xi}^T\boldsymbol{a})\,. \quad (409)$$

These derivatives allow to apply the chain rule if a Hopfield layer is integrated into a deep neural network.

**A.1.7.2 Learning an Association Matrix – Only One Set is Mapped in an Associative Space.**
Only one of the sets $\boldsymbol{R}$ or $\boldsymbol{Y}$ is mapped in the space of the patterns of the other set. Case (a): the state patterns are equal to the raw patterns $\boldsymbol{\xi}_n = \boldsymbol{r}_n$ and raw stored patterns are mapped via $\boldsymbol{W}$ to the space of the state patterns, i.e. $\boldsymbol{x}_s = \boldsymbol{W}\boldsymbol{y}_s$. Case (b): the stored patterns are equal to the raw patterns $\boldsymbol{x}_s = \boldsymbol{y}_s$ and raw state patterns are mapped via $\boldsymbol{W}$ to the space of the stored patterns, i.e. $\boldsymbol{\xi}_n = \boldsymbol{W}^T \boldsymbol{r}_n$. Case (c): the matrix $\boldsymbol{W}$ associates the sets $\boldsymbol{R}$ and $\boldsymbol{Y}$. This case also includes that $\boldsymbol{W}^T = \boldsymbol{W}_K^T \boldsymbol{W}_Q$, which is treated in next subsection. The next subsection focuses on a low rank approximation of $\boldsymbol{W}$ by defining the dimension $d_k$ of associative space and use the matrices $\boldsymbol{W}_K^T$ and $\boldsymbol{W}_Q$ to define $\boldsymbol{W}$, or equivalently to map $\boldsymbol{R}$ and $\boldsymbol{Y}$ into the associative space.

From a mathematical point of view all these case are equal as they lead to the same update rule. Therefore, we consider in the following Case (a) with $\boldsymbol{x}_s = \boldsymbol{W}\boldsymbol{y}_s$ and $\boldsymbol{\xi}_n = \boldsymbol{r}_n$. Still, the following formula are valid for all three cases (a)–(c).

The update rule is

$$\boldsymbol{\xi}^{\text{new}} = \boldsymbol{W}\,\boldsymbol{Y}\,\boldsymbol{p}\,, \tag{410}$$

where we used

$$\boldsymbol{p} = \text{softmax}(\beta\,\boldsymbol{Y}^T \boldsymbol{W}^T \boldsymbol{\xi})\,. \tag{411}$$

We consider the state (query) pattern $\boldsymbol{\xi}$ with result $\boldsymbol{\xi}^{\text{new}}$:

$$\boldsymbol{\xi}^{\text{new}} = \boldsymbol{W}\,\boldsymbol{Y}\,\boldsymbol{p} = \boldsymbol{W}\,\boldsymbol{Y}\,\text{softmax}(\beta\,\boldsymbol{Y}^T \boldsymbol{W}^T \boldsymbol{\xi}) \tag{412}$$

For multiple updates this update rule has to be used. However for a single update, or the last update we consider a simplified update rule.

Since new state vector $\boldsymbol{\xi}^{\text{new}}$ is projected by a weight matrix $\boldsymbol{W}_V$ to another vector, we consider the simplified update rule:

$$\boldsymbol{\xi}^{\text{new}} = \boldsymbol{Y}\,\boldsymbol{p} = \boldsymbol{Y}\,\text{softmax}(\beta\,\boldsymbol{Y}^T \boldsymbol{W}^T \boldsymbol{\xi}) \tag{413}$$

The derivative with respect to $\boldsymbol{W}$ is

$$\frac{\partial \boldsymbol{a}^T \boldsymbol{\xi}^{\text{new}}}{\partial \boldsymbol{W}} = \frac{\partial \boldsymbol{\xi}^{\text{new}}}{\partial \boldsymbol{W}} \frac{\partial \boldsymbol{a}^T \boldsymbol{\xi}^{\text{new}}}{\partial \boldsymbol{\xi}^{\text{new}}} = \frac{\partial \boldsymbol{\xi}^{\text{new}}}{\partial (\boldsymbol{W}^T \boldsymbol{\xi})} \frac{\partial (\boldsymbol{W}^T \boldsymbol{\xi})}{\partial \boldsymbol{W}} \frac{\partial \boldsymbol{a}^T \boldsymbol{\xi}^{\text{new}}}{\partial \boldsymbol{\xi}^{\text{new}}}\,. \tag{414}$$

$$\frac{\partial \boldsymbol{\xi}^{\text{new}}}{\partial (\boldsymbol{W}^T \boldsymbol{\xi})} = \beta\,\boldsymbol{Y}\,\left(\text{diag}(\boldsymbol{p}) - \boldsymbol{p}\boldsymbol{p}^T\right)\,\boldsymbol{Y}^T \tag{415}$$

$$\frac{\partial \boldsymbol{a}^T \boldsymbol{\xi}^{\text{new}}}{\partial \boldsymbol{\xi}^{\text{new}}} = \boldsymbol{a}\,. \tag{416}$$

We have the product of the 3-dimensional tensor $\frac{\partial (\boldsymbol{W}^T \boldsymbol{\xi})}{\partial \boldsymbol{W}}$ with the vector $\boldsymbol{a}$ which gives a 2-dimensional tensor, i.e. a matrix:

$$\frac{\partial (\boldsymbol{W}^T \boldsymbol{\xi})}{\partial \boldsymbol{W}} \frac{\partial \boldsymbol{a}^T \boldsymbol{\xi}^{\text{new}}}{\partial \boldsymbol{\xi}^{\text{new}}} = \frac{\partial (\boldsymbol{W}^T \boldsymbol{\xi})}{\partial \boldsymbol{W}}\,\boldsymbol{a} = \boldsymbol{\xi}^T \boldsymbol{a} \boldsymbol{I}\,. \tag{417}$$

$$\frac{\partial \boldsymbol{a}^T \boldsymbol{\xi}^{\text{new}}}{\partial \boldsymbol{W}} = \beta\,\boldsymbol{Y}\,\left(\text{diag}(\boldsymbol{p}) - \boldsymbol{p}\boldsymbol{p}^T\right)\,\boldsymbol{Y}^T (\boldsymbol{\xi}^T \boldsymbol{a}) = \text{J}\,(\boldsymbol{\xi}^T \boldsymbol{a})\,, \tag{418}$$

where J is the Jacobian of the update rule defined in Eq. (59).

To obtain the derivative of the full update rule Eq. (412) we have to add the term

$$\boldsymbol{a}\,\boldsymbol{p}^T \boldsymbol{Y}^T \tag{419}$$

and include the factor $\boldsymbol{W}$ to get

$$\frac{\partial \boldsymbol{a}^T \boldsymbol{\xi}^{\text{new}}}{\partial \boldsymbol{W}} = \boldsymbol{a}\,\boldsymbol{p}^T \boldsymbol{Y}^T + \beta\,\boldsymbol{W}\,\boldsymbol{Y}\,\left(\text{diag}(\boldsymbol{p}) - \boldsymbol{p}\boldsymbol{p}^T\right)\,\boldsymbol{Y}^T (\boldsymbol{\xi}^T \boldsymbol{a}) \tag{420}$$

$$= \boldsymbol{a}\,\boldsymbol{p}^T \boldsymbol{Y}^T + \boldsymbol{W}\,\text{J}\,(\boldsymbol{\xi}^T \boldsymbol{a})\,.$$

### A.1.7.3 Learning Two Association Mappings – Both Sets are Mapped in an Associative Space.

Both sets $\boldsymbol{R}$ and $\boldsymbol{Y}$ are mapped in an associative space. Every raw state pattern $\boldsymbol{r}_n$ is mapped via $\boldsymbol{W}_Q$ to a state pattern (query) $\boldsymbol{\xi}_n = \boldsymbol{W}_Q \boldsymbol{r}_n$. Every raw stored pattern $\boldsymbol{y}_s$ is mapped via $\boldsymbol{W}_K$ to a stored pattern (key) $\boldsymbol{x}_s = \boldsymbol{W}_K \boldsymbol{y}_s$. In the last subsection we considered a single matrix $\boldsymbol{W}$. For $\boldsymbol{W}^T = \boldsymbol{W}_K^T \boldsymbol{W}_Q$ we have the case of the last subsection. However in this subsection we are looking for a low rank approximation of $\boldsymbol{W}$. Toward this end we define the dimension $d_k$ of associative space and use the matrices $\boldsymbol{W}_K^T$ and $\boldsymbol{W}_Q$ to map to the associative space.

The update rule is

$$\boldsymbol{\xi}^{\text{new}} = \boldsymbol{X}\, \boldsymbol{p}\,, \tag{421}$$

where we used

$$\boldsymbol{p} = \text{softmax}(\beta\, \boldsymbol{X}^T \boldsymbol{\xi})\,. \tag{422}$$

We consider raw state patterns $\boldsymbol{r}_n$ that are mapped to state patterns $\boldsymbol{\xi}_n = \boldsymbol{W}_Q \boldsymbol{r}_n$ with $\boldsymbol{Q}^T = \boldsymbol{\Xi} = \boldsymbol{W}_Q \boldsymbol{R}$ and raw stored pattern $\boldsymbol{y}_s$ that are mapped to stored patterns $\boldsymbol{x}_s = \boldsymbol{W}_K \boldsymbol{y}_s$ with $\boldsymbol{K}^T = \boldsymbol{X} = \boldsymbol{W}_K \boldsymbol{Y}$. The update rule is

$$\boldsymbol{\xi}^{\text{new}} = \boldsymbol{W}_K\, \boldsymbol{Y}\, \boldsymbol{p} = \boldsymbol{W}_K\, \boldsymbol{Y}\, \text{softmax}(\beta\, \boldsymbol{Y}^T \boldsymbol{W}_K^T \boldsymbol{W}_Q\, \boldsymbol{r})\,. \tag{423}$$

Since new state vector $\boldsymbol{\xi}^{\text{new}}$ is projected by a weight matrix $\boldsymbol{W}_V$ to another vector, we consider the simplified update rule:

$$\boldsymbol{\xi}^{\text{new}} = \boldsymbol{Y}\, \boldsymbol{p} = \boldsymbol{Y}\, \text{softmax}(\beta\, \boldsymbol{Y}^T \boldsymbol{W}_K^T \boldsymbol{W}_Q\, \boldsymbol{r})\,. \tag{424}$$

For the simplified update rule, the vector $\boldsymbol{\xi}^{\text{new}}$ does not live in the associative space but in the space of raw stored pattern $\boldsymbol{y}$. However $\boldsymbol{W}_K$ would map it to the associative space.

•*Derivative with respect to $\boldsymbol{W}_Q$.* The derivative with respect to $\boldsymbol{W}_Q$ is

$$\frac{\partial \boldsymbol{a}^T \boldsymbol{\xi}^{\text{new}}}{\partial \boldsymbol{W}_Q} = \frac{\partial \boldsymbol{\xi}^{\text{new}}}{\partial \boldsymbol{W}_Q} \frac{\partial \boldsymbol{a}^T \boldsymbol{\xi}^{\text{new}}}{\partial \boldsymbol{\xi}^{\text{new}}} = \frac{\partial \boldsymbol{\xi}^{\text{new}}}{\partial (\boldsymbol{W}_Q\, \boldsymbol{r})} \frac{\partial (\boldsymbol{W}_Q\, \boldsymbol{r})}{\partial \boldsymbol{W}_Q} \frac{\partial \boldsymbol{a}^T \boldsymbol{\xi}^{\text{new}}}{\partial \boldsymbol{\xi}^{\text{new}}}\,. \tag{425}$$

$$\frac{\partial \boldsymbol{\xi}^{\text{new}}}{\partial (\boldsymbol{W}_Q\, \boldsymbol{r})} = \beta\, \boldsymbol{Y}\, \left(\text{diag}(\boldsymbol{p}) - \boldsymbol{p}\boldsymbol{p}^T\right)\, \boldsymbol{Y}^T \boldsymbol{W}_K^T \tag{426}$$

$$\frac{\partial \boldsymbol{a}^T \boldsymbol{\xi}^{\text{new}}}{\partial \boldsymbol{\xi}^{\text{new}}} = \boldsymbol{a}\,. \tag{427}$$

We have the product of the 3-dimensional tensor $\frac{\partial (\boldsymbol{W}_Q \boldsymbol{r})}{\partial \boldsymbol{W}_Q}$ with the vector $\boldsymbol{a}$ which gives a 2-dimensional tensor, i.e. a matrix:

$$\frac{\partial (\boldsymbol{W}_Q\, \boldsymbol{r})}{\partial \boldsymbol{W}_Q} \frac{\partial \boldsymbol{a}^T \boldsymbol{\xi}^{\text{new}}}{\partial \boldsymbol{\xi}^{\text{new}}} = \frac{\partial (\boldsymbol{W}_Q\, \boldsymbol{r})}{\partial \boldsymbol{W}_Q}\, \boldsymbol{a} = \boldsymbol{r}^T \boldsymbol{a}\, \boldsymbol{I}\,. \tag{428}$$

$$\frac{\partial \boldsymbol{a}^T \boldsymbol{\xi}^{\text{new}}}{\partial \boldsymbol{W}_Q} = \beta\, \boldsymbol{Y}\, \left(\text{diag}(\boldsymbol{p}) - \boldsymbol{p}\boldsymbol{p}^T\right)\, \boldsymbol{Y}^T\, \boldsymbol{W}_K^T (\boldsymbol{r}^T \boldsymbol{a}) = \text{J}\, \boldsymbol{W}_K^T (\boldsymbol{r}^T \boldsymbol{a})\,, \tag{429}$$

where J is the Jacobian of the update rule defined in Eq. (59).

To obtain the derivative of the full update rule Eq. (423) we have to include the factor $\boldsymbol{W}_K$, then get

$$\frac{\partial \boldsymbol{a}^T \boldsymbol{\xi}^{\text{new}}}{\partial \boldsymbol{W}_Q} = \beta\, \boldsymbol{W}_K\, \boldsymbol{Y}\, \left(\text{diag}(\boldsymbol{p}) - \boldsymbol{p}\boldsymbol{p}^T\right)\, \boldsymbol{Y}^T\, \boldsymbol{W}_K^T (\boldsymbol{r}^T \boldsymbol{a}) = \boldsymbol{W}_K\, \text{J}\, \boldsymbol{W}_K^T (\boldsymbol{r}^T \boldsymbol{a})\,. \tag{430}$$

•*Derivative with respect to $\boldsymbol{W}_K$.* The derivative with respect to $\boldsymbol{W}_K$ is

$$\frac{\partial \boldsymbol{a}^T \boldsymbol{\xi}^{\text{new}}}{\partial \boldsymbol{W}_K} = \frac{\partial \boldsymbol{\xi}^{\text{new}}}{\partial \boldsymbol{W}_K} \frac{\partial \boldsymbol{a}^T \boldsymbol{\xi}^{\text{new}}}{\partial \boldsymbol{\xi}^{\text{new}}} = \frac{\partial \boldsymbol{\xi}^{\text{new}}}{\partial (\boldsymbol{W}_K^T \boldsymbol{W}_Q\, \boldsymbol{r})} \frac{\partial (\boldsymbol{W}_K^T \boldsymbol{W}_Q\, \boldsymbol{r})}{\partial \boldsymbol{W}_K} \frac{\partial \boldsymbol{a}^T \boldsymbol{\xi}^{\text{new}}}{\partial \boldsymbol{\xi}^{\text{new}}}\,. \tag{431}$$

$$\frac{\partial \boldsymbol{\xi}^{\text{new}}}{\partial(\boldsymbol{W}_K^T \boldsymbol{W}_Q \, \boldsymbol{r})} = \beta \, \boldsymbol{Y} \, \left( \text{diag}(\boldsymbol{p}) - \boldsymbol{p}\boldsymbol{p}^T \right) \, \boldsymbol{Y}^T \tag{432}$$

$$\frac{\partial \boldsymbol{a}^T \boldsymbol{\xi}^{\text{new}}}{\partial \boldsymbol{\xi}^{\text{new}}} = \boldsymbol{a} \, . \tag{433}$$

We have the product of the 3-dimensional tensor $\frac{\partial(\boldsymbol{W}\boldsymbol{r})}{\partial \boldsymbol{W}_K}$ with the vector $\boldsymbol{a}$ which gives a 2-dimensional tensor, i.e. a matrix:

$$\frac{\partial(\boldsymbol{W}_K^T \boldsymbol{W}_Q \, \boldsymbol{r})}{\partial \boldsymbol{W}_K} \frac{\partial \boldsymbol{a}^T \boldsymbol{\xi}^{\text{new}}}{\partial \boldsymbol{\xi}^{\text{new}}} = \frac{\partial(\boldsymbol{W}_K^T \boldsymbol{W}_Q \, \boldsymbol{r})}{\partial \boldsymbol{W}_K} \, \boldsymbol{a} = \boldsymbol{W}_Q^T \boldsymbol{r}^T \boldsymbol{a} \, \boldsymbol{I} \, . \tag{434}$$

$$\frac{\partial \boldsymbol{a}^T \boldsymbol{\xi}^{\text{new}}}{\partial \boldsymbol{W}_K} = \beta \, \boldsymbol{Y} \, \left( \text{diag}(\boldsymbol{p}) - \boldsymbol{p}\boldsymbol{p}^T \right) \, \boldsymbol{Y}^T \, (\boldsymbol{W}_Q^T \boldsymbol{r}^T \boldsymbol{a}) = \text{J} \, (\boldsymbol{W}_Q^T \boldsymbol{r}^T \boldsymbol{a}) \, , \tag{435}$$

where J is the Jacobian of the update rule defined in Eq. (59).

To obtain the derivative of the full update rule Eq. (423) we have to add the term

$$\boldsymbol{a} \, \boldsymbol{p}^T \boldsymbol{Y}^T \tag{436}$$

and to include the factor $\boldsymbol{W}_K$, then get

$$\begin{aligned} \frac{\partial \boldsymbol{a}^T \boldsymbol{\xi}^{\text{new}}}{\partial \boldsymbol{W}_K} &= \boldsymbol{a} \, \boldsymbol{p}^T \boldsymbol{Y}^T + \beta \, \boldsymbol{W}_K \, \boldsymbol{Y} \, \left( \text{diag}(\boldsymbol{p}) - \boldsymbol{p}\boldsymbol{p}^T \right) \, \boldsymbol{Y}^T (\boldsymbol{W}_Q^T \boldsymbol{r}^T \boldsymbol{a}) \\ &= \boldsymbol{a} \, \boldsymbol{p}^T \boldsymbol{Y}^T + \boldsymbol{W}_K \, \text{J} \, (\boldsymbol{W}_Q^T \boldsymbol{r}^T \boldsymbol{a}) \, . \end{aligned} \tag{437}$$

## A.1.8   Infinite Many Patterns and Forgetting Patterns

In the next subsection we show how the new Hopfield networks can be used for auto-regressive tasks by causal masking. In the following subsection, we introduce forgetting to the new Hopfield networks by adding a negative value to the softmax which is larger if the pattern was observed more in the past.

**A.1.8.1   Infinite Many Patterns.**   The new Hopfield networks can be used for auto-regressive tasks, that is time series prediction and similar. Causal masking masks out the future by a large negative value in the softmax.

We assume to have infinite many stored patterns (keys) $\boldsymbol{x}_1, \boldsymbol{x}_2, \ldots$ that are represented by the infinite matrix

$$\boldsymbol{X} = (\boldsymbol{x}_1, \boldsymbol{x}_2, \ldots, ) \, . \tag{438}$$

The pattern index is now a time index, that is, we observe $\boldsymbol{x}_t$ at time $t$.

The pattern matrix at time $t$ is

$$\boldsymbol{X}_t = (\boldsymbol{x}_1, \boldsymbol{x}_2, \ldots, \boldsymbol{x}_t) \, . \tag{439}$$

The query at time $t$ is $\boldsymbol{\xi}_t$.

For $M_t = \max_{1 \leqslant i \leqslant t} \|\boldsymbol{x}_t\|$, the energy function at time $t$ is $\text{E}_t$

$$\text{E}_t = - \text{lse}(\beta, \boldsymbol{X}_t^T \boldsymbol{\xi}_t) + \frac{1}{2} \boldsymbol{\xi}_t^T \boldsymbol{\xi}_t + \beta^{-1} \ln t + \frac{1}{2} M_t^2 \tag{440}$$

$$= - \beta^{-1} \ln \left( \sum_{i=1}^{t} \exp(\beta \boldsymbol{x}_i^T \boldsymbol{\xi}_t) \right) + \frac{1}{2} \boldsymbol{\xi}_t^T \boldsymbol{\xi}_t + \beta^{-1} \ln t + \frac{1}{2} M_t^2 \, . \tag{441}$$

The update rule is

$$\boldsymbol{\xi}_t^{\text{new}} = \boldsymbol{X}_t \, \boldsymbol{p}_t = \boldsymbol{X}_t \, \text{softmax}(\beta \, \boldsymbol{X}_t^T \boldsymbol{\xi}_t) \, , \tag{442}$$

where we used

$$\boldsymbol{p}_t \;=\; \mathrm{softmax}(\beta\,\boldsymbol{X}_t^T\boldsymbol{\xi}_t)\,. \tag{443}$$

We can use an infinite pattern matrix with an infinite softmax when using causal masking. The pattern matrix at time $t$ is

$$\boldsymbol{X}_t \;=\; (\boldsymbol{x}_1, \boldsymbol{x}_2, \ldots, \boldsymbol{x}_t, -\alpha\boldsymbol{\xi}_t, -\alpha\boldsymbol{\xi}_t, \ldots)\,, \tag{444}$$

with the query $\boldsymbol{\xi}_t$ and $\alpha \to \infty$. The energy function at time $t$ is $\mathrm{E}_t$

$$\mathrm{E}_t \;=\; -\,\mathrm{lse}(\beta, \boldsymbol{X}_t^T\boldsymbol{\xi}_t) \;+\; \frac{1}{2}\boldsymbol{\xi}_t^T\boldsymbol{\xi}_t \;+\; \beta^{-1}\ln t \;+\; \frac{1}{2}M_t^2 \tag{445}$$

$$=\; -\,\beta^{-1}\ln\left(\sum_{i=1}^{t}\exp(\beta\boldsymbol{x}_i^T\boldsymbol{\xi}_t) \;+\; \sum_{i=t+1}^{\lfloor\alpha\rfloor}\exp(-\beta\alpha\|\boldsymbol{\xi}_t\|^2)\right) \;+\; \frac{1}{2}\boldsymbol{\xi}_t^T\boldsymbol{\xi}_t \;+ \tag{446}$$

$$\beta^{-1}\ln t \;+\; \frac{1}{2}M_t^2 \,.$$

For $\alpha \to \infty$ and $\|\boldsymbol{\xi}_t\| > 0$ this becomes

$$\mathrm{E}_t \;=\; -\,\mathrm{lse}(\beta, \boldsymbol{X}_t^T\boldsymbol{\xi}_t) \;+\; \frac{1}{2}\boldsymbol{\xi}_t^T\boldsymbol{\xi}_t \;+\; \beta^{-1}\ln t \;+\; \frac{1}{2}M_t^2 \tag{447}$$

$$=\; -\,\beta^{-1}\ln\left(\sum_{i=1}^{t}\exp(\beta\boldsymbol{x}_i^T\boldsymbol{\xi}_t)\right) \;+\; \frac{1}{2}\boldsymbol{\xi}_t^T\boldsymbol{\xi}_t \;+\; \beta^{-1}\ln t \;+\; \frac{1}{2}M_t^2 \,. \tag{448}$$

**A.1.8.2 Forgetting Patterns.** We introduce forgetting to the new Hopfield networks by adding a negative value in the softmax which increases with patterns that are more in the past.

We assume to have infinite many patterns $\boldsymbol{x}_1, \boldsymbol{x}_2, \ldots$ that are represented by the infinite matrix

$$\boldsymbol{X} \;=\; (\boldsymbol{x}_1, \boldsymbol{x}_2, \ldots,)\,. \tag{449}$$

The pattern index is now a time index, that is, we observe $\boldsymbol{x}_t$ at time $t$.

The pattern matrix at time $t$ is

$$\boldsymbol{X}_t \;=\; (\boldsymbol{x}_1, \boldsymbol{x}_2, \ldots, \boldsymbol{x}_t)\,. \tag{450}$$

The query at time $t$ is $\boldsymbol{\xi}_t$.

The energy function with forgetting parameter $\gamma$ at time $t$ is $\mathrm{E}_t$

$$\mathrm{E}_t \;=\; -\,\mathrm{lse}(\beta, \boldsymbol{X}_t^T\boldsymbol{\xi}_t \;-\; \gamma(t-1, t-2, \ldots, 0)^T) \;+\; \frac{1}{2}\boldsymbol{\xi}_t^T\boldsymbol{\xi}_t \;+\; \beta^{-1}\ln t \;+\; \frac{1}{2}M_t^2 \tag{451}$$

$$=\; -\,\beta^{-1}\ln\left(\sum_{i=1}^{T}\exp(\beta\boldsymbol{x}_i^T\boldsymbol{\xi}_t \;-\; \gamma(t-i))\right) \;+\; \frac{1}{2}\boldsymbol{\xi}_t^T\boldsymbol{\xi}_t \;+\; \beta^{-1}\ln t \;+\; \frac{1}{2}M_t^2 \,. \tag{452}$$

The update rule is

$$\boldsymbol{\xi}_t^{\mathrm{new}} \;=\; \boldsymbol{X}_t\,\boldsymbol{p}_t \;=\; \boldsymbol{X}_t\,\mathrm{softmax}(\beta\boldsymbol{X}_t^T\boldsymbol{\xi}_t)\,, \tag{453}$$

where we used

$$\boldsymbol{p}_t \;=\; \mathrm{softmax}(\beta\boldsymbol{X}_t^T\boldsymbol{\xi}_t)\,. \tag{454}$$

A.1.9   NUMBER OF SPURIOUS STATES

The energy E is defined as

$$\mathrm{E} \;=\; -\,\mathrm{lse}(\beta, \boldsymbol{X}^T\boldsymbol{\xi}) \;+\; \frac{1}{2}\boldsymbol{\xi}^T\boldsymbol{\xi} \;+\; \beta^{-1}\ln N \;+\; \frac{1}{2}M^2 \tag{455}$$

$$=\; -\,\beta^{-1}\ln\left(\sum_{i=1}^{N}\exp(\beta\boldsymbol{x}_i^T\boldsymbol{\xi})\right) \;+\; \beta^{-1}\ln N \;+\; \frac{1}{2}\boldsymbol{\xi}^T\boldsymbol{\xi} \;+\; \frac{1}{2}M^2 \,. \tag{456}$$

Since the negative exponential function is strict monotonic decreasing, $\exp(-\mathrm{E})$ has minima, where E has maxima, and has maxima, where as has minima E.

$$\exp(-\mathrm{E}) = \exp(\mathrm{lse}(\beta, \boldsymbol{X}^T\boldsymbol{\xi})) \; \exp(-\frac{1}{2}\boldsymbol{\xi}^T\boldsymbol{\xi}) \, C \tag{457}$$

$$= \left( \sum_{i=1}^{N} \exp(\beta\boldsymbol{x}_i^T\boldsymbol{\xi}) \right)^{\beta^{-1}} \exp(-\frac{1}{2}\boldsymbol{\xi}^T\boldsymbol{\xi}) \, C$$

$$= \left( \sum_{i=1}^{N} \exp(\beta\boldsymbol{x}_i^T\boldsymbol{\xi}) \right)^{\beta^{-1}} \left( \exp(-\beta\,\frac{1}{2}\boldsymbol{\xi}^T\boldsymbol{\xi}) \right)^{\beta^{-1}} C$$

$$= \left( \sum_{i=1}^{N} \exp(\beta\,(\boldsymbol{x}_i^T\boldsymbol{\xi} - \frac{1}{2}\boldsymbol{\xi}^T\boldsymbol{\xi})) \right)^{\beta^{-1}} C$$

$$= \left( \sum_{i=1}^{N} \exp(\frac{1}{2}\,\beta\,\boldsymbol{x}_i^T\boldsymbol{x}_i - \frac{1}{2}\,\beta\,(\boldsymbol{\xi} - \boldsymbol{x}_i)^T(\boldsymbol{\xi} - \boldsymbol{x}_i)) \right)^{\beta^{-1}} C$$

$$= \left( \sum_{i=1}^{N} \lambda(\boldsymbol{x}_i, \beta) \, G(\boldsymbol{\xi}; \boldsymbol{x}_i, \beta^{-1}\,\boldsymbol{I}) \right)^{\beta^{-1}} C \; ,$$

where $C$ is a positive constant, $\lambda(\boldsymbol{x}_i, \beta) = \exp(\frac{1}{2}\beta\boldsymbol{x}_i^T\boldsymbol{x}_i)$ and $G(\boldsymbol{\xi}; \boldsymbol{x}_i, \beta^{-1}\boldsymbol{I})$ is the Gaussian with mean $\boldsymbol{x}_i$ and covariance matrix $\beta^{-1}\boldsymbol{I}$.

Since $C$ is a positive constant and $x^{\beta^{-1}} = \exp(\beta^{-1}\ln x)$ is strict monotonic for positive $x$, the minima of E are the maxima of

$$\sum_{i=1}^{N} \lambda(\boldsymbol{x}_i, \beta) \, G(\boldsymbol{\xi}; \boldsymbol{x}_i, \beta^{-1}\,\boldsymbol{I}) \; . \tag{458}$$

In Carreira-Perpiñán & Williams (2003) it was shown that Eq. (458) can have more than $N$ modes, that is, more than $N$ maxima.

### A.2 PROPERTIES OF SOFTMAX, LOG-SUM-EXPONENTIAL, LEGENDRE TRANSFORM, LAMBERT W FUNCTION

For $\beta > 0$, the *softmax* is defined as

**Definition A1** (Softmax).

$$\boldsymbol{p} = \mathrm{softmax}(\beta\boldsymbol{x}) \tag{459}$$

$$p_i = [\mathrm{softmax}(\beta\boldsymbol{x})]_i = \frac{\exp(\beta x_i)}{\sum_k \exp(\beta x_k)} \; . \tag{460}$$

We also need the *log-sum-exp function* (lse), defined as

**Definition A2** (Log-Sum-Exp Function).

$$\mathrm{lse}(\beta, \boldsymbol{x}) = \beta^{-1}\ln\left( \sum_{i=1}^{N} \exp(\beta x_i) \right) \; . \tag{461}$$

We can formulate the lse in another base:

$$\beta_a \;=\; \frac{\beta}{\ln a} \;,\tag{462}$$

$$\text{lse}(\beta, \boldsymbol{x}) \;=\; \beta^{-1} \ln \left( \sum_{i=1}^{N} \exp(\beta\, x_i) \right)\tag{463}$$

$$=\; (\beta_a\, \ln a)^{-1} \ln \left( \sum_{i=1}^{N} \exp(\beta_a\, \ln a\, x_i) \right)$$

$$=\; (\beta_a)^{-1} \log_a \left( \sum_{i=1}^{N} a^{\beta_a\, x_i} \right) \;.$$

In particular, the base $a = 2$ can be used to speed up computations.

Next, we give the relation between the softmax and the lse function.

**Lemma A18.** *The softmax is the gradient of the* lse*:*

$$\text{softmax}(\beta\boldsymbol{x}) \;=\; \nabla_{\boldsymbol{x}} \text{lse}(\beta, \boldsymbol{x}) \;.\tag{464}$$

In the next lemma we report some important properties of the lse function.

**Lemma A19.** *We define*

$$\text{L} \;:=\; \boldsymbol{z}^T \boldsymbol{x} \;-\; \beta^{-1} \sum_{i=1}^{N} z_i \ln z_i\tag{465}$$

*with* $\text{L} \geq \boldsymbol{z}^T \boldsymbol{x}$*. The* lse *is the maximum of* L *on the $N$-dimensional simplex $D$ with $D = \{\boldsymbol{z} \mid \sum_i z_i = 1, 0 \leqslant z_i\}$:*

$$\text{lse}(\beta, \boldsymbol{x}) \;=\; \max_{\boldsymbol{z} \in D} \boldsymbol{z}^T \boldsymbol{x} \;-\; \beta^{-1} \sum_{i=1}^{N} z_i \ln z_i \;.\tag{466}$$

*The softmax $\boldsymbol{p} = \text{softmax}(\beta\boldsymbol{x})$ is the argument of the maximum of* L *on the $N$-dimensional simplex $D$ with $D = \{\boldsymbol{z} \mid \sum_i z_i = 1, 0 \leqslant z_i\}$:*

$$\boldsymbol{p} \;=\; \text{softmax}(\beta\boldsymbol{x}) \;=\; \arg\max_{\boldsymbol{z} \in D} \boldsymbol{z}^T \boldsymbol{x} \;-\; \beta^{-1} \sum_{i=1}^{N} z_i \ln z_i \;.\tag{467}$$

*Proof.* Eq. (466) is obtained from Equation (8) in Gao & Pavel (2017) and Eq. (467) from Equation (11) in Gao & Pavel (2017). □

From a physical point of view, the lse function represents the "free energy" in statistical thermodynamics (Gao & Pavel, 2017).

Next we consider the Jacobian of the softmax and its properties.

**Lemma A20.** *The Jacobian $\text{J}_s$ of the softmax $\boldsymbol{p} = \text{softmax}(\beta\boldsymbol{x})$ is*

$$\text{J}_s \;=\; \frac{\partial \text{softmax}(\beta\boldsymbol{x})}{\partial \boldsymbol{x}} \;=\; \beta \left( \text{diag}(\boldsymbol{p}) - \boldsymbol{p}\boldsymbol{p}^T \right) \;,\tag{468}$$

*which gives the elements*

$$[\text{J}_s]_{ij} \;=\; \begin{cases} \beta p_i (1 - p_i) & \text{for } i = j \\ -\beta p_i p_j & \text{for } i \neq j \end{cases} \;.\tag{469}$$

Next we show that $\text{J}_s$ has eigenvalue $0$.

**Lemma A21.** *The Jacobian $\text{J}_s$ of the softmax function $\boldsymbol{p} = \text{softmax}(\beta\boldsymbol{x})$ has a zero eigenvalue with eigenvector $\mathbf{1}$.*

*Proof.*

$$[J_s \mathbf{1}]_i \;=\; \beta \left( p_i(1 - p_i) \;-\; \sum_{j, j \neq i} p_i p_j \right) \;=\; \beta\, p_i (1 - \sum_j p_j) \;=\; 0 \,. \tag{470}$$

$\square$

Next we show that $0$ is the smallest eigenvalue of $J_s$, therefore $J_s$ is positive semi-definite but not (strict) positive definite.

**Lemma A22.** *The Jacobian $J_s$ of the softmax $\boldsymbol{p} = \mathrm{softmax}(\beta \boldsymbol{\xi})$ is symmetric and positive semi-definite.*

*Proof.* For an arbitrary $\boldsymbol{z}$, we have

$$\boldsymbol{z}^T \left( \mathrm{diag}(\boldsymbol{p}) - \boldsymbol{p}\boldsymbol{p}^T \right) \boldsymbol{z} \;=\; \sum_i p_i z_i^2 - \left( \sum_i p_i z_i \right)^2 \tag{471}$$

$$= \; \left( \sum_i p_i z_i^2 \right) \left( \sum_i p_i \right) - \left( \sum_i p_i z_i \right)^2 \;\geq\; 0 \,.$$

The last inequality hold true because the Cauchy-Schwarz inequality says $(\boldsymbol{a}^T \boldsymbol{a})(\boldsymbol{b}^T \boldsymbol{b}) \geq (\boldsymbol{a}^T \boldsymbol{b})^2$, which is the last inequality with $a_i = z_i \sqrt{p_i}$ and $b_i = \sqrt{p_i}$. Consequently $\left( \mathrm{diag}(\boldsymbol{p}) - \boldsymbol{p}\boldsymbol{p}^T \right)$ is positive semi-definite.

Alternatively $\sum_i p_i z_i^2 - \left( \sum_i p_i z_i \right)^2$ can be viewed as the expected second moment minus the mean squared which gives the variance that is larger equal to zero.

The Jacobian is $0 < \beta$ times a positive semi-definite matrix, which is a positive semi-definite matrix. $\square$

Moreover, the softmax is a monotonic map, as described in the next lemma.

**Lemma A23.** *The softmax $\mathrm{softmax}(\beta \boldsymbol{x})$ is monotone for $\beta > 0$, that is,*

$$\left( \mathrm{softmax}(\beta \boldsymbol{x}) - \mathrm{softmax}(\beta \boldsymbol{x}') \right)^T (\boldsymbol{x} - \boldsymbol{x}') \;\geq\; 0 \,. \tag{472}$$

*Proof.* We use the version of mean value theorem Lemma A32 with the symmetric matrix $J_s^m = \int_0^1 J_s(\lambda \boldsymbol{x} + (1 - \lambda)\boldsymbol{x}')\, \mathrm{d}\lambda$:

$$\mathrm{softmax}(\boldsymbol{x}) - \mathrm{softmax}(\boldsymbol{x}') \;=\; J_s^m\, (\boldsymbol{x} - \boldsymbol{x}') \,. \tag{473}$$

Therefore

$$\left( \mathrm{softmax}(\boldsymbol{x}) - \mathrm{softmax}(\boldsymbol{x}') \right)^T (\boldsymbol{x} - \boldsymbol{x}') \;=\; (\boldsymbol{x} - \boldsymbol{x}')^T J_s^m\, (\boldsymbol{x} - \boldsymbol{x}') \;\geq\; 0 \,, \tag{474}$$

since $J_s^m$ is positive semi-definite. For all $\lambda$ the Jacobians $J_s(\lambda \boldsymbol{x} + (1 - \lambda)\boldsymbol{x}')$ are positive semi-definite according to Lemma A22. Since

$$\boldsymbol{x}^T J_s^m \boldsymbol{x} \;=\; \int_0^1 \boldsymbol{x}^T J_s(\lambda \boldsymbol{x} + (1 - \lambda)\boldsymbol{x}')\, \boldsymbol{x}\, \mathrm{d}\lambda \;\geq\; 0 \tag{475}$$

is an integral over positive values for every $\boldsymbol{x}$, $J_s^m$ is positive semi-definite, too. $\square$

Next we give upper bounds on the norm of $J_s$.

**Lemma A24.** *For a softmax $\boldsymbol{p} = \mathrm{softmax}(\beta \boldsymbol{x})$ with $m = \max_i p_i(1 - p_i)$, the spectral norm of the Jacobian $J_s$ of the softmax is bounded:*

$$\|J_s\|_2 \;\leqslant\; 2\, m\, \beta \,, \tag{476}$$

$$\|J_s\|_1 \;\leqslant\; 2\, m\, \beta \,, \tag{477}$$

$$\|J_s\|_\infty \;\leqslant\; 2\, m\, \beta \,. \tag{478}$$

*In particular everywhere holds*

$$\|\mathrm{J}_s\|_2 \ \leqslant \ \frac{1}{2}\,\beta\,. \tag{479}$$

*If $p_{\max} = \max_i p_i \geq 1 - \epsilon \geq 0.5$, then for the spectral norm of the Jacobian holds*

$$\|\mathrm{J}_s\|_2 \ \leqslant \ 2\,\epsilon\,\beta \ - \ 2\,\epsilon^2\,\beta \ < \ 2\,\epsilon\,\beta\,. \tag{480}$$

*Proof.* We consider the maximum absolute column sum norm

$$\|\boldsymbol{A}\|_1 \ = \ \max_j \sum_i |a_{ij}| \tag{481}$$

and the maximum absolute row sum norm

$$\|\boldsymbol{A}\|_\infty \ = \ \max_i \sum_j |a_{ij}|\,. \tag{482}$$

We have for $\boldsymbol{A} = \mathrm{J}_s = \beta\left(\mathrm{diag}(\boldsymbol{p}) - \boldsymbol{p}\boldsymbol{p}^T\right)$

$$\sum_j |a_{ij}| \ = \ \beta\left(p_i(1-p_i) + \sum_{j,j\neq i} p_i p_j\right) \ = \ \beta\,p_i\,(1 \ - \ 2p_i \ + \ \sum_j p_j) \tag{483}$$

$$= \ 2\,\beta\,p_i\,(1-p_i) \ \leqslant \ 2\,m\,\beta\,,$$

$$\sum_i |a_{ij}| \ = \ \beta\left(p_j\,(1-p_j) + \sum_{i,i\neq j} p_j p_i\right) \ = \ \beta\,p_j\,(1 \ - \ 2p_j \ + \ \sum_i p_i) \tag{484}$$

$$= \ 2\,\beta\,p_j\,(1-p_j) \ \leqslant \ 2\,m\,\beta\,.$$

Therefore, we have

$$\|\mathrm{J}_s\|_1 \ \leqslant \ 2\,m\,\beta\,, \tag{485}$$
$$\|\mathrm{J}_s\|_\infty \ \leqslant \ 2\,m\,\beta\,, \tag{486}$$

$$\|\mathrm{J}_s\|_2 \ \leqslant \ \sqrt{\|\mathrm{J}_s\|_1\|\mathrm{J}_s\|_\infty} \ \leqslant \ 2\,m\,\beta\,. \tag{487}$$

The last inequality is a direct consequence of Hölder's inequality.

For $0 \leqslant p_i \leqslant 1$, we have $p_i(1-p_i) \leqslant 0.25$. Therefore, $m \leqslant 0.25$ for all values of $p_i$.

If $p_{\max} \geq 1 - \epsilon \geq 0.5$ ($\epsilon \leqslant 0.5$), then $1 - p_{\max} \leqslant \epsilon$ and for $p_i \neq p_{\max}$ $p_i \leqslant \epsilon$. The derivative $\partial x(1-x)/\partial x = 1 - 2x > 0$ for $x < 0.5$, therefore $x(1-x)$ increases with $x$ for $x < 0.5$. Using $x = 1 - p_{\max}$ and for $p_i \neq p_{\max}$ $x = p_i$, we obtain $p_i(1-p_i) \leqslant \epsilon(1-\epsilon)$ for all $i$. Consequently, we have $m \leqslant \epsilon(1-\epsilon)$. $\qquad\square$

Using the bounds on the norm of the Jacobian, we give some Lipschitz properties of the softmax function.

**Lemma A25.** *The softmax function $\boldsymbol{p} = \mathrm{softmax}(\beta\boldsymbol{x})$ is $(\beta/2)$-Lipschitz. The softmax function $\boldsymbol{p} = \mathrm{softmax}(\beta\boldsymbol{x})$ is $(2\beta m)$-Lipschitz in a convex environment $U$ for which $m = \max_{\boldsymbol{x}\in U}\max_i p_i(1 - p_i)$. For $p_{\max} = \min_{\boldsymbol{x}\in U}\max_i p_i = 1-\epsilon$, the softmax function $\boldsymbol{p} = \mathrm{softmax}(\beta\boldsymbol{x})$ is $(2\beta\epsilon)$-Lipschitz. For $\beta < 2m$, the softmax $\boldsymbol{p} = \mathrm{softmax}(\beta\boldsymbol{x})$ is contractive in $U$ on which $m$ is defined.*

*Proof.* The version of mean value theorem Lemma A32 states for the symmetric matrix $\mathrm{J}_s^m = \int_0^1 \mathrm{J}(\lambda\boldsymbol{x} + (1-\lambda)\boldsymbol{x}')\,\mathrm{d}\lambda$:

$$\mathrm{softmax}(\boldsymbol{x}) \ - \ \mathrm{softmax}(\boldsymbol{x}') \ = \ \mathrm{J}_s^m\,(\boldsymbol{x} \ - \ \boldsymbol{x}')\,. \tag{488}$$

According to Lemma A24 for all $\tilde{\boldsymbol{x}} = \lambda\boldsymbol{x} + (1-\lambda)\boldsymbol{x}'$

$$\|\mathrm{J}_s(\tilde{\boldsymbol{x}})\|_2 \ \leqslant \ 2\,\tilde{m}\,\beta\,, \tag{489}$$

where $\tilde{m} = \max_i \tilde{p}_i(1 - \tilde{p}_i)$. Since $\boldsymbol{x} \in U$ and $\boldsymbol{x}' \in U$ we have $\tilde{\boldsymbol{x}} \in U$, since $U$ is convex. For $m = \max_{\boldsymbol{x} \in U} \max_i p_i(1 - p_i)$ we have $\tilde{m} \leqslant m$ for all $\tilde{m}$. Therefore, we have

$$\left\| \mathrm{J}_s(\tilde{\boldsymbol{x}}) \right\|_2 \ \leqslant \ 2\,m\,\beta \tag{490}$$

which also holds for the mean:

$$\left\| \mathrm{J}_s^m \right\|_2 \ \leqslant \ 2\,m\,\beta \,. \tag{491}$$

Therefore,

$$\left\| \mathrm{softmax}(\boldsymbol{x}) \ - \ \mathrm{softmax}(\boldsymbol{x}') \right\| \ \leqslant \ \left\| \mathrm{J}_s^m \right\|_2 \left\| \boldsymbol{x} \ - \ \boldsymbol{x}' \right\| \ \leqslant \ 2\,m\,\beta \left\| \boldsymbol{x} \ - \ \boldsymbol{x}' \right\| \,. \tag{492}$$

From Lemma A24 we know $m \leqslant 1/4$ globally. For $p_{\max} = \min_{\boldsymbol{x} \in U} \max_i p_i = 1 - \epsilon$ we have according to Lemma A24: $m \leqslant \epsilon$. $\qquad\square$

For completeness we present a result about cocoercivity of the softmax:

**Lemma A26.** *For $m = \max_{\boldsymbol{x} \in U} \max_i p_i(1 - p_i)$, softmax function $\boldsymbol{p} = \mathrm{softmax}(\beta\boldsymbol{x})$ is $1/(2m\beta)$-cocoercive in $U$, that is,*

$$\left( \mathrm{softmax}(\boldsymbol{x}) \ - \ \mathrm{softmax}(\boldsymbol{x}') \right)^T \left( \boldsymbol{x} \ - \ \boldsymbol{x}' \right) \ \geqslant \ \frac{1}{2\,m\,\beta} \| \mathrm{softmax}(\boldsymbol{x}) \ - \ \mathrm{softmax}(\boldsymbol{x}') \|. \tag{493}$$

*In particular the softmax function $\boldsymbol{p} = \mathrm{softmax}(\beta\boldsymbol{x})$ is $(2/\beta)$-cocoercive everywhere. With $p_{\max} = \min_{\boldsymbol{x} \in U} \max_i p_i = 1 - \epsilon$, the softmax function $\boldsymbol{p} = \mathrm{softmax}(\beta\boldsymbol{x})$ is $1/(2\beta\epsilon)$-cocoercive in $U$.*

*Proof.* We apply the Baillon-Haddad theorem (e.g. Theorem 1 in Gao & Pavel (2017)) together with Lemma A25. $\qquad\square$

Finally, we introduce the Legendre transform and use it to describe further properties of the lse. We start with the definition of the convex conjugate.

**Definition A3** (Convex Conjugate). *The* Convex Conjugate (Legendre-Fenchel transform) *of a function $f$ from a Hilbert Space $X$ to $[-\infty, \infty]$ is $f^*$ which is defined as*

$$f^*(\boldsymbol{x}^*) \ = \ \sup_{\boldsymbol{x} \in X} (\boldsymbol{x}^T \boldsymbol{x}^* \ - \ f(\boldsymbol{x})) \,, \quad \boldsymbol{x}^* \in X \tag{494}$$

See page 219 Def. 13.1 in Bauschke & Combettes (2017) and page 134 in Garling (2017). Next we define the Legendre transform, which is a more restrictive version of the convex conjugate.

**Definition A4** (Legendre Transform). *The* Legendre transform *of a convex function $f$ from a convex set $X \subset \mathbb{R}^n$ to $\mathbb{R}$ ($f : X \to \mathbb{R}$) is $f^*$, which is defined as*

$$f^*(\boldsymbol{x}^*) \ = \ \sup_{\boldsymbol{x} \in X} (\boldsymbol{x}^T \boldsymbol{x}^* \ - \ f(\boldsymbol{x})) \,, \quad \boldsymbol{x}^* \in X^* \,, \tag{495}$$

$$X^* \ = \ \left\{ \boldsymbol{x}^* \in \mathbb{R}^n \mid \sup_{\boldsymbol{x} \in X} (\boldsymbol{x}^T \boldsymbol{x}^* \ - \ f(\boldsymbol{x})) < \infty \right\} \,. \tag{496}$$

See page 91 in Boyd & Vandenberghe (2009).

**Definition A5** (Epi-Sum). *Let $f$ and $g$ be two functions from $X$ to $(-\infty, \infty]$, then the infimal convolution (or epi-sum) of $f$ and $g$ is*

$$f \square g : X \to [-\infty, \infty] \,, \ \boldsymbol{x} \mapsto \inf_{\boldsymbol{y} \in X} \left( f(\boldsymbol{y}) + g(\boldsymbol{x} - \boldsymbol{y}) \right) \tag{497}$$

See Def. 12.1 in Bauschke & Combettes (2017).

**Lemma A27.** *Let $f$ and $g$ be functions from $X$ to $(-\infty, \infty]$. Then the following hold:*

    *1. Convex Conjugate of norm squared*

$$\left( \frac{1}{2} \|\cdot\|^2 \right)^* \ = \ \frac{1}{2} \|\cdot\|^2 \,. \tag{498}$$

2. *Convex Conjugate of a function multiplied by scalar $0 < \alpha \in \mathbb{R}$*

$$(\alpha\, f)^* \;=\; \alpha\; f^*(./\alpha)\,. \tag{499}$$

3. *Convex Conjugate of the sum of a function and a scalar $\beta \in \mathbb{R}$*

$$(f \,+\, \beta)^* \;=\; f^* \,-\, \beta\,. \tag{500}$$

4. *Convex Conjugate of affine transformation of the arguments. Let $\boldsymbol{A}$ be a non-singular matrix and $\boldsymbol{b}$ a vector*

$$(f\,(\boldsymbol{Ax} \,+\, \boldsymbol{b}))^* \;=\; f^*\left(\boldsymbol{A}^{-T}\boldsymbol{x}^*\right) \,-\, \boldsymbol{b}^T\boldsymbol{A}^{-T}\boldsymbol{x}^*\,. \tag{501}$$

5. *Convex Conjugate of epi-sums*

$$(f\square g)^* \;=\; f^* + g^*\,. \tag{502}$$

*Proof.*    1. Since $h(t) := \frac{t^2}{2}$ is a non-negative convex function and $h(t) = 0 \iff t = 0$ we have because of Proposition 11.3.3 in Garling (2017) that $h\left(\|x\|\right)^* = h^*\left(\|x^*\|\right)$. Additionally, by example (a) on page 137 we get for $1 < p < \infty$ and $\frac{1}{p} + \frac{1}{q} = 1$ that $\left(\frac{|t|^p}{p}\right)^* = \frac{|t^*|^q}{q}$. Putting all together we get the desired result. The same result can also be deduced from page 222 Example 13.6 in Bauschke & Combettes (2017).

2. Follows immediately from the definition since

$$\alpha f^*\left(\frac{\boldsymbol{x}^*}{\alpha}\right) = \alpha \sup_{\boldsymbol{x}\in X}\left(\boldsymbol{x}^T\frac{\boldsymbol{x}^*}{\alpha} \,-\, f(\boldsymbol{x})\right) = \sup_{\boldsymbol{x}\in X}(\boldsymbol{x}^T\boldsymbol{x}^* - \alpha f(\boldsymbol{x})) = (\alpha f)^*(\boldsymbol{x}^*)$$

3. $(f + \beta)^* := \sup_{\boldsymbol{x}\in X}\left(\boldsymbol{x}^T\boldsymbol{x}^* - f(\boldsymbol{x}) - \beta\right) =: f^* - \beta$

4.

$$\begin{aligned}
(f\,(\boldsymbol{Ax} + \boldsymbol{b}))^*\,(\boldsymbol{x}^*) &= \sup_{\boldsymbol{x}\in X}\left(\boldsymbol{x}^T\boldsymbol{x}^* - f\,(\boldsymbol{Ax} + \boldsymbol{b})\right)\\
&= \sup_{\boldsymbol{x}\in X}\left((\boldsymbol{Ax} + \boldsymbol{b})^T\,\boldsymbol{A}^{-T}\boldsymbol{x}^* - f\,(\boldsymbol{Ax} + \boldsymbol{b})\right) - \boldsymbol{b}^T\boldsymbol{A}^{-T}\boldsymbol{x}^*\\
&= \sup_{\boldsymbol{y}\in X}\left(\boldsymbol{y}^T\boldsymbol{A}^{-T}\boldsymbol{x}^* - f\,(\boldsymbol{y})\right) - \boldsymbol{b}^T\boldsymbol{A}^{-T}\boldsymbol{x}^*\\
&= f^*\left(\boldsymbol{A}^{-T}\boldsymbol{x}^*\right) - \boldsymbol{b}^T\boldsymbol{A}^{-T}\boldsymbol{x}^*
\end{aligned}$$

5. From Proposition 13.24 (i) in Bauschke & Combettes (2017) and Proposition 11.4.2 in Garling (2017) we get

$$\begin{aligned}
(f\square g)^*\,(\boldsymbol{x}^*) &= \sup_{\boldsymbol{x}\in X}\left(\boldsymbol{x}^T\boldsymbol{x}^* - \inf_{\boldsymbol{y}\in X}\left(f(\boldsymbol{y}) - g(\boldsymbol{x} - \boldsymbol{y})\right)\right)\\
&= \sup_{\boldsymbol{x},\boldsymbol{y}\in X}\left(\boldsymbol{x}^T\boldsymbol{x}^* - f(\boldsymbol{y}) - g(\boldsymbol{x} - \boldsymbol{y})\right)\\
&= \sup_{\boldsymbol{x},\boldsymbol{y}\in X}\left(\left(\boldsymbol{y}^T\boldsymbol{x}^* - f(\boldsymbol{y})\right) + \left((\boldsymbol{x} - \boldsymbol{y})^T\,\boldsymbol{x}^* - g(\boldsymbol{x} - \boldsymbol{y})\right)\right)\\
&= f^*(\boldsymbol{x}^*) + g^*(\boldsymbol{x}^*)
\end{aligned}$$

$\square$

**Lemma A28.** *The Legendre transform of the* lse *is the negative entropy function, restricted to the probability simplex and vice versa. For the log-sum exponential*

$$f(\boldsymbol{x}) \;=\; \ln\left(\sum_{i=1}^{n}\exp(x_i)\right)\,, \tag{503}$$

the Legendre transform is the negative entropy function, restricted to the probability simplex:

$$f^*(\boldsymbol{x}^*) \;=\; \begin{cases} \sum_{i=1}^n x_i^* \ln(x_i^*) & \text{for } 0 \leqslant x_i^* \text{ and } \sum_{i=1}^n x_i^* = 1 \\ \infty & \text{otherwise} \end{cases} . \tag{504}$$

*For the negative entropy function, restricted to the probability simplex:*

$$f(\boldsymbol{x}) \;=\; \begin{cases} \sum_{i=1}^n x_i \ln(x_i) & \text{for } 0 \leqslant x_i \text{ and } \sum_{i=1}^n x_i = 1 \\ \infty & \text{otherwise} \end{cases} . \tag{505}$$

*the Legendre transform is the log-sum exponential*

$$f^*(\boldsymbol{x}^*) \;=\; \ln\left(\sum_{i=1}^n \exp(x_i^*)\right) , \tag{506}$$

*Proof.* See page 93 Example 3.25 in Boyd & Vandenberghe (2009) and (Gao & Pavel, 2017). If $f$ is a regular convex function (lower semi-continuous convex function), then $f^{**} = f$ according to page 135 Exercise 11.2.3 in Garling (2017). If $f$ is lower semi-continuous and convex, then $f^{**} = f$ according to Theorem 13.37 (Fenchel-Moreau) in Bauschke & Combettes (2017). The log-sum-exponential is continuous and convex. $\qquad\square$

**Lemma A29.** *Let $\boldsymbol{X}\boldsymbol{X}^T$ be non-singular and $X$ a Hilbert space. We define*

$$X^* \;=\; \left\{ \boldsymbol{a} \mid 0 \leqslant \boldsymbol{X}^T\left(\boldsymbol{X}\boldsymbol{X}^T\right)^{-1}\boldsymbol{a} , \; \mathbf{1}^T\boldsymbol{X}^T\left(\boldsymbol{X}\boldsymbol{X}^T\right)^{-1}\boldsymbol{a} = 1 \right\} . \tag{507}$$

*and*

$$X^v \;=\; \left\{ \boldsymbol{a} \mid \boldsymbol{a} = \boldsymbol{X}^T\boldsymbol{\xi} , \; \boldsymbol{\xi} \in X \right\} . \tag{508}$$

*The Legendre transform of $\mathrm{lse}(\beta, \boldsymbol{X}^T\boldsymbol{\xi})$ with $\boldsymbol{\xi} \in X$ is*

$$\left(\mathrm{lse}(\beta, \boldsymbol{X}^T\boldsymbol{\xi})\right)^*(\boldsymbol{\xi}^*) \;=\; \left(\mathrm{lse}(\beta, \boldsymbol{v})\right)^*\left(\boldsymbol{X}^T\left(\boldsymbol{X}\boldsymbol{X}^T\right)^{-1}\boldsymbol{\xi}^*\right) , \tag{509}$$

*with $\boldsymbol{\xi}^* \in X^*$ and $\boldsymbol{v} \in X^v$. The domain of $\left(\mathrm{lse}(\beta, \boldsymbol{X}^T\boldsymbol{\xi})\right)^*$ is $X^*$.*

*Furthermore we have*

$$\left(\mathrm{lse}(\beta, \boldsymbol{X}^T\boldsymbol{\xi})\right)^{**} \;=\; \mathrm{lse}(\beta, \boldsymbol{X}^T\boldsymbol{\xi}) . \tag{510}$$

*Proof.* We use the definition of the Legendre transform:

$$\begin{aligned} \left(\mathrm{lse}(\beta, \boldsymbol{X}^T\boldsymbol{\xi})\right)^*(\boldsymbol{\xi}^*) \;&=\; \sup_{\boldsymbol{\xi}\in X} \boldsymbol{\xi}^T\boldsymbol{\xi}^* \;-\; \mathrm{lse}(\beta, \boldsymbol{X}^T\boldsymbol{\xi}) \\ &=\; \sup_{\boldsymbol{\xi}\in X} \left(\boldsymbol{X}^T\boldsymbol{\xi}\right)^T \boldsymbol{X}^T\left(\boldsymbol{X}\boldsymbol{X}^T\right)^{-1}\boldsymbol{\xi}^* \;-\; \mathrm{lse}(\beta, \boldsymbol{X}^T\boldsymbol{\xi}) \\ &=\; \sup_{\boldsymbol{v}\in X^v} \boldsymbol{v}^T\boldsymbol{X}^T\left(\boldsymbol{X}\boldsymbol{X}^T\right)^{-1}\boldsymbol{\xi}^* \;-\; \mathrm{lse}(\beta, \boldsymbol{v}) \\ &=\; \sup_{\boldsymbol{v}\in X^v} \boldsymbol{v}^T\boldsymbol{v}^* \;-\; \mathrm{lse}(\beta, \boldsymbol{v}) \\ &=\; \left(\mathrm{lse}(\beta, \boldsymbol{v})\right)^*(\boldsymbol{v}^*) \;=\; \left(\mathrm{lse}(\beta, \boldsymbol{v})\right)^*\left(\boldsymbol{X}^T\left(\boldsymbol{X}\boldsymbol{X}^T\right)^{-1}\boldsymbol{\xi}^*\right) , \end{aligned} \tag{511}$$

where we used $\boldsymbol{v}^* = \boldsymbol{X}^T\left(\boldsymbol{X}\boldsymbol{X}^T\right)^{-1}\boldsymbol{\xi}^*$.

According to page 93 Example 3.25 in Boyd & Vandenberghe (2009), the equations for the maximum $\max_{\boldsymbol{v}\in X^v} \boldsymbol{v}^T\boldsymbol{v}^* \;-\; \mathrm{lse}(\beta, \boldsymbol{v})$ are solvable if and only if $0 < \boldsymbol{v}^* = \boldsymbol{X}^T\left(\boldsymbol{X}\boldsymbol{X}^T\right)^{-1}\boldsymbol{\xi}^*$ and $\mathbf{1}^T\boldsymbol{v}^* = \mathbf{1}^T\boldsymbol{X}^T\left(\boldsymbol{X}\boldsymbol{X}^T\right)^{-1}\boldsymbol{\xi}^* = 1$. Therefore, we assumed $\boldsymbol{\xi}^* \in X^*$.

The domain of $\left(\mathrm{lse}(\beta, \boldsymbol{X}^T\boldsymbol{\xi})\right)^*$ is $X^*$, since on page 93 Example 3.25 in Boyd & Vandenberghe (2009) it was shown that outside $X^*$ the $\sup_{\boldsymbol{v}\in X^v} \boldsymbol{v}^T\boldsymbol{v}^* \;-\; \mathrm{lse}(\beta, \boldsymbol{v})$ is not bounded.

Using

$$\boldsymbol{p} \ = \ \mathrm{softmax}(\beta \boldsymbol{X}^T \boldsymbol{\xi}) \,, \tag{512}$$

the Hessian of $\mathrm{lse}(\beta, \boldsymbol{X}^T \boldsymbol{\xi})$

$$\frac{\partial^2 \mathrm{lse}(\beta, \boldsymbol{X}^T \boldsymbol{\xi})}{\partial \boldsymbol{\xi}^2} \ = \ \beta \ \boldsymbol{X} \left( \mathrm{diag}(\boldsymbol{p}) - \boldsymbol{p}\boldsymbol{p}^T \right) \boldsymbol{X}^T \tag{513}$$

is positive semi-definite since $\mathrm{diag}(\boldsymbol{p}) - \boldsymbol{p}\boldsymbol{p}^T$ is positive semi-definite according to Lemma A22. Therefore, $\mathrm{lse}(\beta, \boldsymbol{X}^T \boldsymbol{\xi})$ is convex and continuous.

If $f$ is a regular convex function (lower semi-continuous convex function), then $f^{**} = f$ according to page 135 Exercise 11.2.3 in Garling (2017). If $f$ is lower semi-continuous and convex, then $f^{**} = f$ according to Theorem 13.37 (Fenchel-Moreau) in Bauschke & Combettes (2017). Consequently we have

$$\left( \mathrm{lse}(\beta, \boldsymbol{X}^T \boldsymbol{\xi}) \right)^{**} \ = \ \mathrm{lse}(\beta, \boldsymbol{X}^T \boldsymbol{\xi}) \,. \tag{514}$$

$\square$

We introduce the Lambert $W$ function and some of its properties, since it is needed to derive bounds on the storage capacity of our new Hopfield networks.

**Definition A6** (Lambert Function). *The Lambert $W$ function (Olver et al., 2010, (4.13)) is the inverse function of*

$$f(y) \ = \ y e^y \,. \tag{515}$$

*The Lambert W function has an upper branch $W_0$ for $-1 \leqslant y$ and a lower branch $W_{-1}$ for $y \leqslant -1$. We use $W$ if a formula holds for both branches. We have*

$$W(x) \ = \ y \ \Rightarrow \ y e^y \ = \ x \,. \tag{516}$$

We present some identities for the Lambert $W$ function (Olver et al., 2010, (4.13)):

**Lemma A30.** *Identities for the Lambert $W$ function are*

$$W(x) \, e^{W(x)} \ = \ x \,, \tag{517}$$

$$W(x e^x) \ = \ x \,, \tag{518}$$

$$e^{W(x)} \ = \ \frac{x}{W(x)} \,, \tag{519}$$

$$e^{-W(x)} \ = \ \frac{W(x)}{x} \,, \tag{520}$$

$$e^{nW(x)} \ = \ \left( \frac{x}{W(x)} \right)^n \,, \tag{521}$$

$$W_0 \left( x \ln x \right) \ = \ \ln x \quad for \ x \geq \frac{1}{e} \,, \tag{522}$$

$$W_{-1} \left( x \ln x \right) \ = \ \ln x \quad for \ x \leqslant \frac{1}{e} \,, \tag{523}$$

$$W(x) \ = \ \ln \frac{x}{W(x)} \quad for \ x \geq - \frac{1}{e} \,, \tag{524}$$

$$W \left( \frac{n \, x^n}{W(x)^{n-1}} \right) \ = \ n \, W(x) \quad for \ n, x \, > \, 0 \,, \tag{525}$$

$$W(x) \ + \ W(y) \ = \ W \left( x \, y \left( \frac{1}{W(x)} + \frac{1}{W(y)} \right) \right) \quad for \ x, y \, > \, 0 \,, \tag{526}$$

$$W_0 \left( - \frac{\ln x}{x} \right) \ = \ - \ln x \quad for \ 0 < x \leqslant e \,, \tag{527}$$

$$W_{-1} \left( - \frac{\ln x}{x} \right) \ = \ - \ln x \quad for \ x \, > \, e \,, \tag{528}$$

$$e^{- \, W(- \, \ln x)} \ = \ \frac{W(- \, \ln x)}{- \, \ln x} \quad for \ x \, \neq \, 1 \,. \tag{529}$$

We also present some special values for the Lambert $W$ function (Olver et al., 2010, (4.13)):

**Lemma A31.**

$$W(0) \;=\; 0 \,, \tag{530}$$

$$W(e) \;=\; 1 \,, \tag{531}$$

$$W\left(-\frac{1}{e}\right) \;=\; -1 \,, \tag{532}$$

$$W\left(e^{1+e}\right) \;=\; e \,, \tag{533}$$

$$W\left(2\ln 2\right) \;=\; \ln 2 \,, \tag{534}$$

$$W(1) \;=\; \Omega \,, \tag{535}$$

$$W(1) \;=\; e^{-W(1)} \;=\; \ln\left(\frac{1}{W(1)}\right) \;=\; -\ln W(1) \,, \tag{536}$$

$$W\left(-\frac{\pi}{2}\right) \;=\; \frac{i\pi}{2} \,, \tag{537}$$

$$W(-1) \;\approx\; -0.31813 + 1.33723i \,, \tag{538}$$

*where the Omega constant $\Omega$ is*

$$\Omega \;=\; \left(\int_{-\infty}^{\infty} \frac{\mathrm{d}t}{(e^t - t)^2 + \pi^2}\right)^{-1} - 1 \;\approx\; 0.56714329 \,. \tag{539}$$

We need in some proofs a version of the mean value theorem as given in the next lemma.

**Lemma A32** (Mean Value Theorem). *Let $U \subset \mathbb{R}^n$ be open, $f : U \to \mathbb{R}^m$ continuously differentiable, and $\boldsymbol{x} \in U$ as well as $\boldsymbol{h} \in \mathbb{R}^n$ vectors such that the line segment $\boldsymbol{x} + t\boldsymbol{h}$ for $0 \leqslant t \leqslant 1$ is in $U$. Then the following holds:*

$$f(\boldsymbol{x} + \boldsymbol{h}) - f(\boldsymbol{x}) \;=\; \left(\int_0^1 J(\boldsymbol{x} + t\,\boldsymbol{h})\,\mathrm{d}t\right) \boldsymbol{h} \,, \tag{540}$$

*where $J$ is the Jacobian of $f$ and the integral of the matrix is component-wise.*

*Proof.* Let $f_1, \dots, f_m$ denote the components of $f$ and define $g_i : [0,1] \to \mathbb{R}$ by

$$g_i(t) \;=\; f_i(\boldsymbol{x} + t\,\boldsymbol{h}) \,, \tag{541}$$

then we obtain

$$f_i(\boldsymbol{x} + \boldsymbol{h}) - f_i(\boldsymbol{x}) \;=\; g_i(1) - g_i(0) \;=\; \int_0^1 g'(t)\,\mathrm{d}t \tag{542}$$

$$\int_0^1 \left(\sum_{j=1}^n \frac{\partial f_i}{\partial x_j}(\boldsymbol{x} + t\,\boldsymbol{h})\,h_j\right) \mathrm{d}t \;=\; \sum_{j=1}^n \left(\int_0^1 \frac{\partial f_i}{\partial x_j}(\boldsymbol{x} + t\,\boldsymbol{h})\,\mathrm{d}t\right) h_j \,.$$

The statement follows since the Jacobian $J$ has as entries $\frac{\partial f_i}{\partial x_j}$. $\qquad\square$

## A.3 MODERN HOPFIELD NETWORKS: BINARY STATES (KROTOV AND HOPFIELD)

### A.3.1 MODERN HOPFIELD NETWORKS: INTRODUCTION

**A.3.1.1 Additional Memory and Attention for Neural Networks.** Modern Hopfield networks may serve as additional memory for neural networks. Different approaches have been suggested to equip neural networks with an additional memory beyond recurrent connections. The neural Turing machine (NTM) is a neural network equipped with an external memory and an attention process (Graves et al., 2014). The NTM can write to the memory and can read from it. A memory network (Weston et al., 2014) consists of a memory together with the components: (1) input feature map (converts the incoming input to the internal feature representation) (2) generalization (updates old memories given the new input), (3) output feature map (produces a new output), (4) response

(converts the output into the response format). Memory networks are generalized to an end-to-end trained model, where the $\arg\max$ memory call is replaced by a differentiable $\operatorname{softmax}$ (Sukhbaatar et al., 2015a;b). Linear Memory Network use a linear autoencoder for sequences as a memory (Carta et al., 2020).

To enhance RNNs with additional associative memory like Hopfield networks have been proposed (Ba et al., 2016a;b). The associative memory stores hidden states of the RNN, retrieves stored states if they are similar to actual ones, and has a forgetting parameter. The forgetting and storing parameters of the RNN associative memory have been generalized to learned matrices (Zhang & Zhou, 2017). LSTMs with associative memory via Holographic Reduced Representations have been proposed (Danihelka et al., 2016).

Recently most approaches to new memories are based on attention. The neural Turing machine (NTM) is equipped with an external memory and an attention process (Graves et al., 2014). End to end memory networks (EMN) make the attention scheme of memory networks (Weston et al., 2014) differentiable by replacing $\arg\max$ through a $\operatorname{softmax}$ (Sukhbaatar et al., 2015a;b). EMN with dot products became very popular and implement a key-value attention (Daniluk et al., 2017) for self-attention. An enhancement of EMN is the transformer (Vaswani et al., 2017a;b) and its extensions (Dehghani et al., 2018). The transformer had great impact on the natural language processing (NLP) community as new records in NLP benchmarks have been achieved (Vaswani et al., 2017a;b). MEMO uses the transformer attention mechanism for reasoning over longer distances (Banino et al., 2020). Current state-of-the-art for language processing is a transformer architecture called "the Bidirectional Encoder Representations from Transformers" (BERT) (Devlin et al., 2018; 2019).

**A.3.1.2   Modern Hopfield networks: Overview.**   The storage capacity of classical binary Hopfield networks (Hopfield, 1982) has been shown to be very limited. In a $d$-dimensional space, the standard Hopfield model can store $d$ uncorrelated patterns without errors but only $Cd/\ln(d)$ random patterns with $C < 1/2$ for a fixed stable pattern or $C < 1/4$ if all patterns are stable (McEliece et al., 1987). The same bound holds for nonlinear learning rules (Mazza, 1997). Using tricks-of-trade and allowing small retrieval errors, the storage capacity is about $0.138d$ (Crisanti et al., 1986; Hertz et al., 1991; Torres et al., 2002). If the learning rule is not related to the Hebb rule then up to $d$ patterns can be stored (Abu-Mostafa & StJacques, 1985). Using Hopfield networks with non-zero diagonal matrices, the storage can be increased to $Cd\ln(d)$ (Folli et al., 2017). In contrast to the storage capacity, the number of energy minima (spurious states, stable states) of Hopfield networks is exponentially in $d$ (Tanaka & Edwards, 1980; Bruck & Roychowdhury, 1990; Wainrib & Touboul, 2013).

Recent advances in the field of binary Hopfield networks (Hopfield, 1982) led to new properties of Hopfield networks. The stability of spurious states or metastable states was sensibly reduced by a Hamiltonian treatment for the new relativistic Hopfield model (Barra et al., 2018). Recently the storage capacity of Hopfield networks could be increased by new energy functions. Interaction functions of the form $F(x) = x^n$ lead to storage capacity of $\alpha_n d^{n-1}$, where $\alpha_n$ depends on the allowed error probability (Krotov & Hopfield, 2016; 2018; Demircigil et al., 2017) (see (Krotov & Hopfield, 2018) for the non-binary case). Interaction functions of the form $F(x) = x^n$ lead to storage capacity of $\alpha_n \frac{d^{n-1}}{c_n \ln d}$ for $c_n > 2(2n-3)!!$ (Demircigil et al., 2017).

Interaction functions of the form $F(x) = \exp(x)$ lead to *exponential* storage capacity of $2^{d/2}$ where all stored patterns are fixed points but the radius of attraction vanishes (Demircigil et al., 2017). It has been shown that the network converges with high probability after one update (Demircigil et al., 2017).

A.3.2   ENERGY AND UPDATE RULE FOR BINARY MODERN HOPFIELD NETWORKS

We follow (Demircigil et al., 2017) where the goal is to store a set of input data $\boldsymbol{x}_1, \ldots, \boldsymbol{x}_N$ that are represented by the matrix

$$\boldsymbol{X} = (\boldsymbol{x}_1, \ldots, \boldsymbol{x}_N) \ . \tag{543}$$

The $\boldsymbol{x}_i$ is pattern with binary components $x_{ij} \in \{-1, +1\}$ for all $i$ and $j$. $\boldsymbol{\xi}$ is the actual state of the units of the Hopfield model. Krotov and Hopfield (Krotov & Hopfield, 2016) defined the energy function E with the interaction function $F$ that evaluates the dot product between patterns $\boldsymbol{x}_i$ and the

actual state $\boldsymbol{\xi}$:

$$\mathrm{E} \;=\; -\sum_{i=1}^{N} F\left(\boldsymbol{\xi}^T \boldsymbol{x}_i\right) \tag{544}$$

with $F(a) = a^n$, where $n = 2$ gives the energy function of the classical Hopfield network. This allows to store $\alpha_n d^{n-1}$ patterns (Krotov & Hopfield, 2016). Krotov and Hopfield (Krotov & Hopfield, 2016) suggested for minimizing this energy an asynchronous updating dynamics $T = (T_j)$ for component $\xi_j$:

$$T_j(\boldsymbol{\xi}) \;:=\; \mathrm{sgn}\left[\sum_{i=1}^{N}\left(F\left(x_{ij} + \sum_{l \neq j} x_{il}\, \xi_l\right) - F\left(-x_{ij} + \sum_{l \neq j} x_{il}\, \xi_l\right)\right)\right] \tag{545}$$

While Krotov and Hopfield used $F(a) = a^n$, Demircigil et al. (Demircigil et al., 2017) went a step further and analyzed the model with the energy function $F(a) = \exp(a)$, which leads to an exponential storage capacity of $N = 2^{d/2}$. Furthermore with a single update the final pattern is recovered with high probability. These statements are given in next theorem.

**Theorem A10** (Storage Capacity for Binary Modern Hopfield Nets (Demircigil et al. 2017))**.** *Consider the generalized Hopfield model with the dynamics described in Eq. (545) and interaction function $F$ given by $F(x) = e^x$. For a fixed $0 < \alpha < \ln(2)/2$ let $N = \exp(\alpha d) + 1$ and let $\boldsymbol{x}_1, \ldots, \boldsymbol{x}_N$ be $N$ patterns chosen uniformly at random from $\{-1, +1\}^d$. Moreover fix $\varrho \in [0, 1/2)$. For any $i$ and any $\widetilde{\boldsymbol{x}}_i$ taken uniformly at random from the Hamming sphere with radius $\varrho d$ centered in $\boldsymbol{x}_i$, $\mathcal{S}(\boldsymbol{x}_i, \varrho d)$, where $\varrho d$ is assumed to be an integer, it holds that*

$$\Pr\left(\exists i\, \exists j:\, T_j\left(\widetilde{\boldsymbol{x}}_i\right) \neq x_{ij}\right) \to 0,$$

*if $\alpha$ is chosen in dependence of $\varrho$ such that*

$$\alpha \;<\; \frac{I(1 - 2\varrho)}{2}$$

*with*

$$I:\, a \;\mapsto\; \frac{1}{2}\left((1 + a)\ln(1 + a) + (1 - a)\ln(1 - a)\right).$$

*Proof.* The proof can be found in Demircigil et al. (2017). $\qquad\square$

The number of patterns $N = \exp(\alpha d) + 1$ is exponential in the number $d$ of components. The result
$$\Pr\left(\exists i\, \exists j:\, T_j\left(\widetilde{\boldsymbol{x}}_i\right) \neq x_{ij}\right) \to 0$$
means that one update for each component is sufficient to recover the pattern with high probability. The constraint $\alpha < \frac{I(1 - 2\varrho)}{2}$ on $\alpha$ gives the trade-off between the radius of attraction $\varrho d$ and the number $N = \exp(\alpha d) + 1$ of pattern that can be stored.

Theorem A10 in particular implies that
$$\Pr\left(\exists i\, \exists j:\, T_j\left(\boldsymbol{x}_i\right) \neq x_{ij}\right) \to 0$$
as $d \to \infty$, i.e. with a probability converging to 1, all the patterns are fixed points of the dynamics. In this case we can have $\alpha \to \frac{I(1)}{2} = \ln(2)/2$.

Krotov and Hopfield define the update dynamics $T_j(\boldsymbol{\xi})$ in Eq. (545) via energy differences of the energy in Eq. (544). First we express the energy in Eq. (544) with $F(a) = \exp(a)$ (Demircigil et al., 2017) by the lse function. Then we use the mean value theorem to express the update dynamics $T_j(\boldsymbol{\xi})$ in Eq. (545) by the softmax function. For simplicity, we set $\beta = 1$ in the following. There exists a $v \in [-1, 1]$ with

$$
\begin{aligned}
T_j(\boldsymbol{\xi}) \;&=\; \mathrm{sgn}\left[-\mathrm{E}(\xi_j = 1) + \mathrm{E}(\xi_j = -1)\right] \;=\; \mathrm{sgn}\left[\exp(\mathrm{lse}(\xi_j = 1)) - \exp(\mathrm{lse}(\xi_j = -1))\right] \\
&=\; \mathrm{sgn}\left[-(2\boldsymbol{e}_j)^T \nabla_{\boldsymbol{\xi}} \mathrm{E}(\xi_j = v)\right] \;=\; \mathrm{sgn}\left[\exp(\mathrm{lse}(\xi_j = v))\, (2\boldsymbol{e}_j)^T \frac{\mathrm{lse}(\xi_j = v)}{\partial \boldsymbol{\xi}}\right] \\
&=\; \mathrm{sgn}\left[\exp(\mathrm{lse}(\xi_j = 1))\, (2\boldsymbol{e}_j)^T \boldsymbol{X} \mathrm{softmax}(\boldsymbol{X}^T \boldsymbol{\xi}(\xi_j = v))\right] \\
&=\; \mathrm{sgn}\left[[\boldsymbol{X} \mathrm{softmax}(\boldsymbol{X}^T \boldsymbol{\xi}(\xi_j = v))]_j\right] \;=\; \mathrm{sgn}\left[[\boldsymbol{X} \boldsymbol{p}(\xi_j = v)]_j\right],
\end{aligned} \tag{546}
$$

where $\boldsymbol{e}_j$ is the Cartesian unit vector with a one at position $j$ and zeros elsewhere, $[.]_j$ is the projection to the $j$-th component, and

$$\boldsymbol{p} \;=\; \mathrm{softmax}(\boldsymbol{X}^T\boldsymbol{\xi}) \,. \tag{547}$$

### A.4 Hopfield Update Rule is Attention of The Transformer

The Hopfield network update rule is the attention mechanism used in transformer and BERT models (see Fig. A.2). To see this, we assume $N$ stored (key) patterns $\boldsymbol{y}_i$ and $S$ state (query) patterns $\boldsymbol{r}_i$ that are mapped to the Hopfield space of dimension $d_k$. We set $\boldsymbol{x}_i = \boldsymbol{W}_K^T\boldsymbol{y}_i$, $\boldsymbol{\xi}_i = \boldsymbol{W}_Q^T\boldsymbol{r}_i$, and multiply the result of our update rule with $\boldsymbol{W}_V$. The matrices $\boldsymbol{Y} = (\boldsymbol{y}_1,\ldots,\boldsymbol{y}_N)^T$ and $\boldsymbol{R} = (\boldsymbol{r}_1,\ldots,\boldsymbol{r}_S)^T$ combine the $\boldsymbol{y}_i$ and $\boldsymbol{r}_i$ as row vectors. We define the matrices $\boldsymbol{X}^T = \boldsymbol{K} = \boldsymbol{Y}\boldsymbol{W}_K$, $\boldsymbol{\Xi}^T = \boldsymbol{Q} = \boldsymbol{R}\boldsymbol{W}_Q$, and $\boldsymbol{V} = \boldsymbol{Y}\boldsymbol{W}_K\boldsymbol{W}_V = \boldsymbol{X}^T\boldsymbol{W}_V$, where $\boldsymbol{W}_K \in \mathbb{R}^{d_y \times d_k}$, $\boldsymbol{W}_Q \in \mathbb{R}^{d_r \times d_k}$, $\boldsymbol{W}_V \in \mathbb{R}^{d_k \times d_v}$. If $\beta = 1/\sqrt{d_k}$ and $\mathrm{softmax} \in \mathbb{R}^N$ is changed to a row vector, we obtain for the update rule Eq. (3) multiplied by $\boldsymbol{W}_V$:

$$\mathrm{softmax}\left(1/\sqrt{d_k}\,\boldsymbol{Q}\,\boldsymbol{K}^T\right)\,\boldsymbol{V} \;=\; \mathrm{softmax}\left(\beta\,\boldsymbol{R}\boldsymbol{W}_Q\,\boldsymbol{W}_K^T\boldsymbol{Y}^T\right)\,\boldsymbol{Y}\boldsymbol{W}_K\boldsymbol{W}_V \,. \tag{548}$$

The left part of Eq. (548) is the transformer attention. Besides the attention mechanism, Hopfield networks allow for other functionalities in deep network architectures, which we introduce via specific layers in the next section. The right part of Eq. (548) serves as starting point for these specific layers.

Figure A.2: We generalized the energy of binary modern Hopfield networks for allowing continuous states while keeping fast convergence and storage capacity properties. We defined for the new energy also a new update rule that minimizes the energy. The new update rule is the attention mechanism of the transformer. Formulae are modified to express $\mathrm{softmax}$ as row vector as for transformers. "="-sign means "keeps the properties".

### A.5 Experiments

#### A.5.1 Experiment 1: Attention in Transformers described by Hopfield dynamics

**A.5.1.1 Analysis of operating modes of the heads of a pre-trained BERT model.** We analyzed pre-trained BERT models from Hugging Face Inc. (Wolf et al., 2019) according to these operating classes. In Fig. A.3 in the appendix the distribution of the pre-trained bert-base-cased model is depicted (for other models see appendix Section A.5.1.4). Operating classes (II) (large metastable states) and (IV) (small metastable states) are often observed in the middle layers. Operating class (I) (averaging over a very large number of patterns) is abundant in lower layers. Similar observations have been reported in other studies (Toneva & Wehbe, 2019a;b; Tay et al., 2020). Operating class (III) (medium metastable states) is predominant in the last layers.

**A.5.1.2 Experimental Setup.** Transformer architectures are known for their high computational demands. To investigate the learning dynamics of such a model and at the same time keeping training time manageable, we adopted the BERT-small setting from ELECTRA (Clark et al., 2020). It has 12 layers, 4 heads and a reduced hidden size, the sequence length is shortened from 512 to 128 tokens and the batch size is reduced from 256 to 128. Additionally, the hidden dimension is reduced from 768 to 256 and the embedding dimension is reduced from 768 to 128 (Clark et al., 2020). The training of such a BERT-small model for 1.45 million update steps takes roughly four days on a single NVIDIA V100 GPU.

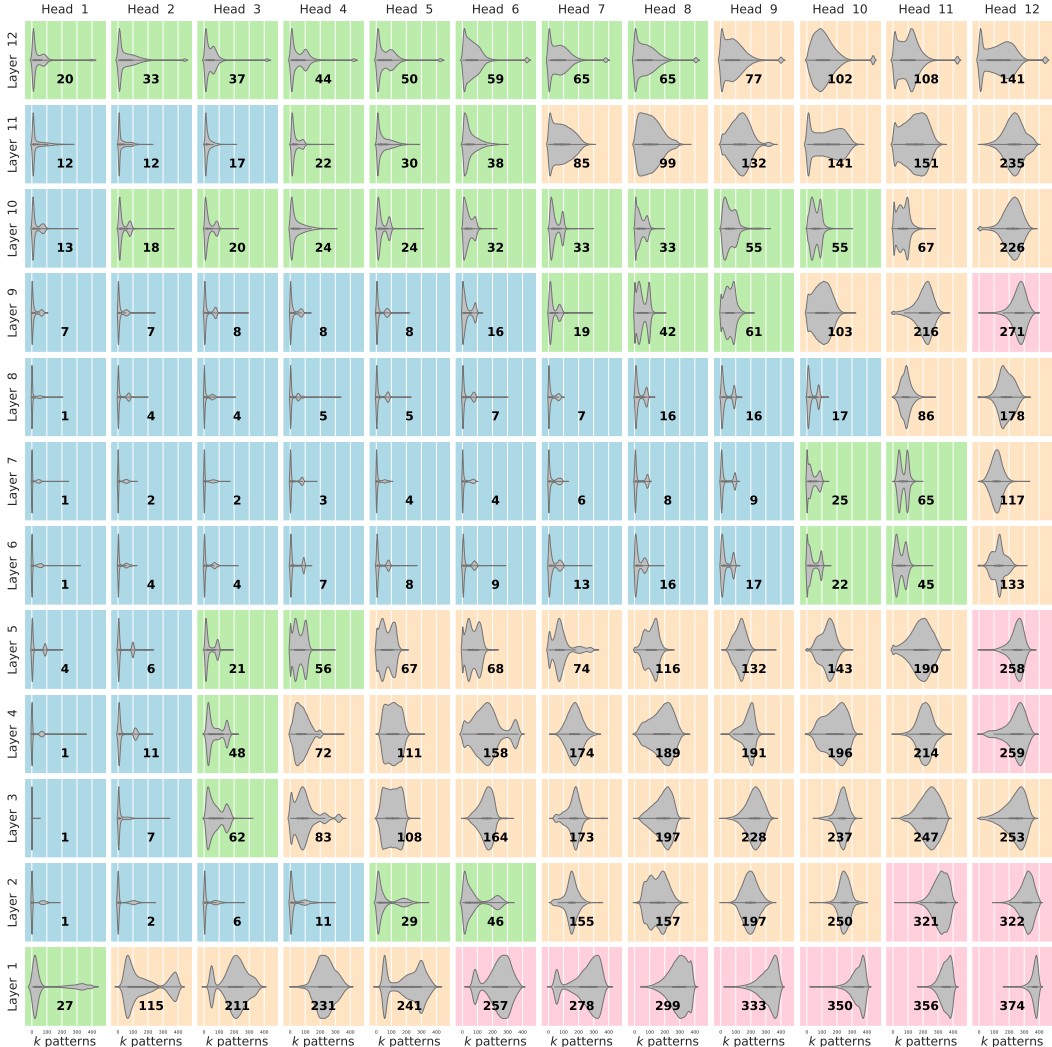

Figure A.3: Analysis of operating modes of the heads of a pre-trained BERT model. For each head in each layer, the distribution of the minimal number $k$ of patterns required to sum up the softmax values to $0.90$ is displayed as a violin plot in a panel. $k$ indicates the size of a metastable state. The bold number in the center of each panel gives the median $\bar{k}$ of the distribution. The heads in each layer are sorted according to $\bar{k}$. Attention heads belong to the class they mainly operate in. **Class (IV) in blue:** Small metastable state or fixed point close to a single pattern, which is abundant in the middle layers (6, 7, and 8). **Class (II) in orange:** Large metastable state, which is prominent in middle layers (3, 4, and 5). **Class (I) in red:** Very large metastable state or global fixed point, which is predominant in the first layer. These heads can potentially be replaced by averaging operations. **Class (III) in green:** Medium metastable state, which is frequently observed in higher layers. We hypothesize that these heads are used to collect information required to perform the respective task. These heads should be the main target to improve transformer and BERT models.

As the code base we use the *transformers* repository from Hugging Face, Inc (Wolf et al., 2019). We aim to reproduce the dataset of Devlin et al. (2019) as close as possible, which consists of the English Wikipedia dataset and the Toronto BookCorpus dataset (Zhu et al., 2015). Due to recent copyright claims the later is not publicly available anymore. Therefore, the pre-training experiments use an uncased snapshot of the original BookCorpus dataset.

### A.5.1.3 Hopfield Operating Classes of Transformer and BERT Models.

To better understand how operation modes in attention heads develop, we tracked the distribution of counts $k$ (see main paper) over time in a BERT-small model. At the end of training we visualized the count distribution, grouped into four classes (see Figure A.4). The thresholds for the classes were chosen according to the thresholds of Figure 2 in the main paper. However, they are divided by a factor of 4 to adapt to the shorter sequence length of 128 compared to 512. From this plot it is clear, that the attention in heads of **Class IV** commit very early to the operating class of small metastable states.

### A.5.1.4 Learning Dynamics of Transformer and BERT Models.

To observe this behavior in the early phase of training, we created a ridge plot of the distributions of counts $k$ for the first $20,000$ steps (see Figure A.5 (a)). This plot shows that the attention in heads of middle layers often change the operation mode to **Class IV** around $9,000$ to $10,000$ steps. At the same time the second big drop in the loss occurs. The question arises whether this is functionally important or whether it is an artefact which could be even harmful. To check if the attention mechanism is still able to learn after the change in the operation mode we analyzed the gradient flow through the $\mathrm{softmax}$ function. For every token we calculate the Frobenius norm of the Jacobian of the $\mathrm{softmax}$ over multiple samples. Then, for every head we plot the distribution of the norm (see Figure A.5(b)). The gradients with respect to the weights are determined by the Jacobian J defined in Eq. (59) as can be seen in Eq. (418), Eq. (429), and Eq. (435). We can see that the attention in heads of **Class IV** remain almost unchanged during the rest of the training.

### A.5.1.5 Attention Heads Replaced by Gaussian Averaging Layers.

The self-attention mechanism proposed in Vaswani et al. (2017a) utilizes the $\mathrm{softmax}$ function to compute the coefficients of a convex combination over the embedded tokens, where the $\mathrm{softmax}$ is conditioned on the input. However, our analysis showed that especially in lower layers many heads perform averaging over a very large number of patterns. This suggests that at this level neither the dependency on the input nor a fine grained attention to individual positions is necessary. As an alternative to the original mechanism we propose Gaussian averaging heads which are computationally more efficient. Here, the $\mathrm{softmax}$ function is replaced by a discrete Gaussian kernel, where the location $\mu$ and the scale $\sigma$ are learned. In detail, for a sequence length of $N$ tokens we are given a vector of location parameters $\boldsymbol{\mu} = (\mu_1, \ldots, \mu_N)^T$ and a vector of corresponding scale parameters $\boldsymbol{\sigma} = (\sigma_1, \ldots, \sigma_N)^T$. We subdivide the interval $[-1, 1]$ into $N$ equidistant supporting points $\{s_j\}_{j=1}^N$, where

$$s_j = \frac{(j-1) - 0.5\,(N-1)}{0.5\,(N-1)}.$$

The attention $[A]_{i,j}$ from the $i$-th token to the $j$-th position is calculated as

$$[A]_{i,j} = \frac{1}{z_i} \exp\left\{ -\frac{1}{2}\left(\frac{s_j - \mu_i}{\sigma_i}\right)^2 \right\},$$

where $z_i$ normalizes the $i$-th row of the attention matrix $A$ to sum up to one:

$$z_i = \sum_{j=1}^{N} \exp\left\{ -\frac{1}{2}\left(\frac{s_j - \mu_i}{\sigma_i}\right)^2 \right\}.$$

For initialization we uniformly sample a location vector $\boldsymbol{\mu} \in [-1, 1]^N$ and a scale vector $\boldsymbol{\sigma} \in [0.75, 1.25]^N$ per head. A simple way to consider the individual position of each token at initialization is to use the supporting points $\mu_i = s_i$ (see Figure A.6). In practice no difference to the random initialization was observed.

•*Number of parameters.* Gaussian averaging heads can reduce the number of parameters significantly. For an input size of $N$ tokens, there are $2 \cdot N$ parameters per head. In contrast, a standard self-attention head with word embedding dimension $d_y$ and projection dimension $d_k$ has two weight matrices

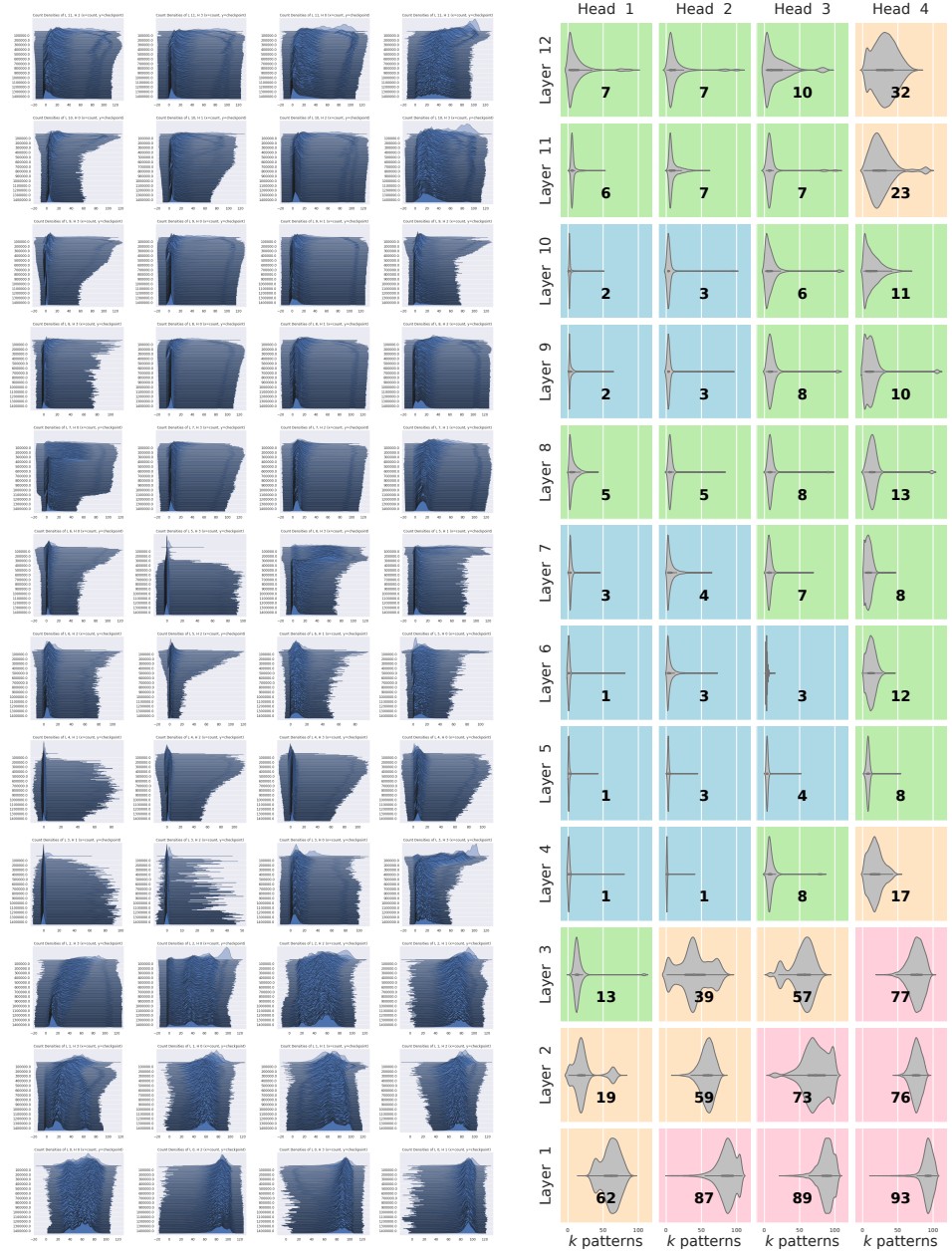

Figure A.4: **Left**: Ridge plots of the distribution of counts $k$ over time for BERT-small **Right**: Violin plot of counts $k$ after $1,450000$ steps, divided into the four classes from the main paper. The thresholds were adapted to the shorter sequence length.

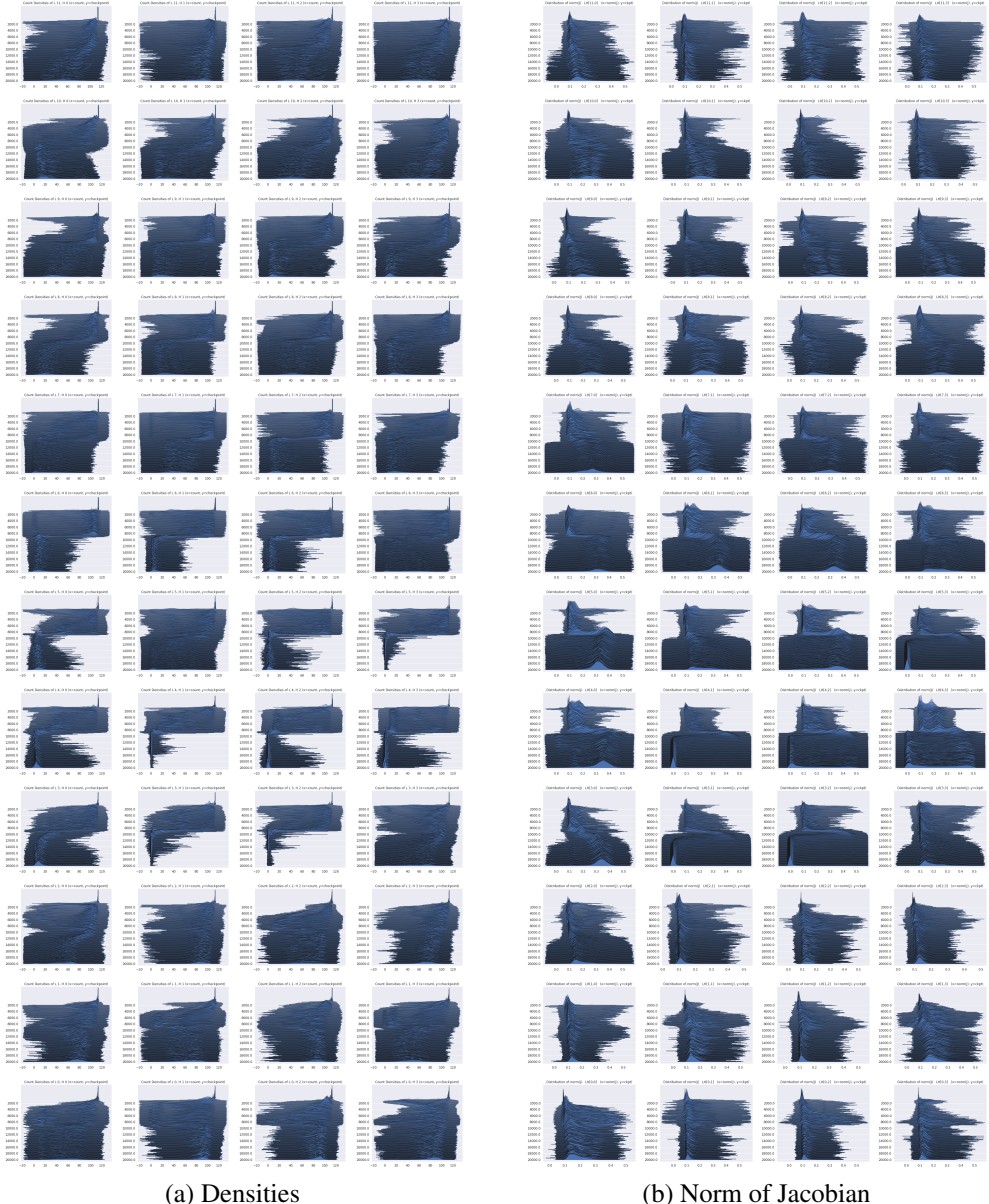

(a) Densities            (b) Norm of Jacobian

Figure A.5: **(a)**: change of count density during training is depicted for the first $20,000$ steps. **(b)**: the corresponding distribution of the Frobenius norm of the Jacobian of the $\mathrm{softmax}$ function is depicted. The gradients with respect to the weights are determined by the Jacobian J defined in Eq. (59) as can be seen in Eq. (418), Eq. (429), and Eq. (435).

$W_Q, W_K \in \mathbb{R}^{d_k \times d_y}$, which together amount to $2 \cdot d_k \cdot d_y$ parameters. As a concrete example, the BERT-base model from Devlin et al. (2019) has an embedding dimension $d_y = 768$, a projection dimension $d_k = 64$ and a sequence length of $N = 512$. Compared to the Gaussian head, in this case $(2 \cdot 768 \cdot 64)/(2 \cdot 512) = 95.5$ times more parameters are trained for the attention mechanism itself. Only for very long sequences (and given that the word embedding dimension stays the same) the dependence on $N$ may become a disadvantage. But of course, due to the independence from the input the Gaussian averaging head is less expressive in comparison to the original attention mechanism. A recently proposed input independent replacement for self-attention is the so called Random Synthesizer (Tay et al., 2020). Here the $\mathrm{softmax}$-attention is directly parametrized with an $N \times N$ matrix. This amounts to $0.5 \cdot N$ more parameters than Gaussian averaging.

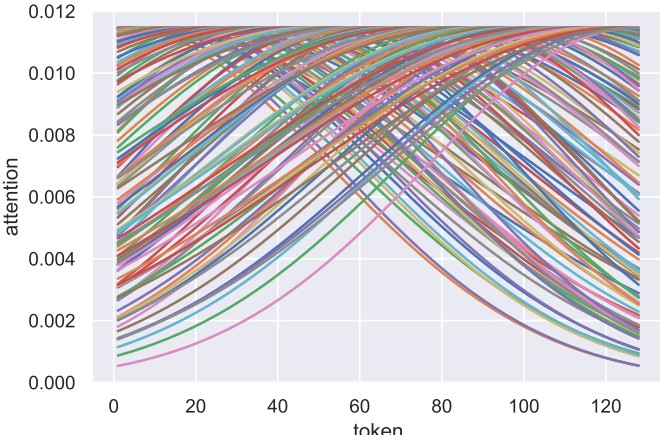

Figure A.6: Attentions of a Gaussian averaging head at initialization for sequence length $N = 128$. Every line depicts one Gaussian kernel. Here, the location parameters are initialized with the value of the supporting points $\mu_i = s_i$.

### A.5.2   EXPERIMENT 2: MULTIPLE INSTANCE LEARNING DATASETS.

**A.5.2.1   Immune Repertoire Classification.** An architecture called DeepRC, is based on our modern Hopfield networks, for immune repertoire classification and compared to other machine learning approaches. For DeepRC, we consider immune repertoires as input objects, which are represented as bags of instances. In a bag, each instance is an immune receptor sequence and each bag can contain a large number of sequences. At its core, DeepRC consists of a modern Hopfield network that extracts information from each repertoire. The stored patterns (keys) are representations of the immune amino acid sequences (instances) that are obtained by an 1D convolutional network with position encoding. Each state pattern (query) is static and learned via backpropagation. For details see Widrich et al. (2020a;b).

Our new Hopfield network has been integrated into a deep learning architecture for immune repertoire classification, a massive multiple instance learning task (Widrich et al., 2020a;b). Theorem 3 states that modern Hopfield networks possess an exponential storage capacity which enables to tackle massive multiple instance learning (MIL) problems (Dietterich et al., 1997). Immune repertoire classification (Emerson et al., 2017) typically requires to extract few patterns from a large set of sequences, the repertoire, that are indicative for the respective immune status. Most MIL methods fail due the large number of instances.

Data is obtained by experimentally observed immune receptors as well as simulated sequences sequence motifs (Akbar et al., 2019; Weber et al., 2020) with low yet varying degrees of frequency are implanted. Four different categories of datasets are constructed: (a) Simulated immunosequencing data with implanted motifs, (b) immunosequencing data generated by long short-term memory (LSTM) with implanted motifs, (c) real-world immunosequencing data with implanted motifs, and (d) real-world immunosequencing data with known immune status (Emerson et al., 2017). Categories (a), (b), and (d) contain approx. 300,000 instances per immune repertoire. With over 30 billion sequences in total, this represents one of the largest multiple instance learning experiments ever conducted (Carbonneau et al., 2018). Despite the massive number of instances as well as the low frequency

of sequences indicative of the respective immune status, deep learning architectures with modern Hopfield networks outperform all competing methods with respect to average area under the ROC curve in all four categories, (a), (b), (c) and (d) (for details see Widrich et al. (2020a)).

We evaluate and compare the performance of DeepRC to a set of machine learning methods that serve as baseline, were suggested, or can readily be adapted to immune repertoire classification. The methods comprise (i) known motif, which counts how often the known implanted motifs occur, (ii) Support Vector Machine (SVM) approach that uses a fixed mapping from a bag of sequences to the corresponding $k$-mer counts and used the MinMax and Jaccard kernel, (iii) $k$-Nearest Neighbor (KNN) with $k$-mer representation, transforming MinMax and Jaccard kernel to distances, (iv) logistic regression on the $k$-mer representation, (v) burden test that first identifies sequences or $k$-mers and then computes a burden score per individual, and (vi) logistic multiple instance learning (lMIL). On the real-world dataset DeepRC achieved an AUC of $0.832 \pm 0.022$, followed by the SVM with MinMax kernel (AUC $0.825 \pm 0.022$) and the burden test with an AUC of $0.699 \pm 0.041$. Overall on all datasets, DeepRC outperformed all competing methods with respect to average AUC (see Widrich et al. (2020a;b)).

Table A.1 reports the average performance in the simulated immunosequencing datasets (last column) and the performance on datasets of the remaining three categories. DeepRC outperforms all competing methods with respect to average AUC. Across categories, the runner-up methods are either the SVM for MIL problems with MinMax kernel or the burden test.

| | Real-world | Real-world data with implanted signals | | | | LSTM-generated data | | | | | Simulated |
| --- | --- | --- | --- | --- | --- | --- | --- | --- | --- | --- | --- |
| | CMV | OM 1% | OM 0.1% | MM 1% | MM 0.1% | 10% | 1% | 0.5% | 0.1% | 0.05% | avg. |
| DeepRC | **0.832** ± 0.022 | **1.00** ± 0.00 | **0.98**± 0.01 | **1.00**± 0.00 | **0.94**±0.01 | **1.00**± 0.00 | **1.00**± 0.00 | **1.00**± 0.00 | **1.00**± 0.00 | **1.00**± 0.00 | **0.846**± 0.223 |
| SVM (MM) | 0.825 ± 0.022 | **1.00** ± 0.00 | 0.58± 0.02 | **1.00**± 0.00 | 0.53±0.02 | **1.00**± 0.00 | **1.00**± 0.00 | **1.00**± 0.00 | **1.00**± 0.00 | 0.99± 0.01 | 0.827± 0.210 |
| SVM (J) | 0.546 ± 0.021 | 0.99 ± 0.00 | 0.53± 0.02 | **1.00**± 0.00 | 0.57±0.02 | 0.98± 0.04 | **1.00**± 0.00 | **1.00**± 0.00 | 0.90± 0.04 | 0.77± 0.07 | 0.550± 0.080 |
| KNN (MM) | 0.679 ± 0.076 | 0.74 ± 0.24 | 0.49± 0.03 | 0.67± 0.18 | 0.50±0.02 | 0.70± 0.27 | 0.72± 0.26 | 0.73± 0.26 | 0.54± 0.16 | 0.52± 0.15 | 0.634± 0.129 |
| KNN (J) | 0.534 ± 0.039 | 0.65 ± 0.16 | 0.48± 0.03 | 0.70± 0.20 | 0.51±0.03 | 0.70± 0.29 | 0.61± 0.24 | 0.52± 0.16 | 0.55± 0.19 | 0.54± 0.19 | 0.501± 0.007 |
| Log. regr. | 0.607 ± 0.058 | **1.00** ± 0.00 | 0.54± 0.04 | 0.99± 0.00 | 0.51±0.04 | **1.00**± 0.00 | **1.00**± 0.00 | 0.93± 0.15 | 0.60± 0.19 | 0.43± 0.16 | 0.826± 0.211 |
| Burden test | 0.699 ± 0.041 | **1.00** ± 0.00 | 0.64± 0.05 | **1.00**± 0.00 | 0.89±0.02 | **1.00**± 0.00 | **1.00**± 0.00 | **1.00**± 0.00 | **1.00**± 0.00 | 0.79± 0.28 | 0.549± 0.074 |
| Log. MIL (KMER) | 0.582 ± 0.065 | 0.54 ± 0.07 | 0.51± 0.03 | 0.99± 0.00 | 0.62±0.15 | **1.00**± 0.00 | 0.72± 0.11 | 0.64± 0.14 | 0.57± 0.15 | 0.53± 0.13 | 0.665± 0.224 |
| Log. MIL (TCRβ) | 0.515 ± 0.073 | 0.50 ± 0.03 | 0.50± 0.02 | 0.99± 0.00 | 0.78±0.03 | 0.54± 0.09 | 0.57± 0.16 | 0.47± 0.09 | 0.51± 0.07 | 0.50± 0.12 | 0.501± 0.016 |
| Known motif b. | – | 1.00 ± 0.00 | 0.70± 0.03 | 0.99± 0.00 | 0.62±0.04 | 1.00± 0.00 | 1.00± 0.00 | 1.00± 0.00 | 1.00± 0.00 | 1.00± 0.00 | 0.890± 0.168 |
| Known motif c. | – | 0.92 ± 0.00 | 0.56± 0.03 | 0.65± 0.03 | 0.52±0.03 | 1.00± 0.00 | 1.00± 0.00 | 0.99± 0.01 | 0.72± 0.09 | 0.63± 0.09 | 0.738± 0.202 |

Table A.1: Results immune repertoire classification across all datasets. Results are given in terms of AUC of the competing methods on all datasets. The reported errors are standard deviations across 5 cross-validation (CV) folds (except for the column "Simulated"). **Real-world CMV:** Average performance over 5 CV folds on the *cytomegalovirus (CMV) dataset* Emerson et al. (2017). **Real-world data with implanted signals:** Average performance over 5 CV folds for each of the four datasets. A signal was implanted with a frequency (=wittness rate) of $1\%$ or $0.1\%$. Either a single motif ("OM") or multiple motifs ("MM") were implanted. **LSTM-generated data:** Average performance over 5 CV folds for each of the 5 datasets. In each dataset, a signal was implanted with a frequency of $10\%$, $1\%$, $0.5\%$, $0.1\%$, and $0.05\%$, respectively. **Simulated:** Here we report the mean over 18 simulated datasets with implanted signals and varying difficulties. The error reported is the standard deviation of the AUC values across the 18 datasets.

### A.5.2.2 Multiple Instance Learning Benchmark Datasets.

Classical benchmarking datasets comprise UCSB breast cancer classification (Kandemir et al., 2014), and the Elephant, Fox, Tiger datasets (Andrews et al., 2003).

Elephant, Fox and Tiger are MIL datasets for image annotation which comprise color images from the Corel dataset that have been preprocessed and segmented. An image consists of a set of segments (or blobs), each characterized by color, texture and shape descriptors. The datasets have 100 positive and 100 negative example images. The latter have been randomly drawn from a pool of photos of other animals. Elephant has 1391 instances and 230 features. Fox has 1320 instances and 230 features. Tiger has 1220 instances and 230 features. Furthermore, we use the UCSB breast cancer classification (Kandemir et al., 2014) dataset, which consists of 2,002 instances across 58 input objects. An instance represents a patch of a histopathological image of cancerous or normal tissue. The layer `HopfieldPooling` is used, which allows for computing a per-input-object representation by

| parameter | values |
|---|---|
| learning rates | $\{10^{-3}, 10^{-5}\}$ |
| learning rate decay ($\gamma$) | $\{0.98, 0.96, 0.94\}$ |
| embedding layers | $\{1, 2, 3\}$ |
| layer widths | $\{32, 64, 256, 1024, 2048\}$ |
| number of heads | $\{8, 12, 16, 32\}$ |
| head dimensions | $\{16, 32, 64\}$ |
| scaling factors | $\{0.1, 1.0, 10.0\}$ |
| hidden dimensions | $\{32, 64, 128\}$ |
| bag dropout | $\{0.0, 0.75\}$ |

Table A.2: Hyperparameter search-space of a manual hyperparameter selection on the respective validation sets of the Elephant, Fox, Tiger and UCSB breast cancer datasets.

extracting an average of instances that are indicative for one of the two classes. The input to the `HopfieldPooling` layer is a set of embedded instances $Y$ and a trainable but fixed state (query) pattern $Q$ used for averaging of class-indicative instances. This averaging enables a compression of variable-sized bags to a fixed-sized representation to discriminate the bags. We performed a manual hyperparameter search on a validation set. In detail, we used the following architecture to perform the given task on the Elephant, Fox, Tiger and UCSCB breast cancer datasets: (I) we apply fully connected linear embedding layers with ReLU activation. (II) The output of this embedding serves as the input to our `HopfieldPooling` layer where the above described pooling operation is performed. (III) Thereafter we use 'ReLU - Linear blocks' as the final linear output layers that perform the classification. Among other hyperparameters, different hidden layer widths (for the fully connected pre- and post-`HopfieldPooling` layers), learning rates and batch sizes were tried. Additionally our focus resided on the hyperparameters of the `HopfieldPooling` layer. Among those were the number of heads, the head dimension and the scaling factor $\beta$. All models were trained for 160 epochs using the AdamW optimizer (Loshchilov & Hutter, 2017) with exponential learning rate decay (see Table A.2), and validated by 10-fold nested cross validation repeated five times with different splits on the data sets. The reported ROC AUC scores are the average of these repetitions. As overfitting imposed quite a problem, bag dropout was applied as the regularization technique of choice.

**A.5.3.1 Motivation.** Datasets with a small number of samples, like the UCI benchmark datasets, are particularly difficult for neural networks to generalize on. In contrast to their performance on larger datasets, they are consistently outperformed by methods like e.g. gradient boosting, random forests (RF) and support vector machines (SVMs). Finding samples or even learning prototypes that are highly indicative for the class of a sample (query) suggest the use of Hopfield networks. We applied a modern Hopfield network via the layer `Hopfield`. The input vector is mapped to $\boldsymbol{R}$ using a self-normalizing net (SNN) and $\boldsymbol{W}_K$ is learned, where the dimension of $\boldsymbol{W}_K$ (the number of stored fixed pattern) is a hyperparameter. The output $\boldsymbol{Z}$ of `Hopfield` enters the output layer.

**A.5.3.2 Methods compared.** Modern Hopfield networks via the layer Hopfield are compared to 17 groups of methods (Fernández-Delgado et al., 2014; Klambauer et al., 2017a):

1. Support Vector Machines
2. Random Forest
3. Multivariate adaptive regression splines (MARS)
4. Boosting
5. Rule-based Methods
6. Logistic and Multinomial Regression (LMR)
7. Discriminant Analysis (DA)
8. Bagging
9. Nearest Neighbor
10. Decision Trees
11. Other Ensembles
12. Neural Networks (standard NN, BatchNorm, WeighNorm, MSRAinit, LayerNorm, ResNet, Self-Normalizing Nets)
13. Bayesian Methods
14. Other Methods
15. Generalized linear models (GLM)
16. Partial Least Squares and Principal Component Regression (PLSR)
17. Stacking (Wolpert)

**A.5.3.3 Experimental design and implementation details.** As specified in the main paper, we consider 75 datasets of the *UC Irvine Machine Learning Repository*, which contain less than $1,000$ samples per dataset, following the dataset separation into large and small dataset in Klambauer et al. (2017a). On each dataset, we performed a grid-search to determine the best hyperparameter setting and model per dataset. The hyperparameter search-space of the grid-search is listed in Table A.3. All models were trained for 100 epochs with a mini-batch size of 4 samples using the cross entropy loss and the PyTorch SGD module for stochastic gradient descent without momentum and without weight decay or dropout. After each epoch, the model accuracy was computed on a separated validation set. Using early stopping, the model with the best validation set accuracy averaged over 16 consecutive epochs was selected as final model. This final model was then evaluated against a separated test set to determine the accuracy, as reported in Tables 2 and Table `uci_detailed_results.csv` in the supplemental materials.

As network architecture, we use $\{0, 1, 7\}$ fully connected embedding layers with SELU Klambauer et al. (2017a) activation functions and $\{32, 128, 1024\}$ hidden units per embedding layer. These embedding layers are followed by the layer `Hopfield`. The number of hidden units is also used as number of dimensions for the Hopfield association space with a number of $\{1, 32\}$ heads. The layer `Hopfield` is followed by a mapping to the output vector, which has as dimension the number of classes. Finally, the softmax function is applied to obtain the predicted probability for a class.

| parameter | values |
|---|---|
| learning rates | $\{0.05\}$ |
| embedding layers | $\{0, 1, 7\}$ |
| hidden units | $\{32, 128, 1024\}$ |
| heads | $\{1, 32\}$ |
| $\beta$ | $\{1.0, 0.1, 0.001\}$ |
| # stored patterns | $\{1, 8\} \cdot n\_classes$ |

Table A.3: Hyperparameter search-space for grid-search on small UCI benchmark datasets. All models were trained for 100 epochs using stochastic gradient descent with early stopping based on the validation set accuracy and a minibatch size of 4 samples. The number of stored patterns is depending on the number of target classes of the individual tasks.

**A.5.3.4   Results.**   We compared the performance of 25 methods based on their method rank. For this we computed the rank per method per dataset based on the accuracy on the test set, which was then averaged over all 75 datasets for each method to obtain the method rank. For the baseline methods we used the scores summarized by (Klambauer et al., 2017a).

| parameter | values |
|---|---|
| beta | $\{0.0001, 0.001, 0.01, 0.1, 0.2, 0.3\}$ |
| learning rates | $\{0.0002\}$ |
| heads | $\{1, 32, 128, 512\}$ |
| dropout | $\{0.0, 0.1, 0.2\}$ |
| state-pattern bias | $\{0.0, -0.1, -0.125, 0.15, -0.2\}$ |
| association-activation | {None, LeakyReLU } |
| state- and stored-pattern static | {False, True} |
| normalize state- and stored-pattern | {False, True} |
| normalize association projection | {False, True} |
| learnable stored-pattern | {False, True} |

Table A.4: Hyperparameter search-space for grid-search on HIV, BACE, BBBP and SIDER. All models were trained if applicable for $4$ epochs using Adam and a batch size of 1 sample.

### A.5.4  EXPERIMENT 4: DRUG DESIGN BENCHMARK DATASETS

**A.5.4.1  Experimental design and implementation details.**   We test Hopfield layers on 4 classification datasets from MoleculeNet (Wu et al., 2017), which are challenging for deep learning methods. The first dataset is HIV, which was introduced by the Drug Therapeutics Program (DTP) AIDS Antiviral Screen. The second dataset is BACE, which has IC50 measurements for binding affinities of inhibitors (molecules) to the human $\beta$-secretase 1 (BACE-1). The third dataset is BBBP (blood-brain barrier permeability), which stems from modeling and predicting the blood-brain barrier permeability (Martins et al., 2012). The fourth dataset is SIDER (Side Effect Resource) Kuhn et al. (2016) and contains 1427 approved drugs. These datasets represent four areas of modeling tasks in drug discovery, concretely to develop accurate models for predicting a) new anti-virals (HIV), b) new protein inhibitors (BACE), c) metabolic effects (BBBP), and d) side effects of a chemical compound (SIDER).

We implemented a Hopfield layer `HopfieldLayer`, in which we used the training-input as stored-pattern $\boldsymbol{Y}$ or key, the training-label as pattern-projection $\boldsymbol{YW}_V$ or value and the input as state-pattern $\boldsymbol{R}$ or query. As described in section A.6 by concatenation of input $\boldsymbol{z}_i$ and target $\boldsymbol{t}_i$ the matrices $\boldsymbol{W}_K$ and $\boldsymbol{W}_V$ can be designed such that inside the softmax the input $\boldsymbol{z}_i$ is used and outside the softmax the target $\boldsymbol{t}_i$.

All hyperparameters were selected on separate validation sets and we selected the model with the highest validation AUC on five different random splits.

**A.5.4.2  Results.**   We compared the Hopfield layer `Hopfieldlayer` to Support Vector Machines (SVMs) (Cortes & Vapnik, 1995; Schölkopf & Smola, 2002), Extreme Gradient Boosting (XGBoost) (Chen & Guestrin, 2016), Random Forest (RF) (Breiman, 2001), Deep Neural Networks (DNNs) (LeCun et al., 2015; Schmidhuber, 2015), and to graph neural networks (GNN) like Graph Convolutional Networks (GCNs) (Kipf & Welling, 2016), Graph Attention Networks (GATs) (Veličković et al., 2018), Message Passing Neural Networks (MPNNs) (Gilmer et al., 2017), and Attentive FP (Xiong et al., 2020). Our architecture with `HopfieldLayer` has reached state-of-the-art for predicting side effects on SIDER $0.672 \pm 0.019$ as well as for predicting $\beta$-secretase BACE $0.902 \pm 0.023$. See Table A.5 for all results, where the results of other methods are taken from Jiang et al. (2020).

Table A.5: Results on drug design benchmark datasets. Predictive performance (ROCAUC) on test set as reported by Jiang et al. (2020) for 50 random splits

| Model | HIV | BACE | BBBP | SIDER |
|---|---|---|---|---|
| SVM | $0.822 \pm 0.020$ | $0.893 \pm 0.020$ | $0.919 \pm 0.028$ | $0.630 \pm 0.021$ |
| XGBoost | $0.816 \pm 0.020$ | $0.889 \pm 0.021$ | $\mathbf{0.926 \pm 0.026}$ | $0.642 \pm 0.020$ |
| RF | $0.820 \pm 0.016$ | $0.890 \pm 0.022$ | $\mathbf{0.927 \pm 0.025}$ | $0.646 \pm 0.022$ |
| GCN | $\mathbf{0.834 \pm 0.025}$ | $0.898 \pm 0.019$ | $0.903 \pm 0.027$ | $0.634 \pm 0.026$ |
| GAT | $0.826 \pm 0.030$ | $0.886 \pm 0.023$ | $0.898 \pm 0.033$ | $0.627 \pm 0.024$ |
| DNN | $0.797 \pm 0.018$ | $0.890 \pm 0.024$ | $0.898 \pm 0.033$ | $0.627 \pm 0.024$ |
| MPNN | $0.811 \pm 0.031$ | $0.838 \pm 0.027$ | $0.879 \pm 0.037$ | $0.598 \pm 0.031$ |
| Attentive FP | $0.822 \pm 0.026$ | $0.876 \pm 0.023$ | $0.887 \pm 0.032$ | $0.623 \pm 0.026$ |
| Hopfield (ours) | $0.815 \pm 0.023$ | $\mathbf{0.902 \pm 0.023}$ | $0.910 \pm 0.026$ | $\mathbf{0.672 \pm 0.019}$ |

## A.6 PyTorch Implementation of Hopfield Layers

The implementation is available at: https://github.com/ml-jku/hopfield-layers

### A.6.1 Introduction

In this section, we describe the implementation of Hopfield layers in PyTorch (Paszke et al., 2017; 2019) and, additionally, provide a brief usage manual. Possible applications for a Hopfield layer in a deep network architecture comprise:

- multiple instance learning (MIL) (Dietterich et al., 1997),

- processing of and learning with point sets (Qi et al., 2017a;b; Xu et al., 2018),

- set-based and permutation invariant learning (Guttenberg et al., 2016; Ravanbakhsh et al., 2016; Zaheer et al., 2017; Korshunova et al., 2018; Ilse et al., 2018; Zhai et al., 2020),

- attention-based learning (Vaswani et al., 2017a),

- associative learning,

- natural language processing,

- sequence analysis and time series prediction, and

- storing and retrieving reference or experienced data, e.g. to store training data and retrieve it by the model or to store experiences for reinforcement learning.

The Hopfield layer in a deep neural network architecture can implement:

- a memory (storage) with associative retrieval (Danihelka et al., 2016; Ba et al., 2016a),

- conditional pooling and averaging operations (Wang et al., 2018; Ilse et al., 2020),

- combining data by associations (Agrawal et al., 1993),

- associative credit assignment (e.g. Rescorla-Wagner model or value estimation) (Sutton & Barto, 2018), and

- attention mechanisms (Vaswani et al., 2017a; Bahdanau et al., 2014).

In particular, a Hopfield layer can substitute attention layers in architectures of transformer and BERT models. The Hopfield layer is designed to be used as plug-in replacement for existing layers like

- pooling layers (max-pooling or average pooling),

- permutation equivariant layers (Guttenberg et al., 2016; Ravanbakhsh et al., 2016),

- GRU & LSTM layers, and

- attention layers.

In contrast to classical Hopfield networks, the Hopfield layer is based on the modern Hopfield networks with continuous states that have increased storage capacity, as discussed in the main paper. Like classical Hopfield networks, the dynamics of the single heads of a Hopfield layer follow a energy minimization dynamics. The energy minimization empowers our Hopfield layer with several advantages over other architectural designs like memory cells, associative memory, or attention mechanisms. For example, the Hopfield layer has more functionality than a transformer self-attention layer (Vaswani et al., 2017a) as described in Sec. A.6.2. Possible use cases are given in Sec. A.6.3. Source code will be provided under `github`.

### A.6.2 FUNCTIONALITY

Non-standard functionalities that are added by a Hopfield layer are

- *Association of two sets*,
- *Multiple Updates* for precise fixed points,
- *Variable Beta* that determines the kind of fixed points,
- *Dimension of the associative space* for controlling the storage capacity,
- *Static Patterns* for fixed pattern search, and
- *Pattern Normalization* to control the fixed point dynamics by norm of the patterns and shift of the patterns.

A functional sketch of our Hopfield layer is shown in Fig. A.7.

•*Association of two sets.* The Hopfield layer makes it possible to associate two sets of vectors. This general functionality allows

- for transformer-like self-attention,
- for decoder-encoder attention,
- for time series prediction (maybe with positional encoding),
- for sequence analysis,
- for multiple instance learning,
- for learning with point sets,
- for combining data sources by associations,
- for constructing a memory,
- for averaging and pooling operations, and
- for many more.

The first set of vectors consists of $S$ *raw state patterns* $\boldsymbol{R} = (\boldsymbol{r}_1, \ldots, \boldsymbol{r}_S)^T$ with $\boldsymbol{r}_s \in \mathbb{R}^{d_r}$ and the second set of vectors consists of $N$ *raw stored patterns* $\boldsymbol{Y} = (\boldsymbol{y}_1, \ldots, \boldsymbol{y}_N)^T$ with $\boldsymbol{y}_i \in \mathbb{R}^{d_y}$. Both the $S$ raw state patterns and $N$ raw stored patterns are mapped to an associative space in $\mathbb{R}^{d_k}$ via the matrices $\boldsymbol{W}_Q \in \mathbb{R}^{d_r \times d_k}$ and $\boldsymbol{W}_K \in \mathbb{R}^{d_y \times d_k}$, respectively. We define a matrix $\boldsymbol{Q}$ ($\boldsymbol{\Xi}^T$) of *state patterns* $\boldsymbol{\xi}_n = \boldsymbol{W}_Q \boldsymbol{r}_n$ in an associative space $\mathbb{R}^{d_k}$ and a matrix $\boldsymbol{K}$ ($\boldsymbol{X}^T$) of *stored patterns* $\boldsymbol{x}_i = \boldsymbol{W}_K \boldsymbol{y}_s$ in the associative space $\mathbb{R}^{d_k}$:

$$\boldsymbol{Q} \ = \ \boldsymbol{\Xi}^T \ = \ \boldsymbol{R}\,\boldsymbol{W}_Q \,, \tag{549}$$

$$\boldsymbol{K} \ = \ \boldsymbol{X}^T \ = \ \boldsymbol{Y}\,\boldsymbol{W}_K \,. \tag{550}$$

In the main paper, Eq. (3) defines the novel update rule:

$$\boldsymbol{\xi}^{\mathrm{new}} \ = \ f(\boldsymbol{\xi}) \ = \ \boldsymbol{X}\,\mathrm{softmax}(\beta\,\boldsymbol{X}^T \boldsymbol{\xi}) \,, \tag{551}$$

For multiple patterns, Eq. (3) becomes:

$$\boldsymbol{\Xi}^{\mathrm{new}} \ = \ f(\boldsymbol{\Xi}) \ = \ \boldsymbol{X}\,\mathrm{softmax}(\beta\,\boldsymbol{X}^T \boldsymbol{\Xi}) \,, \tag{552}$$

where $\boldsymbol{\Xi} = (\boldsymbol{\xi}_1, \ldots, \boldsymbol{\xi}_N)$ is the matrix of $N$ state (query) patterns, $\boldsymbol{X}$ is the matrix of stored (key) patterns, and $\boldsymbol{\Xi}^{\mathrm{new}}$ is the matrix of new state patterns, which are averages over stored patterns. A new state pattern can also be very similar to a single stored pattern, in which case we call the stored pattern to be retrieved.

These matrices allow to rewrite Eq. (552) as:

$$(\boldsymbol{Q}^{\mathrm{new}})^T = \boldsymbol{K}^T \mathrm{softmax}(\beta\, \boldsymbol{K}\, \boldsymbol{Q}^T)\,. \tag{553}$$

For $\beta = 1/\sqrt{d_k}$ and changing in Eq. (553) softmax $\in \mathbb{R}^N$ to a row vector (and evaluating a row vector), we obtain:

$$\boldsymbol{Q}^{\mathrm{new}} = \mathrm{softmax}(1/\sqrt{d_k}\, \boldsymbol{Q}\, \boldsymbol{K}^T)\, \boldsymbol{K}\,, \tag{554}$$

where $\boldsymbol{Q}^{\mathrm{new}}$ is again the matrix of new state patterns. The new state patterns $\boldsymbol{\Xi}^{\mathrm{new}}$ are projected via $\boldsymbol{W}_V$ to the result patterns $\boldsymbol{Z} = \boldsymbol{\Xi}^{\mathrm{new}}\boldsymbol{W}_V$, where $\boldsymbol{W}_V \in \mathbb{R}^{d_k \times d_v}$. With the pattern projection $\boldsymbol{V} = \boldsymbol{K}\boldsymbol{W}_V$, we obtain the update rule Eq. (10) from the main paper:

$$\boldsymbol{Z} = \mathrm{softmax}(1/\sqrt{d_k}\, \boldsymbol{Q}\, \boldsymbol{K}^T)\, \boldsymbol{V}\,. \tag{555}$$

•*Multiple Updates.* The update Eq. (553) can be iteratively applied to the initial state $\boldsymbol{\xi}$ of every Hopfield layer head. After the last update, the new states $\boldsymbol{\Xi}^{\mathrm{new}}$ are projected via $\boldsymbol{W}_V$ to the result patterns $\boldsymbol{Z} = \boldsymbol{\Xi}^{\mathrm{new}}\boldsymbol{W}_V$. Therefore, the Hopfield layer allows multiple update steps in the forward pass without changing the number of parameters. The number of update steps can be given for every Hopfield head individually. Furthermore, it is possible to set a threshold for the number of updates of every Hopfield head based on $\|\boldsymbol{\xi} - \boldsymbol{\xi}^{\mathrm{new}}\|_2$. In the general case of multiple initial states $\boldsymbol{\Xi}$, the maximum over the individual norms is taken.

•*Variable $\beta$.* In the main paper, we have identified $\beta$ as a crucial parameter for the fixed point dynamics of the Hopfield network, which governs the operating mode of the attention heads. In appendix, e.g. in Lemma A7 or in Eq. (102) and Eq. (103), we showed that the characteristics of the fixed points of the new modern Hopfield network are determined by: $\beta$, $M$ (maximal pattern norm), $m_{\mathrm{max}}$ (spread of the similar patterns), and $\|\boldsymbol{m_x}\|$ (center of the similar patterns). Low values of $\beta$ induce global averaging and higher values of $\beta$ metastable states. In the transformer attention, the $\beta$ parameter is set to $\beta = 1/\sqrt{d_k}$ as in Eq. (555). The Hopfield layer, however, allows to freely choose $\beta > 0$, since the fixed point dynamics does not only depend on the dimension of the associative space $d_k$. Additionally, $\beta$ heavily influences the gradient flow to the matrices $\boldsymbol{W}_Q$ and $\boldsymbol{W}_K$. Thus, finding the right $\beta$ for the respective application can be crucial.

•*Variable dimension of the associative space.* Theorem A5 says that the storage capacity of the modern Hopfield network grows exponentially with the dimension of the associative space. However higher dimension of the associative space also means less averaging and smaller metastable states. The dimension of the associative space trades off storage capacity against the size of metastable states, e.g. over how many pattern is averaged. In Eq. (550) and in Eq. (549), we assumed $N$ raw state patterns $\boldsymbol{R} = (\boldsymbol{r}_1, \ldots, \boldsymbol{r}_N)^T$ and $S$ raw stored patterns $\boldsymbol{Y} = (\boldsymbol{y}_1, \ldots, \boldsymbol{y}_S)^T$ that are mapped to a $d_k$-dimensional associative space via the matrices $\boldsymbol{W}_Q \in \mathbb{R}^{d_r \times d_k}$ and $\boldsymbol{W}_K \in \mathbb{R}^{d_y \times d_k}$, respectively. In the associative space $\mathbb{R}^{d_k}$, we obtain the state patterns $\boldsymbol{Q} = \boldsymbol{\Xi}^T = \boldsymbol{R}\boldsymbol{W}_Q$ and the stored patterns $\boldsymbol{K} = \boldsymbol{X}^T = \boldsymbol{Y}\,\boldsymbol{W}_K$. The Hopfield view relates the dimension $d_k$ to the number of input patterns $N$ that have to be processed. The storage capacity depends exponentially on the dimension $d_k$ (the dimension of the associative space) and the size to metastable states is governed by this dimension, too. Consequently, $d_k$ should be chosen with respect to the number $N$ of patterns one wants to store and the desired size of metastable states, which is the number of patterns one wants to average over. For example, if the input consists of many low dimensional input patterns, it makes sense to project the patterns into a higher dimensional space to allow a proper fixed point dynamics. Intuitively, this coincides with the construction of a richer feature space for the patterns.

•*Static Patterns.* In Eq. (550) and Eq. (549), the $N$ raw state patterns $\boldsymbol{R} = (\boldsymbol{r}_1, \ldots, \boldsymbol{r}_N)^T$ and $S$ raw stored patterns $\boldsymbol{Y} = (\boldsymbol{y}_1, \ldots, \boldsymbol{y}_S)^T$ are mapped to an associative space via the matrices $\boldsymbol{W}_Q \in \mathbb{R}^{d_r \times d_k}$ and $\boldsymbol{W}_K \in \mathbb{R}^{d_y \times d_k}$, which gives the state patterns $\boldsymbol{Q} = \boldsymbol{\Xi}^T = \boldsymbol{R}\boldsymbol{W}_Q$ and the stored patterns $\boldsymbol{K} = \boldsymbol{X}^T = \boldsymbol{Y}\,\boldsymbol{W}_K$. We allow for static state and static stored patterns. Static pattern means that the pattern does not depend on the network input, i.e. it is determined by the bias weights and remains constant across different network inputs. Static state patterns allow to determine whether

particular fixed patterns are among the stored patterns and vice versa. The static pattern functionality is typically needed if particular patterns must be identified in the data, e.g. as described for immune repertoire classification in the main paper, where a fixed $d_k$-dimensional state vector $\boldsymbol{\xi}$ is used.

•*Pattern Normalization.* In the appendix, e.g. in Lemma A7 or in Eq. (102) and Eq. (103), we showed that the characteristics of the fixed points of the new modern Hopfield network are determined by: $\beta$, $M$ (maximal pattern norm), $m_{\max}$ (spread of the similar patterns), and $\|\boldsymbol{m_x}\|$ (center of the similar patterns). We already discussed the parameter $\beta$ while the spread of the similar patterns $m_{\max}$ is given by the data. The remaining variables $M$ and $\boldsymbol{m_x}$ that both control the fixed point dynamics are adjusted pattern normalization. $M$ is the maximal pattern norm and $\boldsymbol{m_x}$ the center of the similar patterns. Theorem A5 says that larger $M$ allows for more patterns to be stored. However, the size of metastable states will decrease with increasing $M$. The vector $\boldsymbol{m_x}$ says how well the (similar) patterns are centered. If the norm $\|\boldsymbol{m_x}\|$ is large, then this leads to smaller metastable states. The two parameters $M$ and $\boldsymbol{m_x}$ are controlled by pattern normalization and determine the size and convergence properties of metastable states. These two parameters are important for creating large gradients if heads start with global averaging which has small gradient. These two parameters can shift a head towards small metastable states which have largest gradient as shown in Fig. A.5(b). We allow for three different pattern normalizations:

- pattern normalization of the input patterns,
- pattern normalization after mapping into the associative space,
- no pattern normalization.

The default setting is a pattern normalization of the input patterns.

### A.6.3 USAGE

As outlined in Sec. A.6.1, there are a variety of possible use cases for the Hopfield layer, e.g. to build memory networks or transformer models. The goal of the implementation is therefore to provide an easy to use Hopfield module that can be used in a wide range of applications, be it as part of a larger architecture or as a standalone module. Consequently, the focus of the Hopfield layer interface is set on its core parameters: the association of two sets, the scaling parameter $\beta$, the maximum number of updates, the dimension of the associative space, the possible usage of static patterns, and the pattern normalization. The integration into the PyTorch framework is built such that with all the above functionalities disabled, the "HopfieldEncoderLayer" and the "HopfieldDecoderLayer", both extensions of the Hopfield module, can be used as a one-to-one plug-in replacement for the *TransformerEncoderLayer* and the *TransformerDecoderLayer*, respectively, of the PyTorch transformer module.

The Hopfield layer can be used to implement or to substitute different layers:

- **Pooling layers:** We consider the Hopfield layer as a pooling layer if only one static state (query) pattern exists. Then, it is de facto a pooling over the sequence, which results from the softmax values applied on the stored patterns. Therefore, our Hopfield layer can act as a pooling layer.

- **Permutation equivariant layers:** Our Hopfield layer can be used as a plug-in replacement for permutation equivariant layers. Since the Hopfield layer is an associative memory it assumes no dependency between the input patterns.

- **GRU & LSTM layers:** Our Hopfield layer can be used as a plug-in replacement for GRU & LSTM layers. Optionally, for substituting GRU & LSTM layers, positional encoding might be considered.

- **Attention layers:** Our Hopfield layer can act as an attention layer, where state (query) and stored (key) patterns are different, and need to be associated.

- Finally, the extensions of the Hopfield layer are able to operate as a self-attention layer (HopfieldEncoderLayer) and as cross-attention layer (HopfieldDecoderLayer), as described in (Vaswani et al., 2017a). As such, it can be used as building block of transformer-based or general architectures.

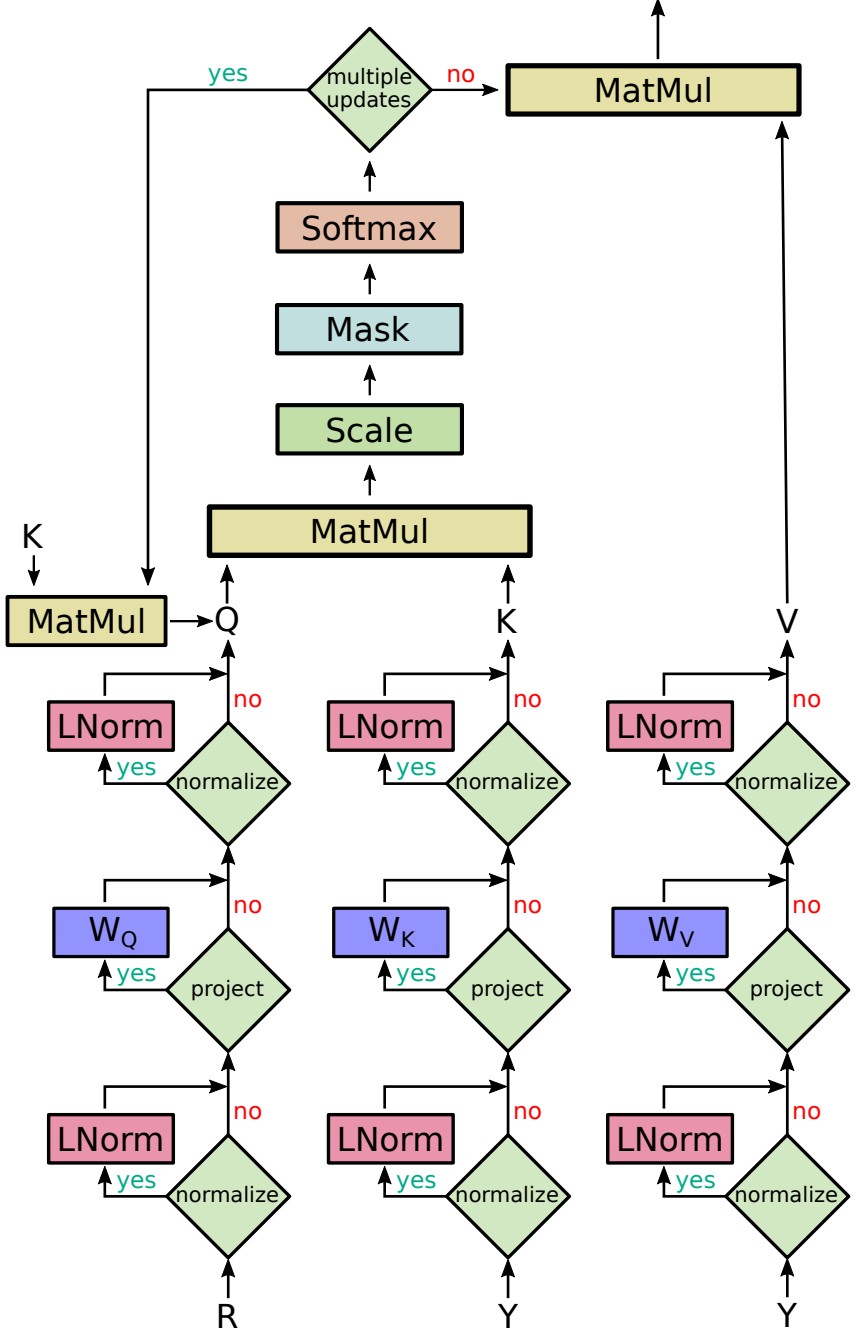

Figure A.7: A flowchart of the Hopfield layer. First, the raw state (query) patterns $R$ and the raw stored (key) patterns $Y$ are optionally normalized (with layer normalization), projected and optionally normalized (with layer normalization) again. The default setting is a layer normalization of the input patterns, and no layer normalization of the projected patterns. The raw stored patterns $Y$ can in principle be also two different input tensors. Optionally, multiple updates take place in the projected space of $Q$ and $K$. This update rule is obtained e.g. from the full update Eq. (423) or the simplified update Eq. (424) in the appendix.

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
