# OpenReview forum: "Hopfield Networks is All You Need"
_ICLR.cc/2021/Conference — ICLR 2021 Poster_

### Official Review · AnonReviewer3 · 2020-10-27
**Very interesting but missing critical content**

**Rating:** 7
**Confidence:** 4

**Review:**

This work extends the binary Hopfield network (Demircigil et al., 2017) to continuous patterns and states. Connections are drawn between the result model to the attention layers of the transformers, the pooling operation of LSTM, similarity search, and fully connected layers. Experimental results are briefly described for analyzing the attention of Bert models, multiple instance learning, and small UCI classification tasks.

The proposed model seems very interesting, and the proposed applications seem reasonable at a very high level. However, there is just not enough detail in this paper for me to understand how the models are implemented or why the model works better than other approaches.

For example, section 3 declared 3 types of Hopfield layers, but without any formal definitions to them, or how they are integrated to the proposed models. The experiment section compares performances with existing models, but lacks any analysis of why the proposed models work better. Similarly, there is a lack of motivation/intuition in the introduction section.

## After author feedback ##
Thanks for the paper update, and now I have a better understanding of the proposed approach. I have updated my review to the following:

Previously Widrich+ (2020) showed that integrating transformer-like attention (or equivalently modern Hopfield networks based on softmax)  into deep learning architectures outperforms existing methods (kNN and logistic regression) for massive MIL such as immune repertoire classification. More specifically a pooling layer can be formed by attending over a repertoire of instances with a fixed (but learnable) query vector.

This work provides theoretical analysis of such a layer for its energy function, convergence of updates, and storage capacity, and points to directions of how such a layer can be understood and controlled. It extends the previous experiment:
1) apply HopfieldPooling (attention with fixed learnable query Q) to more MIL datasets (animal image and breast cancer)  and achieve state of the art results.
2) apply Hopfield (attention) to 75 small UCI benchmarks replacing feedforward nets. Here Selu units (Klambauer+ 2017) are used to map input to storage Y  and query R. The result is quite positive beating previous approaches including SVM, random forest, and SNN (Klambauer+ 2017)
3) apply HopfieldLayer (attention with fixed training data Y as storage) to 4 drug design tasks  acting as an instance-based learning approach.

The result seems quite interesting indicating that general purpose layers such as  feedforward, pooling and nearest neighbors can be improved (in terms of robustness, learnability, or controllability) by adding attention like operations.

I think the paper can talk less about existing results, and focus more on the new results and their analysis:
- remove [Immune Repertoire Classification] result since it is from previous work.
- move the Drug Design experiment details to the main text, and add some comment about under what condition Hopfield outperforms/underperforms RF.
- for the UCI benchmark experiment the transformer layer (Vaswani+ 2017) seems to be a natural baseline and should be compared to.

Suggestions for the presentation:
- Should only in the future work section state that Hopfield can potentially substitute LSTMs or GRUs, since it is all hypothetical with no experiment result at this point.
- The word "implemented"  in Section 4 seems misleading as there is nothing changed in the Bert model structure? "Transformer and BERT models can be implemented by the layer Hopfield."
- Can be more specific in descriptions. For example in the description of (2) Layer HopfieldPooling and (3) Layer HopfieldLayer in Section 3, R and  W_K can be  referenced again for "state (query) patterns " and "The stored (key) patterns" respectively.
- It is probably more informative to replace figure 1 with a table to directly compare the energy function and updating rules of different Hopfield nets--i.e., classical, exponential and attention.
- Avoid using "x" in equation 1, since the symbol has already been used for the stored patterns.
- "HopfieldLayer" seems to be a very strange name.

---

> ### Author Response · Authors · 2020-11-17
> **Response to Reviewer 3**
>
> Thank you for your helpful review. Please, let us expand a bit on the weaknesses you pointed out.
>
>
> * “not enough detail in this paper for me to understand how the models are implemented or why the model works better than other approaches.” & “For example, section 3 declared 3 types of Hopfield layers, but without any formal definitions to them”: Sorry for the bad description. We massively extended Section 3, to describe (i) our main goal, (ii) how Hopfield networks are integrated into deep learning architectures, and (iii) how the layers are designed. We also give possible applications of the layers. Then we refer to the experiments, where the layers are used.
> Since gradients have to be propagated through these layers, we aim at obtaining continuous Hopfield networks that are differentiable and can retrieve by one update step. One update is equivalent to updating a layer in a neural network.
>
> * “why the model works better than other approaches.” & “but lacks any analysis of why the proposed models work better”:
>  We now give explanations why the proposed models work better. In particular, we show that Hopfield layers can realize k-nearest neighbor (with a learned distance metric), SVM-like methods (storing support vectors or reference vectors), similarity-based methods (similarity to stored patterns), learning vector quantization (stored patterns are the centers of clusters), etc. We also show that the transformer model can readily be implemented. Furthermore, the layers can perform simple pooling operations or store the elements of a time series, therefore, can replace LSTM or GRU layers.
>
> * “lack of motivation in the introduction section.”:
> We added a contribution paragraph to the introduction and massively extended Section 3. We now write in the abstract: “These Hopfield layers enable new ways of deep learning, beyond fully-connected, convolutional, or recurrent networks, and provide pooling, memory, association, and attention mechanisms.

---

### Official Review · AnonReviewer4 · 2020-10-30
**Paper makes good technical contribution draws interesting connections between classical Hopfield networks and Attention Mechanism in transformers**

**Rating:** 6
**Confidence:** 3

**Review:**

The paper introduces a new Hopfield network which have continuous states and propose update rules for optimizing it. It also draws connections between the new model and attention mechanism used in transformers. Small scale empirical study is presented.

Overall I like the technical contribution of the work but feel the paper could be revised to improve clarity about the optimization in the new proposed variant of hopfield networks. Below some specific comments:

Pros:
- connecting hopfield networks to attention mechanism and drawing out the variants in section 3 (as hopfield layers) is useful
- The exposition in section 1 and 2 where the authors describe the hopfield network with continuous states is written well (although I do feel the motivation behind update equations could be explained a bit better)

Cons:
- As I mentioned earlier, I don't fully understand the intuition behind convergence in one update. Can the authors clarify this? Also the paper mentions update rule in eqn (5) converges after one update for well separated patterns. What happens to the updates / optimization when the patterns are not well separated? This should be discussed after equation (5). Maybe present different scenarios to make it clear.

- Empirical study is limited in my opinion and can be improved. Is the trend in Fig 2 observed across more or less across all datasets? Can the authors comment on this? I like the visualization in the figure but it is bit hard to interpret (perhaps a more clearer label for it could help with that).

Other comments:
- The idea of separated patterns leads me to ask this question: is there any connection of this work to max-margin classifiers / kernel methods?

- Did the authors consider what would happen if non-linear transformations (e.g. activation functions in DNNs) are applied on top of the inputs? How does the existing network change in that case?

- Can the authors comment on the utility / challenges in applying their proposed method on datasets / tasks beyond the small scale UCI datasets used in their experiments? e.g. using them in large scale language modeling tasks where transformers are popular right now.

---

> ### Author Response · Authors · 2020-11-17
> **Response to Reviewer 4**
>
> Thank you for a very insightful review that helps us to improve our paper.
>
>
> * “clarity about the optimization in the new proposed variant of hopfield networks”:
>  We massively extended Section 3, to describe (i) our main goal, (ii) how Hopfield networks are integrated into deep learning architectures, and (iii) how the layers are designed.
>
> * “motivation behind update equations”:
> In the appendix “A.1.3 NEW UPDATE RULE“, we give in Eq. (29) for comparison, the synchronous update rule for the classical Hopfield network with threshold zero, which is very similar to our update rule but without the softmax. In appendix “Lemma A18”, we show that the softmax is the derivative of the Log-Sum-Exp Function.
>
> * “intuition behind convergence in one update”:
> Sorry, we mixed up mathematical convergence and retrieval, which is the convergence in praxis. We now separated mathematical convergence from retrieval (being close to a fixed point). We now define retrieval by an update that comes epsilon-close to the fixed point. Random patterns are mentioned.
>
> * What happens to the updates / optimization when the patterns are not well separated?:
> We discuss this case in the paragraph “Metastable states and one global fixed point.” We write: “If some vectors are similar to each other and well separated from all other vectors, then a metastable state near the similar vectors exists. Iterates that start near the metastable state converge to this metastable state, also if initialized by one of the similar patterns.” In this case these similar patterns are retrieved collectively. We write further: “If no pattern is well separated from the others, then the iteration converges to a global fixed point close to the arithmetic mean of the vectors.”
>
> * “Is the trend in Fig 2 observed across more or less across all datasets?”:
> Fig. 2 (now moved into the appendix) is specific to the transformer architecture and exemplary NLP tasks. We give in the appendix additional examples for this trend.
>
> * Other comments “max-margin classifiers / kernel methods”:
> We now give the connections to SVMs as the stored patterns can serve as support vectors. However, we do not see an obvious relation to max-margin classifiers.
>
> * Other comments “non-linear transformations”:
> Non-linear activation functions been used for the experiments in immune repertoire classification. We clarify that more, in particular in the appendix.
>
> * “using them in large scale language modeling tasks where transformers are popular right now.”:
> We have shown that the transformer attention mechanism is exactly the update rule of a modern Hopfield network. The transformer architecture is one example of applying our approach with modern Hopfield networks. The layer Hopfield together with residual connections (skip connections) gives the self-attention layer of the transformer. Also the encoder-decoder attention layer of the transformer can be realized, where Y comes from the encoder and R from the decoder. Also the layernorm is supplied automatically for the Hopfied layer by our Pytorch implementation. Since there are already many experiments with transformers, we focus on new tasks that can be solved with Hopfield networks.
> The attention mechanism of transformers is just an associative memory, where queries serve to retrieve keys that are stored. However, we can supply more functionalities by other Hopfield layers. Therefore, we focus on experiments, where these new architectures have not been tested. We achieved many new state-of-the-art for different datasets.

---

### Official Review · AnonReviewer1 · 2020-10-31
**Interesting results but some questions arise**

**Rating:** 7
**Confidence:** 3

**Review:**

This paper considers a continuous version of the classical Hopfield network (HN) model.In contrast to well studied discrete models where the patterns (vectors) that are
stored are discrete, this paper studied continuous vectors and a new continuous energy function.
Convergence results to a fixed point are proven for the new rule, and it is shown that for the case of random patterns, the Hopfield network can memorize exponentially many patterns (with high probability).  Finally several implementations are given showing how incorporating the new Hopfield net in classification tasks can improve classification accuracy in regimes where
data is scarce and where neural networks do not fare well.

The paper is rather long and I did not verify all results. The description appears sound.The proofs appear non-trivial and rather technical. While the results here are nontrivial I was left me wondering about the
added value of this new model. One of the biggest advantages of HN was its simplicity and elegance. More recent results of Hopfield and others with higher degree energy functions managed to maintain this clarity and brevity. The new model however is significantly more involved. It was not clear to me what is gained by this greater complexity and whether the gains
justify the larger complexity. In actual implementations very limited precision is often necessary.How does this discretization influence the continuous model? How robust is it to rounding errors? Don't we get "old" discrete models in disguise?

The (impressive) empirical results raise similar questions. Can't we use old discrete HN instead of the new model and achieve similar results? It would be perhaps more informative to compare different HN to the new model presented in this paper. It seems a bit strange that previous uses of HN (discrete ) did not achieve such an improvement in previous studies. It would be beneficial to add more on related work in this area.

 The authors might consider breaking their long paper to two different sections, one presenting the theoretical advantages of their new model and the other focusing on practical benefits.

Finally, the nature of convergence to a fixed point wasn't clear to me. It seems likely that if patterns are not random convergence can take a long time as is the case for discrete HN.
Some recent work about the complexity of finding fixed points of continuous functions may be relevant here:A converse to Banach's fixed point theorem and its CLS-completeness.
More specific comments:
1) The paper starts with a rather lengthy discussion of previous work.
I would recommend outlining the contributions of this paper earlier on.
2) "converge in one update step with exponentially low error and have storage capacity proportional to..." It was not clear to me that random patterns are considered here.
3) "proven for c= 1.37andc= 3.15 in Theorem 3" for what c exactly is the result proven?
4) "Furthermore, with a single update, the fixed point recovered with high probability"I presume this is true for random patterns?
5) Is beta>0?

---

> ### Author Response · Authors · 2020-11-17
> **Response to Reviewer 1**
>
> Thank you for a very elaborate review that helps us to improve our paper. We address the points individually.
>
>
> * “I was left me wondering about the added value of this new model”:
> The main goal of the paper is to integrate associative memories (the Hopfield layers) into deep learning architectures as layers. Therefore each layer can store and access raw input data, reference data, intermediate results, or (learned) prototypes. These Hopfield layers enable new ways of deep learning, beyond e.g. CNNs or recurrent networks. Hopfield layers can be used for multiple instance learning, point sets, or learning to process sequences. They enable the substitution of k-nearest neighbor, support vector machine models, or learning vector quantization in each layer separately. Also the transformer’s  self-attention and encoder-decoder attention are examples of Hopfield layers, but the interpretation as an associative memory is novel.
>
> * “It was not clear to me what is gained by this greater complexity and whether the gains justify the larger complexity.”:
> A continuous Hopfield network (not discrete) is necessary in order to enable end-to-end differentiable models. The continuous Hopfield network is integrated as a special layer in deep learning architectures, where backpropagation requires this layer to be differentiable. Therefore, we also investigate if one update is sufficient for being close to the fixed point. Integrated into a deep learning architecture, only one Hopfield update step should be performed, which is equivalent to updating a layer in a neural network. The reviewer might be right in their assumption that discrete networks might also do the job and the continuous models are the discrete models in disguise. However, it is not clear how to learn the weights that map to the embedding space, where the Hopfield network stores and retrieves patterns. We massively extended Section 3, to describe (i) our main goal, (ii) how Hopfield networks are integrated into deep learning architectures, and (iii) how the layers are designed.
>
> * “breaking their long paper to two different sections, one presenting the theoretical advantages of their new model and the other focusing on practical benefits”:
> Thanks for this advice. We do that: Section 2 is dedicated to theoretical considerations and Section 3 to practical / implementation details. We now massively extended Section 3.
>
> * “Comment 2” and “the nature of convergence to a fixed point wasn't clear to me” & “converge in one update step”:
> Sorry, we mixed up mathematical convergence and retrieval, which is the convergence in praxis. We now separated mathematical convergence from retrieval (being close to a fixed point). We now define retrieval by an update that comes epsilon-close to the fixed point. Random patterns are mentioned.
>
> * "Comment 3: proven for c= 1.37 and c= 3.15 in Theorem 3": Sorry for the ambiguous formulation. It is proven if the assumptions in Theorem 3 are fulfilled, reasonable settings which fulfill the assumptions are given for c=1.37 and c=3.15. This has been corrected.
>
> * “Comment 4: true for random patterns”:  Yes. We mention that now.
>
> * “Comment 5: Is beta>0”: Yes. We mention that now.

---

### Author Response · Authors · 2020-11-17
**Thank you for your feedback!**

We thank all reviewers for their time and for their constructive feedback. It helped us a lot to improve our paper. We hope to answer all questions and provide clarifications in individual responses to the respective reviewers. Further, we uploaded a rebuttal revision of our paper incorporating your sound suggestions. Concretely, we massively extended Section 3, to describe (i) our main goal, (ii) how Hopfield networks are integrated into deep learning architectures, and (iii) how the layers are designed.

---

### Decision · Program_Chairs · 2021-01-07
**Final Decision**

**Decision:**

Accept (Poster)

**Comment:**

The novelty of the paper are:
+ introduces a new Hopfield network with continuous states, hence can be learned end-to-end differentiation and back propagation.
+ derives efficient update rules
+ reveals a connection between the update rules and transformers
+ illustrate how the network can be used as a layer in deep neural network that can perform different functions

The presentation was clear enough for the reviewers to understand and appreciate the novelty, although there were a few points of confusion. I would recommend the authors to address several suggestions that came up in the discussions including:
- additional analysis to highlight when and how the networks is able to outperform other competing models
- intuitions about the proofs for the theorems (okay to leave the detailed derivation in the appendix)